# Combining agent-based, trait-based and demographic approaches to model coral-community dynamics

Bruno Sylvain Carturan[1]*, Jason Pither[1,2,3]†*, Jean-Philippe Maréchal[4], Corey JA Bradshaw[5], Lael Parrott[1,2,3]†*

[1]Department of Biology, University of British Columbia, Kelowna, Canada; [2]Institute for Biodiversity, Resilience, and Ecosystem Services, University of British Columbia, Kelowna, Canada; [3]Department of Earth, Environmental and Geographic Sciences, University of British Columbia, Kelowna, Canada; [4]Nova Blue Environment, Schoelcher, France; [5]Global Ecology, College of Science and Engineering, Flinders University, Adelaide, Australia

**Abstract** The complexity of coral-reef ecosystems makes it challenging to predict their dynamics and resilience under future disturbance regimes. Models for coral-reef dynamics do not adequately account for the high functional diversity exhibited by corals. Models that are ecologically and mechanistically detailed are therefore required to simulate the ecological processes driving coral reef dynamics. Here, we describe a novel model that includes processes at different spatial scales, and the contribution of species' functional diversity to benthic-community dynamics. We calibrated and validated the model to reproduce observed dynamics using empirical data from Caribbean reefs. The model exhibits realistic community dynamics, and individual population dynamics are ecologically plausible. A global sensitivity analysis revealed that the number of larvae produced locally, and interaction-induced reductions in growth rate are the parameters with the largest influence on community dynamics. The model provides a platform for virtual experiments to explore diversity-functioning relationships in coral reefs.

*For correspondence:
bruno.carturan@alumni.ubc.ca (BSC);
jason.pither@ubc.ca (JP);
lael.parrott@ubc.ca (LP)

†These authors contributed equally to this work

Competing interests: The authors declare that no competing interests exist.

## Introduction

Corals are the foundation species of coral-reef ecosystems and provide essential ecological functions for ecosystem resilience, as well as services to millions of people worldwide (*Moberg and Folke, 1999*; *Woodhead et al., 2019*). The loss of coral cover and the change in species composition occurring in communities around the globe (*Hughes et al., 2018*; *Torda et al., 2018*) therefore alter the functioning of the entire ecosystem. For instance, the replacement of morphologically complex, but highly sensitive species, by simpler and more resilient species reduces the overall architectural complexity of reef habitats (*Alvarez-Filip et al., 2009*)—this flattening reduces the diversity and abundance of fish (*Darling et al., 2017*; *Newman et al., 2015*) and macroinvertebrates (*Nelson et al., 2016*) and the functions they provide (e.g. grazing, bioerosion) (*Pratchett et al., 2018*).

Disentangling the identity effect (effects of individual species on processes) from the diversity effect (shared effects of a collection of species on processes) is necessary to define which species, functional groups, or aspects of diversity are essential for maintaining ecological processes (*Bellwood et al., 2019*; *Brandl et al., 2019*). While the effect of such essential species on ecosystem functions and the consequences of their loss has been widely reported (e.g. *Hughes, 1994*; *Alvarez-Filip et al., 2009*; *Hoey and Bellwood, 2009*), the relationship between species richness or functional diversity and ecosystem functioning is still poorly understood. Diversity is hypothesized to

enhance functioning because of niche complementarity and facilitation (*Bulleri et al., 2016*; *Loreau, 2000*), but tests of this hypothesis with corals are scarce (but see *McWilliam et al., 2018a*).

Even less understood is the effect of diversity on the temporal variability (i.e. stability) and resilience (i.e. resistance and recovery) of community-level aggregate properties such as biomass, percentage cover, and calcification rate (*Griffin et al., 2009*). Higher diversity is hypothesized to promote ecosystem stability and resilience via asynchronous, independent population dynamics and compensatory population dynamics resulting from functional redundancy and response diversity—the latter is the insurance hypothesis (reviewed in *Griffin et al., 2009* and *McCann, 2000*). Tests of the insurance hypothesis are rare for coral reefs (but see *Mellin et al., 2014* and *Nash et al., 2016* for fish, and *Zhang et al., 2014* and *Clements and Hay, 2019* for coral diversity).

Predicting how the reassembly of coral species in reefs alters ecosystem functioning is therefore a research priority to inform conservation management (*Bellwood et al., 2019*; *Graham et al., 2014*). To do this requires identifying the main ecosystem processes and associated functional traits affecting coral dynamics, and quantifying the relationship between the two (*Bellwood et al., 2019*; *Carturan et al., 2018*). Both virtual and physical experiments are also required (*Brandl et al., 2019*; *Mcleod et al., 2019*), where different aspects of diversity such as species and functional richness, and external factors such as disturbance regimes, larval connectivity, and grazing, are varied independently from one another to determine how they influence these dynamics. Unfortunately, physical experiments with coral species are unwieldy because of their slow growth rates and challenging environment in which they exist. However, models use simulation to overcome these challenges. To simulate the effect of species and functional diversity on ecosystem functioning realistically, models should include: (*i*) links between functional traits and processes; (*ii*) biotic interactions such as spatial competition, and (*iii*) population dynamics representing demographic structure. Although some coral models have been developed (reviewed in *Kubicek and Borell, 2011*; *Weijerman et al., 2015*), most implement only one or two of these aspects at most, and often with limited ecological detail. Importantly, no models have yet been developed to represent the breadth of species richness and functional diversity found in coral reefs in different regions of the world.

Trait-based approaches (eventually combined with demographic approaches) have been used in conceptual, statistical, equation-based, and agent-based models to address diverse theoretical and practical questions (*Zakharova et al., 2019*). Agent-based approaches are particularly suited to simulate coral-community dynamics as a function of diversity, functional traits and demography because (*i*) spatial processes can easily be implemented explicitly, (*ii*) it is possible to describe the functional (by implementing effect, resistance, and recovery traits) and demographic (by implementing age or size-related fecundity and survival effects on ecosystem functions) characteristics of each individual in the community, and (*iii*) they are flexible and adaptable frameworks that can implement different types of submodels (e.g. statistical, equation-based, or algorithmic), which can be evaluated separately (*DeAngelis and Grimm, 2014*; *Grimm et al., 2005*; *Grimm and Railsback, 2005*). Agent-based models have been criticized for being complex, and difficult to parameterize, analyze and communicate. However, empirical, trait-based approaches and traits databases have become more prevalent, facilitating model parameterization. Additionally, standardized protocols—such as the Overview, Design concepts and Details (ODD) protocol and hierarchically structured validation—are now well-defined to help communicate and validate agent-based models (*Grimm et al., 2020*; *Grimm et al., 2006*; *Kubicek et al., 2015*).

We present a new, spatially explicit, agent-based model representing benthic communities in tropical reefs composed of coral species and six functional groups of algae. Individual colonies grow, reproduce, compete for space, and respond to disturbances as a function of their size and trait-process relationships, which we defined using 11 functional traits (*Table 1*) informed by published empirical data (*Figure 1*, *Figure 2*). The number of coral species present in the community can be varied without impacting model complexity and processing time. Functional diversity can be varied by sampling species from a set of 798 functionally realistic species, which we obtained by imputing missing trait data based on values measured for real species (*Madin et al., 2016a*). Importantly, we used empirical data and previously established models to implement most of the processes represented in the model (*Table 2*). Our aim is to provide the full description of our model's design, concepts, and capabilities. In the main text, we present a streamlined description, and use the Appendices for details regarding: (1) traits and imputation of missing data (Appendix 1), (2) the Overview, Design concepts and Details protocol (Appendices 2 and 4), (3) calibration with empirical

**Table 1.** The 11 functional traits we used to implement ecological processes in the model.

| Traits | Related processes and details |
|---|---|
| Age at maturity (yr) | The minimum age required for a coral colony to reproduce (Appendix 2: §7.2.1.1.a) |
| Aggressiveness (0 to 100) | Spatial direct competition for space between coral species; the trait is only used for species not considered in the *Precoda et al., 2017* study on probability of species-pair interactions (Appendix 1: §1.2) |
| Colony max diameter (cm) | Initial colony size distributions (Appendix 2: §5.2.2); colony fecundity (for the species with small colonies; Appendix 2: §7.2.1.1.b); bleaching (Appendix 2: §7.4.2.1; Appendix 4); colony vegetative growth (to define maximum planar area; Appendix 2: §7.5.1) |
| Corallite area (cm$^2$) | Colony fecundity (Appendix 2: §7.2.1.1.b); bleaching (Appendix 2: §7.4.2.1; Appendix 4) |
| Egg diameter (mm) | Time to motility of coral larvae (§7.2.1.1.d) |
| Polyp fecundity | Colony fecundity (Appendix 2: §7.2.1.1.b) |
| Growth form | Formation of reef rugosity (Appendix 2: §7.1.2.2); colony fecundity (Appendix 2: §7.2.1.1.b); dislodgement (Appendix 2: §7.3.1.2); spatial competition (overtopping; Appendix 2: §7.5) |
| Growth rate (mm.yr$^{-1}$) | Bleaching (Appendix 2: §7.4.2.1; Appendix 4); vegetative growth (Appendix 2: §7.5.1) |
| Mode of larval development | Coral reproduction (Appendix 2: §7.2.1.1.a) |
| Microscopic reduced scattering coefficient ($\mu_{S,m}$, mm$^{-1}$) | Bleaching (Appendix 2: §7.4.2.1; Appendix 4) |
| Sexual system | Colony fecundity (Appendix 2: §7.2.1.1.b) |

data (Appendix 3), (4) hierarchically structured validation (Appendix 5), and (5) global sensitivity analysis (Appendix 6). For complete transparency and reproducibility, the model, the R scripts and instructions are available on the Open Science Framework (OSF) (*Carturan et al., 2020*).

# Materials and methods

## Sources and software

We collected coral-trait data from coraltraits.org (*Madin et al., 2016a*) and other resources from the peer-reviewed literature (Appendix 1). We systematically verified and corrected coral-species nomenclature using the World Register of Marine Species as a reference. We used R (version 3.5.0, *R Development Core Team, 2017*) to manipulate datasets, for statistical analyses, and to manage model simulations. We developed the model with the open-source, Java object-oriented programming language *Repast Simphony* 2.5.0 (*North et al., 2013*). We launched simulations using the R package rrepast 0.7.0 (*García and Rodríguez-Patón, 2016*) and rJava 0.9–10 (*Urbanek, 2018*). We used the R package `missForest` 1.4 (*Stekhoven and Bühlmann, 2012*) to impute missing trait data. We included phylogenetic information as a predictor using *Huang and Roy (2015)* phylogenetic supertrees to improve predictions; we manipulated the supertrees using the R packages `ape` 5.0 (*Paradis and Schliep, 2019*) and `phytools` 0.5–38 (*Revell, 2012*) (Appendix 1). We defined coral bleaching probabilities using the R packages `MuMIn` 1.40.0 (*Bartón, 2017*), `betareg` 3.1–0 (*Cribari-Neto and Zeileis, 2010*), `lme4` 1.1–15 (*Bates et al., 2015*), and `lmtest` 0.9–35 (*Zeileis and Hothorn, 2002*) (Appendix 4). For the global sensitivity analysis, we drew a Latin hypercube sample from the parameter space using the *randomLHS* function from the R package `lhs` 0.16 (*Carnell, 2018*), and we measured the influence of parameters on different response variables by fitting boosted regression trees using the *gbm.step* function from the R package `dismo` 1.1–4 (*Hijmans et al., 2017*).

## Model description

We provide here a brief description of the model following the Overview, Design concepts, and Details protocol (*Grimm et al., 2020*; *Grimm et al., 2010*; *Grimm et al., 2006*). A complete protocol that contains all the details about parameterization and process implementation along with a review of the supporting literature is available in Appendices 2 and 4.

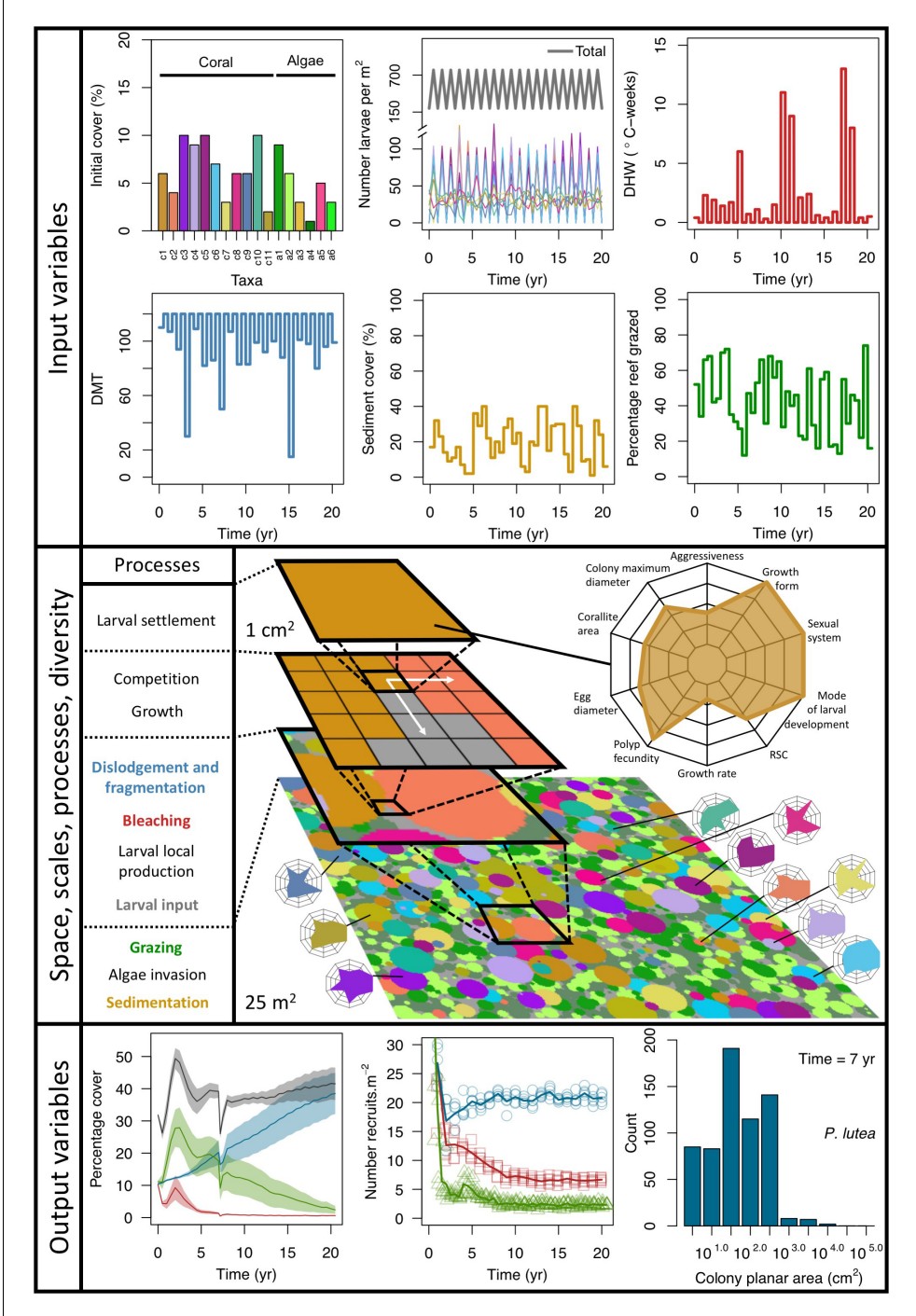

**Figure 1.** Description of the agent-based model. Six different variables as model inputs determine (*i*) initial community composition, (*ii*) number of larvae coming from the regional pool (total number divided among different species, with annual supply for spawning species, and biannual supply for brooding species), (*iii*) thermal stress in degree-heating weeks, (*iv*) hydrodynamic regime intensity expressed as dislodgment mechanical threshold (unitless), (*v*) sedimentation, and (*vi*) the percentage of reef grazed. All variables are inputs at every time period except for the initial community composition that is determined during initialization. The model represents a 25 m$^2$ coral reef community and is composed of 1 cm$^2$ cell agents. Once every time step, living agents (algae and corals) grow by converting their neighbouring agents within a certain radius (white arrows in middle panel). Different processes affect the community at different spatial scales. For instance, the grazing process lasts until the imposed percentage cover grazed over the entire reef is reached. In contrast, coral colonies are individually

*Figure 1 continued on next page*

*Figure 1 continued*

considered for dislodgment during hydrodynamic disturbance and a single agent is potentially converted into a new coral recruit when larvae settle successfully. Radar charts represent the functional characteristics of coral species (defined by a specific colour): each vertex corresponds to a functional trait and the coloured polygon indicates the trait values of the species (higher values are farther away from the centre of the web). At the end of each time step, the model provides the percentage cover, the number of coral recruits, and the size of each colony for every taxon, and optionally, the reef rugosity created by coral colonies (bottom panel). The benthic community at the largest scale is a screenshot of the model output.

## Purpose and patterns

The purpose of the model is to predict coral population dynamics as a function of hydrodynamic (i.e. waves and cyclones) and thermal disturbances, grazing pressure, larval connectivity, sedimentation (i.e. sand import and export), interspecific competitive interactions, and benthic community diversity (species richness and functional diversity). Time series defining disturbance regimes, sand cover and the diversity and number of external coral larvae are imposed and need to be defined before launching simulations. The grazing regime is imposed but can also be determined by activating the feedback process linking reef rugosity (created by colonies) to herbivore fish density to grazing pressure. Patterns in species cover, colony size distributions, recruitment rates and rugosity are used to understand the model's dynamics and its accuracy.

## Entities, state variables, and scales

The model consists of grid-cell agents each representing 1 cm$^2$, so that the benthic community is represented at a scale of organization smaller than the colony (equivalent to that of a polyp, although polyp size varies among species by several orders of magnitude) (*Figure 1*). During a simulation, an agent can be temporally part of a coral colony (798 species), a patch of algae (i.e. macroalgae, allopathic macroalgae, *Halimeda* spp., turf, articulated coralline algae or crustose coralline algae), sand, or bare substratum. Each agent is characterized by 33 variables that describe where the agent is in space (its position is fixed), its species identity (i.e. one of the 798 coral species or six functional groups of algae) and related functional characteristics, its age, the colony's planar area and identification number, if it is bleached or was grazed recently, *et cetera* (*Appendix 2—table 1*). Coral colonies and patches of algae are entities composed of multiple agents sharing the same variable values (except for their spatial coordinates) and changing their state simultaneously during certain processes. For instance, dislodgement is simulated by converting all the agents forming the dislodged colony into barren ground; a turf algae overgrowing a colony is simulated by converting the coral agents constituting the overgrown part into turf, but conserving the information about the colony (i.e. identification number, size, species, growth form).

The size of the reef and the length of a time step are changeable. We defined a 25 m$^2$ reef for our simulations (i.e. 250,000 agents), which is usually the scale at which benthic communities are assessed in detail (e.g. *Holbrook et al., 2018*; *Torda et al., 2018*). We defined a 6-month time step because the empirical data we used to calibrate the model were collected biannually. We acknowledge that six months is a coarse time step, potentially preventing the simulation of subtle dynamics, for instance changes triggered by mild disturbances. We opted for this time step considering that (*i*) corals grow slowly (<180 mm yr$^{-1}$), and (*ii*) their reproductive cycles, and (*iii*) thermal and hydrodynamic disturbance regimes are seasonal. It is, however, possible to define shorter periods (i.e., 3- and 4-month time steps) as time steps in the model.

The model estimates three-dimensional colony surface areas using geometric formulae (*Appendix 2—table 5*) to determine the number of larvae produced in each colony (Appendix 2: §7.2.1.1), and (optionally) the rugosity of the reef (Appendix 2: §7.1.2.2). The model also accounts for colony and algae heights in overtopping processes (Appendix 2: §7.5.2.2 and §7.5.3.2). Algae have constant heights and colony heights equal to the radius of the colony planar area, assuming the latter is circular (*Appendix 2—table 18*).

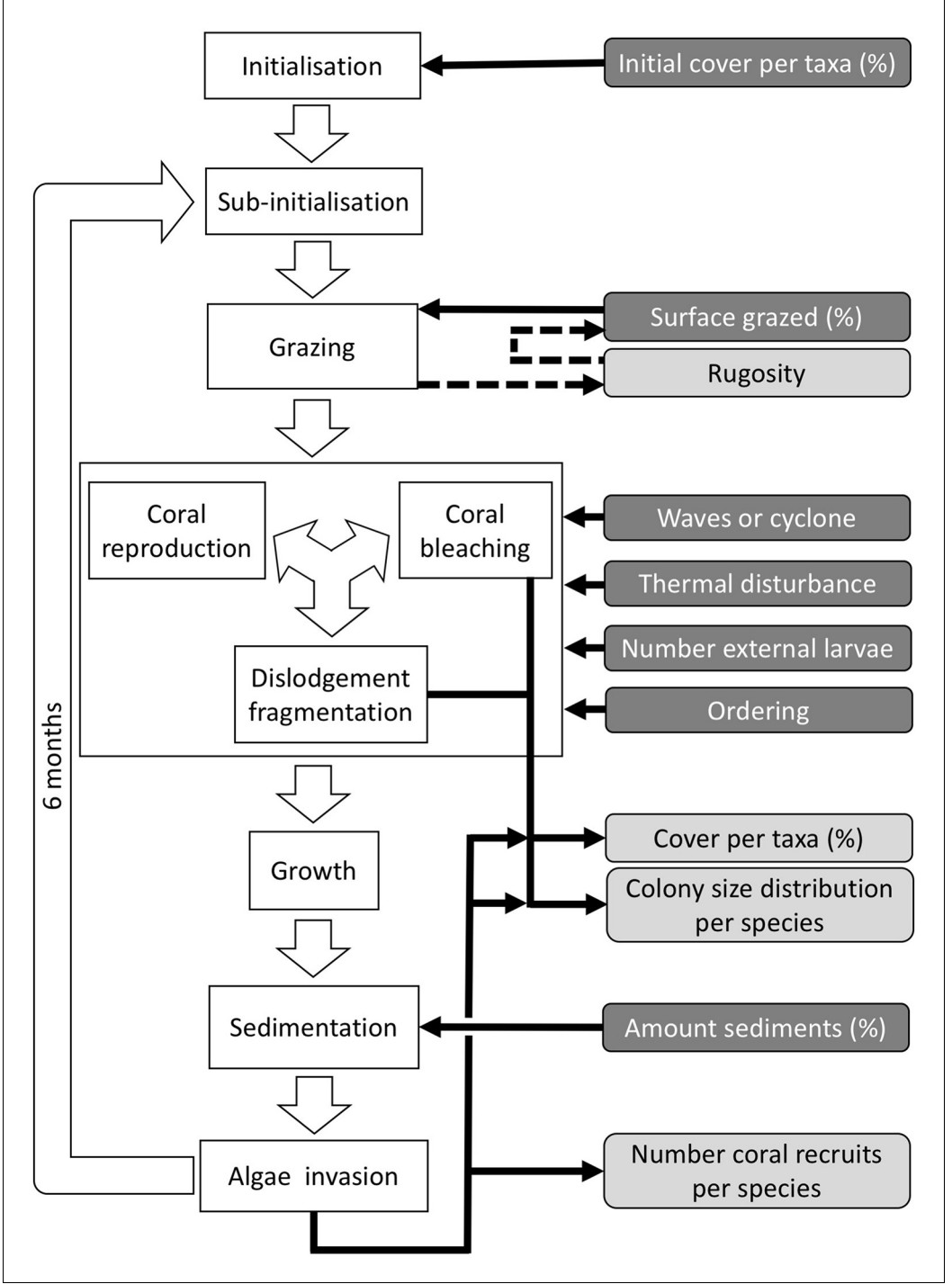

**Figure 2.** Ordering of processes in the coral agent-based model: white rectangles represent processes, dark grey rectangles with white text are input data, and light grey rectangles with black text are outputs. Large white arrows define the ordering of processes and black arrows show the direction of data transfer; dashed black arrows are optional processes (not activated for the analyses we present here). The order of occurrence of coral reproduction, bleaching, and colony dislodgement and fragmentation is imposed to simulate recruitment failure due to the occurrence of a disturbance prior to reproduction. The intensity of waves and cyclones is expressed as a dimensionless dislodgment mechanical threshold; thermal stress is expressed in degree-heating weeks.

**Table 2.** Empirical data and models we used to implement ecological processes in the model.

| Processes/variables | Comments | References |
|---|---|---|
| Colony size (initialization) | We used colony size distributions measured for eleven species and *maximum colony diameter* to define colony size distributions for each species (Appendix 2: §5.2.2) | E. H. Meesters and R. P. M. Bak, personal communication, May 2017 |
| Herbivorous fish density supported by the reef rugosity | We used an empirical model to determine the density of herbivore fish present in the reef as a function of reef rugosity (Appendix 2: §7.1.2.2) | *Bozec et al., 2013* |
| Grazing intensity due to herbivorous fish density | We defined a model using empirical data to determine the surface of the reef grazed as a function of herbivorous fish density (Appendix 2: §7.1.2.2) | *Williams and Polunin, 2001* |
| Polyp maturity in colonies | We defined a model from models established empirically to determine the proportion of mature polyps in a colony as a function of colony planar area using data for eight species (Appendix 2: §7.2.1.1.b) | *Álvarez-Noriega et al., 2016* |
| Larval competency | We used a model established empirically to determine time to motility of coral larvae as a function of *egg diameter* (Appendix 2: §7.2.1.1.d) | *Figueiredo et al., 2013* |
| Larval retention | We used models established empirically to determine the proportion of competent larvae remaining in the reef as a function of time to motility and water retention time (Appendix 2: §7.2.1.1.d) | *Figueiredo et al., 2013* |
| Larval competency loss | We defined a model from models established empirically to determine the proportion of external competent larvae settling on the focal reef as a function of the distance travelled (Appendix 2: §7.2.1.2.b) | *Connolly and Baird, 2010* |
| Larval post-settlement survival | We defined a model using empirical data to determine the proportion of surviving settled larvae as a function of time (Appendix 2: §7.2.1.3.b) | *Ritson-Williams et al., 2016* |
| Colony dislodgement | We used models established empirically to determine if a colony is dislodged as a function of colony growth form, planar area and the intensity of the hydrodynamic disturbance (Appendix 2: §7.3.1.2.a) | *Madin and Connolly, 2006* |
| Survival of dislodged branching colonies | We defined a model using a model established empirically to determine the proportion of a dislodged branching colony that survives dislodgement (Appendix 2: §7.3.1.2.b) | *Highsmith et al., 1980* |
| Coral bleaching | We used the empirically established bleaching-response index to determine species bleaching susceptibility from functional traits (Appendix 2: §7.4.2; Appendix 4) | *Swain et al., 2016b* |
| Coral competition | We used species-pair probabilities of interaction outcomes established from mix-effect models and a review of empirical data (Appendix 2: §7.5.2.2.a) | *Precoda et al., 2017* |
| Coral-algae competition | We defined probabilities of interaction outcomes using proportions of interaction won and lost between coral species and the different functional group of algae implemented measured experimentally (Appendix 2: §7.5.3) | *Brown et al. (2017)* and K. T. Brown, personal communication, October 2017 |

## Process overview and scheduling

Each time step includes the following consecutive processes: (*i*) grazing—patches of agents are randomly selected and grazed until a certain proportion of the reef is reached, (*ii*) coral reproduction—locally and regionally produced larvae attempt to settle, (*iii*) thermal disturbance, which, if triggered, eventually causes colonies to bleach and/or die; (*iv*) dislodgement and fragmentation—the effect of waves and cyclones on certain colonies, (*v*) growth—each living agent, selected in a random order, attempts to convert its neighbouring agents within a certain radius to its own state, (*vi*) sedimentation—barren ground agents are converted to sand and *vice versa* until the desired sand cover is reached, (*vii*) algae invasion—the remaining ungrazed, barren-ground agents are converted into algae agents (*Figure 2*) (note that the process differs from species invasion). The order at which processes *iii*, *iv*, and *v* happen must be defined beforehand.

During each time step, the model exports response variables: the cover of each benthic group (coral, algae, and sand), the planar area of each colony present per species, the number of recruited coral larvae m$^{-2}$ species$^{-1}$, and the reef rugosity (in cases when rugosity-grazing feedback is activated). The first two variables are collected after processes *iii*, *iv*, and *vii*, and the third and fourth variable after process *vii*. There are six variables imported each time step; their values respectively determine the cover to be grazed, the intensity of waves or cyclone, and of thermal stress, the number of external larvae m$^{-2}$ entering the reef, the order that reproduction, bleaching, and wave or

cyclone events happen, and the cover of sand to be achieved. We present a complete schedule that includes additional model-related processes in Appendix 2: §3.

## Design concepts

**Basic principles**: the model combines agent-based, trait-based, and demographic approaches to simulate coral reef community dynamics in imposed environmental scenarios. The model captures fundamental principles in ecology: (*i*) biodiversity influences ecosystem resilience and (*ii*) functioning, (*iii*) disturbance regimes filter species and mediate interspecific competition, (*iv*) interspecific functional differences (or strategies) mediate competitive exclusion and coexistence, and (*v*) source-sink dynamics regulate species coexistence in metacommunities.

**Emergence**: The dynamics of the benthic community emerge from species traits and the imposed disturbance regime (waves, cyclone, thermal stress), larval connectivity, sedimentation, and grazing pressure intensity (which can also emerge from reef rugosity). Cell agents do not make decisions and their behavior results from imposed deterministic or probabilistic rules.

**Interactions**: Agents on the edge of coral colonies and patches of algae interact with one another when competing for space. The outcome of a coral-coral interaction is determined by its specific pairwise outcome probabilities—the probability of coral-algae interactions are the same for all coral species, and algae-algae interactions result in a stand-off except when competing against crustose coralline algae. Branching and plating species also have the capacity to overtop other colonies and algae depending on their size (Appendix 2: §7.5).

**Stochasticity**: The model draws success or failure outcomes each time a patch of algae is grazed, a larva attempts to settle and survive the first 6 months, an agent tries to convert another living agent (Appendix 2: §7.1, 7.2, 7.5), and a colony is thermally stressed (i.e. bleaching and bleaching-induced mortality; Appendix 2: §7.4). Each of these random events is based on a probability of success specific to the process and the species involved. During initialization, colonies are created and placed randomly in space; their sizes are drawn from right-skewed frequency distributions (Appendix 2: §5.1). Finally, grazing and larval settlement happen randomly in space.

**Collectives**: Collective behaviour of agents happens when a colony is (*i*) dislodged—agents sharing the same colony (i.e. coral and algae agents growing on a colony) are converted to barren ground (Appendix 2: §7.3), (*ii*) bleaches or dies—the coral agents of the colony are converted to a bleached or dead state, respectively (Appendix 4: §7.1, 7.2), or (*iii*) reproduces—the number of larvae or gametes produces by a colony depends on certain coral traits and the size and age of the colony (Appendix 2: §7.2).

**Observation**: Four types of data are collected during a simulation: (*i*) percentage cover of each taxon, (*ii*) number of recruits for each coral species m$^{-2}$, (*iii*) planar area of each colony species$^{-1}$, and (*iv*) optionally, the rugosity created by the coral colonies (**Figure 2**).

## Initialization

The initial composition of the benthic community (i.e. the cover of coral species, algae, barren ground, and sand) is defined by the user and is imported from a comma-delimited text file. The space is filled first by creating circular coral colonies randomly in space. The colony diameters are drawn from skewed distributions that we defined using empirical data (E. H. Meesters and R. P. M. Bak, personal communication, May 2017) and as a function of the trait *colony maximum diameter.* Circular patches of algae (314 cm$^2$) are then created and the remaining agents are converted into barren ground and sand (Appendix 2: §5).

## Input data

Predefined time series (recorded in the text files) of input data are used to define the environmental context of the reef (**Figure 2**). At each time step, the model imports values for the corresponding period of the (*i*) surface grazed (%), (*ii*) number of external larvae settling, (*iii*) intensity of waves of cyclones (in dislodgement mechanical threshold, a dimensionless measure of the mechanical threshold imposed by waves and cyclones), (*vi*) thermal stress intensity (in degree-heating weeks), and (*v*) sand cover (%).

## Submodels

### Grazing

The reef is grazed by randomly selecting circular patches of agents (29 cm$^2$) until a certain percentage cover is reached. The cover to reach can be either exclusively imposed (imported from a file) or it can result from the rugosity that coral colonies create if the rugosity-grazing feedback process is activated. We used the empirically established *Bozec et al. (2013)* model to determine herbivorous fish density from reef rugosity and data from *Williams and Polunin (2001)* to estimate grazing pressure from herbivorous fish density (Appendix 2: §7.1).

### Reproduction and recruitment

Coral larvae locally produced and arriving from the regional pool attempt to settle in the reef at a random location. The number of larvae produced locally for each species depends on species traits (i.e. *polyp fecundity*, *corallite area*, *growth form*, *sexual system*)—we used geometric formulae from *McWilliam et al. (2018b)* to calculate colony surface area from planar area—and the distribution of colony planar areas in their population—we used models from *Álvarez-Noriega et al. (2016)* to determine the proportion of fecund polyps in colonies as a function of planar area. We used models from *Figueiredo et al. (2013)* to determine the proportion of spawned eggs remaining in the reef from water retention time and *egg diameter*—species producing larger eggs also produce larvae having a greater time to motility and a higher chance of being exported outside the reef. Species with a brooding *mode of larval development* release larvae ready to settle and are not affected by water retention time (Appendix 2: §7.2.1.1). The number of external larvae arriving at the reef can either be defined beforehand and imported from a file, or is calculated as a function of the connectivity imposed—we used the models of *Connolly and Baird (2010)* to determine the proportion of alive and competent larvae as a function of distance travelled (Appendix 2: §7.2.1.2). Larvae have a chance of settling successfully on barren ground, crustose coralline algae, and dead coral agents. We used data from *Ritson-Williams et al. (2016)* to define the proportion of settled larvae surviving the duration represented by a time step (Appendix 2: §7.2.1.3). Algae recruit at the end of a time step by filling up the remaining available space (i.e. ungrazed barren ground and dead coral agents) (Appendix 2: §7.2.2).

### Wave and cyclone damage

We modelled colony dislodgment using colony shape factor from *Madin and Connolly (2006)*, which is compared for each colony to the intensity of the disturbance (expressed as dislodgement mechanical threshold). We implemented branching-colony fragmentation by modifying the relationship between fragment size and survival established by *Highsmith et al. (1980)*. We defined our own models to simulate the effect on the algae community because no relationships have been established empirically (Appendix 2: §7.3).

### Bleaching

We first defined a species-specific index of bleaching susceptibility using bleaching-resistance traits and the bleaching response index from *Swain et al. (2016b)*. We then used this index to establish species-specific logistic bleaching responses as a function of the intensity of the thermal stress (in degree heating-week) using data from *Eakin et al. (2010)*. Finally, we defined a bleaching-induced mortality logistic-response model (Appendix 2: §7.4; Appendix 4).

### Growth and spatial competition

Coral and algae agents on the edge of their colony or patch attempt to convert neighbouring agents within a certain radius. The size of the radius depends on the species *growth rate* and the state of the neighbouring agents—we simulated the effect of direct competition with a living agent on *growth rate* by reducing the length of the radius. We used competitive outcome probabilities from *Precoda et al. (2017)* to simulate between coral interactions (we used the trait *aggressiveness* if the species were not present in their list; see Appendix 1) (Appendix 2: §7.5.2). Branching and plating colonies can also overtop other colonies and algae. We used empirical estimates of competitive outcome probability to simulate competition between coral and algae (*Brown et al., 2017*; K. T. Brown, personal communication, October 2017; Appendix 2: §7.5.3). We considered algal functional groups

as equal competitors and therefore they cannot overgrow each other, except for crustose coralline algae, which is a weaker competitor (Appendix 2: §7.5.4).

## Model calibration

Here, we provide a short description of the calibration. All details are presented in Appendix 3.

### Study sites and related data

We used data collected between November 2001 and July 2011 in three sites located in Martinique in the Caribbean: Fond Boucher (14° 39′ 21.07″ N, 61° 09′ 38.98″ W), Pointe Borgnesse (14° 26′ 48.74″ N, 60° 54′ 12.72″ W), and Ilet à Rats (14° 40′ 58.04″ N, 60° 54′ 1.18″ W). The data were collected biannually (once per dry and wet seasons) by the *Observatoire du Milieu Marin Martiniquais* (OMMM) for the program *Initiative Française pour les REcifs COralliens* (IFRECOR). These data describe the benthic, macroinvertebrate, and fish communities at the species or genus levels, as well as sand cover for each site and at each sampling time (*Appendix 3—figure 1*, *2*). We downloaded values of degree-heating weeks for the corresponding location from the US National Oceanic and Atmospheric Administration data server ERDDAP (Environmental Research Division's Data Access Program; coastwatch.pfeg.noaa.gov/erddap) (*Appendix 3—figure 3*). We identified cyclone tracks using the National Oceanic and Atmospheric Administration Historical Hurricane Tracks website (coast.noaa.gov/hurricanes).

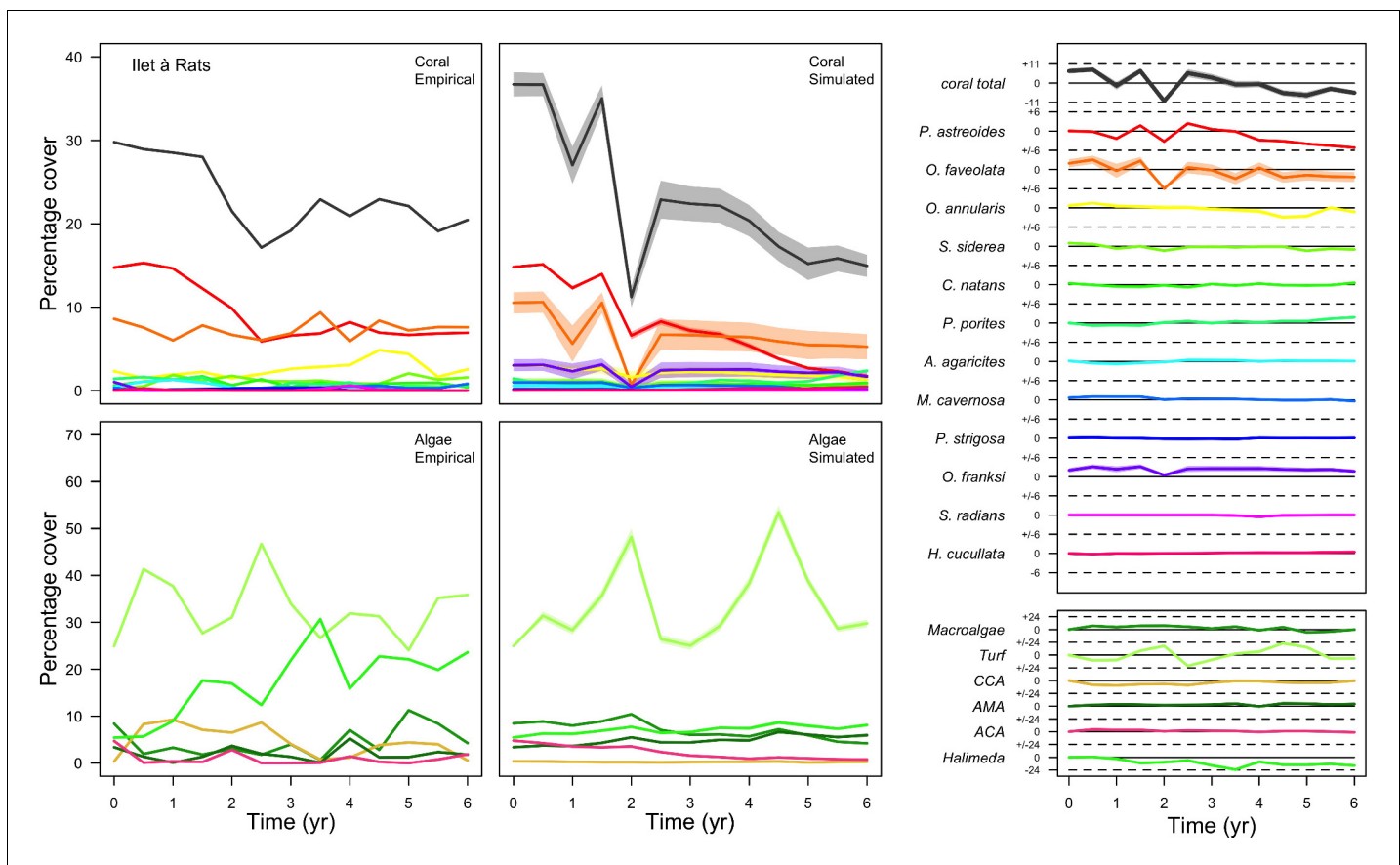

**Figure 3.** Comparison of empirical and simulated taxa cover for the combination of parameter values providing the best fit for site Ilet à Rats. Solid lines in the simulated time series are the percentage cover means (averaged over five replicates) and the shaded areas show the standard error. The right panels display the cover difference between simulated and empirical time series.

## Definition of the environmental context

We modelled thermal stress by inputting at each time step the maximum degree-heating week value found for the corresponding period. We represented the intensity of hydrodynamic regimes by inputting values of the dislodgement mechanical threshold (*Madin and Connolly, 2006*). We imposed a constant value in the absence of cyclone and a lower value when Hurricane Dean affected the reefs in August 2007 (its intensity changed from Category 1 to 2 while passing over Martinique). We chose threshold values arbitrarily considering wave exposure and cyclone intensity. Because of this uncertainty, we defined three different hydrodynamic regimes that we included in the calibration procedure for each site (*Appendix 3—figure 5*).

To estimate the percentage of the reef grazed at each time step, we first defined models predicting grazing intensity (i.e. percentage cover maintained in a cropped state) as a function of herbivorous fish and urchin density (we did not activate the rugosity-grazing feedback process). We defined these models using the empirical data from *Williams and Polunin (2001)* and *Sammarco (1980)* for fish and urchins, respectively (*Appendix 3—figure 6*). We then used these models and the population densities of *Acanthuridae* spp., *Scaridae* spp., and sea urchins measured in the three sites to predict their respective grazing regimes. Finally, we defined three additional similar regimes of different intensities, which we included in the calibration procedure (*Appendix 3—figure 8*).

The model adjusts the amount of sand cover (i.e. by removing or adding sand patches) at each time step according to the observed cover measured in each site (*Appendix 3—figure 4*). Having no information about larval connectivity at the three sites, we set the number of larvae m$^{-2}$ to 700 during each reproductive time period (i.e. once a year). This number corresponds to our estimate of competent larvae arriving on a hypothetical reef 20 km from an upstream reef having a 50% coral cover (Appendix 2: §7.2.1.2). The number is realistic considering that the distance separating the three sites from other coral communities is lower, but the average coral cover in the French West Indies is on average <40% (*Wilkinson, 2008*).

## General procedure

We calibrated the model for each site independently. We selected 12 parameters, for which we defined between two to five potential values (*Appendix 3—table 1*). We defined an algorithm to explore the parameter space optimally. The algorithm first selects the centroid, the most extreme values, and the values situated at mid-distance between the centroid and the extremes. A simulation with each parameter value is launched and replicated five times. We measured the fit between the empirical and simulated cover time series using an objective function. The objective function measures the performance of a given run by calculating the Euclidian distance between the empirical and simulated cover time series (averaged over five replicates), averaged over all the taxa (Appendix 3: §3.2). Performance is thus a positive value, with smaller values indicating higher performance (lower difference between simulated and empirical values). The algorithm then selects the 10 runs providing the best performance and generates for each of them the five closest (using Gower's distance metric; *Gower, 1971*) and untested parameter combinations. The algorithm then launches these new simulations and repeats the procedure once more.

To compare the performance of model runs to a null expectation, we generated a null distribution of performance values for each empirical dataset by randomizing cover values within each row and calculating the distance from the original datasets.

## Hierarchically structured validation

Models are often validated by comparing outputs of a single level of organization (i.e. individual, population, community) to equivalent empirical datasets (individual species coverages in our case), but this approach only examines lower dimensionality for more complex models. Following the recommendation of *Kubicek et al. (2015)*, and aligned with the approach of pattern-oriented modeling (*Grimm et al., 2005*), we used a hierarchical approach to assess whether the different processes implemented in our model—starting from the those occurring at the lowest scales, to those affecting the entire system—produce ecologically realistic patterns by comparing them to expectations formulated *a priori*. We based several of the expectations using the classification of life-history strategies of *Grime (1977)* into competitive, stress-tolerant and ruderal (CSR) (or weedy) functional groups— a classification which was adapted to corals (*Darling et al., 2012*). This 'CSR' classification

is independent from the effect, resistance and recovery trait classification that *Carturan et al. (2018)* adapted to corals, and which we used to select the traits to implement in the model.

We assessed the following processes of our model: (*i*) we expected colony lateral growth to equal the species growth rate in absence of spatial interaction and to decrease as space becomes saturated by colonies; (*ii*) recruitment rate should increase as a population grows from low initial cover, and then decrease as space saturates; for competition under different (*iii*) disturbance-regime intensities—we expected the competitive species to dominate the community under low-disturbance regimes, and ruderal or stress-tolerant species otherwise; (*iv*) larval connectivity—we expected species with higher colony fecundity or brooding mode of larval development to dominate the community under low connectivity, and the competitive ones otherwise; (*v*) grazing—under low grazing pressure, the benthic community should be dominated by algae, and by corals otherwise. In procedures *iii*, *iv*, and *v*, we varied the intensity of one factor at time while maintaining the other factors at intermediate values. This design generated factor combinations that are not realistic (e.g. grazing pressure and larval connectivity usually decrease after a strong disturbance), but is more rigorous because it prevented us from subjectively defining time series of grazing and larval connectivity that would yield more realistic population dynamics. We did not activate the rugosity-grazing feedback process in the analysis.

Because we expected the community dynamics to depend on species-specific trait differences, we did procedures *iii*, *iv*, and *v* with two different communities, each composed of a competitive, a ruderal, and a stress-tolerant species, originating from the Eastern Pacific and Western Atlantic, respectively. Note that our goal was not to use suites of species that accurately reflect the taxonomic composition of particular reefs or species pools, but rather to select species based on their functional trait attributes. Although we refer to species by name, the names themselves therefore matter less than their functionality. It is well known that reefs with different biogeographic or evolutionary histories host species that are functionally similar (*McWilliam et al., 2018b*). All details are in Appendix 5.

## Global sensitivity analysis

Our goal was to estimate the sensitivity of the predicted dynamics of the model to parameter variation during a process of recovery after a strong pulse disturbance. We constructed a global sensitivity analysis for 10 of the calibrated parameters and six additional parameters with high uncertainty (*Appendix 6—table 1*). For each parameter, we defined a range around the value(s) calibrated (for the 10 parameters considered in the calibration) or the value used in the simulations (for the six additional parameters). These parameters are all continuous but vary in their type (i.e. probabilities, ratio, heights, sub-model coefficients). We defined their respective ranges considering parameter uncertainty, realistic boundaries, and what values might improve the model performance based on model calibration and hierarchically structured validation. We did the procedure for each site independently because certain parameters were calibrated on different values between sites and because the coral communities differ.

We simulated our model for 10 years with a bleaching event of an intensity of 12 degree-heating weeks occurring after 4 years. We consequently assessed model sensitivity 6 years after the disturbance, a time when the communities in most runs were still recovering. We kept the following processes constant: grazing (50%; we did not activate the rugosity-grazing feedback process), wave hydrodynamic regime (dislodgement mechanical threshold = 120, which is equivalent to strong wave regimes that colonies experience at the reef crest), and larval input from the regional pool (700 larvae m$^{-2}$). We defined the same initial benthic composition as the one observed in the Caribbean sites.

We defined five response variables that represent the ecological state of the community at the end of the simulation: (*i*) total coral cover, (*ii*) difference of total coral cover at year 10 and just after the bleaching event, (*iii*) Pielou's evenness, (*iv*) coral species richness (only the species with $\geq$1% cover), and (*v*) number of recruits m$^{-2}$.

We estimated the relative importance of the parameters selected on each response variable following the efficient protocol of *Prowse et al. (2016)*. For each site, we sampled 1000 combinations of parameter values from a continuous parameter space using Latin hypercube sampling and uniform-sampling distributions. We launched each combination once (no replicates). We then fitted boosted regression trees on the input parameter values for each response variable—the procedure

provides the respective influence of each predictor (i.e. model parameter) on the variation of the response variable in question. We ensured the sampling was sufficient by comparing the influence of the parameters obtained with $n = 1000$ samples with values obtained with subsamples ($n = 100$, 250, 500 and 750); sampling is estimated to be sufficient when the influence of the parameters converge to similar values as sample size increases. All the details of the procedure are in Appendix 6.

## Results

### Model calibration

Model performance (the Euclidian distance between the empirical and simulated cover time series averaged over all taxa) varies between 28 and 10 (lower values = better performance), and were all lower than the lower 95% confidence bound of the random distribution (*Appendix 3—figure 9*). This shows that, despite the model's complexity and parameter uncertainty, the model outputs population dynamics closer to the empirical data compared to random. The best performance values converged toward 10 among the three sites (i.e. minimum ± standard error: 10.93 ± 3.677, 10.89 ± 2.872, 10.39 ± 3.119 for Fond Boucher, Pointe Borgnesse and Ilet à Rats, respectively).

With the combination of parameter estimates yielding the best fit, the model produces time series of total coral cover similar to the empirical ones for each site (*Figure 3*; *Appendix 3—figure 14*; *Appendix 3—figure 15*). The difference between the simulated and real total coral cover does not exceed 15, 20, and 11% for Fond Boucher, Pointe Borgnesse and Ilet à Rats, respectively. Results at the species level are more variable, but the cover difference of individual coral populations never exceeds 8%. For some species, the simulated cover closely predicts the empirical data—for instance, *O. faveolata* and *O. annularis* at Ilet à Rats (*Figure 3*) and *A. agaricites* and *S. siderea* at Fond Boucher (*Appendix 3—figure 14*). The model failed to predict the population dynamics of some other species accurately; for instance, in the simulated reefs, *M. mirabilis*, *M. decactis* and *P. furcata* became the dominant species, while the cover of *P. atreoides* and *M. meandrites* approached zero at Fond Boucher; *M. mirabilis* outcompeted *O. annularis*, *O. faveolata*, *O. franksi* and *P. astreoides* at Pointe Borgnesse (*Appendix 3—figure 14*; *Appendix 3—figure 15*), while the *P. astreoides*'s population decreased at Ilet à Rats (*Figure 3*).

The simulated cover of algae also closely mimics the empirical data for most algal groups (*Figure 3*; *Appendix 3—figure 14*; *Appendix 3—figure 15*). The difference of percentage cover is the highest for turf and reaches a maximum of 29, 22% and 24% for Fond Boucher, Pointe Borgnesse and Ilet à Rats, respectively. These percentages are high compared to other groups or taxa, but this can be explained partially by the high variance in algal turf cover observed at the reefs. Turf cover generally fluctuates by >20%, a pattern that our model was able to reproduce at all three sites (*Figure 3*; *Appendix 3—figure 14*; *Appendix 3—figure 15*). Notably, crustose coralline algae are systematically less abundant in the simulated reefs compared to the observed data, a phenomenon we attempted to correct in the calibration procedure (Appendix 3: §3.3). Finally, the model could not reproduce the high cover of *Halimeda* spp. observed at Ilet à Rats (*Figure 3*) compared to the other sites. See Appendix 3 for more detailed results and discussion regarding the between-site comparison.

### Hierarchically structured validation

The hierarchically structured validation shows that the model produces ecologically realistic population dynamics under different environmental conditions. Here, we provide a summary of the results, but a more-complete description and explanation are available in Appendix 5.

#### Growth

Coral colonies grew *ipso facto* at their species-specific growth rate at low population density. However, as space filled up, colonies began constraining each other spatially and their growth rates decreased until eventual stasis (*Appendix 5—figure 4*).

#### Recruitment

For a single coral population, the different patterns of recruitment observed among three functionally distinct species (*Appendix 5—figure 6*) results from the interaction of several factors: (*i*)

individual colony fecundity determined by its planar area, species-specific polyp fecundity, corallite area (polyp size), growth form, sexual system, and mode of larval development; (*ii*) the distribution of colony size in the population, which depends on maximum colony diameter; and (*iii*) the amount of surface available for larval settlement. Weedy (*Agaricia tenuifolia*) and stress-tolerant species (*Echinophyllia orpheensis*) produced bell-shape recruitment patterns (*Appendix 5—figure 8*). Recruitment rate was initially low because populations were composed of small, low-fecundity colonies, but the rate increased as colonies grew and became more fertile (*Appendix 5—figure 8*). Recruitment subsequently decreased as space became saturated. In contrast, recruitment rate for competitive species (*Acropora gemmifera*) was initially high and only decreased as cover occupancy increased. This pattern is essentially due to a higher vegetative growth rate associated with a population initially composed of fewer but larger, more fecund colonies (*Appendix 5—figure 8*).

## Disturbance intensity

In both the Western Atlantic and Eastern Pacific communities (we compared the two functionally distinct coral communities in the rest of the analysis), the competitive species dominated the coral community under low wave exposure (*Appendix 5—figure 10*, *11, 14*). The success of the competitive species was due mainly to two interacting processes—with a higher vegetative growth rate, competitive species (*i*) overcame free space before other species, and (*ii*) enhanced recruitment by achieving large colony size rapidly. Higher wave exposure reduced the cover of competitive species because colonies were dislodged at a certain colony size, which reduced recruitment rate and provided other species with more available space to grow and recruit.

In the Western Atlantic community, increased availability of space favoured the weedy species (*Madracis pharencis*) over the stress-tolerant species (*Orbicella annularis*), principally because of the former's brooding mode of larval development (twice a year), faster growth rate, and high wave-resistance of its growth form (digitate). In contrast, species coexisted in the Eastern Pacific community under the highest-intensity disturbances (*Appendix 5—figure 14*). Both the competitive (*Pocillopora elegans*) and stress-tolerant species (*Porites lutea*) recruited more than weedy species (*P. damicornis*) due to their spawning mode of reproduction; spawning species received three times more larvae from the regional pool (Appendix 2: §7.2.1.2). However, the weedy species recruited twice as frequently and is slightly more aggressive than the other two species. The stress-tolerant species has a massive growth form, which conferred higher resistance to waves compared to the other two branching species. Nonetheless, this advantage barely compensated for its slower growth rate and lower colony fecundity.

Population(s) recovered to pre-disturbance cover after only one year, regardless of the intensity of the event. This recovery is faster than most dynamics observed in real reef systems and arises because we imposed a constant and high number of larvae (7000 m$^{-2}$) coming from the regional pool. In reality, larval supplies are reduced because a strong bleaching disturbance would also affect the surrounding reefs (*Hughes et al., 2019*). Another reason for this outcome was that recruitment preceded growth in the model (*Figure 2*), which inflated the former process because more space was available for settlement.

## Larval connectivity

Low larval connectivity influenced the two coral communities differently (Appendix 5: §5). In the Western Atlantic, the weedy species thrived under zero to moderate larval input (0, 66, 700 larvae m$^{-2}$) while the other two species went locally extinct (*Appendix 5—figure 17*). The weedy species produced ready-to-settle larvae twice a year, while the other two species reproduced annually and only a portion of their larvae were able to settle because of their time to motility (Appendix 2: §7.2.1.1.d). In contrast, the stress-tolerant species dominated in the Eastern Pacific community (*Appendix 5—figure 18*) due to its higher wave-resistance compared to the other two branching species.

Under the highest larval connectivity (7000 and 35,000 larvae m$^{-2}$), the competitive species dominated in both communities, principally because of their higher growth rates, spawning mode of reproduction, and their capacity to overtop smaller colonies.

## Grazing

Population dynamics were similar between the two communities (*Appendix 5—figure 20*, *25*). The total coral cover corresponded approximately to the imposed percentage of reef grazed, and the remaining ungrazed part of the reef was occupied by algae (also, ungrazed coral agents potentially exist). We observed no hysteresis because we did not implement feedback processes. Turf dominated the algae community in all simulated grazing regimes, despite having the highest palatability among algae (*Appendix 2—table 3*). The success of turf was due to its much higher growth rate compared to other algae (*Appendix 2—table 19*).

Coral recruitment rates at the steady state were the highest under medium grazing pressure (50%) because under lower and higher grazing intensities, space was saturated by turf and coral colonies, respectively. Under the lowest grazing pressure, most of the colonies were $\leq$100 cm$^2$ in surface area (*Appendix 5—figure 22*) and coral populations were rescued by external larval input. At intermediate pressures (30% and 50%), the competitive species dominated the coral community mainly because of their higher external larval input compared to the brooding (weedy) species, and their higher growth rate than the stress-tolerant species. Having a high growth rate was particularly important under low grazing pressure because this trait compensated better for the cover lost in competition with turf, which wins all its interactions with corals in the model (*Appendix 2—table 19*).

Under higher grazing pressures (70% and 90%), there were more coral-coral and fewer coral-algae interactions, which changed coral species dominance. The Western Atlantic community was dominated by the weedy species, followed by the stress-tolerant species and the competitive species was competitively excluded, mainly because of its lower aggressiveness and highest vulnerability to waves (*Appendix 5—figure 10*). In the Eastern Pacific community, the stress-tolerant species dominated the coral community and slowly outcompeted the other two species mainly because of its much higher wave resistance (*Appendix 5—figure 27*).

## Global sensitivity analysis

The parameters that had the most important effects on the response variables (i.e. total coral cover, Pielou's evenness, difference cover, coral species richness and the number of coral recruits m$^{-2}$, all measured 10.5 years after the disturbance) were *growth rate reduction interaction* (the reduction of lateral growth rate of an organism overgrowing another one) and *otherProportions* (coefficient controlling the number of larvae produced locally), followed by probabilities for larvae to settle on different substrata, and the probabilities of algal grazing. The remaining 10 parameters did not have an important influence on any of the five response variables (*Appendix 6—figure 1*).

Globally, all the influential parameters affected the response variables according to expectations. For instance, increasing growth-rate reduction when organisms interact (mostly turf over corals) reduced the competitive advantage of the dominant taxa, which increased coral *richness*, *total coral cover* (due to reduced competitiveness of algae), and consequently enhanced the *difference in coral cover* and the *number of coral recruits* (*Appendix 6—figure 1*). Increasing *otherProportions* increased the number of larvae produced by each coral population, which positively affected *coral cover*, *cover difference* and the *number of coral recruits*. The parameter was negatively correlated with *richness* and *evenness* because higher values disproportionately benefitted species capable of higher recruitment (e.g. brooder; *Appendix 6—figure 3*).

In general, the parameters influenced the response variables in consistent ways among sites (*Appendix 6—figure 1*). Differences were mainly due to different ranges of values tested for particular parameters. For instance, *probability of grazing allopathic macroalgae* had a stronger effect at Pointe Borgnesse compared to the other two sites because its range included smaller values, implying lower palatability, higher abundance (*Appendix 6—table 1*, *Appendix 6—figure 4*), and a larger effect.

Ten of the parameters had a negligible effect on the response variables, because the processes they contributed to did not occur in these simulations. For instance, *probability of algae to cover crustose coralline algae* did not have an effect because crustose coralline algae was not present in high enough abundance (*Appendix 6—figure 4*), and *height of big algae* and *height of turf* did not have an effect because most of the branching colonies did not reach sizes large enough to overtop these algae (*Appendix 6—figure 5*).

## Discussion

Our primary goal was to develop a model that captured the spatiotemporal dynamics of community composition in coral reefs as component coral and algal species responded to inter-species competitive interactions and external disturbances. Our trait-based and demographic approaches provided a combination that yielded better predictions and a better understanding of coral ecosystem dynamics relative to single-component models (*Edmunds et al., 2014*; *Salguero-Gómez et al., 2018*; *Violle et al., 2007*). The spatial structure we imposed—a grid of 1 cm$^2$ agents that collectively comprise a sizeable reef (tens of m$^2$) as inspired by previous models (e.g. *Langmead and Sheppard, 2004*; *Sleeman et al., 2005*; *Tam and Ang, 2009*; *Sandin and McNamara, 2012*)—yielded emergence, scaling, self-organization, and unpredictability, each of which is a property of complex systems (*Parrott, 2002*) including coral reefs (*Dizon and Yap, 2006*; *Hatcher, 1997*). Operating at such a small spatial grain, processes can be modelled at the appropriate scale (e.g. dislodgement removes entire colonies while spatial competition affects colony edges) (*Figure 1*) to generate distributions of colony size, and in turn, colony fitness and performance. Overall, the population dynamics resulted from the collective performance of each colony, which implies that at the scale of the community, a given species' fitness depended on its capacity to persist under a certain environmental context and compete with functionally dissimilar species. As in the real world, macro-scale community dynamics emerged from finer-scale processes and interactions, a phenomenon clearly demonstrated by the hierarchically structured validation (Appendix 5). The model structure can also accommodate the initialization of a specific spatial colony arrangement based on empirical data. This feature is absent in previous models (but see *Wakeford et al., 2008*), despite the importance of spatial patterns for herbivory (*Eynaud et al., 2016*) and coral population dynamics (*Brito-Millán et al., 2019*).

Our model is unique in being designed to simulate the effects of coral species richness and functional diversity on ecosystem dynamics. Most coral models have been developed to describe the effect of external drivers (mainly disturbances) on the state of the coral community (usually total cover) (e.g., *Bozec and Mumby, 2015*; *Kubicek et al., 2019*; *Kubicek and Reuter, 2016*; *Madin et al., 2012*; *Melbourne-Thomas et al., 2011a*). In contrast, few models exist that assess the influences of aspects of diversity on community or ecosystem dynamics (e.g. *Tam and Ang, 2012*; *Ortiz et al., 2014*; *Fabina et al., 2015*), but these have represented diversity with limited detail, and consequently have limited capacity to evaluate the effects of identity and diversity on ecosystem functioning (*Brandl et al., 2019*), or the effects of functional redundancy and response diversity on ecosystem resilience (*Mcleod et al., 2019*). In contrast, our model represents diversity in detail; we considered eleven functional traits and included their influence over eight ecological processes applied to 798 functionally realistic species. In addition, we ensured that species richness can be varied without affecting computation time. Our model therefore enables exploration of many realistic assemblage scenarios within an easily modified experimental setting.

Our calibrated model was able to reproduce similar total coral-cover dynamics in the three sites (*Figure 3*; *Appendix 3—figure 14*; *Appendix 3—figure 15*). At the population level, results were more varied, with several populations well predicted, others less so. Overall, these results are remarkable considering model complexity, the large number of parameters, and the limited data describing the environmental context and diversity at the three sites. Note that we validated the population dynamics of the species within an imposed environmental context. Specifically, we determined the external larval supply, hydrodynamic, thermal, grazing and sand input regimes before the simulations. In reality, feedback processes emerge and contribute in shaping community dynamics (*van de Leemput et al., 2016*). Implementing additional feedback processes would have increased model complexity, and we estimated that the empirical data we had were insufficient to validate these. For instance, the model offers the option to activate the feedback process between structural complexity and grazing pressure (Appendix 2: §7.1.2.2), but validating this model with this process requires better population density estimates of the major herbivorous fishes.

A useful model should ideally be calibrated and validated with empirical data at each level of organization (e.g. colony, population, community) (*Kubicek et al., 2015*). However, empirical data are usually lacking for some or even all of these levels. Coral models have therefore been validated against one or a few community-aggregated variables, and rarely at the species level. Sampling additional data specifically for the model and at the sites used for calibration and benefiting from

the opinion of local experts can improve the capacity of the model considerably to reproduce realistic dynamics. For instance *Mumby (2006)* developed a spatially explicit, mechanistic model to reproduce the total coral and macroalgae cover observed in Jamaican reefs. In two subsequent developments of the model, *Ortiz et al. (2014)* reproduced accurate recovery rates and final community composition of six coral taxa at 14 reefs in the Great Barrier Reef, and *Bozec et al. (2015)* reproduced the cover of seven coral species and the rugosity in reefs in Cozumel (Mexico). With a similar model, *Kubicek et al. (2012)* generated time series of major coral taxa cover at Chumbe Island (Tanzania) similar to real data. Further, *Kayal et al. (2018)* accurately reproduced colony density distributions of three coral species in four different sites in Moorea, French Polynesia, using integral-projection models.

In contrast with conventional approaches, we developed our model independently of the empirical data on which calibration was based. Instead, our model included the ecological details required for achieving our primary objective. Below we discuss the primary sources of uncertainty in our model calibration and suggest realistic ways for improvement.

*Grazing*: We estimated average grazing pressure over six months (% of reef grazed) based on a biannual assessment of sea urchin and herbivorous fish (*Scaridae* spp. and *Acanthuridae* spp.) populations. *Acanthuridae* spp. are mobile herbivores (*Thibaut et al., 2012*), so frequent assessments are necessary to obtain accurate estimates of mean population size. More data collection could improve the accuracy of the modeled processes, has others have done for several fish species (e.g. *Bozec et al., 2016*; *Mumby, 2006*).

*Hydrodynamic regime*: We defined time series of dislodgement mechanical threshold as a function of site exposure and cyclone intensity. Measuring the real dislodgement mechanical threshold over time in each site would improve the precision of the simulations. This would require measuring horizontal water velocity and tensile strength of the substratum (*Madin et al., 2013*; *Madin and Connolly, 2006*).

*Recruitment*: There is high uncertainty in our implementation of recruitment because we did not have estimates of recruitment rates and of the proportion of recruits originating from the local reef *versus* the regional pool. We therefore fixed the number of external larvae coming into the reef and controlled recruitment rate with one parameter (*otherProportion*s). The sensitivity analysis revealed that the parameter has a strong influence on the model's predictions. Reducing this uncertainty requires better estimates of recruitment rates, which can be achieved with tile experiments (e.g. *Ritson-Williams et al., 2016*) or visual assessment of new recruits along transects (e.g. *Gilmour et al., 2013*; *Holbrook et al., 2018*). The proportion of locally *versus* regionally recruited larvae can be estimated with population genetics (e.g. *Almany et al., 2017*; *Johnson et al., 2018*) or by modeling larvae plumes (e.g. *Golbuu et al., 2012*; *Wolanski and Kingsford, 2014*).

*Trait data*: The hierarchically structured validation showed that between-species trait differences influenced community dynamics. Considerable gaps in the coral-trait database (*Madin et al., 2016a*) limited our capacity to estimate traits accurately for many coral species. Collecting reliable trait data is critical to predict coral-community dynamics and ecosystem functioning (*Madin et al., 2012*). Further precision in the prediction would be gained by measuring traits locally, because traits can vary substantially among populations in different locations (e.g. *Diaz-Pulido et al., 2009*), and factors such as nutrient concentration affect both algae and corals growth (*Wear and Thurber, 2015*; *Zaneveld et al., 2016*).

*The third dimension*: The model represents flat benthic communities and estimates the height and surface area of colonies using simple geometric formulae. These approximations potentially misrepresent certain processes such as larval production, formation of reef rugosity, overtopping, and their interspecific differences. Recent efforts to quantify physical attributes of the colony from planar areas and growth forms (*Zawada et al., 2019*) provide potential opportunities to improve our model's accuracy.

Our model is flexible and can be tailored to represent coral communities around the world, and to explore many different questions pertaining to the links between diversity and ecosystem dynamics. This version of the model focusses primarily on coral diversity and the effect of two disturbance types, but other disturbance types, and additional processes and aspects of reef diversity, could be easily implemented provided sufficient data are available. Examples include functions related to herbivory, algal diversity, disturbance types, and feedback processes, which we elaborate below.

Herbivores differ in their foraging behaviour (reviewed in Appendix 2: §7.1.1), which affects benthic diversity, coral reef recovery, and functioning (*Burkepile and Hay, 2010*; *Cheal et al., 2013*; *Cheal et al., 2010*; *Nash et al., 2016*; *Pratchett et al., 2014*). A few models have described aspects of herbivore diversity; for instance, *Sandin and McNamara (2012)* modelled the effect of spatially differentiated foraging behaviour between fish and urchins on the dynamics of a coral community, and *Bozec et al. (2016)* modelled the population dynamics of several parrot fish species and their respective species and size-specific contribution to grazing. However, herbivore diversity has generally been neglected in coral-reef models. Accommodating herbivore diversity and its effect on the benthic community in our model is feasible, provided associations between population densities and processes (e.g. grazing, bioerosion) are empirically established for different taxonomic or functional groups (e.g. Appendix 3: §2.3).

Algal diversity is potentially as important as coral and herbivore diversity for reef functioning and recovery (e.g., *Roff et al., 2015*). Yet, most coral-reef models describe the algal community with no more than three functional groups (macroalgae, crustose coralline algae, turf). Our model is the first to implement six functional groups, which accommodated additional ecological details such as grazing preferences and coral-algae interactions (Appendix 2: §7.1.2 and §7.5.3, respectively). To date, trait-based research on tropical reef algae is modest compared to fishes and corals (*Brandl et al., 2019*) and an algal-traits database has not yet been created.

We implemented the effects of hydrodynamic variation, thermal disturbances, and changes in grazing pressure, but reefs are also affected by other disturbances, and some of these have been implemented in previous models—including ocean acidification (e.g. *Anthony et al., 2011*; *Madin et al., 2012*), predation by *Acanthaster planci* (e.g. *Hogeweg and Hesper, 1990*; *van der Laan and Bradbury, 1990*), disease (e.g. *Brandt and McManus, 2009*), destructive fishing (e.g. *Kubicek et al., 2012*), and pollution (e.g., *Wolanski et al., 2004*; *Melbourne-Thomas et al., 2011b*; *Kennedy et al., 2013*). We are currently not able to model the species-specific effects of these disturbances on coral assemblages because it is not clear what traits are relevant, nor how these relate to ecological processes and responses. Such information is necessary to parameterize mechanistic models such as ours, as exemplified by our trait-based model of the response of corals to bleaching (Appendix 4). Nevertheless, our model would benefit from further validation, and is missing important variables (e.g. symbiont diversity) for which data are lacking (*Carturan et al., 2018*).

Feedback processes affect population dynamics by generating thresholds, hysteresis, and by shaping basins of attraction (*Scheffer et al., 2001*; *Scheffer and Carpenter, 2003*). Coral reefs are notorious for feedback processes (*Hughes et al., 2010*; *Mumby and Steneck, 2008*), some of which have been implemented in models (e.g. *Mumby et al., 2007*; *Muthukrishnan et al., 2016*; *Kubicek and Reuter, 2016*). *van de Leemput et al. (2016)* reviewed over 20 different feedback processes observed in reefs and demonstrated with a simple model that the combination of several feedback processes, although weak individually, can have important effects on system dynamics. However, the empirical quantification of these processes remains to be established (*van de Leemput et al., 2016*).

The model we present here is suitable for simulating the local response of benthic coral reef communities to disturbances over short time periods (<2 decades). Predicting community dynamics over longer periods (e.g. under different climate-change scenarios) requires calibrating the model with longer empirical time series because we cannot guarantee that the actual calibration will yield realistic community dynamics beyond the periods we considered. For testing and demonstration purposes, we implemented the model for small spatial extents. Consequently, the size-class distributions of certain coral species comprising large colonies might not be realistic, and certain influential processes happening at larger scales (e.g. connectivity along environmental gradients and from refuges) are not implemented in our simulations. However, the model can be run for larger spatial extents, but such simulations require substantial computational power due to the high ecological detail and 1 cm$^2$ spatial resolution of the model. Ongoing model development includes improving computational efficiency to accommodate the simulation of larger spatial scales and related processes.

Minimalist models of coral reef systems (e.g. differential equation systems) have generally been developed to simulate the response of state variables to different processes (i.e. pulse and press disturbances, feedback processes) (*Weijerman et al., 2015*). Our model, while developed for the same objectives, provides the possibility to represent realistic benthic diversity and its effect on

community dynamics. Comparing the results of our model to those obtained from minimalist models would help establish the degree to which ecological details are necessary.

## Conclusion

We have constructed a dynamic and customizable model that allows coral species richness and functional diversity to be manipulated independently. The model combines trait-based, demographic, and agent-based approaches to implement many ecological processes that drive coral reef dynamics. Its structure is flexible, and more processes, traits and taxa can be incorporated, provided the data are available. To that end, we highlighted several knowledge gaps that impede the modelling of important details or components of coral reef ecosystems. Our model can be used as a platform for virtual experiments aimed at testing hypotheses about the effects of species identity and diversity on ecosystem functioning, and about the effects of functional redundancy and response diversity on resilience.

## Acknowledgements

We received constructive feedback from V Grimm, H Reuter, an anonymous reviewer, J Caley, K Brown, S Kim, W Hare, and technical support from W Klaver and F Cid Yañez.

## Additional information

### Funding

| Funder | Grant reference number | Author |
| --- | --- | --- |
| Canada Foundation for Innovation | 23065 | Jason Pither |
| Natural Sciences and Engineering Research Council of Canada | 2019-05190 | Lael Parrott |
| Natural Sciences and Engineering Research Council of Canada | 2014-04176 | Jason Pither |

The funders had no role in study design, data collection and interpretation, or the decision to submit the work for publication.

### Author contributions

Bruno Sylvain Carturan, Conceptualization, Resources, Data curation, Software, Formal analysis, Validation, Investigation, Visualization, Methodology, Writing - original draft, Project administration, Writing - review and editing; Jason Pither, Lael Parrott, Conceptualization, Formal analysis, Supervision, Funding acquisition, Methodology, Writing - original draft, Project administration, Writing - review and editing; Jean-Philippe Maréchal, Resources, Writing - review and editing; Corey JA Bradshaw, Resources, Methodology, Writing - review and editing

### Author ORCIDs

Bruno Sylvain Carturan https://orcid.org/0000-0001-6811-1063
Jason Pither https://orcid.org/0000-0002-7490-6839
Jean-Philippe Maréchal https://orcid.org/0000-0002-9413-5242
Corey JA Bradshaw https://orcid.org/0000-0002-5328-7741

### Decision letter and Author response

Decision letter https://doi.org/10.7554/eLife.55993.sa1
Author response https://doi.org/10.7554/eLife.55993.sa2

## Additional files

### Supplementary files

• Transparent reporting form

### Data availability

All data generated and associated scripts have been deposited in OSF under the https://doi.org/10.17605/OSF.IO/CTQ43.

The following dataset was generated:

| Author(s) | Year | Dataset title | Dataset URL | Database and Identifier |
|---|---|---|---|---|
| Carturan BS, Pither J, Parrott L | 2020 | Combining agent-based, trait-based and demographic approaches to model coral community dynamics - Data and scripts | https://doi.org/10.17605/OSF.IO/CTQ43 | Open Science Framework, 10.17605/OSF.IO/CTQ43 |

The following previously published datasets were used:

| Author(s) | Year | Dataset title | Dataset URL | Database and Identifier |
|---|---|---|---|---|
| Huang D, Roy K | 2015 | Data from: The future of evolutionary diversity in reef corals | https://doi.org/10.5061/dryad.178n3 | Dryad Digital Repository, 10.5061/dryad.178n3 |

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

## Appendix 1

## Functional traits, phylogeny, and trait imputation

Assembling diverse virtual coral communities composed of species occupying different locations in functional space requires a complete dataset of the traits implemented in the model. The Coral Trait Database (*Madin et al., 2016a*), from which we downloaded most of the trait values has many data gaps (*Madin et al., 2016b*). We consequently applied a data-imputation method to fill in these gaps, including phylogenetic information to improve prediction. We present here the traits we used, the phylogenetic information and finally, the trait in-filling procedure.

### 1. Functional trait data

### 1.1. Trait summary

We collected trait values from coraltraits.org (*Madin et al., 2016a*) and other sources from the primary literature (*Appendix 1—table 1*). We limited our analysis to zooxanthellate scleractinian coral species, because our main focus is on species forming typical tropical-reef habitats. We first assembled a total of 828 coral species after correcting for nomenclature using the World Register of Marine Species (marinespecies.org) as a reference. We then removed 30 species that had only categorical trait values available (e.g. growth form, model of larval development, sexual system) and for which we did not have phylogenetic information. We included the categorical traits 'coloniality' and 'sexual system' to increase the number of predictors in the random-forest imputation procedure, because these traits have been defined for many species and have different degrees of phylogenetic conservation (i.e. sexual system is conserved, whereas coloniality has evolved several times within different lineages; *Baird et al., 2009b*; *Kitahara et al., 2010*).

### Related code

*Manuscript/Rscripts/ Appendix S1 - Functional traits.R* (*Carturan et al., 2020*).
All the R scripts related to a functional trait and *trait_data_compilation.R inside Traits_and_imputation/Rscripts* (*Carturan et al., 2020*).

**Appendix 1—table 1.** Summary of the sources and the methods of compilation of the functional-trait data.
We downloaded all trait information from coraltrait.org between 30 March 2017 and 19 April 2018 (taxon-BRI: Bleaching Response Index).

| Traits | Source(s) | No. sp. | Used in model | Used for imputation | Comments |
|---|---|---|---|---|---|
| Age at maturity (yr) | Coral Trait Database | 3 | Yes | No | This trait is not used in the trait in-filling process because it is defined for too few species |
| Aggressiveness (0 to 100) | *Abelson and Loya, 1999* *Connell et al., 2004* *Dai, 1990* *Lang, 1973* *Logan, 1984* *Sheppard, 1979* | 116 | Yes | Yes | Values 0 and 100 correspond to the lowest and highest possible aggressiveness, respectively (see §1.2 for details) |
| Coloniality | Coral Trait Database | 743 | No | Yes | Used to increase the number of predictors in the imputation random forest procedure |
| Colony max diameter (cm) | Coral Trait Database | 307 | Yes | Yes | We considered the maximum value in case of duplicated species |
| Corallite area (cm$^2$) | Coral Trait Database | 712 | Yes | Yes | Obtained from 'corallite width maximum' and 'corallite width minimum'; we calculated the average when both values were available |

*Continued on next page*

*Appendix 1—table 1 continued*

| Traits | Source(s) | No. sp. | Used in model | Used for imputation | Comments |
|---|---|---|---|---|---|
| Egg diameter (mm) | (*Figueiredo et al., 2013*) Coral Trait Database | 25 | Yes | Yes | We obtained values from the coral trait database from 'egg size' (two species) and 'mature egg diameter' (four species); values were averages when possible |
| Polyp fecundity | Coral Trait Database | 13 | Yes | Yes | Values obtained from 'polyp fecundity' (10 species) and 'mesentery fecundity' (three species); we averaged values when possible |
| Growth form | Coral Trait Database | 791 | Yes | Yes | We used 'growth form typical'. We grouped under 'branching' the 'open' and 'closed branching' and 'hispidose'; 'massive' also comprises 'submassive' |
| Growth rate (mm.yr$^{-1}$) | Coral Trait Database | 125 | Yes | Yes | We converted radial to linear/diametral measurements |
| Mode of larval development | Coral Trait Database | 312 | Yes | Yes | *Pocillopora ankeli* and *P. damicornis* can be both brooder and spawner, but are considered brooder in the model |
| Microscopic reduced scattering coefficient ($\mu_{S,m}$, mm$^{-1}$) | *Marcelino et al., 2013*; *Swain et al., 2016a* | 93 | Yes | Yes | - |
| Taxon-BRI (0–100) | *Marcelino et al., 2013*; *Swain et al., 2016a*; *Swain et al., 2016b* | 304 | No | No | We removed observations for which only the genera was known and for non-scleractinian coral |
| Sexual system | Coral Trait Database | 306 | No | Yes | To increase the number of predictors in the imputation random-forest procedure. This trait is particularly well-conserved (*Brandt and McManus, 2009*) |
| Total number of species | | 798 | | | |

## 1.2 Aggressiveness ranking

For most species, the outcome of direct interactions between coral colonies is determined using the probability of species-pair interactions from *Precoda et al. (2017)* (Appendix 2: §7.5.2). We also defined for each species an aggressiveness ranking value, which is used to determine if a colony can overgrow another species' colony. Aggressiveness values are only used for the species not considered by *Precoda et al. (2017)*.

We defined aggressiveness ranking values combining six lists of coral species whose interspecific dominance relationships were assessed (*Appendix 1—table 2*). We first ranked the species in each study based on the metrics the authors used. Because each list had at least one species in common with another list, we could combine the six lists and rank all species considered from the least to the most competitive. We used the iterative partial rank-aggregation pivoting algorithm (IPRAPA) developed and explained in detail by *Swain et al. (2017)* (the authors used the method to rank 110 symbiodinium phylotypes aggregated from 35 reports based on their thermotolerance). The IPRAPA implements the ranking method based on consensus-based, Borda-rank aggregation (developed in voting systems) and updates ranks by a pivot element (i.e. a species shared between input lists). When more than one species is present in both lists, the species with the least uncertainty is designated as pivot element. In case of equal uncertainty, we selected the first species of the first list. We repeated the process 10 times. Each iteration provides an updated ranking score from the previous iteration. Ranking scores are decimal values comprised between 0 and 100, which represent the lowest and highest aggressiveness, respectively.

We selected the final ranking scores of a particular iteration based on two metrics: (1) the percentage of species-pair associations that were reversed after the aggregation compared to the

original ranking in each list (i.e. *Ppa*, a change in dominance to equality or vice versa was not counted as reversed), and (2) the Kendall's $\tau$, which is a measure of ordinal association between two quantities ($\tau = 0$ if all the species-pair associations are discordant, and $\tau = 1$ if they are preserved). Finally, we measured the match between the final global ranking and the individual ranking of each study with the Spearman rank $\rho$. By ranking species in each list separately using the ranking score, we can see that the procedure conserves most of the initial ranking in each list (*Appendix 1—figure 1*).

## Related code

*Traits_and_imputation/Rscripts/ aggressiveness.R* (*Carturan et al., 2020*).

**Appendix 1—table 2.** References used to define the aggressiveness ranking index per species (CI: Coral Index).

| References | Metrics | Original no. of taxa | Final no. of species | Remarks |
|---|---|---|---|---|
| *Lang, 1973* | Ranking number based on number of subordinates | 27 | 22 | |
| *Sheppard, 1979* | CI | 26 | 22 | |
| *Logan, 1984* | Number of subordinates | 17 | 15 | Paired-interactions showing opposite result between field and lab experiment were not considered |
| *Dai, 1990* | Classification in five categories based on CI | 76 | 76 | |
| *Abelson and Loya, 1999* | Ranking number based on frequency of wins and losses and composition of losing and winning species | 33 | 30 | |
| *Connell et al., 2004* | % of winning interactions | 13 | 11 | |
| Total number of species: | | 147 | 116 | |

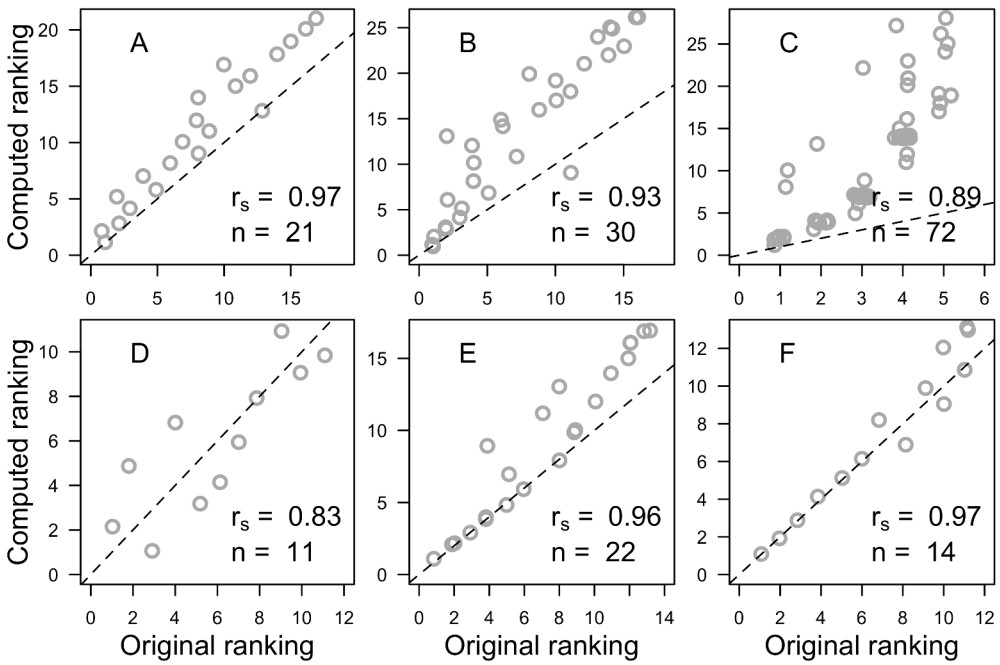

**Appendix 1—figure 1.** Correlation analyses between the original aggressiveness rankings and the

selected global computed ranking for each study. (**A**: *Sheppard, 1979*; **B**: *Abelson and Loya, 1999*; **C**: *Dai, 1990*; **D**: *Connell et al., 2004*; **E**: *Lang, 1973*; **F**: *Logan, 1984*). Also displayed are the Spearman $\rho$ ($r_s$), the number of species (n) and the identity line (dashed lines) for visual aid. All p<0.001 except for D, where p=0.003.

## 2. Phylogeny

We used the phylogenetic supertrees of *Huang and Roy (2015)* to include phylogenetic information as a predictor for the imputation of missing trait data. They reconstructed phylogenetic supertrees of the scleractinian clade from different trees (i.e. a molecular phylogeny, 13 morphological trees and 1 taxonomic tree). Their methodology resulted in 1000 fully resolved supertrees, comprising 1547 coral species We updated the name of 113 species and removed 47 species whose corrected name was already present in the tree. We used the *drop.tip* function from the R package `ape` 5.0 (*Paradis and Schliep, 2019*). We then removed 712 azooxanthellae species (based on the trait 'zooxanthellate' from the Coral Trait Database). Finally, we added 10 species at the genus level using the *add.species.to.genus* function from the R package `phytools` 0.5–38 (*Revell, 2012*). We added these species randomly along the edges to maintain binary nodes in the trees and avoid polytomies. The resulting supertrees are composed of 798 species.

### Related code

*Traits_and_imputation/Rscripts/phylogeny_coral.R* and *phylogeny_coral_and_functional_traits.R* (*Carturan et al., 2020*).

## 3.Imputation of missing trait data

### 3.1. Including phylogenetic information in the functional-trait table

We combined phylogenetic information with species functional traits to improve the prediction of missing trait data. The reasoning is based on the following assumptions: (1) closely related species are generally more functionally similar because of phylogenetic conservation of traits and (2) species having similar values for certain traits might have similar values for other traits because of functional trade-offs (*Darling et al., 2012*).

The phylogenetic information is included in the form of eigenvectors obtained from doing a principal component analysis on the distance matrix representing the phylogenetic branch length separating each species in the phylogeny (*Swenson, 2014a*). We first calculated the distance matrix of each of the 1000 trees using the function *cophenetic* from the R package `stats` (*R Development Core Team, 2017*). We averaged the 1000 matrices to obtain a final distance matrix. We then did a principal components analysis using the function *prcomp* from the R package `stats`. We selected the first nine eigenvectors using the broken stick method and used them as predictors (*Appendix 1—figure 2*). We retained the nine principal components because each describe different levels of the phylogeny (*Swenson, 2014b*).

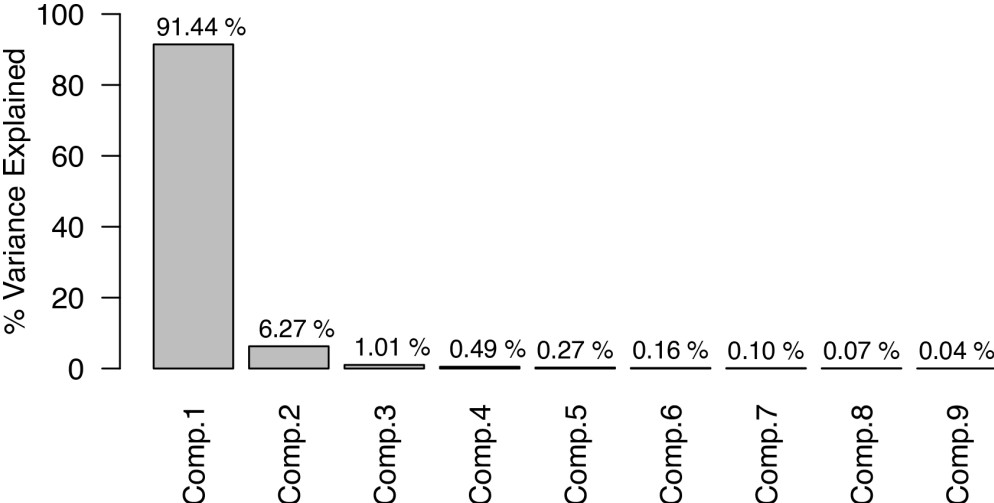

**Appendix 1—figure 2.** Percentage of variance explained by the nine first eigenvectors produced by the PCA on the averaged phylogenetic distance matrix.

## 3.2. Imputation of missing data

Among the different statistical methods available to perform imputation of missing data in functional-trait datasets (*Penone et al., 2014*; *Schrodt et al., 2015*), we chose the random-forest algorithm (*Breiman, 2001*) because it can handle highly dimensional datasets, does not rely on distributional assumptions, and is particularly appropriate for modeling complex interactions and non-linear relationships among variables. It is one of the best-performing techniques and allows the inclusion of phylogenetic information (*Penone et al., 2014*). We used the R package `missForest` 1.4 (*Stekhoven and Bühlmann, 2012*) to do the imputation. We set the maximum number of iterations to 10 and the number of trees generated by iteration to 100 (default values). Performance is evaluated with the normalized root mean squared error for numerical traits and proportion of falsely classified entries for categorical traits (good performance leads to a value close to 0, and bad performance to a value close to one). The method stopped after five iterations. It produced little error (normalized root mean squared error = 0.0985; proportion of falsely classified = 0.0735), and conserved the shape of the trait distributions (*Appendix 1—figure 3*, *Appendix 1—figure 4*, *Appendix 1—figure 5*).

## Related code

*Traits_and_imputation/Rscripts/imputation_traits_missForest.R* (*Carturan et al., 2020*).

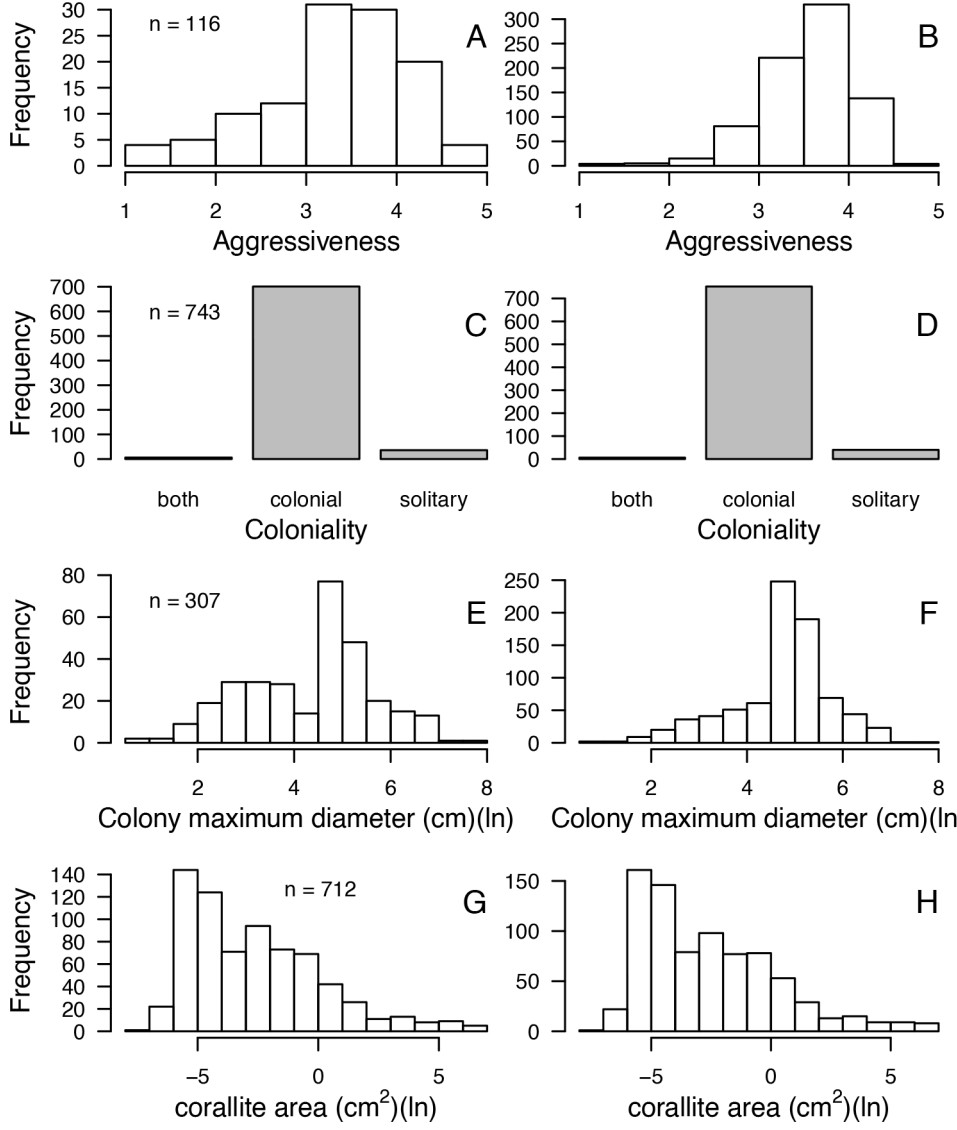

**Appendix 1—figure 3.** Comparisons of frequency distributions of trait values between the original dataset (left column) and the imputed dataset (right column) for aggressiveness (**A, B**); coloniality (**C, D**); colony maximum diameter (**E, F**); corallite area (**G, H**) (total *n* = 798).

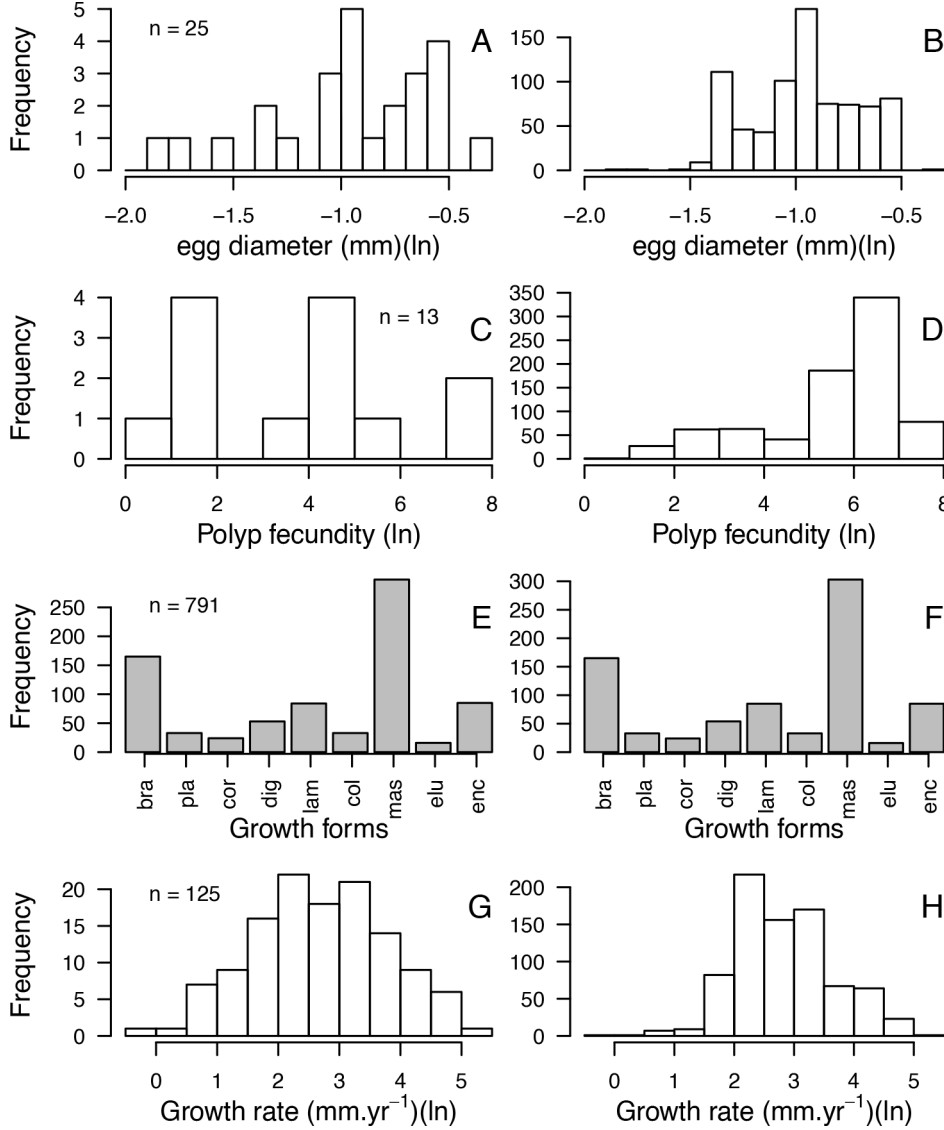

**Appendix 1—figure 4.** Comparisons of frequency distributions of trait values between the original dataset (left column) and the imputed dataset (right column) for egg diameter (**A, B**); polyp fecundity (**C, D**); growth forms: branching (bra), plate (pla), corymbose (cor), digitate (dig), laminar (lam), columnar (col), massive (mas), encrusting long upright (elu), encrusting (enc) (**E, F**); growth rate (**G, H**) (total *n* = 798).

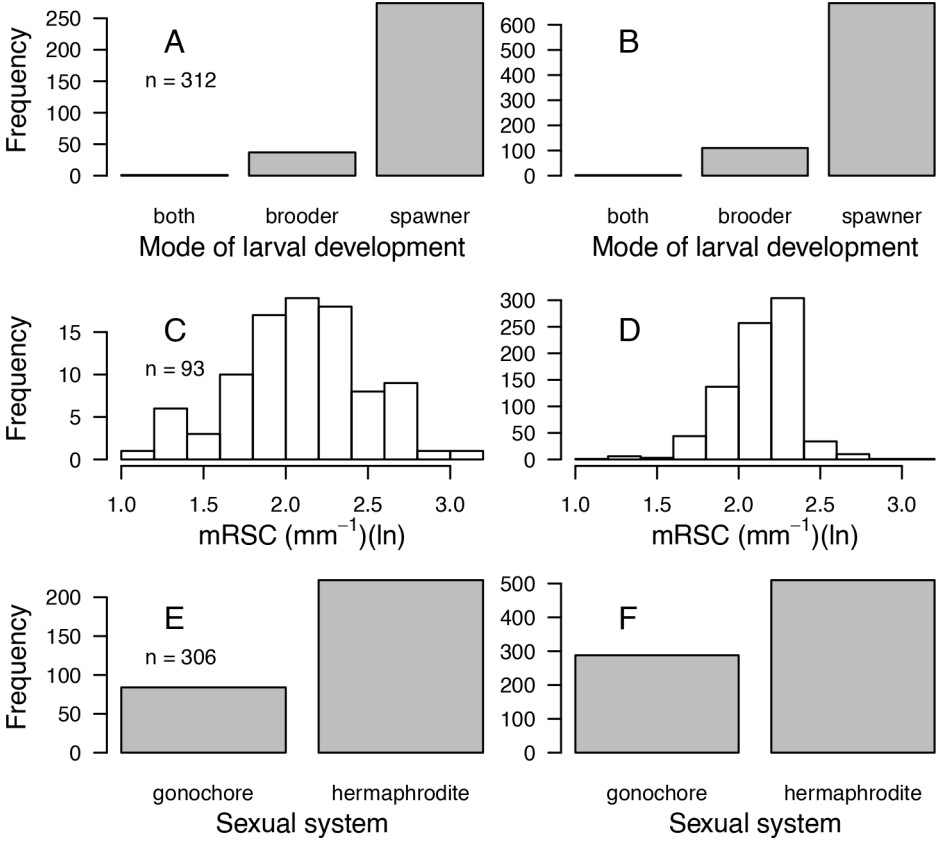

**Appendix 1—figure 5.** Comparisons of frequency distributions of trait values between the original dataset (left column) and the imputed dataset (right column) for model of larval development (**A, B**); microscopic reduced scattering coefficient (mRSC; **C, D**); sexual system (**E, F**) (total *n* = 798).

## Appendix 2

### 1. Purpose and patterns

The purpose of the model is to predict coral population dynamics as a function of cyclone and bleaching intensity and frequency, grazing pressure, larval connectivity, interspecific competitive interactions, and benthic community diversity (species richness and functional diversity). The higher level purpose is to understand what aspects of diversity contribute to the coral ecosystem functioning and resilience, which is achievable by analysing patterns of population percentage cover, larval recruitment rates, colony class distributions, reef rugosity and grazing pressure, and by comparing these patterns between functionally distinct benthic communities and across different environmental scenarios. These patterns are essential in understanding the emerging community dynamics because they reflect the different biodiversity-related processes at play (see Appendix 5 for detailed expected vs. observed patterns as a function of both functional diversity and the environmental context). Additionally, the model accuracy can be evaluated by comparing these patterns to real data (see Appendix 3 for the model calibration with empirical data).

### 2. Entities, state variables, and scales

The model consists of an ensemble of grid-cell agents, which are both spatial units and agents. Each grid-cell agent represents 1 cm$^2$, so that the benthic community is represented at a scale of organization smaller than the colony (equivalent to that of a polyp, although polyp size varies among species by several orders of magnitude) (see *Figure 1* in main manuscript; *Appendix 2—figure 1*). We chose such a small grain to represent certain processes at the appropriate scale (e.g. settlement, growth and spatial interactions), which permits to simulate higher level dynamics as the emergent outcome of agent level processes (e.g. colony size distribution). Each agent is characterized by 33 variables that describe where the agent is in space (its position is fixed), its species identity and related functional characteristics, its age, the colony's planar area and identification number, if it is bleached or was grazed recently (*Appendix 2—table 1*). Coral colonies and patches of algae are entities composed of multiple agents sharing several variable values (except for their spatial coordinates) and changing their state simultaneously during certain processes. For instance, dislodgement is simulated by converting all the agents forming the dislodged colony into barren ground; a turf algae overgrowing a colony is simulated by converting the coral agents constituting the overgrown part into turf, but conserving the information about the colony (i.e. identification number, size, species, growth form).

We decided to use grid-cell agents for two reasons. First for simplicity, because grid-cell agents constitute the spatial unit (i.e. the grid) and can represent both non-living elements that can interact with living entities (i.e. barren ground is suitable for vegetative growth and settlement while sand is not), and living sessile organisms. Second, because grid-cell agents allow to accurately present the dual level of organization at which a coral colony can be described: a set of smaller interacting entities (i.e. polyps) or a single individual in a population. In the latter, grid-cell agents forming a colony form a collective of grid-cell agents that share state variables that are unique to their colony, such as: *IDNumber*, *planar_area_colony* and the *timeRecoveryBleaching* (*Appendix 2—table 1*).

The size of the reef and the length of a time step are changeable. We defined a 25 m$^2$ reef (i.e. 250,000 agents) for the model calibration and analysis (*Appendix 2—figure 1*), which is usually the scale at which benthic communities are assessed in detail (e.g. *Holbrook et al., 2018*; *Torda et al., 2018*). Representing larger areas is possible but more computationally intensive. We used a 6-month time step for our model calibration and analysis because the empirical data we used to calibrate the model were collected biannually. Alternatively, it is possible to define 3-, 4-, 6- or 12-month time steps (using the parameter *yearDivision*).

### Related code

*coralreef2/src/coralreef2/Agent.java* (*Carturan et al., 2020*).

*coralreef2/src/coralreef2/InputData/ FunctionalTraitData.java* (*Carturan et al., 2020*).

**Appendix 2—table 1.** Grid-cell agent attributes, state variables (*) and other variables.

| Variable name | Description |
| --- | --- |
| *context* | The 'context' (a Java object that encapsulates the agents and projections; it is static, and all the agents belong to the same context). |
| *grid* | The 'grid', a spatial Projection associated to the Contextand allowing to place and locate the agents spatially (it is static, and all the agents are part of the same Grid). |
| *x* * | The x coordinate of the agent in the *grid* (static). |
| *y* * | The y coordinate of the agent in the *grid* (static). |
| *substrateCategory* * | First categorization of the agents: *BarrenGround* for barren ground and sand; *Algae* for algae; *Coral* for corals. Value are imported from data/functionalTraitDF_model.csv |
| *substrateSubCategory* * | Second categorization of the agents: *BarrenGround* for barren ground and sand; *Macroalgae, AMA, Halimeda, Turf, ACA* and *CCA* for macroalgae, allopathic macroalgae, *Halimeda* spp., turf, articulated coralline algae and crustose coralline algae, respectively; *BleachedCoral, DeadCoral, LiveCoral* for bleached, dead and living corals, respectively. Value imported from data/functionalTraitDF_model.csv |
| *species* * | Third categorization of the agents: *BarrenGround* for barren ground; *sand* for sand; *Macroalgae, AMA, Halimeda, Turf, ACA* and *CCA* for macroalgae, allopathic macroalgae, *Halimeda* spp., turf, articulated coralline algae and crustose coralline algae, respectively; coral species names for corals (e.g., *Acanthastrea_brevis*). Value imported from data/functionalTraitDF_model.csv |
| *age* * | Age (yr) of living agents. |
| *age_maturity* * | Species traits (see *Appendix 1—table 1*) Value imported from data/functionalTraitDF_model.csv |
| *aggressiveness* * | |
| *bleaching_probability* * | |
| *coloniality* * | |
| *colony_maximum_diameter* * | |
| *corallite_area* * | |
| *egg_diameter* * | |
| *fecundity_polyp* * | |
| *growth_form* * | |
| *mode_larval_development* * | |
| *reduced_scattering_coefficient* * | |
| *sexual_system* * | |
| *size_maturity* * | |
| *growth_rate* * | The maximum radius $r_{max}$ within which an agent can convert neighboring agents (see §7.5.1) |
| *correction_coeff_polypFecundity* * | A coefficient to apply to small species (*colony_maximum_diameter* < 16.7 cm) when determining the proportion of mature polyps in the colony (see §7.2.1.1.b). |
| *timeRecoveryBleaching* * | Time (yr) remaining before the agent totally recovers from bleaching (see §7.4). |
| *IDNumber* * | The identification number; unique to each colony and shared by the agents forming a same colony. |
| *red* * | The Red Green Blue (RGB) colour code to represent each benthic entity. Value imported from data/functionalTraitDF_model.csv |
| *green* * | |
| *blue* * | |

*Continued on next page*

*Appendix 2—table 1 continued*

| Variable name | Description |
|---|---|
| *canIGrow* * | Boolean variable used to allow or forbid an agent to grow. |
| *haveIbeenConverted* * | Boolean variable used to forbid an agent to grow during the *Growth* procedure (*Figure 2*) if it has already been converted (i.e., overgrown) by another agent during the same *Growth* procedure. |
| *haveIbeenGrazed* * | Boolean variable used to forbid an algae agent to overgrow another non-algae agent if the latter has been grazed during the *Grazing* procedure of the same time step (*Figure 2*). |
| *planar_area_colony* | The planar area (cm$^2$) of the colony formed by the agent (the value equals the number of agents forming the colony). |
| *newRecruit* * | Boolean variable used to indicate if the agent is a new recruit (i.e., if it settled during the *Coral reproduction* procedure of the present time step; *Figure 2*). |
| *size_UpDated* * | Boolean variable used to indicate if the agent's *planar_area_colony* has been updated during a procedure of colony size update (i.e., after each disturbance and at the end of a time step; *Figure 2*). |

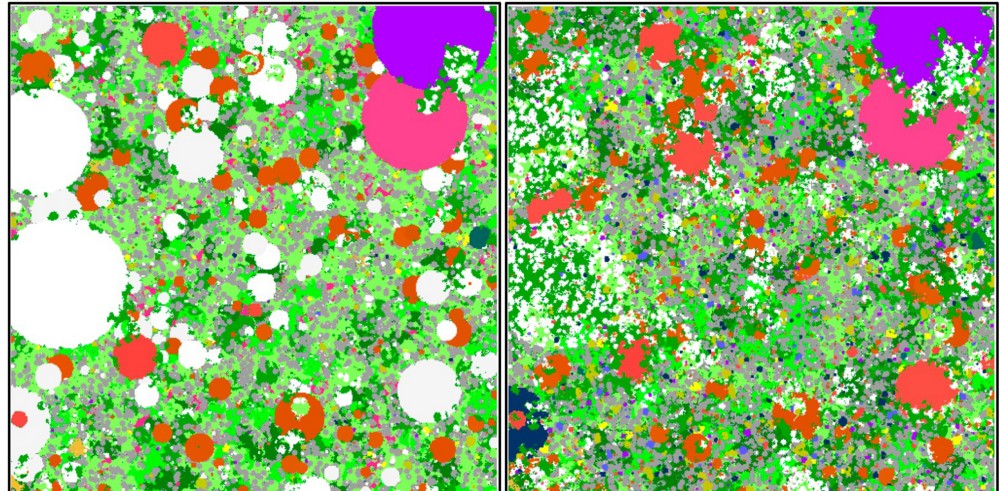

**Appendix 2—figure 1.** Screenshots displaying a 25 m$^2$ of the benthic community at Ilet à Rats taken after the execution of the bleaching process (left) and at the end of the same 6-month time step (right). We used parameter values calibrated with the empirical data describing this specific site (see Appendix 3). Coral colonies (pink, orange, blue, dark blue, white and light grey shapes) and algae (macroalgae, turf, allopathic macroalgae, and crustose coralline algae are in green, light green, dark green and beige shapes, respectively) compete spatially. Colourful coral colonies were not affected by the thermal disturbance, light grey colonies bleached and survived, and white colonies died. Grey and yellow agents represent barren ground and sand, respectively.

## 3. Process overview and scheduling

Each time step is constituted of different consecutive processes (*Appendix 2—figure 2*), which we detail here with the corresponding name of the method and eventually the related files:

1. Initialization
    a. *CoralReef2Builder()*:
        i. All the lists the model uses are erased, then all the necessary data related to the simulation are imported: the values of the parameters defined in the general user interface (GUI); time series of the number and diversity of larvae coming from the regional pool (either a constant or a time series imported from data/Disturbance_larvalConnectivity. csv), the cover of sand (imported from data/Disturbance_sand.csv), of the thermal (in

degree heating week; imported from data/Disturbance_bleaching.csv) and hydrodynamic disturbance regimes (in dislodgement mechanical threshold, imported from data/Disturbance_cyclone.csv), of the grazing pressure (in % cover; imported from data/Disturbance_grazing.csv); the order of occurrence for coral reproduction, hydrodynamic and thermal disturbances (imported from data/Disturbance_priority.csv); the functional traits table (imported from data/functionalTraitDF_model.csv); the coral species and algae functional groups present and their associated initial % cover (imported from data/Initial_benthic_composition.csv); the length (yr) of the simulation.

 ii. All the parameter values used for this specific simulation are exported in output/Parameters_values/Parameters_values_simulations.csv.

 iii. Creation of the *context* and the *grid*, then of the agents, according to the initial % cover of each living and non-living benthic groups previously imported. When an agent is created, its *x* and *y* variables are set relatively to its the position in the *grid*, its functional traits and colour variables are set accordingly to its benthic group (*Appendix 2—table 1*), whose characteristics were imported previously (see §5.2.2 on how the space is filled up).

2. Sub-initialization (i.e. the initialization executed each time step)

 a. Display is updated.

 b. *RunTimer(): year* is checked to see if it corresponds to the end of the simulation.

 c. *checkDisturbance():* the environment imports for the corresponding *year* (*i*) the intensity of the thermal disturbance (in degree-heating weeks) from data/Disturbance_bleaching.csv and update the variable *bleaching_DHW*; (*ii*) the intensity of the hydrodynamic disturbance (in dislodgement mechanical threshold) from data/Disturbance_cyclone.csv and update the variable *cyclone_DMT*; (*iii*) the intensity of the grazing pressure (in % cover) from data/Disturbance_grazing.csv and update the variable *percentage_reef_grazed*; (*iv*) the order at which coral reproduction, thermal and hydrodynamic disturbances are executed, from data/Disturbance_priority.csv and update the variables *priority_1*, *priority_2*, *priority_3*, and *season* (either 'dry' or 'wet').

 d. *calculatePercentateCover_and_NumberRecruits():* calculate the % cover of each living and non-living benthic group and the mean number of recruits m$^{-2}$ for each coral species. The data is exported in the simulation-related file output/PercentageCover/PercentageCover_...csv and NumberRecruits/NumberRecruits_...csv, respectively. This method is executed here (at the beginning of the time step) only at *year* = 0.0; it is executed systematically after a disturbance and at the end of the time step.

 e. *setDefaultParameters()*: each agent updates certain of its variables:

 i. The Boolean variables *canIGrow, haveIBeenConverted, sizeUpDated, haveIbeenGrazed,* and *newRecruit* are set to false.

 ii. If the agent is recovering from bleaching (i.e. *timeRecoveryBleaching* >0.0), then *timeRecoveryBleaching* is updated. If once updated 0.0 < *timeRecoveryBleaching* ≤ 0.5, the agent recovers its normal growth rate and colour (i.e. *growthRate, red, blue* and *green* are set to their original values); else, if *timeRecoveryBleaching* = 0.0, the agent recovers its capacity to produce gametes (i.e. *polyp_fecundity* is set to its initial value).

 f. *updateAgentColonySize()*: each agent associated to a colony (alive and dead coral and algae growing on a dead colony) have their *planar_area_colony* updated, and then their *sizeUpDated* set to true. If *year* = 0.0, the planar area of each colony, associated with its *IDnumber*, *species,* and *year_event* and the name of the event (i.e. 'cyclone') are exported in the simulation-related file output/ColonyPlanarArea/ColonyPlanarArea_... csv.

3. *grazing()*:

 a. Display is updated.

 b. *rugosityToGrazing():* the methods is only activated if the model parameter *Rugosity_Grazing* = true. The rugosity created by coral colonies is calculated and used to determine the grazing pressure (% cover) generated by the herbivorous fish supported by the coral community (see §7.1.2.2). The value of cover grazed is added to *percentage_reef_grazed*. The rugosity, % cover grazed due to the fish community, *percentage_reef_grazed* and *year* are exported in the simulation-related file output/RugosityCoverGrazed/RugosityCoverGrazed_...csv.

 c. Circular clusters of 29 agents are selected randomly and grazed (only the agents having *haveIbeenGrazed* = false can be grazed) until the percentage of the reef grazed = *percentage_reef_grazed*. All agents grazed have their *haveIbeenGrazed* variable

set to true. Algae agent successfully grazed are converted into the substrate on which they were growing (i.e., barren ground or dead coral colony) (see §7.1.2.1).

4. *Disturbance_and_reproduction()*: first the variable *year_event = year*, then the following methods are executed in an order defined by *priority_1, priority_2* and *priority_3*:

   a. Display is updated

   b. *coralReproduction()*: the following methods are executed:

      i. *coralLarvaeTotalProduction()*: the number of larvae produced locally and coming from the regional pool is calculated for each species (see §7.2.1).

      ii. *coralSettlement():* agents are randomly selected and eventually converted into a new coral recruit; new recruits take variables related to their coral species and *planar_area_colony* = 1 cm$^2$, *newRecruits* = true. Agent potentially converted are barren ground, dead coral and crustose coralline algae. Twenty agents are selected at a time for each species; the procedure is repeated until all the number of larvae calculated in *coralLarvaeTotalProduction()* is reached.

   c. *bleaching():*

      i. All the coral colonies are selected one after another and eventually bleach, according to a probability defined as a function of the species bleaching susceptibility (i.e., *bleaching_probability*) and the intensity of the thermal stress (i.e., *bleaching_DHW*). A bleached colony eventually dies, according to a probability that depends uniquely on the intensity of the thermal stress (see §7.4). Agents of a dying colony have their *substrateSubCategory* = 'DeadCoral', *growth_rate* = 0.0, *aggressiveness* = 0, *age* = 0.0, become white (i.e., *red* = 255, *green* = 255, *blue* = 255), *canIGrow* = false, *timeRecoveryBleaching* = 0.0, *newRecruit* = false. Agents of a surviving bleached colony have their *substrateSubCategory* = 'BleachedCoral', *growth_rate* reduced by half during at least six months (depending on the temporal representation of a time step), *fecundity_polyp* = 0.0 during the next year, become light grey (i.e. *red* = 245, *green* = 245, *blue* = 245), *timeRecoveryBleaching* = 1.0.

      ii. *year_event = year_event + 0.1*.

      iii. *calculatePercentageCover_and_NumberRecruits():* see 2) c. for details.

      iv. *updateAgentColonySize():* see 2) e. for details. In addition, the planar area of each colony, associated with its *IDnumber, species,* and *year_event* and the name of the event (i.e. 'bleaching') are exported in the simulation-related file output/ColonyPlanarArea/ColonyPlanarArea_... csv.

   d. *cyclone():*

      i. *cyclone_DMT():* all the coral colonies are selected one after another, their colony shape factor (i.e. fragility) is calculated as a function of their *growth_form* and *planar_area_colony*. If the colony shape factor >*cyclone_DMT* (i.e., intensity of the disturbance), the colony is dislodged. In that case, all the coral agents forming the colony are converted into barren ground (only *context, grid, x* and *y* remain unchanged). Branching colonies can potentially fragment and survive. A proportion of algae are removed by patches of agents (i.e. the algae agents are converted to the substrate supporting them) as a function of the algae susceptibility and the intensity of the disturbance (see §7.3).

      ii. *year_event = year_event + 0.1*.

      iii. *calculatePercentageCover_and_NumberRecruits():* see 2) c. for details.

      iv. *updateAgentColonySize():* see 2) e. for details. The data is exported in the simulation-related file.

5. grow():

   a. Display is updated.

   b. Agents are selected in a random order and have to opportunity to grow – to convert neighbouring agents within a certain radius into their own state: for the majority of interactions, all the agent own variables are updated except *context, grid, x* and *y*; in the case algae agents convert coral agents, the newly converted algae agents keep several of their previous state variables (i.e. *species, age_maturity, coloniality, colony_max_diameter, corallite_area, egg_diameter, fecundity_polyp, growth_form, mode_of_larval_development, reduced_scattering_coefficient, sexual_system, size_maturity, correction_coeff_polypFecundity, IDNumber, planar_area_colony*) so as to simulate the growth of algae on coral colonies. Only living agents have the opportunity to grow and the rules of interactions are complex (see §7.5).

6. *smoothingCoralColonie():* coral colonies are smoothed by converting into barren ground coral agents that have three or four Von Neumann neighbour agents not from the same colony.

7. *sandImport():* sand is added or removed by converting random patches of barren ground agents into sand (i.e., *species* = 'sand'; *red* = 255, *green* = 255, *blue* = 51) or patches of sand agents into barren ground (i.e., *species* = 'BarrenGround'; *red* = 160, *green* = 160, *blue* = 160), respectively. The desired sand cover of a given time step (*sand_cover*) is imported from data/Disturbance_sand.csv.

8. *AlgaeInvasion():* barren ground and dead coral agents that have not been grazed (i.e., *haveIbeenGrazed* = false) during the previously executed *grazing()* method are converted into algae agents. Interactively, patches of these agents are selected randomly in space and converted successively into one of the algae functional groups initially present.

9. *substrateCompositionCSV():*
   a. *year* is incremented.
   b. *calculatePercentageCover_and_NumberRecruits():* see 2) c. for details. Data is exported in the simulation-related files.
   c. *updateAgentColonySize():* see 2) e. for details. Data is exported in the simulation-related file.
   d. If this is the last time step: *rugosityToGrazing()* (see 3) a. for details) and data is exported in the simulation-related file.

## Discussion

We decided to implement coral growth, spatial competition, recruitment, their response to thermal and hydrodynamic disturbances and grazing pressure because they are fundamental ecological processes that shape coral communities and their dynamics. Additionally, these processes have been well studied, and associated functional traits have been collected on numerous species.

Executing these processes simultaneously, like in reality, is not possible because grid-cell agents can only be activated one after another. We consequently had to decide upon the order at which these processes were executed. We placed grazing, thermal and hydrodynamic disturbances before growth so they would affect the growth and interactions between coral colonies and algae. We implemented the possibility to decide on the order of occurrence between disturbances and coral reproduction because the timing of incidence of these events can be critical for coral population dynamics. Once all the organisms grow and compete, sand is imported or exported to a desired % cover. We did not implement a process of sedimentation (associated to hydrodynamic disturbances, for instance) due to insufficient empirical data. We thought it necessary to potentially represent sand cover because sand can occupy a large proportion of the reef and is an unstable substrate that prevents organisms from growing and settling. We placed this sedimentation process between growth and algae invasion to secure a sand cover close to the desired value and to constrain the remaining available space for the last process. Finally, we implemented algae invasion because we assumed that available non-grazed surface would be occupied by an algae within a several-month period in a real reef.

### Related code

*coralreef2/src/coralreef2/CoralReef2 Builder.java* and *ContextCoralReef2.java* (***Carturan et al., 2020***).

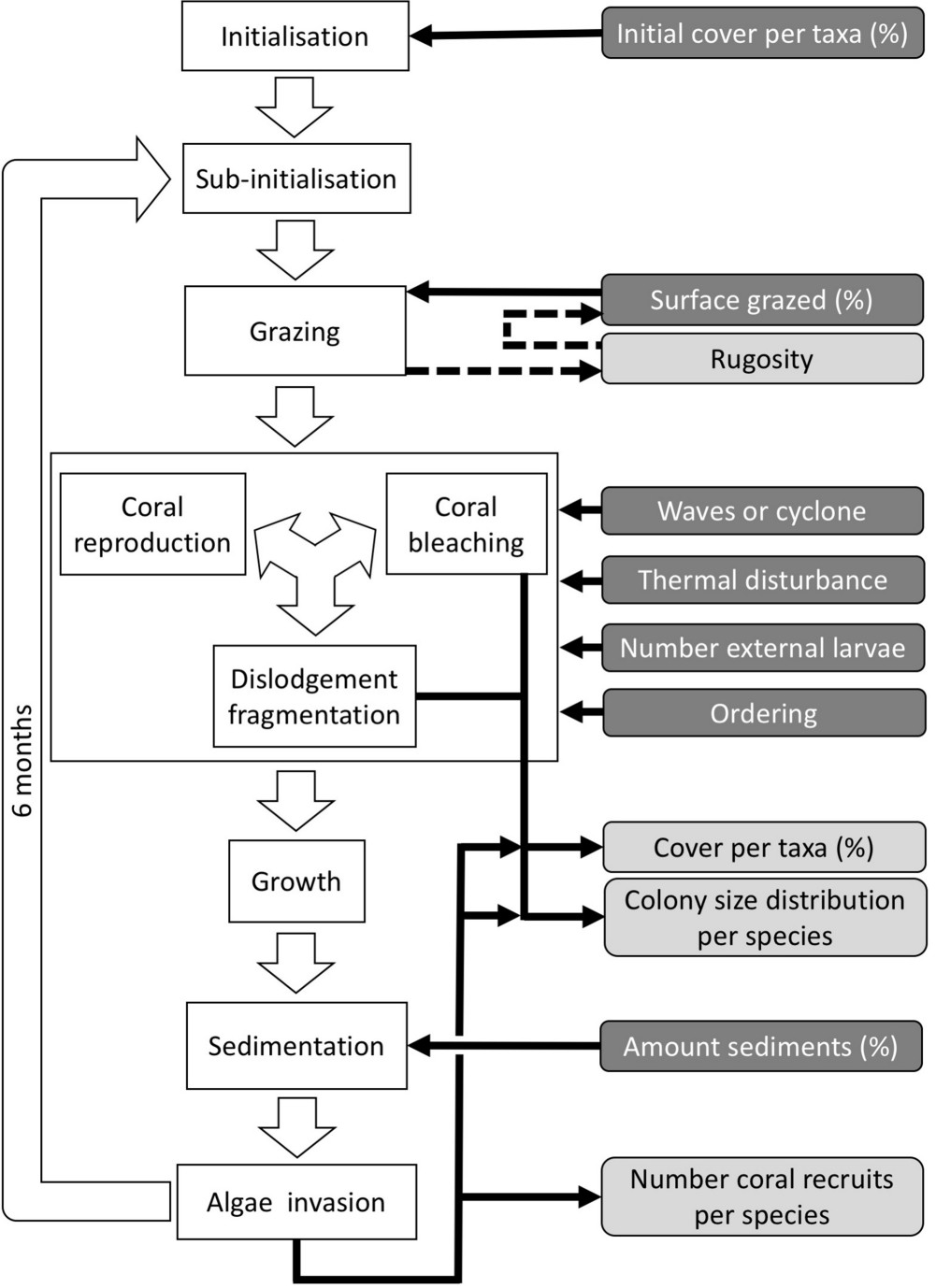

**Appendix 2—figure 2.** Ordering of the most important processes in the coral agent-based model: white rectangles represent processes, dark grey rectangles with white text are input data, and light grey rectangles with black text are outputs. Large white arrows define the ordering of processes and black arrows show the direction of data transfer; dashed black arrows are optional processes. The order of occurrence of coral reproduction, bleaching, and colony dislodgement and fragmentation is imposed to simulate recruitment failure due to the occurrence of a disturbance prior to reproduction. The intensity of waves and cyclones is expressed as a dimensionless dislodgment mechanical threshold; thermal stress is expressed in degree-heating weeks.

## 4. Design concepts

### Basic principles

The model combines three fundamental approaches: (i) a complex agent-based approach (*Breckling et al., 2006*; *Grimm, 2019*; *Grimm and Railsback, 2005*)—the dynamics observed at higher levels of organization (e.g. population percentage cover, recruitment rates, colony size distributions, rugosity) emerge from agent-related processes happening at the lower scales (e.g. conversion of a barren ground agent to a new coral recruit agent, dislodgement of a single colony); (ii) a functional trait-based approach (*Madin et al., 2016b*; *McGill et al., 2006*)—species diversity and dynamics are determined by mechanistically linked trait-process associations; (iii) a demographic approach (*Edmunds et al., 2014*; *Tuljapurkar and Caswell, 1997*)—the dynamics of a population depends on its demographic structure (e.g. colony size distribution) because the size of a colony influences its capacity to reproduce, compete and resist disturbances. We combined these three approaches by implementing evidence-based trait-process mechanistic associations at an appropriate spatial scale (e.g. single agent for a larvae recruiting, all the agents forming a colony during a bleaching event) and accounting for size-process interactions (e.g. proportion of fecund polyps in a colony, colony shape factor for dislodgement).

The model captures several fundamental principles: (i) species diversity influences ecosystem resilience (i.e. the diversity-stability relationship; *Ives and Carpenter, 2007*; *McCann, 2000*; *Nyström, 2006*) and (ii) functioning (i.e. the diversity-ecosystem functioning relationship; *Brandl et al., 2019*; *Loreau, 2000*), (iii) disturbance regimes shape communities by filtering species and mediating interspecific competition (*Kraft et al., 2015a*; *Sommer et al., 2014*), (iv) interspecific functional differences (or strategies) mediate competitive exclusion and coexistence (*Kraft et al., 2015b*; *Vellend, 2010*), (v) source-sink dynamics regulate species coexistence in metacommunities (*Amarasekare and Nisbet, 2001*; *Loreau and Mouquet, 1999*). Note that there are many mechanisms leading to species coexistence (e.g. niche partitioning, spatial heterogeneity, facilitation; *Adler et al., 2013*; *Chesson, 2000*) that we did not implement in the model.

### Emergence

All the model outputs (i.e. population % cover, number coral recruits m$^{-2}$, colony size distributions and eventually rugosity) emerge from the processes implemented and depend on the imposed environmental conditions (i.e. larvae connectivity, grazing pressure, hydrodynamics and thermal disturbance regime, sand input). The outputted percentage of sand cover (if included) results mostly from the imposed sand cover (but is it possible that not enough space is available to reach the desired cover). If the rugosity-grazing feedback process is activated (*Appendix 2—figure 2*), the percentage of reef grazed results from the sum of the % imposed (if the user decides to maintain a certain imposed grazing regime) and the % emerging from the complexity of the habitat created by coral colonies (see §7.1.2.2). Cell agents do not make decisions and their behavior results from imposed deterministic or probabilistic rules.

### Adaptation

Not implemented.

### Objectives

Not implemented.

### Learning

Not implemented.

### Prediction

Not implemented.

## Sensing

Not implemented.

## Interactions

Agents on the edge of coral colonies and patches of algae interact directly with one another when competing for space. The outcome of a coral-coral interaction is determined by its specific pairwise outcome probabilities (or *aggressiveness* if the probability is not available for the species pair), the probability of coral-algae interactions are the same for all coral species, and algae-algae interactions result in a stand-off except when competing against crustose coralline algae. Branching and plating species also have the capacity to overtop other colonies and algae depending on their size (see §7.5). Agent also directly interact by occupied space, which prevents potential new recruits to settle.

## Stochasticity

The model draws success or failure outcomes in several processes, based on probabilities that represent specific ecological phonemes or details. We used (i) grazing probabilities to represent the difference of palatability among algae functional groups (see §7.1.2); (ii) settlement probability to represent the differences of settlement suitability among substrate types (see §7.2.1.3); (iii) a surviving probability for the first few months of the life of new coral recruits in order to control recruitment rates without implementing the causes of new recruit mortality (see §7.2.1.3); (iv) coral-algae, algae-algae and published species-pair probabilities of interaction outcomes to represent the non-transitive and inconsistent nature of direct interactions between these organisms (see §7.5); (v) species-specific bleaching probabilities and a bleaching-induced probability of mortality to account for the complexity and variability of bleaching responses observed within populations (see §7.4); (vi) species-specific positively skewed density distributions of colony diameters to create coral colonies during the initialization of the reef (see §5.2.2). Finally, grazing and larval settlement and the placement of colonies and algae during the initialization of the reef happen randomly in space.

## Collectives

Coral colonies are collectives of coral agents emerging from the growth of individual coral agents (new recruits). Agents being part of a same colony (i.e. alive, bleached and dead coral and algae on a colony) behave collectively when dislodged (see §7.3.1.2.a); living coral agents behave collectively when bleaching, dying from bleaching (see §7.4) and reproducing (see §7.2.1.1). Plating and branching corals can overtop other colonies and algae if their colony is large enough (see §7.5.2.2.b).

## Observation

The model exports data that ecologists typically collect on the field to describe and understand patterns of community dynamics: (i) the percentage cover of each taxon (collected initially, after hydrodynamic and thermal disturbances and at the end of each time step and recorded in output/PercentageCover/PercentageCover_...csv), (ii) the mean number of recruits for each coral species m$^{-2}$ (collected at the end of each time step and recorded in output/NumberRecruits/NumberRecruits_...csv), (iii) the planar area of each colony (collected initially, after hydrodynamic and thermal disturbances and at the end of each time step and recorded in output/ColonyPlanarArea/ColonyPlanarArea_... csv), and optionally (*iv*) the rugosity of the reef created by coral colonies and the associated % cover grazed due to the fish community (collected during the grazing process and recorded in output/RugosityCoverGrazed/RugosityCoverGrazed_...csv).

## 5. Initialization

### 5.1. Initialization of species cover

The principal objective of the model is to simulate community dynamics as a function of species composition in different environmental scenarios. The composition and initial % cover for each coral species and algae functional group is imported from data/Initial_benthic_composition.csv and communities can be selected using their unique *communityNumber* (or '*Community Number*' in the GUI). It is possible to define a % cover of bleached and dead coral species. Once the

*communityNumber* is selected and the corresponding % cover imported, the *grid* is filled by creating agents. First, coral colonies are created at random locations, ordering the species from largest to smallest maximum colony diameter (this avoids having small colonies being enclosed by larger ones). Colonies are created by selecting an empty cell from the list of all available cells. A colony radius is then drawn from a species-specific size distribution (see §5.2) and used to select all the neighboring cells present in the list. These cells are removed from the list, and a cell-agent is created in each one of them. Upon creation, each agent is associated to the *context* and *grid*, and is given its *x* and *y* coordinates; all agent forming a same colony share the same other variables (*Appendix 2—table 1*). The procedure continues until the cover occupied reaches the defined value and is repeated for each coral species. Similarly, patches of algae agents (of a 10 cm radius) are then created in the remaining available cells, one functional group at a time, until the inputted cover is reached. The remaining empty grid-cells are filled by creating barren ground agents and finally patches of sand are created by converting patches of barren grounds agents. The initial % cover of sand is imported from data/Disturbance_sand.csv and it is associated to a certain environmental scenario, which can be selected using its unique *disturbanceScenarioNumber* (or '*Disturbance Scenario*' in the GUI; see §6).

Placing coral colonies randomly in space is probably a limitation as aggregation of colonies influence coral reef dynamics (*Brito-Millán et al., 2019*; *Eynaud et al., 2016*). Providing the option to define the initial spatial arrangement of the colonies could be part of future development of the model. So far, only *Wakeford et al. (2008)* implemented coral colonies spatial arrangements according to empirical data.

The number of agents created (i.e. the size of the *grid*) is defined by the parameters *reef_height* and *reef_width* (in cm).

## Related code

*coralreef2/src/coralreef2/CoralReef2 Builder.java* (*Carturan et al., 2020*).

## 5.2. Initialization of colonies size and age

### 5.2.1. Background

Colony size influences important processes such as colony fecundity (see §7.2), spatial competition (see §7.5) and mortality (see §7.3) (*Connell et al., 2004*; *Hughes et al., 1992*). Initializing colony size properly is consequently important. Size distributions of coral colonies are in nature positively skewed and approximate a symmetric distribution when log-transformed (e.g. *Bak and Meesters, 1999*; *Bak and Meesters, 1998*). The shape of the distribution depends on (*i*) life-history strategy— weedy species (i.e. short life cycle, high recruitment rate, small colony size) display positively skewed log-transformed distributions as opposed to stress resistant species (i.e. larger colonies, long life cycle, low recruitment rate; e.g. *Meesters et al., 2001*); and (*ii*) past disturbances having selective effects on certain colony sizes (e.g. *Bauman et al., 2013*).

The age of a colony also influences fecundity as young colonies need to reach maturity before sexually reproducing. However, colony size can vary enormously within individual colonies of a same cohort due to genetic and environmental differences. Because of its importance (regardless of age) on fecundity, competition and survival, colony size is considered a better predictor of coral fitness (*Hughes, 1984*).

### 5.2.2. Implementation

During the initialization of the reef, we draw colony diameters from species-specific colony size distributions. We generated these distributions using the following custom function:

```
nextSkewDistFun<-function(minVal=0,maxVal,skew,bias,n=10000){
    range<-maxVal-minVal
    mid<-minVal+range/2
    unitGaussian<-rnorm(n,0,1)
    biasFactor<-exp(bias)
    retval<-mid+(range*(biasFactor/(biasFactor+exp(-5))))
    retval
}
```

where *minVal* = minimum and *maxVal* = maximum colony diameter, *skew* and *bias* = two parameters influencing the shape of the distribution, and *n* = sample size. To generate the required species-specific size distributions, we predicted values for *bias* and *skew* as follow. We first defined *skew* and *bias* for 11 Caribbean species for which empirical size distributions were available based on data collected in Curaçao (Netherland Antilles) in 1996 (E. H. Meesters and R. P. M. Bak, personal communication, May 2017; *Appendix 2—figure 3*, *Appendix 2—figure 4*). We iteratively tried different values of *skew* and *bias* until the distributions visually matched the empirical ones.

The *colony maximum diameter* of these species spans a small range (11 to 250 cm) compared to the entire range of diameter found across all species (2 to 2000 cm). Consequently, we selected 20 additional species from the 798 species available in our coral traits imputed dataset (Appendix 1), selecting 10 with larger (706 to 2000 cm) and 10 with smaller (2 to 8 cm) colony maximum diameter. We defined the values of *skew* and *bias* of these species assuming a positively skewed colony size distributions and higher skewness for species having larger *maximum colony diameter* (because reaching very large colonies is unlikely in the intense disturbance regimes that commonly affect reefs). We then fitted least-squares linear regressions for both *skew* and *bias* using *maximum colony diameter* as a predictive variable (*Appendix 2—figure 5*, *Appendix 2—table 2*). Finally, we predicted the *skew* and *bias* values of the 798 species (*Appendix 2—figure 6*).

Given that colony size and age are poorly correlated, there was no basis for establishing age based on the value of colony size drawn from the distributions. Thus, in all cases, we set initial colony age at three years, which is the age at first maturity in the model (Appendix 1; §7.2.1.1.a). The effect on recruitment is minor because we modeled colony fecundity as a function of the size of the colony; age being used only to determine when colonies recruited during simulation can start reproducing (§7.2).

## Related code

*coralreef2/src/coralreef2/CoralReef2 Builder.java* (*Carturan et al., 2020*).

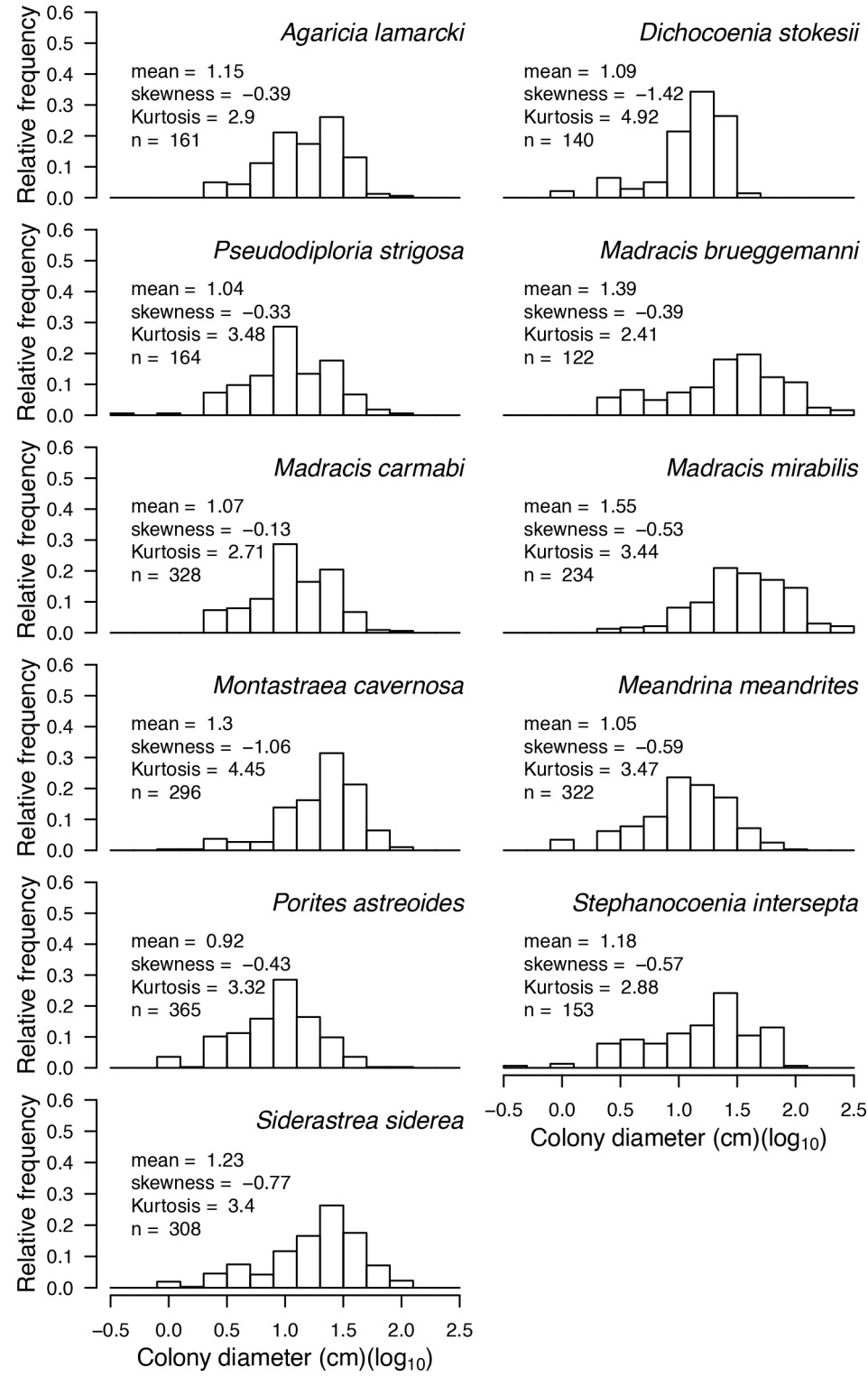

**Appendix 2—figure 3.** Size-class distributions of 11 Caribbean coral species (data collected in Cura-çao in 1996, E. H. Meesters and R. P. M. Bak, personal communication, May 2017).

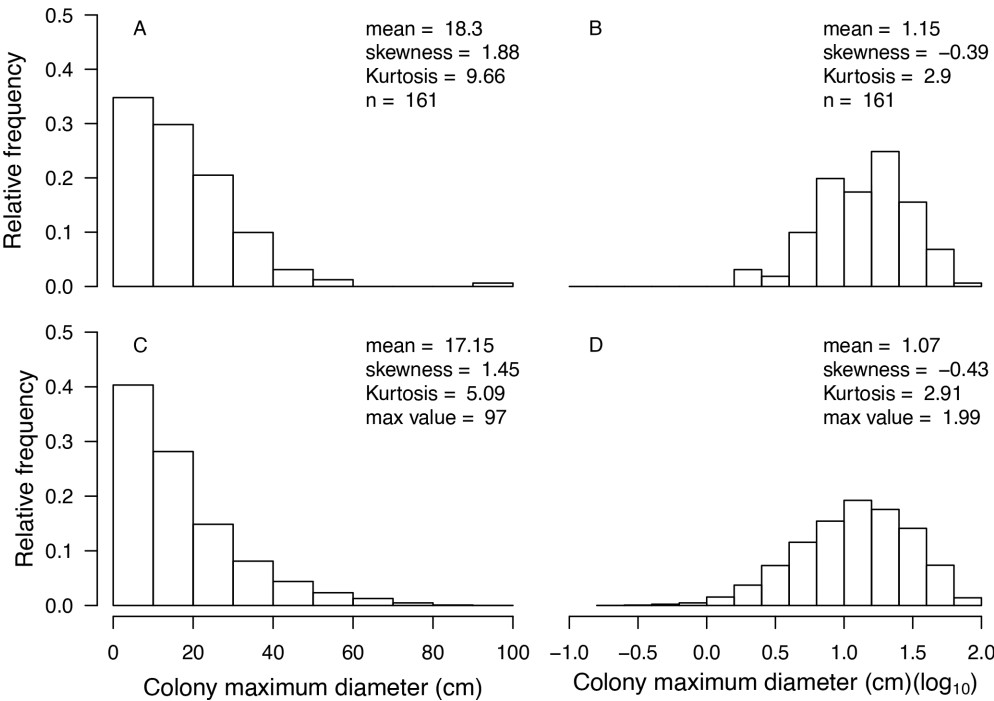

**Appendix 2—figure 4.** Example of a visual comparison of colony size distributions between the real data (**A, B**) and the one obtained with the command *nextSkewDistFun* (**C, D**) for *Agaricia lamarcki* (n = 10,000). Real data were collected in Curaçao in 1996 (E. H. Meesters and R. P. M. Bak, personal communication, May 2017). We obtained the simulated distributions with *bias* = −1.9 and *skew* = 0.88.

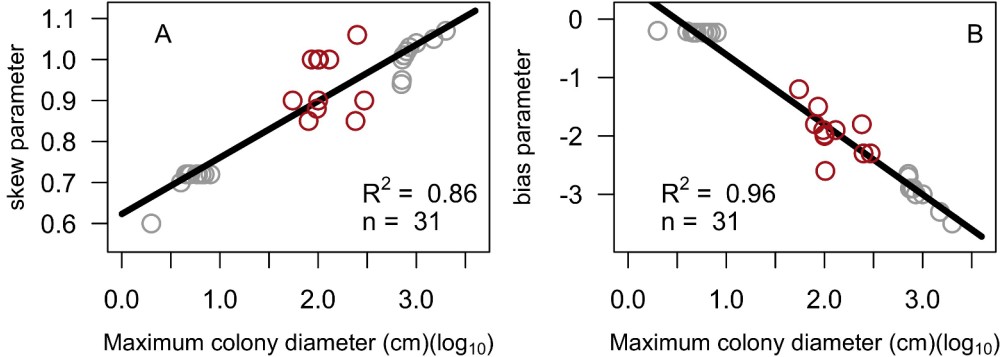

**Appendix 2—figure 5.** Linear regression models for *skew* (**A**) and *bias* (**B**) expressed as a function of *maximum colony diameter*. Red circles are values of *skew* and *bias* manually obtained by comparison with empirical colony size class distributions (E. H. Meesters and R. P. M. Bak, personal communication, May 2017). Grey circles are the values of *skew* and *bias* we defined for the 20 species we added in order to increase the sample size and range of colony sizes. Also displayed are the least squares linear regression and the corresponding coefficient of determination.

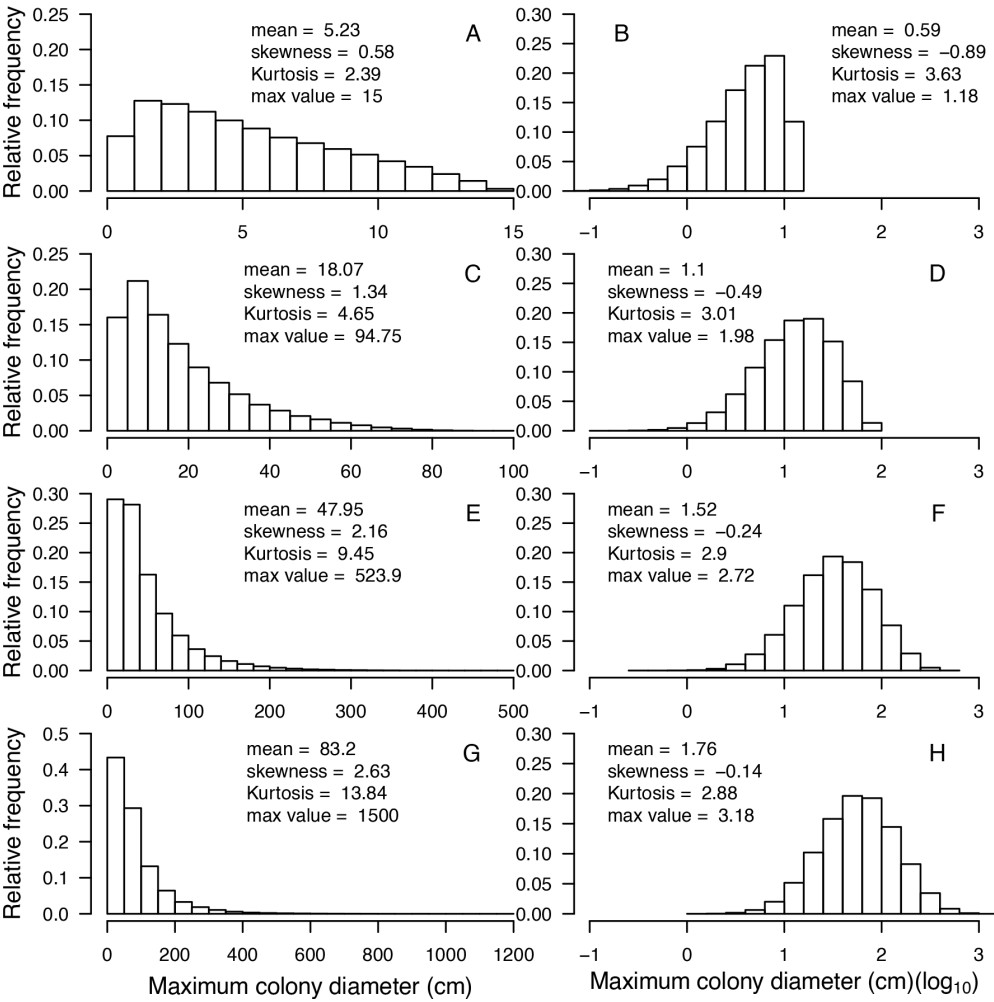

**Appendix 2—figure 6.** Examples of modeled species-specific colony diameter distributions (n = 100,000). From top to bottom: *Acropora nana* (**A, B**), *Acanthastrea echinata* (**C, D**), *Acropora longitcyathus* (**E, F**), *Acropora florida* (**G, H**). *Max value = maximum colony diameter.*

**Appendix 2—table 2.** Parameters of the linear regression models for *skew* and *bias* expressed as a function of *maximum colony diameter* ($\log_{10}$) (n = 31).

| Model | Parameters | Estimate | SE | $R^2$ | p-Value |
|---|---|---|---|---|---|
| *skew* | Intercept | 0.62 | 0.022 | 0.86 | <0.001 |
| | Slope | 0.14 | 0.010 | | <0.001 |
| *bias* | Intercept | 0.59 | 0.094 | 0.96 | <0.001 |
| | Slope | −1.20 | 0.044 | | <0.001 |

## 6. Input data

Five predefined time series of input data are used to define the environmental context (*Appendix 2—figure 2*) and are characterized by a unique *disturbanceScenarioNumber* (or 'Disturbance Scenario' in the GUI). The entire times series of a chosen scenario are imported upon initialization and the values used each time step are selected according to *year*. The time series define: (i) the hydrodynamic disturbance regime, in dislodgement mechanical threshold, which is a dimensionless measure of the mechanical threshold imposed by waves and cyclones on coral colonies (see §7.3.1.1). The data is imported from data/Disturbance_cyclone.csv. We are not aware of any source for this data. Thermal stress intensity (ii), in degree heating week, which is a measure of how much

heat stress has accumulated in an area (50 × 50 km) over the past 12 weeks (it is calculated by adding up any temperature exceeding the bleaching threshold). The data is imported from data/Disturbance_bleaching.csv. For the model calibration (Appendix 3: §1 and 2.2), we downloaded real data from the US National Oceanic and Atmospheric Administration data server ERDDAP (Environmental Research Division's Data Access Program; coastwatch.pfeg.noaa.gov/erddap). The value of degree-heating weeks we imposed each time step was the maximum degree heating week value recorded during the corresponding periods. The % of sand cover (iii) is imported from data/Disturbance_sand.csv. The % of reef grazed (iv) is imported from data/Disturbance_grazing.csv. For the model calibration, we estimated grazing from population densities of *Acanthuridae* spp., *Scaridae* spp. and *Diadema* spp. using submodels we established from empirical data (Appendix 3: §2.3). The number of larvae per coral species m$^{-2}$ coming from the regional pool (v) is defined by the parameter *connectivity* (or 'Connectivity' in the GUI) and is either imported from data/Disturbance_larvalConnectivity.csv (if *connectivity* = 'connectivityCSV') or constant (if *connectivity* = any other value, see §7.2.1.2.b).

### Related code

*coralreef2/src/coralreef2/CoralReef2 Builder.java*
*coralreef2/src/coralreef2/InputData/ BiodiversityData.java,*
*Disturbance_bleaching.java, Disturbance_cyclone.java, Disturbance_grazing.java,*
*Disturbance_larvalConnectivity.java, Disturbance_priority.java, Sand_cover.java* (**Carturan et al., 2020**).

## 7. Sub-models

All the related codes for production of the following figures are in *Manuscript/Rscripts/ Appendix S2 - Overview Design concepts and Details.R* (**Carturan et al., 2020**).

### 7.1. Grazing

#### 7.1.1. Background

The process of grazing is essential for maintaining reef ecosystems in a coral-dominated state (**Mumby et al., 2007**). The outcomes of coral-algae interactions are species- and context- specific (see §7.5.3), but coral species are in general disadvantaged by their slower growth rate when competing for space. Herbivores help corals by maintaining algae populations at low abundance and by influencing algal succession (**Hixon and Brostoff, 1996**; **McClanahan, 1997**). Reduction of grazing (due, for instance, to increased fishing) leads to algal populations increase, potentially exceeding the maximum grazing capacity (**Williams et al., 2001**), and allowing less palatable macroalgae to expand (**Hay and Fenical, 1988**). Negative feedbacks can then establish and maintain the ecosystem in an alternative stable and algae-dominated state. Alternatively, positive feedback processes help in maintaining the ecosystem in a coral-dominated state. The relationship between habitat complexity and grazing pressure is, for instance, one of the most important feedback process involved in reef resilience: coral colonies form complex structures that support a high diversity and biomass of herbivores, which enhances grazing pressure and favors coral growth and recruitment (**Bozec et al., 2013**; **Mumby and Steneck, 2008**; **van de Leemput et al., 2016**; **Vergés et al., 2011**).

Numerous species of herbivore graze on reefs, and they exhibit inter-class differences in grazing preference among algal functional groups (**Steneck and Dethier, 1994**). Sea urchin and nudibranch (*Gastropoda*) are for instance generalists and feed on most algae (**Diaz-Pulido and McCook, 2008**; **Morrison, 1988**). In contrast, fish species are generally more sensitive to allopathic defenses. Variation in allopathic sensitivity and behavior between grazers leads to different algal population dynamics, depending on the identities of the most abundant herbivores. For example, sea urchin populations can maintain a benthic algal community in an intermediate succession stage (**McClanahan, 1997**), whereas parrotfishes and surgeon fishes strongly deflect the trajectory of algal succession, and territorial damsel-fishes merely slow the successional rate (**Hixon and Brostoff, 1996**). Herbivory also affects spatial patterning: contrary to fish, sea urchins have spatially constrained movements and create foraging 'halos' which favor coral recruitment (**Sandin and McNamara, 2012**) and reef recovery (**Eynaud et al., 2016**). Finally, intense grazing regimes can have negative

effects on corals by increasing bioerosion (*Bellwood et al., 2004*) and reducing coral recruitment rate (*Sammarco, 1980*).

Algae differ in their palatability. Despite high interspecific variability, generalities have been made for functional groups. Turf algae are considered highly edible: they are consumed by all grazers (*Steneck and Dethier, 1994*), and their presence under high grazing pressure is due to their fast growth rate rather than their resistance to grazing (*Diaz-Pulido and McCook, 2008*). In comparison, macroalgae have a larger and thicker structure, which provides resistance against small grazing fish and crustaceans (*Hay, 1984*; *Mumby et al., 2007*), but see *Kuempel and Altieri (2017)*. Allelopathic macroalgae (AMA) release secondary metabolites strong enough to significantly reduce their palatability (*Hay and Fenical, 1988*; *Paul et al., 1990*). In addition to secondary chemical compounds, *Halimeda* spp. have calcareous structure, which reduces further their edibility (*Kuempel and Altieri, 2017*; *Lewis, 1985*; *McClanahan et al., 2002*). Crustose coralline algae (CCA) can be considered poorly eatable because of their calcified and encrusting structure, which explain why its cover is usually positively correlated with grazing intensity (*Belliveau and Paul, 2002*; *Steneck, 1997*; *Steneck, 1986*).

## 7.1.2. Implementation

### 7.1.2.1. General procedure

The proportion of the reef grazed at each time step is entered as an input variable and needs to be defined before the start of simulations. Values can be arbitrarily defined in case of an experiment. Simulating real grazing regimes can be achieved differently and depends on the type of data available. For instance, *Mumby (2006)* defined the percentage of the reef maintained in a cropped state as being proportional to *Scaridae* spp. (parrot fish) density and inversely proportional to rugosity. For the calibration of our model with the empirical data of the three sites in Martinique (Caribbean), we modeled grazing of fish (*Acanthuridae* spp. and *Scaridae* spp.) and urchins separately using published data (Appendix 3).

The reef is grazed by randomly selecting circular patches of agents (29 agents at a time). Non-algae agents selected during the grazing process are qualified as 'grazed', and consequently cannot be converted into algae during the present time step. Algae agents successfully grazed are converted into the type of substratum they were covering (barren ground or dead coral) and are qualified as 'grazed'. A selected algae agent avoids being grazed if its functional group-specific grazing probability (*Appendix 2—table 3*) is smaller than a random number generated from a uniform distribution bounded between 0 and one. The grazing process is executed until the desired percentage of cover cropped is reached.

**Appendix 2—table 3.** Probabilities of being grazed of each algal functional group implemented in the model.

We considered several values for calibration (Appendix 3). Bold values are the ones providing best fit in at least one of the three Caribbean sites. More than one bold value are shown when different values maximized the fit in different sites. No bold values are shown when none of the values tested improved the fit.

| Functional groups | Probability of being grazed | Reasoning |
|---|---|---|
| MA | 0.3; 0.5; **0.7** | Thicker structure |
| AMA | **0.3**; **0.5** | Thicker structure, strong secondary metabolites |
| Halimeda | 0.3; **0.5**; 1.0 | Thicker structure (calcareous), strong secondary metabolites |
| ACA | 0.5; **0.7**; 1.0 | Thick structure (calcareous) |
| Turf | 1.0 | Thin structure |
| CCA | 0.05; 0.1; 0.25; 0.5; 0.75 | Harder structure (calcareous), encrusting |

MA: macroalgae; AMA: allelopathic macroalgae; Halimeda: *Halimeda* spp.; ACA: articulate coralline algae; CCA: crustose coralline algae

### 7.1.2.2. The rugosity-grazing feedback (optional)

If the procedure is activated (*Rugosity_Grazing* = TRUE), the % cover grazed obtained is added to the one imported from the file. We implemented the feedback process by linking the rugosity of the reef created by coral colonies to the abundance of herbivorous fish, which we then linked to the percentage of reef maintained in a grazed state during the duration of a time step. Each time the state of the community is updated, the model calculates the linear rugosity of the reef with the formual from *Kubicek and Reuter (2016)*'s formula:

$$\text{Rugosity} = \sqrt{\frac{S_{\text{uncovered}} + \sum\limits_{i=1}^{n} S_{\text{colony } i}}{S_{\text{total}}}}$$

where $S_{uncovered}$ = the surface of the reef not covered by a coral colony, $S_{colony\ i}$ = the surface of the $i^{th}$ colony, $n$ = the total number of colonies present and $S_{total}$ = the surface of the reef (25 m$^2$). The model calculates colony surface areas ($S_{colony\ i}$) using geometric formulas defined for each growth (*Appendix 2—table 5*). We assumed that colonies with a planar surface area <100 cm$^2$ do not contribute to reef rugosity.

Rugosity is then used to determine the density of herbivorous fish using the following empirically established relationship (*Bozec et al., 2013*):

$$\text{Density}_{\text{Fish}} = 19.74 \times (\text{Rugosity} - 1)$$

where $\text{Density}_{\text{Fish}}$ = the density of herbivory fish (indiv.120 m$^{-2}$). Note that *Bozec et al. (2013)* established the relationship with individual fish belonging to eight parrot fish species (*Scaridae*), which are the dominant herbivorous fish in the Caribbean (*Appendix 2—table 4*).

The model then converts fish density in g.m$^{-2}$ using the mean fish length of each species measured by *Bozec et al. (2013)* and the following length-weight relationships:

$$\text{Weight} = a \times \text{Length}^b$$

where $b$ = the isometric growth in body proportions and $a$ = a parameter describing body shape. Values for $a$ and $b$ (*Appendix 2—table 4*) are available in FishBase (*Froese et al., 2014*; *Froese et al., 2014*).

Finally, the model determines the proportion of reef grazed ($Surface_{grazed}$) from herbivorous fish density using a asymptotic model we defined from *Williams and Polunin (2001)* empirical data (*Appendix 2—figure 7*):

$$\text{Surface}_{\text{grazed}} = \frac{70 \times \text{Density}_{\text{Fish}}^2}{\text{Density}_{\text{Fish}}^2 + 90}$$

where $\text{Density}_{\text{Fish}}$ = fish density (g.m$^{-2}$). *Williams and Polunin (2001)* conducted field surveys in 19 Caribbean reefs and analyzed the relationship between the percentage cover cropped (i.e., covered by either turf, crustose coralline algae or bare substratum but not by macroalgae) and the density of *Acanthuridae* spp. and *Scaridae* spp. present. (Other grazers such as the sea urchin *Diadema* spp. were not present in high enough abundance to influence their results.)

## Related code

*coralreef2/src/coralreef2/ContextCoralReef2.java* (*Carturan et al., 2020*).

**Appendix 2—table 4.** Parrot fish density and body length (total length ≥4 cm) collected on the fore reef zone of Glovers Atoll by *Bozec et al. (2013)* and values for a and b parameters of the from the length-weight relationships (LWR) and available from FishBase (*Froese and Pauly, 2014*).

| Species | Mean density (indiv. 120 m$^{-2}$) | Mean body length (cm) | A | B |
|---|---|---|---|---|
| *Scarus iserti* | 4.28 | 9.8 | 0.01096 | 3.02 |
| *Sparisoma aurofrenatum* | 2.41 | 13.4 | 0.01072 | 3.12 |

*Continued on next page*

*Appendix 2—table 4 continued*

| Species | Mean density (indiv. 120 m$^{-2}$) | Mean body length (cm) | A | B |
| --- | --- | --- | --- | --- |
| *Sparisoma viride* | 1.12 | 24.0 | 0.01380 | 3.04 |
| *Sparisoma chrysopterum* | 0.37 | 26.7 | 0.01072 | 3.09 |
| *Sparisoma rubripinne* | 0.12 | 28.4 | 0.00933 | 3.04 |
| *Scarus taeniopterus** | 0.04 | 16.7 | 0.01096 | 3.02 |
| *Scarus vetula* | 0.02 | 27.8 | 0.01445 | 3.04 |
| *Scarus coelestinus* | <0.01 (0.005)[†] | 40.0 | 0.01622 | 3.05 |

*Values for *a* and *b* for *Scarus taeniopterus* were not available so we chose values for *Scarus iserti* because the two species have similar maximum total and common lengths.

[†]We attributed a density of 0.005 indiv.120 m-2 for *Scarus coelestinus*

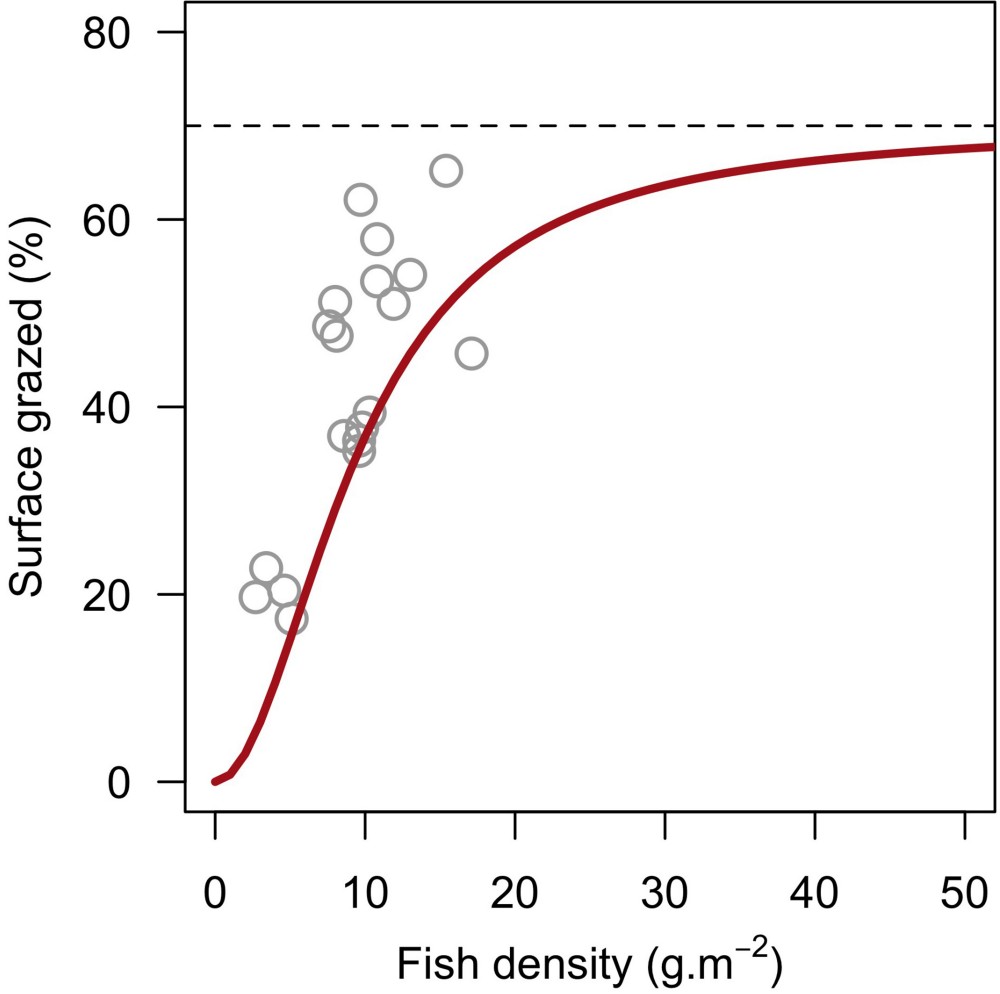

**Appendix 2—figure 7.** Model defining the percentage of reef surface grazed as a function of herbivorous fish density. Grey circles are averaged surface of reefs maintained in a cropped state as a function of pooled *Acanthuridae* spp. and *Scaridae* spp. densities in 19 Caribbean reefs (***Williams and Polunin, 2001***). The red line is the asymptotic model we defined: $y = 70x^2 / (90 + x^2)$.

## 7.2. Reproduction and recruitment

### 7.2.1. Implementation of coral reproduction

Coral recruits are composed of larvae produced locally and immigrating from the regional pool. The proportion of larvae immigrating from the regional pool depends on the parameter *connectivity*.

#### 7.2.1.1. Coral larvae locally produced

#### a. Onset of spawning

Using a three, four or six-month time step allows to implement seasonality (i.e., wet and dry seasons) and a distinction between broadcast spawning and brooding species: the latter release eggs once every season as opposed to spawners, which reproduce only once a year. During a spawning event, all the colonies able to reproduce will release gametes simultaneously. This implementation is in accordance with the simplified generalization that brooding species have multiple reproductive cycles through the year (*Ritson-Williams et al., 2009*) and spawning species only reproduce annually and in synchrony intra and interspecifically (*Baird et al., 2009b*). In reality, coral reproduction is more complex and diverse (e.g., *Glynn et al., 2000*). Importantly, the model offers the possibility to prioritize the onset of disturbances (i.e. cyclone and bleaching) and coral reproduction (*Appendix 2—figure 2*), as the timing of these different processes can be crucial for coral recruitment (*Harrison and Wallace, 1990*).

Juvenile colonies do not reproduce as they invest most of their energy into growth. We set the age at maturity of a colony at three years for all coral species. This value corresponds to the time needed for *Acropora millepora* colonies to become mature (*Guest et al., 2014*). The trait 'age at maturity' certainly varies among species (*Harrison and Wallace, 1990*) but the only three other species for which this trait is available are *Coelastrea aspera* (4.5 years), *Goniastrea favulus* (5.5 years), *Platygyra sinensis* (6.5 years) (coraltraits.org; *Madin et al., 2016a*). We chose the smallest value to avoid penalizing species that strategically invest into early onset of sexual reproduction and because the size of the colony is more important for determining colony fecundity (*Hughes, 1984*).

#### b. Calculation of the total number of oocytes produced on the reef (O$_t$)

The total number of oocytes produced in the reef for each species ($O_t$) is obtained with the formula:

$$O_t = \frac{f_p \times p_f}{C_a} \times \sum_{i=1}^{N} S_i \times p_{m_i}$$

where $S_i$ = three-dimensional surface area of a colony, $p_{mi}$ = the proportion of mature polyps in this colony, $N$ = the total number of colonies in the population, $f_p$ = the trait *polyp fecundity*, $p_f$ = the proportion of female polyps in the population and $C_a$ = the trait *corallite area*.

The surface $S$ is obtained by summing the three-dimensional area of all the colonies of a given species. Given that vertical growth is not explicitly simulated in the model, these surface areas are obtained by calculating colony planar surface area ($S_p$) using geometric models defined for each coral growth form (*McWilliam et al., 2018a*, *Appendix 2—table 5*, *Appendix 2—figure 8*). For simplification, the radius of each colony $r$ is estimated under the assumption that colonies' planar areas are circular.

The proportion of mature polyps in a colony ($p_m$) varies interspecifically and increases with colony size (*Álvarez-Noriega et al., 2016*), and the proportion of the 'sterile zone' (i.e. zone often situated at the extremities of the colonies, where polyps invest more in growth than reproduction) is bigger in smaller size colonies. *Álvarez-Noriega et al. (2016)* defined empirical models predicting polyp maturity probability as a function of *growth form* and colony planar surface area. They defined their models for eight coral species and four growth forms. This limited number of species and growth forms prevented the definition of a model for all 798 species or all nine growth forms used in our coral ABM. In consequence, we defined a single model by averaging the model parameters over the eight species (*Appendix 2—table 6*, *Appendix 2—figure 9A*). The parameterized model is:

$$\text{logit}(p_m) = 8.626 + 1.682 \times \log_e(S_p)$$

with the colony planar surface area ($S_p$) expressed in m$^2$. Several of the species in our set only reach small maximum colony sizes. Applying this model for them would underestimate their

reproductive output. We consequently defined species-specific models by applying a correction coefficient ($C_c$) as follows:

$$\text{logit}(p_m) = 8.626 + 1.682 \times log_e(S_p + C_c)$$

$$C_c = \frac{\text{logit}(0.9) - 8.626}{1.682} - log_e(S_{pmax})$$

$$S_{pmax} = \frac{\pi}{4} \times d_{\max}^2$$

with $S_{pmax}$ = the maximum planar surface area the colony can reach (m$^2$) and $d_{max}$ = the trait *maximum colony diameter* (m). We defined the correction coefficient (*correction_coeff_polypFecundity*) so that a polyp has 0.9 chance of being fecund when its colony has reached its maximum size. To be selected for this correction, a species must have a strictly positive $C_c$, which corresponds to having $S_{pmax}$ < 218.9 cm$^2$ and $d_{max} \leq$ smaller than 16.7 cm (*Appendix 2—figure 9B*). For comparison, the smallest species considered in *Álvarez-Noriega et al. (2016)* (for which $d_{max}$ is known) is *Acropora nasuta*, which has a 80 cm colony maximum diameter, corresponding to a 5026.5 cm$^2$ circular planar surface area.

The proportion of female polyps in a colony (or female colonies in a population) $p_f$ depends on the species sexual system, which can be globally classified as 'hermaphrodite' or 'gonochore' (with numerous variations; *Baird et al., 2009b*). We chose $p_f$ = 1.0 for hermaphrodites species and $p_f$ = 0.5 for gonochoric species in order to consider the proportion of male polyps in a colony or male colonies in the population (both scenarios being observed in reality).

**Appendix 2—table 5.** Formulae and values used to calculate the three dimensional surface area of colonies depending on their growth form (*McWilliam et al., 2018b*).
Note that we corrected the formula for laminar (M. McWilliam, personal communication, July 2019).

| Growth from | Formula surface area | Parameter values |
|---|---|---|
| Branching* | $\pi r^2 \left( N_b \left( 2\pi r_b h_b + \pi r_b^2 \right) \right)$ | $r_b = 1$<br>$h_b = 10$<br>$N_b = 0.225$ |
| Tabular | See branching | $r_b = 0.5$<br>$h_b = 1$<br>$N_b = 2.5$ |
| Laminar | $2\pi r \sqrt{r^2 + h_b^2}$ | $h_b = 20$ |
| Massive | $2\pi r^2$ | - |
| Corymbose | See branching | $r_b = 1$<br>$h_b = 5$<br>$N_b = 0.5$ |
| Digitate | See branching | $r_b = 2$<br>$h_b = 5$<br>$N_b = 0.2$ |
| Columnar | See branching | $r_b = 3$<br>$h_b = 25$<br>$N_b = 0.05$ |
| Encrusting long upright | Encrusting + branching | $r_b = 0.5$<br>$h_b = 5$<br>$N_b = 0.2$ |
| Encrusting | $\pi r^2$ | - |

Parameters are: colony radius (*r*); branch radius (*rb*); branch height (*hb*); number of branches per cm2 (*Nb*).

\* Coefficient values were obtained by averaging values for 'complex' and 'simple' branching.

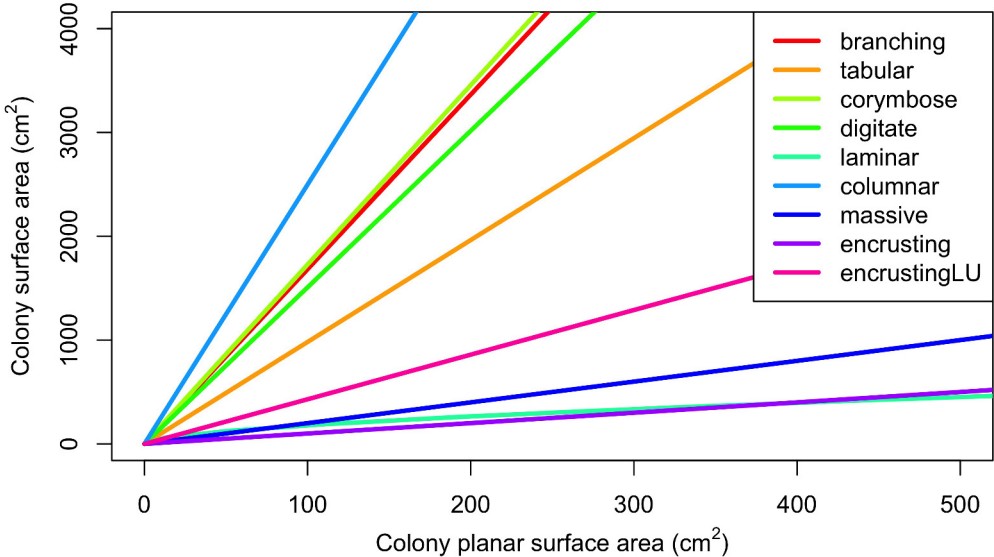

**Appendix 2—figure 8.** Conversion of colony planar area into three-dimensional surface area using geometric models for each growth form (*McWilliam et al., 2018a*). The conversion necessitated considering the planar surface area of each colony as circular.

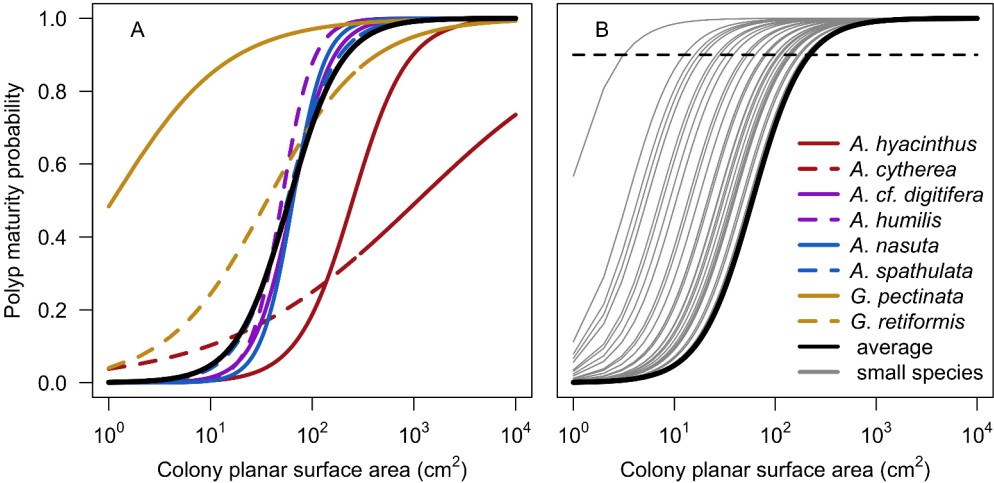

**Appendix 2—figure 9.** Probability of a polyp to be mature depending on the size of the colony. Coloured lines (i.e. red, purple and yellow) in panel A displays the eight species-specific models established by *Álvarez-Noriega et al. (2016)*; the black line in panels A and B represents the model obtained by averaging coefficient over the eight species. Panel B displays the 50 models we defined for the species reaching a maximum planar surface area inferior to 218.9 cm$^2$. The horizontal dashed line indicates 0.9 probability; it intercepts with individual grey lines when the colony of the corresponding species reaches its maximum planar area (assuming the latter is circular).

**Appendix 2—table 6.** Parameter values for the models defining the probability of a polyp to be fecund as a function of colony planar surface area (*Álvarez-Noriega et al., 2016*).

| Species | Growth form | Intercept | | | Slope | | | |
|---|---|---|---|---|---|---|---|---|
| | | LCI | Median | UCI | LCI | Median | UCI | n |
| *A. hyacinthus* | tabular | 4.022 | 5.908 | 8.345 | 0.889 | 1.602 | 2.477 | 24 |
| *A. cytherea* | tabular | 0.446 | 1.023 | 1.657 | 0.107 | 0.462 | 0.841 | 24 |
| *A. digitifera* | digitate | 7.309 | 11.507 | 17.142 | 1.252 | 2.277 | 3.572 | 24 |

*Continued on next page*

*Appendix 2—table 6 continued*

| Species | Growth form | Intercept | | | Slope | | | n |
|---|---|---|---|---|---|---|---|---|
| | | LCI | Median | UCI | LCI | Median | UCI | |
| *A. humilis* | digitate | 9.477 | 14.999 | 22.269 | 1.649 | 2.833 | 4.339 | 24 |
| *A. nasuta* | corymbose | 7.782 | 14.074 | 22.922 | 1.296 | 2.791 | 4.753 | 24 |
| *A. spathulata* | corymbose | 5.933 | 9.496 | 14.107 | 0.973 | 1.838 | 2.916 | 24 |
| *G. pectinata* | massive | 0.802 | 7.036 | 14.169 | −0.594 | 0.771 | 2.126 | 20 |
| *G. retiformis* | massive | 2.796 | 4.966 | 7.423 | 0.376 | 0.883 | 1.450 | 20 |
| *Average* | - | - | 8.626 | - | - | 1.682 | - | - |

### c. Calculation of the number of larvae produced on the reef ($L_c$)

The total number of competent larvae produced on the reef for each species ($L_c$) is given by:

$$L_c = O_t \times f_r \times (1 - p_r)$$

with $O_t$ = the number of oocytes produced, $f_r$ = the fertilization rate and $p_r$ = the predation rate.

Fertilization rate varies among species (**Negri et al., 2007**) and depends on gametes concentration (**Nozawa et al., 2015**; **Oliver and Babcock, 1992**), temperature and environmental conditions (**Ritson-Williams et al., 2009**). For simplicity, we chose $f_r$ = 0.5, a value approximating the average fertilization rates measured in the field for *Montipora digitata* by **Oliver and Babcock (1992)**. We attributed the same value for brooding species because no information is available for this mode of larval developmen—**Brazeau and Lasker (1992)** found a fertilization rate comprised between 5% and 25% in a brooding octocoral species.

Predation of coral larvae by fish is considered as one of the major sources of larval mortality (**Hamner et al., 1988**; **Pratchett et al., 2001**; **Westneat and Resing, 1988**). **Pratchett et al. (2001)** estimated in a study conducted in Lizard Island (Great Barrier Reef) that between 20% and 36% of coral propagules released during a spawning event are consumed by reef fish. According to these results, we fixed predation rate $p_r$ = 0.3. Larval predation by coral is also important (**Fabricius and Metzner, 2004**) but is considered during the settlement process (see §7.2.1.3). We did not implement the difference of palatability of coral propagules to fish (**Baird et al., 2001**) and corals (**Fabricius and Metzner, 2004**).

### d. Calculation of the number of competent larvae settling on the reef ($L_s$)

The number of larvae settling in the reef ($L_s$) is determined by the number of viable larvae becoming competent before being flushed away from the reef (**Connolly and Baird, 2010**):

$$L_s = L_c \times p_s \times p_o$$

with $L_c$ = the number of competent larvae, $p_s$ = the proportion of the latter settling and $p_o$ = a proportional coefficient (i.e. the model parameter *otherProportion*; Appendix 3) accounting for other factors potentially affecting the number of larvae locally settling.

The proportion of larvae settling $p_s$ is the proportion of larvae reaching competency while remaining on the reef. **Figueiredo et al. (2013)** modeled the relationship between time to motility ($t_m$, in days) and the proportion of competent larvae retained on the reef (and settling) for different retention times (using eight spawning species):

$$p_s = \alpha + \beta \times t_m$$

(or)

$$p_s = \alpha + \gamma \times e^{-\rho \times t_m}$$

with $\alpha$, $\beta$, $\gamma$ and $\rho$ being model parameters they empirically defined (**Appendix 2—table 7**). We chose the middle range value of 4.69 days as retention time for our simulations but other values for *retentionTime* = 16.30, 10.24, 7.66, 6.97, 2.14, 1.5, 1.21, 0.90 and 0.70 days. Additionally, the

authors established a significant linear relationship between time to motility ($t_m$, in hours) and the trait *egg diameter* (i.e. $e_d$, μm, n = 20 spawning species), so that the $t_m$ value is species-specific:

$$t_m = 0.059 \times e_d + 0.067$$

For brooding species, larvae are matured and motile when released in the water (**Gleason and Hofmann, 2011**) so their time to motility equals zero.

The proportional coefficient $p_o$ is introduced to account for other potential factors, such as the proportion of non-viable oocytes or spawning disynchrony observed between individual colonies of a same species (**Baird et al., 2000**). Its value is 0.0001 and was obtained during the model calibration (Appendix 3).

**Appendix 2—table 7.** Parameter values for the different models expressing the proportion of larvae retained in the reef ($p_s$) as a function of time to motility ($t_m$, in days) and retention time (**Figueiredo et al., 2013**).

| Retention time (d) | $\alpha$ | $\beta$ | $\gamma$ | $\rho$ |
| --- | --- | --- | --- | --- |
| 16.3 | 0.801 | −0.222 | - | - |
| 10.24 | 0.768 | −0.247 | - | - |
| 7.66 | 0.741 | −0.267 | - | - |
| 6.97 | 0.731 | −0.274 | - | - |
| 4.69 | 0.180 | - | 0.545 | 1.354 |
| 2.14 | 0.090 | - | 0.557 | 2.740 |
| 1.50 | 0.050 | - | 0.551 | 3.400 |
| 1.21 | 0.031 | - | 0.536 | 3.800 |
| 0.90 | 0.014 | - | 0.501 | 4.500 |
| 0.70 | 0.006 | - | 0.461 | 5.100 |

### 7.2.1.2. Larvae immigrating from the regional pool

Numerous processes and factors influence larval connectivity: distance between the reefs, oceanic currents, frictional forces of coastal topography, predation (**Cowen and Sponaugle, 2009**), mortality and loss of competency (**Connolly and Baird, 2010**). We defined the parameter *connectivity* to determine the number of larvae immigrating to the focal reef. By setting *connectivity* = 'noConnectivity', the focal reef is totally isolated, and no larvae immigrate. By choosing *connectivity* = 'connectivityCSV', the model imports the number of larvae m$^{-2}$ ($L_{sr}$) for each coral species and each time step from coralreef2/data/Disturbance_larvalConnectivity.csv. The time series of larvae density have to be defined manually beforehand and are associated to the parameter *disturbanceScenarioNumber*.

Alternatively, *connectivity* can be set to certain distance separating the focal 'sink' reef from a fictional 'source' reef: 'high (5 km)', 'medium (10 km)', 'low (20 km)', 'isolated (100 km)' and 'isolated (200 km)'. Each distance represents a certain number of alive and competent larvae m$^{-2}$ immigrating to the focal reef ($L_{sr}$). We made the following assumptions to determine the number and diversity of immigrating larvae: (i) the remote reef has the same species; (ii) the total number of larvae produced is shared equally between species; (iii) brooding species larvae are three times less abundant than spawning species larvae because their populations are usually more closed in comparison to spawning species populations (**Doropoulos et al., 2015**); (iv) only brooded larvae are produced during the non-reproductive season, in an amount equivalent to the number of brooded larvae produced during the reproductive season. We explain how we determined the number of alive and competent larvae m$^{-2}$ immigrating ($L_{sr}$) for each distance we considered in the following sections.

a. Number of larvae produced in the remote reef

The total number of larvae produced on the remote reef ($L_{cr}$) is determined using the following assumptions: (*i*) the remote and focal reefs have the same surface area ($S_r$, m$^2$); the remote reef (*ii*) produces $10^6$ larvae m$^{-2}$ (**Hall and Hughes, 1996**; **Pratchett et al., 2001**) and (*iii*) has a 50% coral cover. We applied the same predation rate ($p_r$) with the focal reef:

$$L_{cr} = 10^6 \times S_r \times 0.5 \times (1 - p_r)$$

### b. Number of larvae reaching the focal reef

The number of larvae reaching the focal reef m$^{-2}$ ($L_{sr}$) is:

$$L_{sr} = L_{cr} \times p_l \times p_{rf} \times p_{ac}$$

with $L_{cr}$ = the total number of larvae produced on the remote reef m$^{-2}$, $p_l$ = the proportion of larvae leaving the remote reef, $p_{rf}$ = the proportion of larvae reaching the focal reef and $p_{ac}$ = the proportion of them being still alive and competent.

The proportion of larvae leaving the remote reef depends on retention time and time to motility (**Figueiredo et al., 2013**). We arbitrarily chose $p_l$ = 0.5.

The proportion of larvae reaching the focal reef $p_{rf}$ depends on the distance separating the two reefs and the speed and direction of water currents. In **Black (1993)** simulated experiment, between 0.1% and 10% of particles released from an upstream reef were captured by a downstream reef (they had the same size and were spaced by 19 km), depending on the orientation of the water current. We chose $p_{rf}$ = 1.0%, the most commonly observed proportion in the simulations. In the model, the distance 19 km corresponds to a 'low level of connectivity' as it is the maximum reef spacing in the Great Barrier Reef (**Black, 1993**). Based on this value, we arbitrarily estimated the percentage of larvae reaching the focal reef for other levels of connectivity (**Appendix 2—table 8**).

While coral larvae are transported between reefs, their risk of losing competency and dying increases with time. **Connolly and Baird (2010)** established species-specific models for five species to predict the proportion of larvae in a cohort that are competent and alive as a function of time. We defined a unique 'average model' for all the 798 species by averaging the coefficients (because five species was not enough to define species-specific coefficients using traits based predictive models; **Appendix 2—table 9**; **Appendix 2—figure 10**):

$$p_{ac}(t) = p_{\text{alive}}(t) \times p_{\text{competent}}(t)$$

$$p_{alive}(t) = e^{-(\lambda t)^{\upsilon}}$$

$$p_{\text{competent}}(t) = \begin{cases} 0 & t > tc \\ \frac{a\left(e^{-b(t-t_c)} - e^{-a(t-t_c)}\right)}{a-b} & t > tc \end{cases}$$

with $P_{\text{competent}}(t)$ and $P_{\text{alive}}(t)$ the proportions of competent and alive larvae, respectively, as a function of time $t$ (in days) and $t_c$ = the development time required before acquisition of competency. We chose velocity = 0.15 m.s$^{-1}$ (common value observed in the GBR, **Brinkman et al., 2002**), we used the average model to determine the proportion of larvae still alive and competent $p_{ac}$ for the different levels of connectivity (**Appendix 2—table 10**).

**Appendix 2—table 8.** Proportion of larvae from the remote reef reaching the focal reef ($p_{rf}$) depending on the separation distance.

| Distance between reefs | $p_{rf}$ |
|---|---|
| High (5 km) | 0.5 |
| Medium (10 km) | 0.1 |
| Low (20 km) | 0.01 |
| Isolated (100 km) | 0.001 |
| Isolated (200 km) | 0.0001 |
| Not connected | 0.0 |

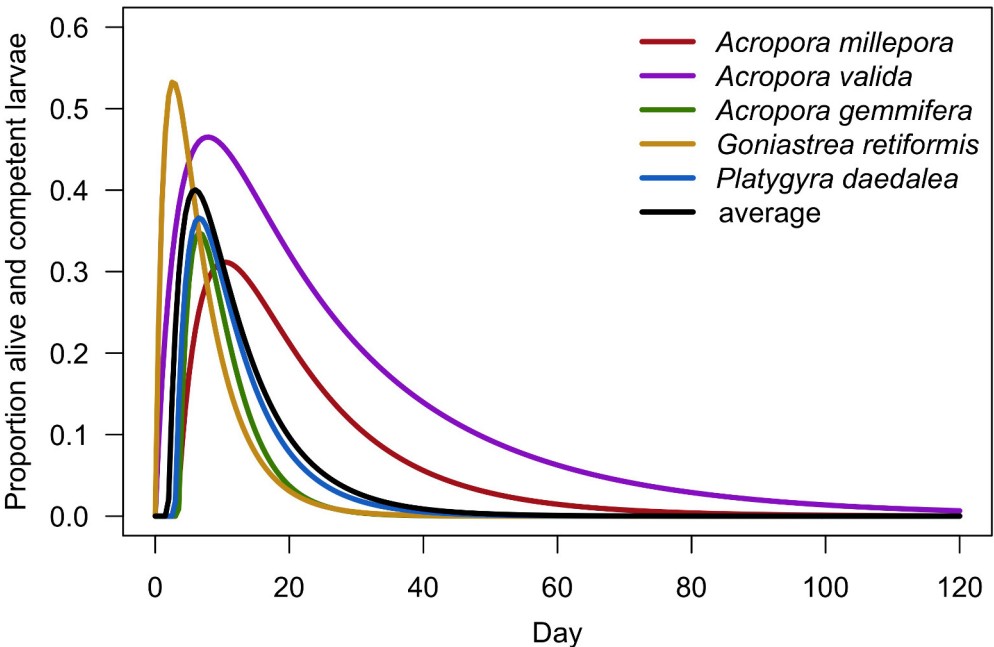

**Appendix 2—figure 10.** Prediction of the dispersal potential depending on time for five coral species (***Connolly and Baird, 2010***). The black line represents the prediction from the average model (i.e. we averaged the coefficients of the five species-specific models).

**Appendix 2—table 9.** Parameter values for the different models predicting the proportion of larvae alive ($P_{alive}$) and competent ($P_{competent}$) as a function of time (***Connolly and Baird, 2010***).

| Species | Competency model ($P_{competent}$) | | | Survival model ($P_{alive}$) | |
|---|---|---|---|---|---|
| | $a$ | $b$ | $t_c$ | $\lambda$ | $\nu$ |
| *Acropora millepora* | 0.180 | 0.050 | 3.239 | 0.043 | 0.57 |
| *Acropora valida* | 0.220 | 0.031 | 0.000 | 0.019 | 0.46 |
| *Acropora gemmifera* | 0.390 | 0.145 | 3.471 | 0.067 | 1.00 |
| *Goniastrea retiformis* | 0.580 | 0.096 | 0.000 | 0.087 | 1.00 |
| *Platygyra daedalea* | 0.390 | 0.099 | 2.937 | 0.060 | 0.72 |
| Average | 0.352 | 0.084 | 1.929 | 0.055 | 0.75 |

*tc*: Development time required before acquisition of competence can begin, in days.

**Appendix 2—table 10.** Proportion of larvae remaining alive and competent ($p_{ac}$) as a function of the distance between the remote and the focal reef.
We defined the traveling time assuming 0.15 m.s$^{-1}$ current velocity and alignment of the reefs in the current direction. Also displayed is the corresponding number of larvae settling by squared meter of the focal reef ($L_{sr}$) and the corresponding values of the model parameter *connectivity*.

| Distance between reefs | Connectivity | Duration journey (d) | $p_{ac}$ | $L_{sr}$ (m$^{-2}$) |
|---|---|---|---|---|
| 5 km | 'high (5 km)' | 0.39 | 0.40 * | 35000.00 |
| 10 km | 'medium (10 km)' | 0.77 | 0.40 * | 7000.00 |
| 20 km | 'low (20 km)' | 1.54 | 0.40 * | 700.00 |
| 100 km | 'isolated (100 km)' | 7.72 | 0.38 | 66.50 |
| 200 km | 'isolated (200 km)' | 15.43 | 0.17 | 2.98 |
| ∞ | "noConnectivity | - | 0.0 | 0.00 |
| - | 'connectivityCSV' | - | - | - |

* At these distances, the larvae have reached the focal reef before being competent. We attributed the maximum proportion as we assumed these larvae remain and settle in the focal reef.

### 7.2.1.3. Larval settlement
#### a. Background

Larval settlement is complex as it varies inter-specifically and depends on environmental factors. Certain species produce larvae capable of habitat selection (*Golbuu and Richmond, 2007*; *Harrington et al., 2004*; *Morse et al., 1988*), which increases post-settlement survival (*Ritson-Williams et al., 2009*). Certain crustose coralline algae species attract larvae for settlement and induce larval development but others use anti-settlement strategies (*Harrington et al., 2004*; *Price, 2010*). Settlement success can also be specific to the coral-algae species associations (*Ritson-Williams et al., 2010*). Finally, additional factors such as topographic cues (*Whalan et al., 2015*), light exposure (*Morse et al., 1988*) and density-dependence (*Doropoulos et al., 2017*) influence the settlement process. In consequence, recruitment probability is highly variable, context and time dependent (*Appendix 2—table 11*).

**Appendix 2—table 11.** Proportions of larvae settling, metamorphosing and surviving ($P_{ss}$) in different laboratory experiments.

| $P_{ss}$ (%) | Substratum type | Duration | No. coral sp. | Reference |
|---|---|---|---|---|
| 36 to 87 | 5 CCA sp. | 1 day | 3 | *Morse et al., 1988* |
| 67 to 91 | 4 CCA and 1 ACA sp. | 8 days | 1 | *Heyward and Negri, 1999* |
| 27 | Rubble | | | |
| 0 to > 60 | Coral skeleton | | | |
| 64.2; 57.2; 47.1 | Tile | 2, 30, 60 days | 2 | *Nishikawa et al., 2003* |
| 81.7, 59.9, 13.1 | | 0, 20, 40 days | | |
| 24.2 | 1 CCA sp. | 240 days | 2 | *Harrington et al., 2004* |
| 20.1 | Tile | | | |
| 0 | four other CCA sp. | | | |
| 0 to 80 | 2 CCA sp., rubble | 24 hr | 2 | *Golbuu and Richmond, 2007* |
| 0 to 18 | 2 CCA sp. | 6 weeks | 2 | *Ritson-Williams et al., 2010* |

CCA: crustose coralline algae; ACA: articulated coralline algae

#### b. Implementation

Larvae settle one after another randomly in the reef. Coral species do not differ in their capacity to settle. We defined the probability of recruitment $p_{lr}$ as:

$$p_{lr} = p_{ls} \times p_{lxs}$$

with $p_{ls}$ = the probability to successfully settle and $p_{lxs}$ = the probability to survive during the number of months represented by a time step.

The probability to successfully settle $p_{ls}$ depends on the type of substratum and was estimated to the best of our judgement (*Appendix 2—table 12*).

We defined $p_{lxs}$ using *Ritson-Williams et al. (2016)* experimental results. The authors first let larvae of two brooding and two spawning species settle and metamorphose on tiles covered with a 'preferred' crustose coralline algae species (*Hydrolithon boergesenii*) under laboratory conditions. They then placed the tiles in a reef and measured the proportion of new recruits surviving at different time intervals. We used their results (combining all four species) to fit a least-squares regression

model that we used to predict the proportion of new recruits surviving after 6 months (**Appendix 2—figure 11**, **Appendix 2—figure 12**, **Appendix 2—table 13**).

**Appendix 2—table 12.** Probabilities of successful coral larvae settlement on different substrata.

| Substratum | $p_{ls}$ | Justification |
|---|---|---|
| Barren ground dead coral CCA | 0.5 | We chose this mid value for all the suitable substratum types because of the high diversity of coral specificity toward CCA species and other substratum (**Birrell et al., 2008**), and the wide ranges of settlement success (**Appendix 2—table 11**), and insufficient knowledge. |
| Sand | 0.0 | Sand does not provide a stable substratum for coral larvae to metamorphose |
| Alive coral | 0.0 | Corals feed on larvae (**Fabricius and Metzner, 2004**) |
| Bleached coral | 0.0 | Bleached corals rely on heterotrophy to compensate for the loss of the symbiont (**Grottoli et al., 2006**) |
| Macroalgae turf AMA | 0.0 | Pre-emption of space, and release of deleterious or lethal chemicals impede larvae metamorphosis (**Birrell et al., 2008**) |
| ACA | 0.0 | ACA do not provide a stable substratum for coral larvae to metamorphose |
| *Halimeda* | 0.0 | Halimeda is ephemeral so larvae settling die through shading (**Nugues and Szmant, 2006**) |

CCA: crustose coralline algae; AMA: allelopathic macroalgae; ACA: articulate coralline algae

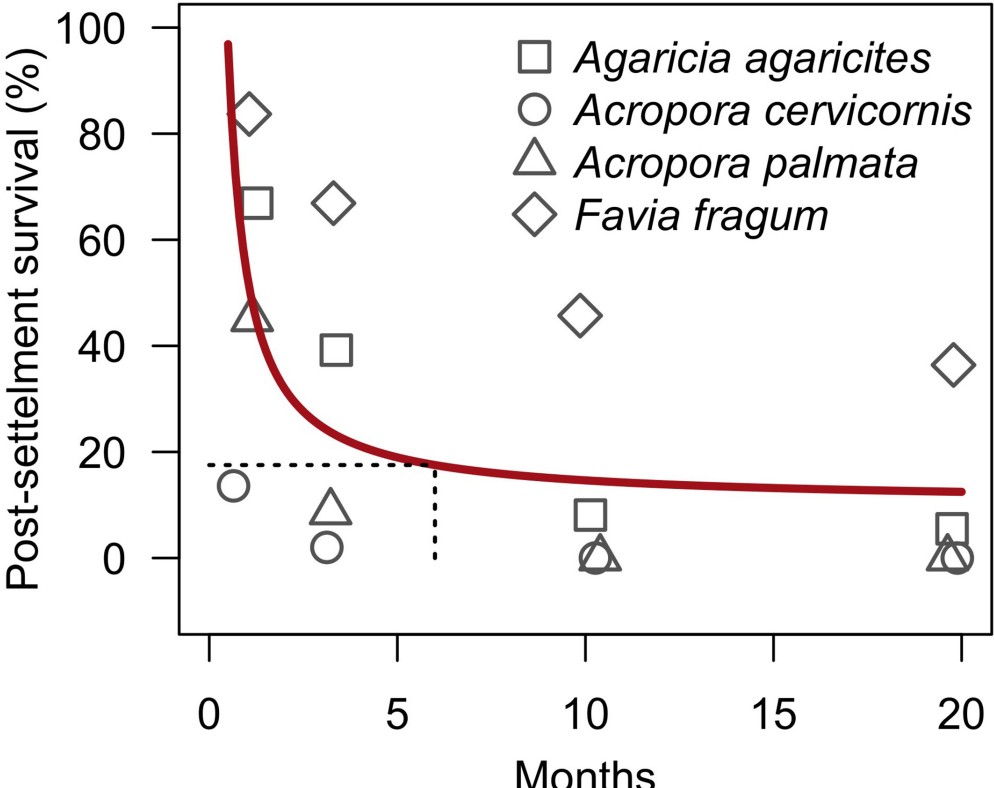

**Appendix 2—figure 11.** Proportion of larvae surviving after settlement as a function of time (from Figure 8b in **Ritson-Williams et al., 2016**). Twelve and 20 replicate tiles (covered with the CCA *Hydrolithon boergesenii*) were used for brooding (*A. agaricites, Favia fragum*) and spawning species (*A. cervicornins, A. palamata*), respectively. Each tile received initially three recruits of a same species. The grey symbols represent the mean percentage of surviving recruits by tile. The red line is the least-squares regression model we established, considering each point as a single observation (**Appendix 2—table 13**). The black dashed lines show the proportion of recruits at six months after settlement ($p_{lxs}$ = 0.18). Points have been offset slightly to improve visibility.

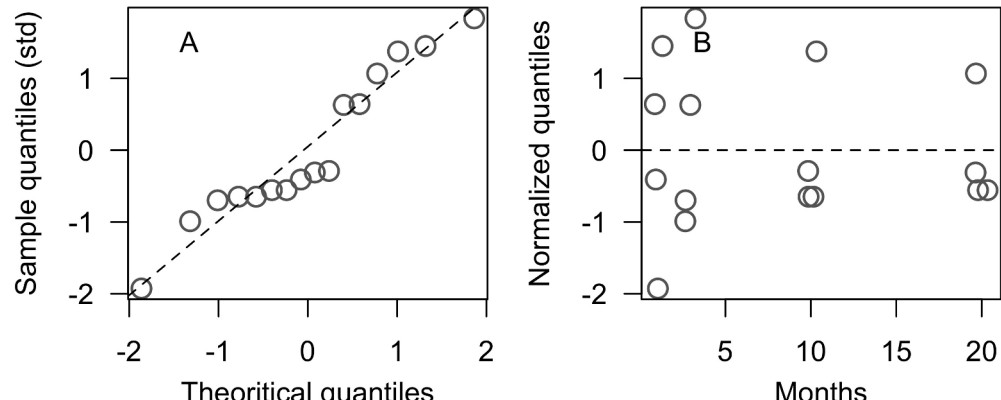

**Appendix 2—figure 12.** Residual diagnostic plots of the coral recruitment model we established based on *Ritson-Williams et al. (2016)*'s results (*Appendix 2—figure 11*): normal quantile plot (**A**) and residual plot (**B**) respectively show that the standardized residuals are normally and homogeneously distributed. Points in B have been offset slightly to improve visibility.

**Appendix 2—table 13.** Parameter values of the least-squares regression model we established based on *Ritson-Williams et al. (2016)*'s results: *y = Intercept + Slope × 1/x* ( *Appendix 2—figure 11*, *Appendix 2—figure 12*).

| Parameters | Estimate | SE | P-value |
|---|---|---|---|
| Intercept | 10.3 | 8.30 | 0.235 |
| Slope | 43.3 | 15.67 | 0.015 |

## Related code

*coralreef2/src/coralreef2/CoralReproduction.java* (*Carturan et al., 2020*).

## 7.2.2. Algae reproduction

### 7.2.2.1. Implementation

Algal functional groups exhibit enormous variation in reproductive mode (i.e. sexual, asexual, vegetative), the onset of spawning, and the amount and types of propagules (e.g. *Bellgrove et al., 2004*). This, combined with a lack of data, prevented us from implementing algae reproduction processes. Instead, we simulated algal reproduction by converting all the ungrazed and available space (barren ground and dead coral skeleton) remaining after all the other processes have been simulated (*Appendix 2—figure 2*)—patches of algae agent with a five centimetres radius are created consecutively and at random locations, for each functional group of algae, until the available space (i.e. ungrazed dead coral and barren ground agents) is filled.

## Related code

*coralreef2/src/coralreef2/ContextCoralReef2.java* (*Carturan et al., 2020*).

## 7.3. Wave and cyclone damage

### 7.3.1. Damage on the coral community

#### 7.3.1.1. Background

Waves and cyclones generate different types of damages: mechanical breakage, dislodgement, increase in sedimentation and turbidity, lower salinity and change in sea level (*Harmelin-Vivien, 1994*). *Madin and Connolly (2006)* defined the colony shape factor (CSF), a dimensionless measure of a colony's mechanical vulnerability to hydrodynamic disturbances (e.g. waves and cyclones). The colony shape factor is expressed as a function of colony planar area and *growth form* and has to be compared with the dislodgement mechanical threshold (DMT), which represents the threshold imposed by a hydrodynamic disturbance. The dislodgement mechanical threshold depends on the

intensity of the disturbance and the exposure of the reef. A colony is dislodged when its colony shape factor surpasses the dislodgement mechanical threshold. But for some species, dislodgement does not mean extirpation: branching species are the most susceptible to dislodgement but their capacity to fragment into small viable pieces enhances their survival rate (*Madin et al., 2014*). Asexual reproduction via fragmentation is even considered to be the main means of reproduction for some species (*Harrison and Wallace, 1990*; *Hughes et al., 1992*).

### 7.3.1.2. Implementation

### a. Dislodgement

We used the colony shape factor to determine the dislodgement susceptibility of colonies, which is calculated for each colony according to *Madin and Connolly (2006)* model:

$$ln(CSF) = \alpha + \beta \times ln(S_{plan})$$

with $S_{plan}$ = the colony planar surface area (m$^2$).

*Madin et al. (2014)* determined empirically $\alpha$ and $\beta$ for five of the nine growth forms implemented in the model. We consequently determined the parameter values for the remaining four growth forms using the best of our judgement. First, the mechanical vulnerability of encrusting and encrusting long upright colonies is independent of their size. The latter are slightly more susceptible to dislodge than the encrusting colonies because they produce vertical features, which create friction. Laminar growth forms are similar to table forms but tend to remain closer to the substratum, which confers slightly more resistance. Finally, columnar colonies form thick vertical features, which are more resistant than laminar colonies but offer more friction than corymbose species (*Appendix 2—table 14*, *Appendix 2—figure 13*).

Values of dislodgement mechanical threshold must be defined for each time step before starting the simulations (and imported from a file). During the cyclone process, the colony shape factor of each colony is calculated and a colony is dislodged the value surpasses the dislodgement mechanical threshold. Dislodgement is simulated by converting all the agents of the colony (i.e. alive or dead coral, algae recovering coral) into barren ground.

**Appendix 2—table 14.** Model parameters determining the colony shape factor of a colony as a function of its size and *growth form* (*Madin et al., 2014*).
Values below the dashed lines were determined arbitrarily (see text, *Appendix 2—figure 13*).

| Morphology | $\beta$ | $\alpha$ | n |
|---|---|---|---|
| Branching | 0.79 | 8.34 | 73 |
| Table_or_plate | 0.39 | 4.47 | 76 |
| Corymbose | 0.16 | 2.28 | 78 |
| Digitate | −0.04 | 1.25 | 68 |
| Massive | −0.23 | −0.94 | 86 |
| Laminar | 0.27 | 3.80 | - |
| Columnar | 0.20 | 3.00 | - |
| Encrusting_long_upright | 0.00 | 1.40 | - |
| Encrusting | 0.00 | 1.00 | - |

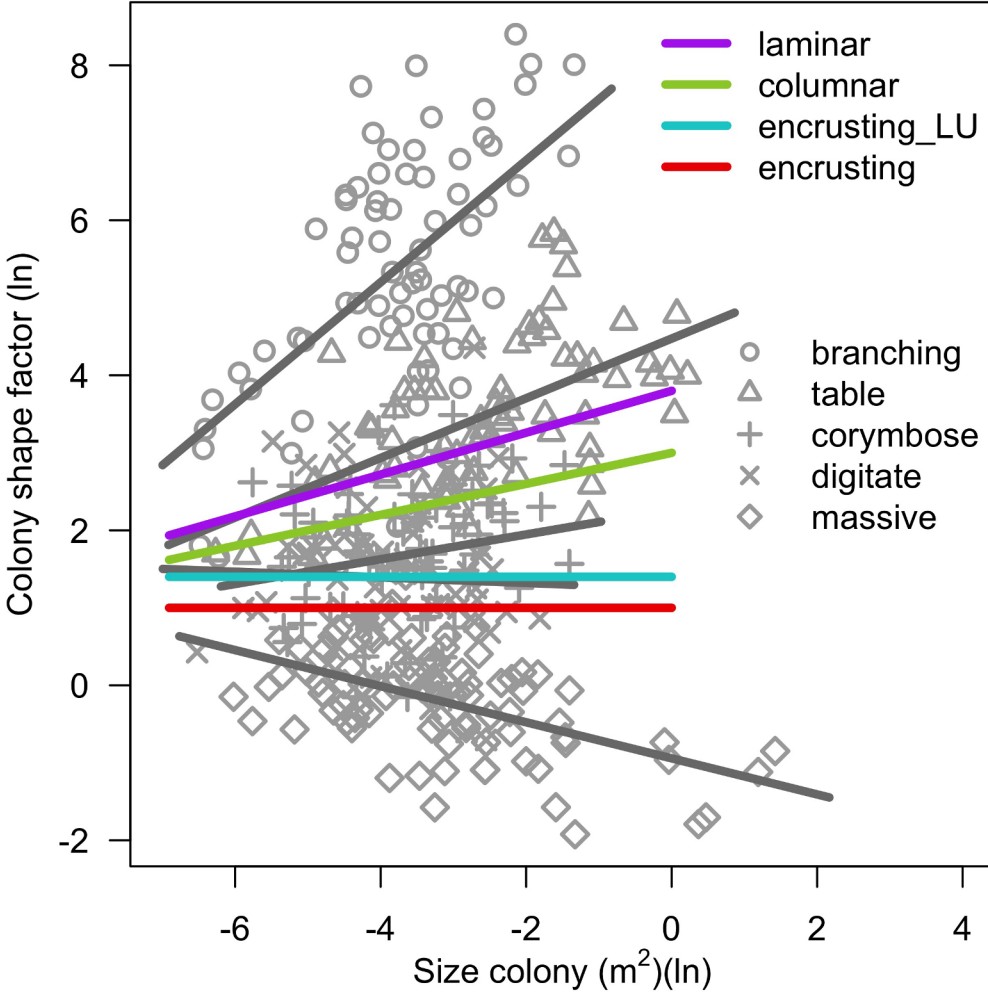

**Appendix 2—figure 13.** Colony shape factor as a function of colony planar area for different colony growth forms. Grey symbols represent a single colony and lines are fitted least-squares regression lines per growth forms (*Madin et al., 2014*). The colored lines represents the model we implemented in the coral ABM for the growth forms not considered in *Madin et al. (2014)* (see text; *Appendix 2—table 14*).

### b. Proportion of surviving branching fragments

We implemented the capacity of a branching colony to survive dislodgement by first defining the size of the fragment potentially surviving as a function of the disturbance intensity. We arbitrarily established a linear model assuming that half of the colony remains if dislodgement mechanical threshold = 120 and none of the colony survive if dislodgement mechanical threshold = 1 (*Appendix 2—figure 14A*). We then determined the survival probability of the fragment using *Highsmith et al. (1980)*'s results: they empirically defined a model predicting the proportion of surviving fragments of *Acropora palmata* (branching) as a function of their length. Their model predicts values superior to one when fragment length >112 cm, so we used a similar but asymptotic model (*Appendix 2—figure 14B*). In case the fragment survives, the colony planar area is reduced accordingly to the proportion obtained by the previous model. The length of a fragmented colony is its diameter, assuming a circular planar area.

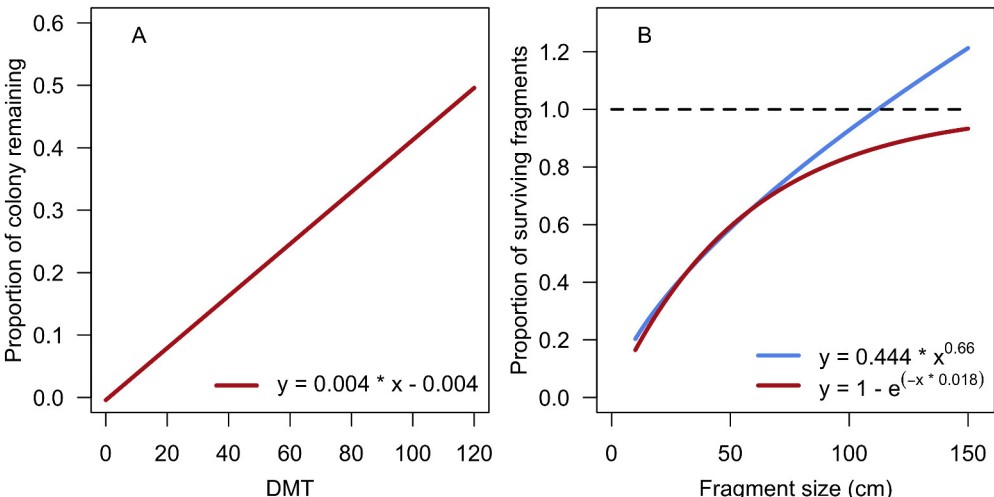

**Appendix 2—figure 14.** Models predicting the proportion of a branching colony remaining and potentially surviving as a function of dislodgement mechanical threshold (DMT, **A**) and the proportion of surviving fragments as a function of their length (**B**). The blue line in panel B corresponds to the empirical relationship established by *Highsmith et al. (1980)*, the red line is the model we defined.

### c. Sedimentation

There is no empirical model determining the amount of sediment cover generated by a cyclone of a given intensity. The amount of sediment at a given time step must be determined before launching a simulation (and imported from a file). Sedimentation is the last process simulated during a time step (*Figure 2*) and consists of adding or removing patches of sand (i.e. converting barren ground agents into sand agents or vice versa) until the desired cover is reached.

## 7.3.2. Damage on the algae community

### 7.3.2.1. Background

Hydrodynamic disturbances such as cyclones can dramatically affect the algal community (e.g., *Blair et al., 1994*). The loss of algae can be caused by three mechanisms: removal, burial or erosion of sediment supporting the community (*Fourqurean and Rutten, 2004*). All functional groups or taxa can be affected (e.g., *Glynn et al., 1964*), but there are notable variations between morphotypes: the loosely attached species with erect or sprawling habits are the most impacted, whereas encrusting species and those with firm or substantial attachment survive better (e.g. *Blair et al., 1994*; *Fourqurean and Rutten, 2004*). There are several examples of macroalgae being dramatically extirpated after a cyclone (>75%; *Blair et al., 1994*; *Fourqurean and Rutten, 2004*; *Lapointe et al., 2006*). Turf and crustose coralline algae are in general less affected (*Diaz-Pulido and McCook, 2008*).

Cyclones can also facilitate the growth of certain functional groups by releasing nutrients and increasing available space, which favour fast growing and opportunistic species (*Diaz-Pulido and McCook, 2008*). For instance, *Trichosolen* spp. (which we classified as turf) are notorious for blooming and pre-empting freed space during the weeks following the disturbance (*Littler and Littler, 1999*; *Pauly et al., 2011*; *Woodley et al., 1981*). Finally, other earlier colonists organisms such as diatoms can overcome available space just after the disturbance (e.g. *Diaz-Pulido et al., 2007*).

### 7.3.2.2. Implementation

We defined cyclone response models for each functional group of algae based on the following assumptions: macroalgae, allopathic macroalgae, *Halimeda* spp. and articulated coralline algae are the most sensitive groups because they sustain the highest friction due to their height. In consequence, they are totally extirpated under intense cyclone intensity (dislodgement mechanical threshold <10). In comparison, turf algae are more resistant and maintain at least 10% of their cover under the highest intensity disturbance. Crustose coralline algae are the least affected because they offer less friction and are mostly impacted by sedimentation and abrasion due to rubble being moved

around. In consequence, at least 20% of the initial cover remains even under the most intense cyclones (*Appendix 2—figure 15*, *Appendix 2—table 15*).

## Related code

*coralreef2/src/ Disturbances/Cyclone.java* (*Carturan et al., 2020*).

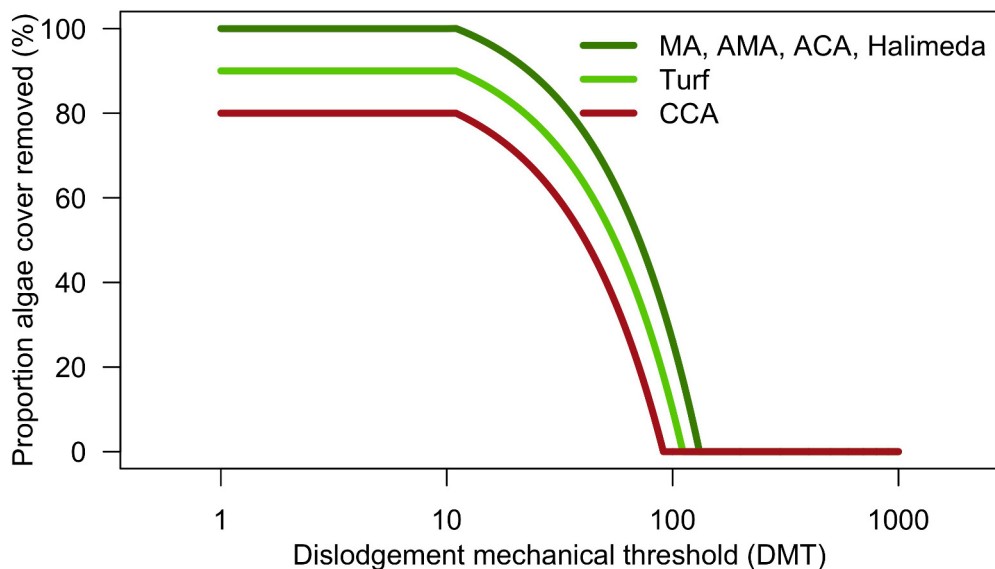

**Appendix 2—figure 15.** Proportions of algae removed from the reef depending on cyclone intensity (DMT) (*Appendix 2—table 15*).

**Appendix 2—table 15.** Models and parameter values defined to determine the proportion of algal cover removed as a function of dislodgement mechanical threshold for each functional group of algae (*Appendix 2—figure 15*).

| Functional groups | Proportion algae cover removed ($P_{ar}$, %) | | |
| --- | --- | --- | --- |
| | DMT $\leq$ 10 | 10 < DMT < $\alpha$ | $\alpha \leq$ DMT |
| MA<br>AMA<br>ACA<br>Halimeda | 100 | $\alpha = 130$<br>$P_{ar} = -0.83 \times DMT + 108.33$ | 0 |
| Turf | 90 | $\alpha = 110$<br>$P_{ar} = -0.90 \times DMT + 99.00$ | 0 |
| CCA | 80 | $\alpha = 90$<br>$P_{ar} = -1.00 \times DMT + 90.00$ | 0 |

DMT: dislodgement mechanical threshold; MA: macroalgae; CCA: crustose coralline algae; AMA: allelopathic macroalgae; ACA: articulate coralline algae

## 7.4. Coral bleaching

### 7.4.1. Background

Coral bleaching is a stress-response corresponding to the expulsion by the host (i.e., the polyp) of its symbiodinium (i.e. 'zooxanthellae') because the latter realized reactive oxygen species (ROS) that are harmful for the host's cells. The disruption of the symbiosis is due to diverse stressful environmental conditions (e.g. water cooling, pollution, reduced salinity), but most commonly because of a combination of temperature and irradiance increase and wind speed decrease (*Brown, 1997*). The bleaching process is complex and coral species show a high interspecific variability in the response: species can bleach and die, bleach and not die, not bleach and die or not bleach and not die (*McClanahan, 2004*). Further, for some species, the stress (i.e. bleaching or dying) is shared by only a subset of the polyps, as opposed to being shared by the whole colony (*Baird and Marshall,*

*2002*). This allows for a quicker recovery via vegetative growth (*Glynn and Fong, 2006*; *Roff et al., 2014*).

Such variation in the bleaching response is partially explained by the high fidelity of most coral species to one main type of symbiodinium (*Hidaka, 2016*) and the pronounced difference in thermo-tolerance between phylotypes of the latter (*Swain et al., 2017*). Interspecific functional differences of the host also contribute to the interspecific variation (*Baird et al., 2009a*). Coral species have developed diverse strategies to maintain a high fitness (*Darling et al., 2012*) and to cope with bleaching events (*Wooldridge, 2014*). Numerous bleaching resistance traits are potentially involved in the response. Evaluating their respective importance and developing predictive species-specific response models is a topical challenge (*Carturan et al., 2018*).

Typical tropical coral reefs are found in shallow oligotrophic waters. In consequence, most coral species depend significantly on symbiotic phototrophic sources of carbon (*Yellowlees et al., 2008*). Thus, bleaching results in starvation and eventually death of the polyp, colony or the population depending on the temporal and spatial scale of the disturbance (*Baker et al., 2008*; *Hughes et al., 2018*). In case of a mild-to-moderate event, polyps usually recover their normal symbiont density a few months after they have bleached (e.g. *Hughes and Grottoli, 2013*; *Jokiel and Coles, 1977*; *Rodrigues and Grottoli, 2007*; *Ward et al., 2000*). It can however take more than eight months to recover normal levels of tissue biomass, lipid, protein and carbohydrate (e.g. *Hughes and Grottoli, 2013*; *Rodrigues and Grottoli, 2007*) as well as normal ratios between heterotrophic and phototro-phic carbon (*Baumann et al., 2014*). These affect coral growth and calcification rate. For instance, *Goreau and Macfarlane (1990)* and *Mendes and Woodley (2002)* measured a reduction of growth rate between 40% and 80% for severely bleached *M. annularis* colonies approximately 6 months after the bleaching event; *Baird and Marshall (2002)* found no growth for two species 9 months after the event (*Appendix 2—table 16*). The effects on coral reproduction can last even longer (*Appendix 2—table 17*). For instance, 9 months after the event, *Ward et al. (2000)* found that the fecundity was reduced by more than 50% for several species and *Baird and Marshall (2002)* observed a half reduction of the proportion of fecund colonies for *Acropora Hyacinthus*. In certain cases, severely bleached colonies did not complete gametogenesis 1 year later (*Szmant and Gass-man, 1990*) or the number of full size gonads was still reduced after 2 years (*Mendes and Woodley, 2002*). Additionally, the reduced gametes concentration and sperm motility can decrease the fertili-zation rate during a mass spawning event, which further compromises coral recruitment (*Omori et al., 2001*).

**Appendix 2—table 16.** Examples of coral growth and calcification rates reduction after a bleaching event.

| Species | Growth rate reduction (%) | Bleaching event | Time post bleaching | Reference |
|---|---|---|---|---|
| *Montastrea annularis* | 60 | Nov. 1987, Jamaica, strong | 6 mo. | *Goreau and Macfarlane, 1990* |
| *Acropora millepora A. hyacinthus* | ~100 | Early 1998, GBR, strong | 9 mo. | *Baird and Marshall, 2002* |
| *Montastraea annularis* | 40 to 80 | Sept. 1995, Jamaica, ? | 5 to 7 mo. | *Mendes and Woodley, 2002* |
| *Porites compressa* | 89* 100* 67* | Experiment | 1.5 mo. 4 mo. 8 mo. | *Rodrigues and Grottoli, 2006* |
| *Montipora capitata* | 80* 80* 27* | Experiment | 1.5 mo. 4 mo. 8 mo. | *Rodrigues and Grottoli, 2006* |

*Calcification rate.

**Appendix 2—table 17.** Examples of different effects of bleaching events on coral reproduction.

| Species | Effect on reproduction | | Bleaching event | Time post bleaching | Reference |
|---|---|---|---|---|---|
| *Acropora millepora* *A. hyacinthus* | % reduction of fecund colonies | - 54 - 7 | Early 1998, GBR, strong | 9 months | ***Baird and Marshall, 2002*** |
| *Montastraea annularis* | No gametogenesis completed for severely bleached colonies | | Sept. 1995, Jamaica | 1 year | ***Mendes and Woodley, 2002*** |
| *Acropora aspera* *A. humilis* *A. millepora* *A. nobilis* *A. palifera* *A. pulchra* *A. valida* *Montipora digitata* *M. spp.* *Symphyllia spp.* | % reduction of polyps fecundity / % reduction of reproductive polyps | 63/59 100/ 100 −24/8 33/33 66/71 34/14 55/59 77/85 100/ 100 100/ 100 | March 1998, GBR, strong | 6 weeks | ***Ward et al., 2000*** |
| *Acropora aspera* *A. millepora* *A. nobilis* *A. pulchra* *A. valida* | % reduction of polyps fecundity / % reduction of reproductive polyps | 53/50 51/66 24/54 11 / −31 92/ 100 | March 1998, GBR, strong | 9 months | ***Ward et al., 2000*** |
| *A. nasuta* | % reduction of fertilization rate | 55 | 1998, Japan, strong | 1 year | ***Omori et al., 2001*** |
| *M. annularis* | No gametogenesis completed | | Summer 1987, Caribbean | 1 year | ***Szmant and Gassman, 1990*** |

GBR: Great Barrier Reef.

## 7.4.2. Implementation

Implementing bleaching response consisted of three successive steps (i) defining a species-specific index of bleaching susceptibility (*bleaching_probability*) using the bleaching response index (***Swain et al., 2016b***) and bleaching resistance coral functional traits; (ii) establishing species-specific logistic bleaching probability models using the *bleaching_probability* as a parameter and degree heating week—a measure of the intensity of the bleaching event (***Kayanne, 2017***)—as predictor; (iii) defining a logistic mortality probability model, which also uses degree heating week as independent variable and determines the risk of a bleached colony to die. We only provide here the final sub-models obtained because the procedures produced a lot of statistical results and the three steps are related to one another. We refer the reader to appendix 4 for more details.

### 7.4.2.1. The species-specific index of bleaching susceptibility

The final averaged beta regression model we defined to predict *bleaching_probability* for each of the 798 coral species is:

$$
\begin{aligned}
cloglog(E(bleaching\_probability_i)) = {} & -1.242 + 0.187 \times ln(colony\ maximum\ diameter)_i \\
& -0.123 \times ln(corallitearea)_i \\
& +0.024 \times ln(growthrate)_i \\
& -0.668 \times ln(mRSC)_i \\
& +0.024 \times ln(colony\ maximum\ diameter)_i : ln(corallitearea)_i \\
& +0.063 \times ln(colony\ maximum\ diameter)_i : ln(growthrate)_i \\
& +0.001 \times ln(colony\ maximum\ diameter)_i^2 \\
& -0.001 \times ln(corallitearea)_i^2 \\
& -0.034 \times ln(growthrate)_i^2 \\
& +0.135 \times ln(mRSC)_i^2
\end{aligned}
$$

with $bleaching\_probability_i \sim B(\mu_i, \phi)$ (i.e. beta distribution with mean $\mu_i$ and dispersion $\phi$), E($bleaching\_probability$) = $\mu$, VAR($bleaching\_probability$) = $\mu(1 - \mu)/(1 + \phi)$; $\phi$ = 4.267; $mRSC$ = the trait microscopic reduced scattering coefficient ($reduced\_scattering\_coefficient$); $growth\ rate$ = the diametral lateral growth rate (in mm.yr$^{-1}$) and not the $growth\_rate$ implemented in the model (see §7.5.1). See Appendix 4: §1 for more details.

### 7.4.2.2. Species-specific bleaching probability models

The probability of a coral species $i$ to bleach ($P_{Bi}$) is expressed as a function of the thermal disturbance intensity ($DHW$), its bleaching susceptibility ($bleaching\_probablity$, $IP_{Bi}$) relatively to the bleaching susceptibility of the other 797 species:

$$
logit(P_{Bi}) = \alpha + \beta_1 \times DHW + \gamma_i
$$

$$
\gamma_i = \frac{IP_{Bi} - mean(IP_B)}{max(IP_B) - min(IP_B)} \times \frac{DHW}{\varphi}
$$

with logit() the logit transformation, $\alpha$ = -2.78, $\beta_1$ = 0.29, $mean(IP_B)$ = 0.27, $min(IP_B)$ = 0.06, and $max(IP_B)$ = 0.60 (the mean, maximum and minimum $bleaching\_probablity$ among the 798 coral species, respectively) and $\varphi = 2$ (value calibrated; see Appendix 3). See Appendix 4: §2 for more details.

### 7.4.2.3. Bleaching-induced mortality probability model

The probability of a bleached coral colony to die ($P_{BD}$) depends on the intensity of the thermal disturbance ($DHW$):

$$
P_{BD} = \frac{1}{1 + e^{-0.4 \times (DHW - 11.6)}}
$$

The probability of mortality is multiplied by two in cases where the coral colony is already bleached when the bleaching event occurs. See Appendix 4: §3 for more details.

### 7.4.2.4. Effect of bleaching on surviving colonies

Surviving bleached colonies have their growth rate divided by two during 6 months after the bleaching event and cannot reproduce for 1 year (this is monitored with $timeRecoveryBleaching$).

## 7.5. Growth and spatial competition

### 7.5.1. Growth

We simulated radial vegetative growth: agents on the edge of a colony or a patch (for algae) attempt to convert their neighbouring agents (outside of their colony or patch) within a certain radius $r_{random}$. The latter is generated each attempt by first sampling a decimal number from the range [0; $r_{max}$[and then rounding this number to the nearest inferior integer. We defined $r_{max}$ (i.e. $growth\_rate$ in *Appendix 2—table 1*) for each taxon so that on averaged $r_{random}$ equals a real growth rate (see Traits_and_imputation/Datasets/growthRate_randomRadiusConversin.xls). This procedure allows to simulate continuous growth rate in a space having a discrete dimension (here with one cm$^2$ for minimum value). We verified the accuracy of the method in the hierarchically structure validation (Appendix 5: §2). A colony cannot grow larger than its species-specific maximum planar surface area = $\pi \times$ (*maximum colony diameter*/2)$^2$.

Related code

*coralreef2/src/ Disturbances/Bleaching.java* (*Carturan et al., 2020*).

## 7.5.2. Competition between corals

### 7.5.2.1. Background

Coral species have different strategies when directly competing for space with one another: extension of mesentrial filaments, specialized sweep tentacles, extension of long polyps, production of mucus, production of cytotoxins and overgrowing or overtopping, depending on the size and shape of the colonies (*Lang and Chornesky, 1990*). Coral interactions are complex as the use of these diverse mechanisms differ greatly between species (*Lang and Chornesky, 1990*). Additionally, species respond differently to environmental factors and consequently the nature of their interactions can change depending on the type of habitat (*Connell et al., 2004*) or the geographic regions (e.g. Caribbean *versus*. Red Sea *versus*. GBR; *Logan, 1984*) or even between laboratory experiments and field observations (*Logan, 1984*). It is consequently challenging to establish a constant linear dominance ranking between species (*Lang and Chornesky, 1990*). Recently, *Precoda et al. (2017)* established species-pair probabilities of interaction outcomes (i.e. winning, standoff and loosing) for 774 species based on a review of 2322 interactions. Fitting these outcomes of interaction to mixed-effect models, they found that (*i*) nearly 80% of the species triple interactions are transitive (purely and non-hierarchical combined), the rest being intransitive. They also found that *corallite area* explained most of the species' competitive ability, followed by geographical range and *growth form*. However, random effect was greater than the trait effect, which can be due to pair-species idiosyncrasy, environmental heterogeneity or omission of important traits.

### 7.5.2.2. Implementation

#### a. Direct encounter

We used the output of *Precoda et al. (2017)* simulations to determine the outcome of an encounter between two coral species. After correcting nomenclature, their dataset contained 741 coral species. In case the species-pair is not present in this list, we used the trait *aggressiveness*, which represents the species-specific competitive ability of coral species when in contact with one another. We constructed *aggressiveness* by combining six ranking lists established empirically (*Abelson and Loya, 1999*; *Connell et al., 2004*; *Dai, 1990*; *Lang, 1973*; *Logan, 1984*; *Sheppard, 1979*) and using an iterative partial rank aggregation pivoting algorithm (IPRAPA; *Swain et al., 2017*; *aggressiveness.R*). The procedure allowed us to attribute an aggressiveness value to 116 species. We predicted the value for the rest of the 798 species with the random-forest trait data imputation (Appendix 1).

The growth rate of a colony overgrowing another colony is reduced by being multiplied by a coefficient (*growth rate reduction interaction*) to account for the energy invested in the process. We calibrated the value of the coefficient with empirical datasets (Appendix 3: §2 and §8).

#### b. Overtopping colonies

Plating and branching colonies can overtop smaller other colonies, a strategy known as the 'escape in height strategy' (*Meesters et al., 1996*; *Swierts and Vermeij, 2016*). The growth rate of an overtopping colony is unchanged, contrary to a direct encounter. The process requires the planar cover of the overtopping colony to surpass the planar cover of the overtopped colony by the ratio *ratio overtop colony* = 2; value calibrated using empirical datasets (Appendix 3).

To overtop encrusting and encrusting long upright colonies, branching and plating colonies have to surpass their height by 5 cm. The height of a non-encrusting colony is its radius, assuming a semi-spherical growth. The height of an encrusting colony = 2 cm (*Appendix 2—table 18*).

**Appendix 2—table 18.** Parameters and rules involved in the overtopping process of a branching or plating colony when growing over a coral colony of a patch of algae.

| Parameters | Value | Rule for overtopping |
|---|---|---|
| *ratioAreaBranchingPlating_ OvertopColonies* | 2 (value calibrated; Appendix 3) | $\frac{So}{Su}>2$ |
| *height_BigAlgae* (i.e. macroalgae, ACA, AMA, Halimeda) | 30 cm | $\sqrt{\frac{So}{\pi}}>30+5$ |

*Continued on next page*

*Appendix 2—table 18 continued*

| Parameters | Value | Rule for overtopping |
|---|---|---|
| *height_Turf* | 10 cm | $\sqrt{\frac{So}{\pi}} > 10 + 5$ |
| *height_CCA_EncrustingCoral* | 2 cm | $\sqrt{\frac{So}{\pi}} > 2 + 5$ |

*So*: planar surface area of the overtopping branching or plating colony; *Su*: planar surface area of the overtopped colony; CCA: crustose coralline algae; ACA: articulated coralline algae; AMA: allopathic macroalgae algae.

**c. Special cases**

An agent (coral or algae) can grow over a dead coral colony without constraint. However, a coral colony is able to grow over a dead branching or plating coral colony only if the ratio of the planar surface areas of the two colonies is <*ratio overtop colony*. Similarly, an encrusting (and encrusting long up right) coral agent can overgrow a dead branching or plating colony only if the difference of heights is <5 cm.

To simulate 're-sheeting', or 'phoenix effect', which is observed in species of different growth forms and enhances recovery rate (*Glynn and Fong, 2006*; *Jordan-Dahlgren, 1992*; *Roff et al., 2014*), a coral colony can grow over dead coral agents of a same species one third faster than its normal growth rate (as polyps do not have to grow skeletons).

## 7.5.3. Competition between corals and algae

### 7.5.3.1. Background

Interactions between corals and algae are highly variable in mechanisms: overgrowth, shading, abrasion, chemical, space pre-emption, recruitment barrier, epithelial sloughing (*McCook et al., 2001*). In addition, certain algae can indirectly affect corals by contaminating their tissues with pathogenic bacteria (*Barott et al., 2012a*; *Smith et al., 2006*). The outcome of the competition depends on the life-history of the species competing and environmental factors such as herbivory, nutrient input and light level (*McCook et al., 2001*). Inhibition is usually reciprocal for both algae and corals (*Jompa and McCook, 2002*; *McCook et al., 2001*; *Titlyanov et al., 2007*).

The outcomes of these interactions are consequently difficult to generalize, even within functional groups (*Jompa and McCook, 2003a*) and are best considered at the level of species (*Jompa and McCook, 2003b*; *Jompa and McCook, 2002*; *Titlyanov et al., 2007*). But doing so is challenging because of the difficulty to identify algae at the species level, their high plasticity, and the multispecies composition of algae assemblages (*McCook et al., 2001*).

More specifically, crustose coralline algae are in general inferior (*Barott et al., 2012b*) or equal (*Vermeij et al., 2010*) competitors against corals, regardless of the level of anthropogenic stresses (e.g. fishing or nutrient input) and little or no apparent stress is observed for the coral (*Barott et al., 2012a*; *Barott et al., 2009*). Observations for filamentous turf algae are varied. Turf has been described as a mixture of a large number of species that are poor competitors and have minor effect on corals (*McCook et al., 2001*; *McCook, 2001*), with however exceptions (e.g. *Jompa and McCook, 2003a*; *Jompa and McCook, 2003b*). But if coral can overgrow turf assemblages, the input of nutrients can reverse the competitive dominance (*Barott et al., 2012b*; *Vermeij et al., 2010*, but see *McCook, 2001*).

Similarly varied observations were made for macroalgae: a variety of primarily upright macroalgae had minor effect on coral except for one species (*Tanner, 1995*), same for sargassum beds that have little or no competitive effects on understory corals (*McCook, 1999*). Contrastingly, the creeping foliose brown alga *Lobophora variegate*, which is a common species in the GBR (*Diaz-Pulido and McCook, 2008*), is a markedly superior competitor against *Porites cylindrical* (*Jompa and McCook, 2002*). Finally, different macroalgae and turf assemblages can cause hypoxia to coral tissue in contact (*Barott et al., 2012a*; *Barott et al., 2009*) and can on average (both functional groups confounded) win the majority of their interactions with corals (*Barott et al., 2009*).

Recently, *Brown et al. (2017)* measured and categorized the interactions between different groups of corals and algae on the field. They found that (i) filamentous algae and cyanobacteria (i.e.

turf) always won; (ii) crustose coralline algae and *Halimeda* spp. algae lost on average approximately 25% and 45% of their interactions, respectively; (iii) macroalgae won 80% of their interactions.

### 7.5.3.2. Implementation

#### a. Direct encounter

We defined probabilities of winning interactions (i.e. overgrowing) between coral and algae using *Brown et al. (2017)*'s results as well as additional unpublished results (K. T. Brown, personal communication, October 2017; *Appendix 2—table 19*). The growth rate of an algae overgrowing a coral colony is reduced by growth rate *reduction interaction* (see §7.5.2.2.a). (For comparison, *De Ruyter van Steveninck et al. (1988)* found that the growth rate of *Lobophora variegata*'s blades was reduced by approximately 35% when in contact with coral colonies).

We did not implement the effect of colony size on the competitive ability against algae because of conflicting empirical results: *Brown et al. (2017)* found that small colonies are less affected; as opposed to *Ferrari et al. (2012)* who found the opposite and *Bonaldo and Hay (2014)*, who did not observe a relationship.

**Appendix 2—table 19.** Functional groups of algae, their competitive outcome probability when competing with corals (based on K. T. Brown, personal communication, October 2017) and radial growth rates.

| Functional group | Description | Probability of winning against corals | Radial growth rate (mm.yr$^{-1}$) |
|---|---|---|---|
| Macroalgae (MA) | Forms 30 cm high canopy (e.g., *Sargassum*, *Hydroclathrus*) | 0.70 | 150 |
| *Halimeda* spp. | Calcified and non-aggressive macroalgae; forms 30 cm high canopy | 0.15 | 150 |
| Allelopathic macroalgae (AMA) | Produces defensive chemicals; forms 30 cm high canopy (e.g. *Chlorodesmis*) | 0.80 | 150 |
| Turf | Filamentous algae and cycnobacteria; forms 10 cm high canopy | 1.00 | 250 |
| Articulate coralline algae (ACA) | Forms 30 cm high canopy (e.g. *Calliarthron tuberculosum*) | 0.40 | 21 |
| Crustose coralline algae (CCA) | Encrusting, 2 cm high (e.g., *Porolithon onkodes*) | 0.10 | 12 |

#### b. Overtopping colonies

Branching and plating corals can overtop algae when their colony is high enough (*Barott et al., 2012b*; *Tanner, 1995*). The height of the branching or plating colony must surpass the height of the algae by 5 cm (*Appendix 2—table 19*). In case algae are on a dead coral colony, the total height that a branching or plating colony has to overtop is the sum of the height of the dead colony and of the algae plus 5 cm.

## 7.5.4. Competition between algae

### 7.5.4.1. Background

Competition between algae is complex and the outcomes depend on (*i*) the nature of the competition: indirect (e.g. depletion of resources, pre-emption of space), direct (i.e. interference interactions), (*ii*) the traits and strategies involved (e.g. growth rate, size and shape, allelopathy) and (*iii*) the implication of other factors such as herbivory, nutrient concentration and disturbances (*Olson and Lubchenco, 1990*). Predicting the outcome between two algae species is challenging because of the difficulty to identify which mechanisms are involved in a specific interaction and the competition can be asymmetric (i.e. species use different mechanisms against one another; *Olson and Lubchenco, 1990*).

Among the algal functional groups defined in the model, crustose coralline algae is the only algae that can be generally considered as an inferior competitor against different macroalgae (e.g. *McClanahan, 1997*) and turf (e.g. *Borowitzka et al., 1978*; *Kendrick, 1991*).

### 7.5.4.2. Implementation

Except for crustose coralline algae, we considered algal functional groups as equal competitors and can consequently not overgrow each other. We defined *prob cover crustose coralline algae* as the probability of crustose coralline algae to lose their interactions against other algae. We attempted to calibrate the parameter using the empirical dataset, but no value could be defined during the procedure (Appendix 3). We consequently chose the smallest value considered (i.e. 0.1) in order to compensate for its lack of competitiveness in the model.

## Related code

*coralreef2/src/coraReef2/agent/ Agent.java* (*Carturan et al., 2020*).

## Appendix 3

### Model calibration

The objective of this calibration is to define general parameter values, so the species simulated interact, recruit and respond to disturbances in an ecologically relevant manner. We calibrated 12 model parameters using empirical data describing the biodiversity and environmental context of three Caribbean reefs over time. For each site, we used the time series of the percentage cover of coral species and algae functional groups to define the initial size of each population in the simulations and to measure the fit between the percentage cover of the simulated and real populations. We used empirical time series of sand cover, degree heating weeks (DHW) and herbivorous fish and urchin densities to determine at each time step the amount of sand to input or output in the virtual reef, the intensity of the thermal stress and the grazing pressure, respectively. Finally, we considered the occurrence and intensity of cyclones that affected the reefs during the period considered. We present here the empirical data used, their implementation and the design and results of the calibration. All the related code for production of the figures is in *Manuscript/Rscripts/ Appendix S3 - Model calibration.R* (*Carturan et al., 2020*).

### 1. Study sites and related data

We used data collected in three sites located in Martinique in the Caribbean: Fond Boucher (14° 39' 21.07' N, 61° 09' 38.98' W), Pointe Borgnesse (14° 26' 48.74' N, 60° 54' 12.72' W), and Ilet à Rats (14° 40' 58.04' N, 60° 54' 1.18' W) and Ilet à Rats (14°40'58.04"N; 60°54'1.18"W) between November 2001 and July 2011. For each site, we only considered a time period where all data were available.

The biodiversity data were collected by the *Observatoir du Milieu Marin Martiniquais* for the program *Initiative Française pour les REcifs COralliens*. Surveys were conducted biannually and describe the benthic, macroinvertebrates and fish communities at the species or genera levels (no data were available for 2010, *Appendix 3—figure 1*, *Appendix 3—figure 2*). The benthic community composition was assessed using a line-intercept transect method (LIT). One permanent 60-m long transect was positioned along the reef crest at each site. The benthic groups recorded are: live coral, dead coral (bare dead coral substrate and fragments), sessile invertebrates (soft coral, sponges, zoanthids), algae (macroalgae, turf, cyanophycae, encrusting and erected calcareous algae) and sand. They assessed the fish community structure at the same transect but over a $4 \times 50$ and $2 \times 50$ m belt transect for mobile and territorial species respectively. Urchin populations density were measured along three $1 \times 50$ m belt transects (*Appendix 3—figure 2*).

We downloaded values of degree-heating weeks for the corresponding location (collected twice every week at a 50 km resolution) from the US National Oceanic and Atmospheric Administration data server ERDDAP (Environmental Research Division's Data Access Program; coastwatch.pfeg. noaa.gov/erddap) (*Appendix 3—figure 3*). We identified cyclones tracks using the National Oceanic and Atmospheric Administration Historical Hurricane Tracks website (coast.noaa.gov/hurricanes). Hurricane Dean affected the reefs in August 2007 and its intensity changed from category one to two while passing over Martinique.

### Related code

*Empirical_datasets_Martinique_calibration/Rscripts/*
*Benthos_data_2001_2010_backup.R, cyclones_bleaching.R, Martinique_algae.R,*
*Martinique_corals.R, Martinique_dead_substratum.R, Martinique_herbivores.R,*
*Martinique_Sites.R, species_selection_model.R* (*Carturan et al., 2020*).

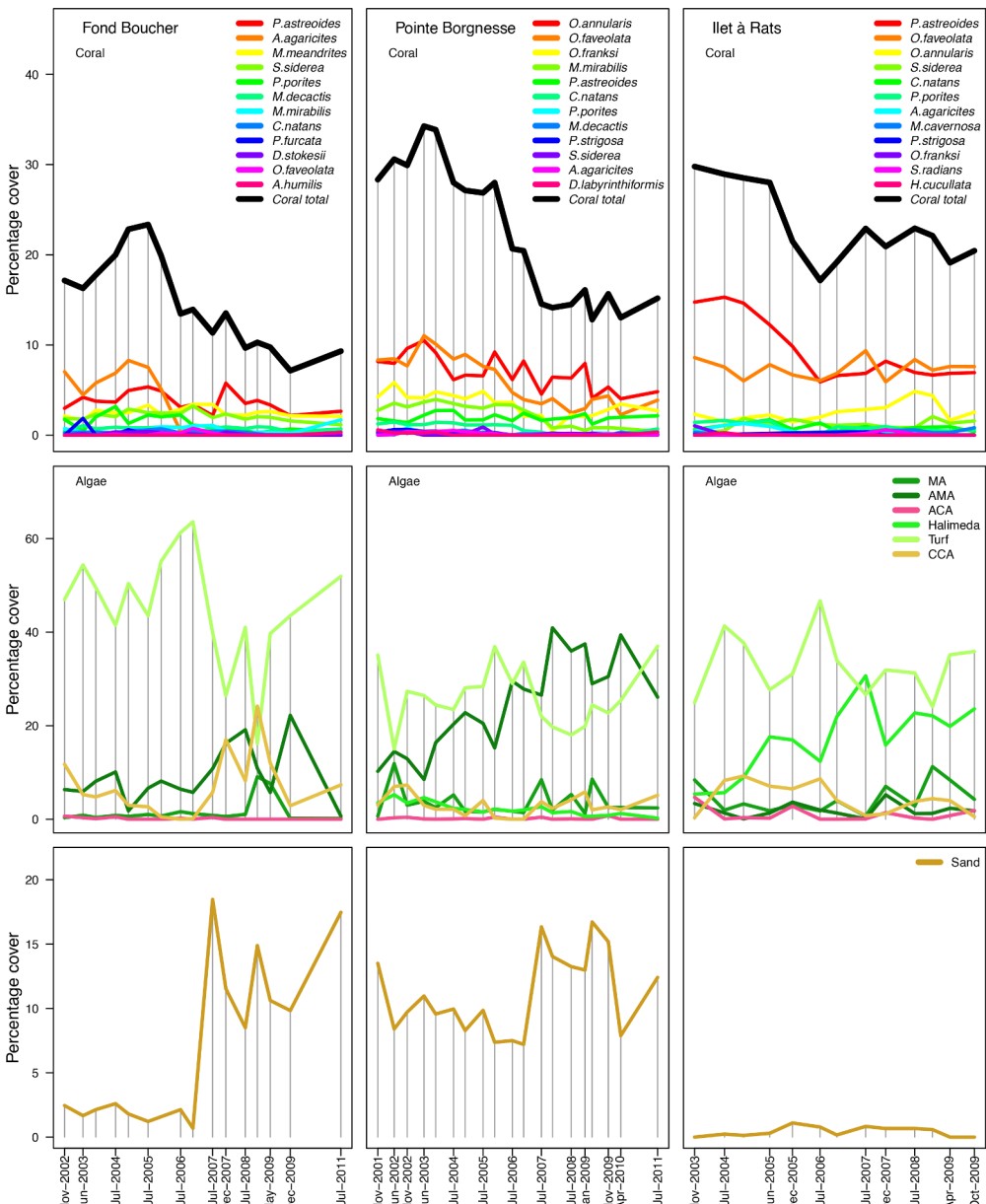

**Appendix 3—figure 1.** Composition of the benthic communities in the three Caribbean sites. Vertical grey bars indicate the dates at which data were collected. The data were collected by the *Observatoir du Milieu Marin Martiniquais* for the program *Initiative Française pour les REcifs COralliens*.

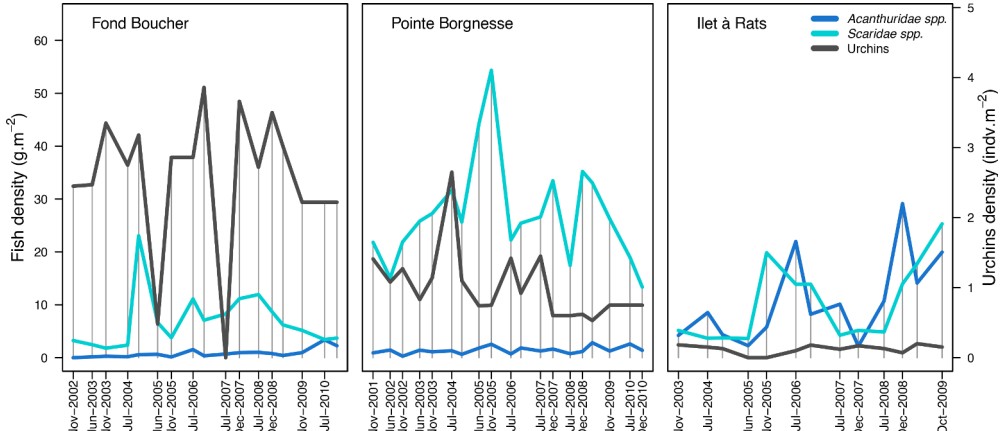

**Appendix 3—figure 2.** Population densities of herbivore fish and sea urchins (*Diadema antillarum* and *Echinometra viridis* combined) measured in the three Caribbean sites. Vertical grey bars indicate the dates at which data were collected. The data were collected by the *Observatoir du Milieu Marin Martiniquais* for the program *Initiative Française pour les REcifs COralliens*.

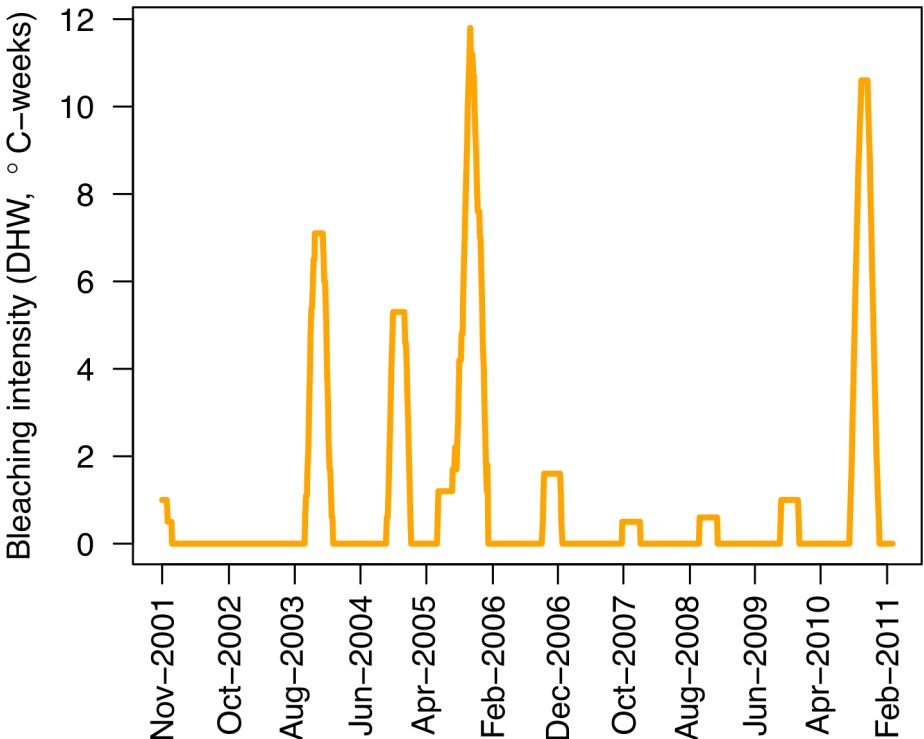

**Appendix 3—figure 3.** Evolution of the thermal stress in degree heating weeks affecting the three Caribbean sites (US National Oceanic and Atmospheric Administration's data server ERDDAP).

## 2. Data implementation

### 2.1. Benthic cover

We manipulated the empirical data so it could be used in our 6-month time step model. Biological data were collected at the end of the dry and wet seasons, which respectively span from December to May and from June to November. Measures of benthic cover were used in the calibration by being compared to the simulated cover values. We adjusted temporally the empirical cover data when necessary, so each measure would fall exactly 6 months after the previous one.

Sand cover was used to determine the target percentage at each time step (*Appendix 3—figure 4*). Small patches of barren ground agents were converted to sand or vice versa in case sand needed to be added or removed.

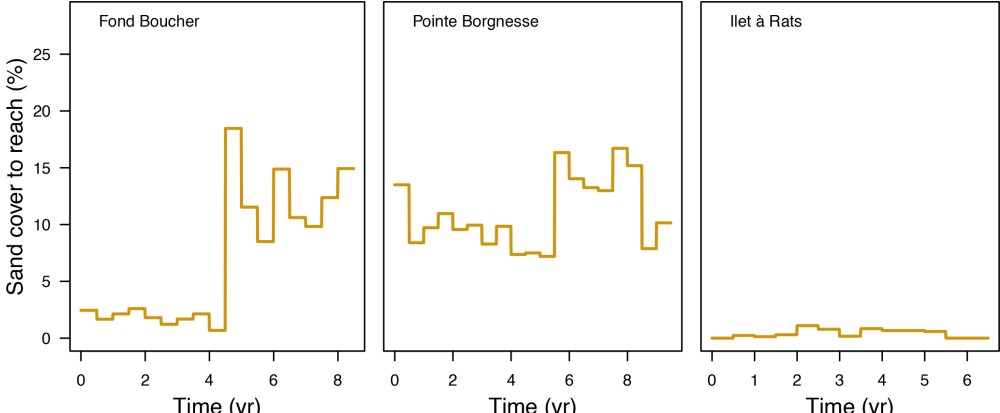

**Appendix 3—figure 4.** Level of sand cover to reach at each time step for three Caribbean sites.

## 2.2. Thermal stress and hydrodynamic regimes

The value of degree-heating weeks to impose during a time step is the maximum DHW value recorded during the corresponding period (*Appendix 3—figure 5*).

The intensity of hydrodynamic regimes is defined by the dislodgement mechanical threshold (DMT), a dimensionless measure of the mechanical threshold imposed by waves and cyclones (Appendix 2: §7.3.1; *Madin and Connolly, 2006*). A lower value represents a more intense disturbance. For instance, *Madin and Connolly (2006)* estimated that cyclone Rona (category three) imposed a DMT approximately equal to 18 at the crest and 88 at the back of Lizard Island in February 1999. In comparison, they also estimated the DMT imposed by a 'moderate cyclone' at the same locations: around 38 and 162 at the crest and back of the reef, respectively. We had no information about the intensity of the hydrodynamic regimes other than the category of the hurricane Dean. Considering that the three sites are not directly in contact with the Atlantic Oceanic currents (Fond Boucher and Pointe Borgnesse are in the Caribbean side and Ilet à Rats is protected in a bight), we first arbitrarily attributed DMT values to the different disturbance regimes: 140 for waves, 120 for tropical storms (TS), 100 for hurricanes of category 1 (H1), 80 for H2, 60 for H3, 40 for H4 and 10 for H5. The sites having potentially different exposure to both wave and cyclone, we defined two additional hydrodynamic disturbance regimes by (1) subtracting 10 and (2) adding 10 to the DMT values at each time step (respectively $cyclone_{model1}$ and $cyclone_{model3}$ in *Appendix 3—figure 5*).

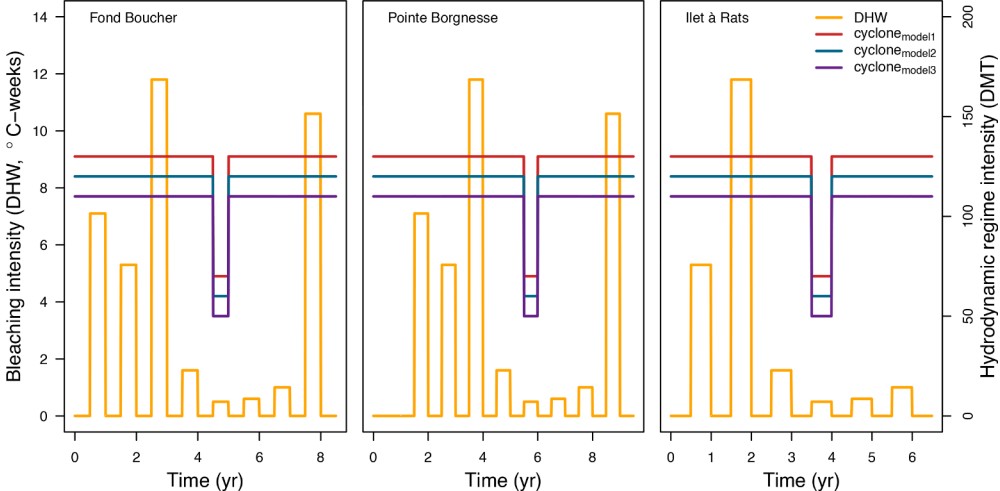

**Appendix 3—figure 5.** Values of thermal stress (in degree heating weeks) and hydrodynamic

thresholds (dislodgment mechanical threshold) imposed in the model at each time step for the three Caribbean sites.

## 2.3. Grazing pressure estimation

We defined four models predicting the proportion of the reef maintained in a grazed state as a function of herbivore densities (*Appendix 3—figure 6*). Among the different species present in the datasets, we only considered *Acanthuridae* spp. and *Scaridae* spp. and sea urchins, as they are considered the most important herbivores in reefs (*Steneck, 1988*). We determined the fish grazing pressure using *Williams and Polunin (2001)*'s empirical data. They conducted field surveys in 19 Caribbean reefs and analysed the relationship between the percentage cover cropped (i.e. covered by either turf, crustose coralline algae or bare substratum but not by macroalgae) and the density of *Acanthuridae* spp. and *Scaridae* spp. present. (Other grazers such as the sea urchin *Diadema* spp. were not present in high enough abundance to influence their results.) We combined abundances of the two genera and established a least-square linear regression between the total fish biomass and the percentage cover grazed (we considered 'cropped' as equivalent to 'grazed' in our model). The fish densities measured in *Williams and Polunin (2001)*'s study do not reach the maximum values observed in the three Caribbean sites (*Appendix 3—figure 2*). In order to avoid predicting percentage values above 100, we defined two asymptotic models approximating the linear regression but plateauing respectively at 90% and 70% for higher fish densities (*Appendix 3—figure 6A*):

$$S_{G\,model1} = \frac{90 \times D_F^2}{D_F^2 + 135}$$

$$S_{G\,model2} = \frac{70 \times D_F^2}{D_F^2 + 90}$$

where $S_G$ = the surface of the reef grazed and $D_F$ = the density of herbivore fish (g.m$^{-2}$). We excluded the possibility for the reef to be grazed at 100% because such a scenario is not realistic even at high fish densities (*Paddack et al., 2006*; *Williams et al., 2001*).

Similarly, we modeled urchin grazing using data from *Sammarco (1980)*, who experimentally manipulated the density of the sea urchin *Diadema antillarum* (excluding other types of grazers) and analysed the relationship with algal cover (the author did not specify which functional groups the term includes) in a reef at Discovery Bay (Jamaica). We defined a least-squares linear regression model to predict the 'percentage of reef grazed' (obtained by subtracting the percentage cover of algae from 100) with the density of *D. antillarum* (*Appendix 3—figure 6*). We log-transformed urchin density to meet the assumptions of normality and equal variance of the linear model (*Appendix 3—figure 7*). Both urchin species *D. antillarum* and *Echinometra viridis* were present in the three Caribbean reefs. We determined the urchin grazing pressure by pooling their respective abundances, assuming a similar function of the two species. This could potentially be a limitation as the two species have little niche overlap (*McClanahan, 1999*).

The total surface grazed is obtained by adding up the surface respectively grazed by fishes and urchins, which provides the grazing regimes grazing$_{model1}$ and grazing$_{model2}$ in *Appendix 3—figure 8*. We defined regimes grazing$_{model3}$ and grazing$_{model4}$ by subtracting 10% and 20% to grazing$_{model2}$ respectively.

### Related code

*Empirical_datasets_Martinique_calibration/Rscripts/Prep_for_model.R, Prep_for_model_Output_comparison.R* (*Carturan et al., 2020*).

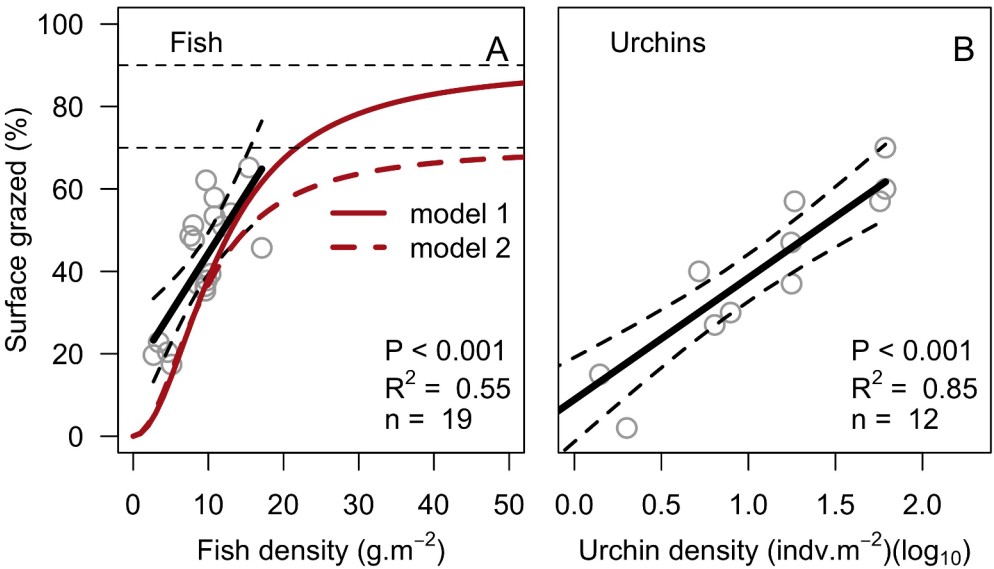

**Appendix 3—figure 6.** Models defining the percentage of reef surface grazed as a function of herbivore density. Fish densities were measured in 19 Caribbean reefs (**A**, *Williams and Polunin, 2001*). Also shown are the surface grazed averaged by site (grey circles), the significant least-squares regression line (black solid line; equation: Y = 15.5 + 2.9X) and the two asymptotic models we defined for modeling fish grazing pressure at higher fish densities (i.e. densities not observed in the study). Sea urchin densities were measured in a cage exclusion field experiment (**B**, *Sammarco, 1980*). Grey circles are surface grazed averaged by site, black lines are least-squares regressions (black solid line; equation: Y = 8.9 + 29.5X). Dashed lines represent the 95% confidence bands.

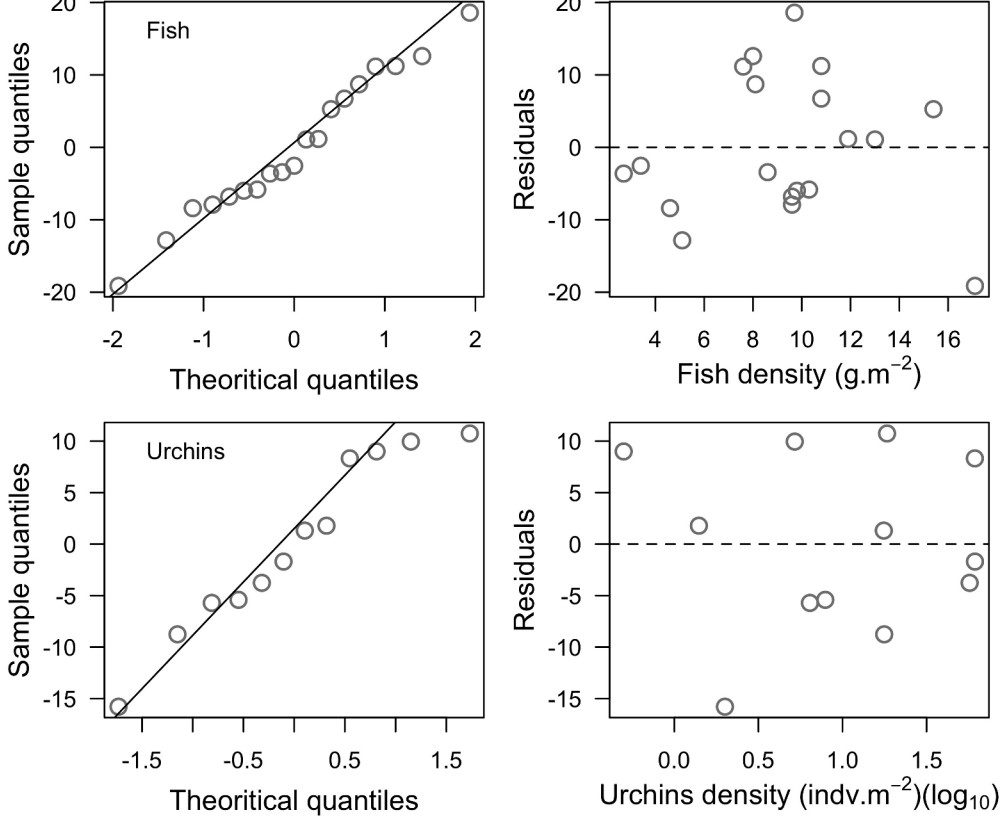

**Appendix 3—figure 7.** Normal quantile plots (left column) and residual plots (right column)

respectively verifying the assumptions of normality and equal variance distribution of the residuals of the least-squares linear regression models established to model fish (top row) and urchin grazing pressure (bottom row).

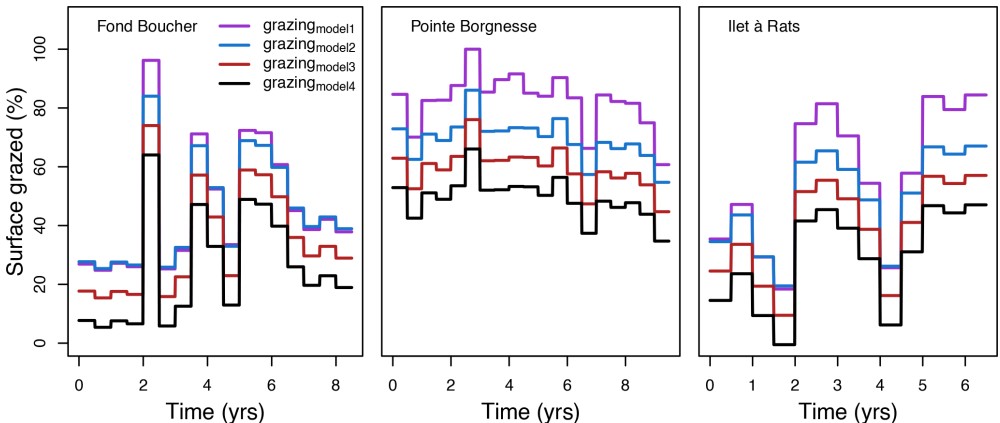

**Appendix 3—figure 8.** Percentage cover grazed imposed in the model at each time step for the three Caribbean sites.

## 3. Calibration

### 3.1. General procedure

We calibrated the model for each site independently and then conducted a between sites comparison of the parameter values that maximized model fit. A common approach for model calibration and validation is to divide the empirical data into a training and a test set. The model parameters are calibrated with the training set and the model generalizability is evaluated with the test set. Alternatively, resampling procedures can be used to train and test the model on the whole dataset iteratively (e.g. k-fold cross-validation). The time series we used were too short to be split and still allow for proper parameter calibration and model testing. Alternatively, we could have calibrated the model with one site and tested the model on the other sites. But we feared that model testing would not be satisfactory because of the paucity of the data describing the environmental context of the sites. By calibrating the model with the three datasets independently and then comparing the selected parameter values, we can dissociate between the parameters that can be generalized from those that need to be adjusted to a specific site. Alternatively, between-site differences in the parameter values offer the opportunity to question how relevant is the implementation of the related processes, if and how the implementation could be improved and if additional processes should be considered.

### 3.2. Sampling algorithm

All the possible combinations of parameter values are generated. Each parameter combination is interpretable as a point, whose parameter values are its coordinates in the parameter space. The algorithm we defined first selects the centroid, the most extreme parameter points, and the points situated at mid-distance between the centroid and the extreme points (1). A simulation with each parameter point is launched and replicated five times for each of the three Caribbean sites (2). The fit between the empirical and the simulated cover time series is measured using an objective function. The algorithm then selects the ten points providing the best performance and samples around each of them to select the five closest (and not yet tested) points in the parameter space (3). Steps (2) and (3) are repeated one more time. Distance between points in the parameter space is measured using the Gower's distance metric (*Gower, 1971*).

The objective function measures the performance of a given run by calculating the Euclidian distance between the empirical (*coverEmp*) and simulated cover time series (*coverSim*, averaged over five replicates), averaged over all the taxa:

$$performance = \frac{\sum\limits_{j=1}^{j=Ntaxa} \sqrt{\sum\limits_{t=0}^{t=t_{end}} \left(coverEmp_{j,t} - coverSim_{j,t}\right)^2}}{Ntaxa}$$

where $j$ = a specific taxon, $Ntaxa$ = the last taxon, $t$ = a specific time step, $t_{end}$ = the last time step. Smaller values indicate better performance, zero being the minimum value and indicating perfect match.

In order to compare the performance of the runs to a second reference value, we generated for each empirical dataset a null distribution of performance values. These values are measures of performance between the empirical datasets and a randomized version of themselves. We randomized by rows (i.e. time periods) to keep the total taxa cover identical to the real data.

## 3.3. Parameter selection and procedure

Being limited by the number of simulations we could perform in a reasonable time, we decided to calibrate the model in several rounds of simulations, each round being composed of one or several of the following steps: (1) select a limited number of parameters, define several possible values (instead of ranges for continuous parameters) and generate all the possible combination of parameter values; (2) launch the previously described sampling algorithm; (3) analyze the results of the calibration to identify the parameter values providing the worst performance; (4) analyze the simulated time series of the runs having the best performance to identify the processes and associated parameters that could improve the performance of the model and decide on the new values to add; (5) generate a new set of all the possible combinations of parameter values with the new parameter values and without those being associated with the worst performance; finally repeat (2 , 3) and (4) and eventually (5) until results are satisfactory. Eventually (7), launch additional simulations with specific parameter values selected by the modeller. This 'hands-on' calibration approach allows exploring specific parts of the parameter space potentially omitted by the algorithm or influencing specific processes that the modeller estimates to be relevant to act upon after having analysed previous results of the calibration. This human-directed search served to accelerate and guide the process, providing a faster convergence to the best parameter values.

We conducted in total four rounds of simulations. In round one, we selected only the parameters related to the different disturbance regimes (i.e. *bleaching*, *cyclone* and *grazing models*) and those whose value could not be found or estimated from the literature (i.e. *growth rate reduction interaction*, *otherProportions*, *prob cover crustose coralline algae*, *ratio overtop colony*, *Appendix 3—table 1*). We limited the number of possible parameter values to a maximum of five and generated the 4860 possible parameter points, from which our algorithm sampled 357 points for each site. This first round of simulations showed that none of the fittest parameter points at one site were sampled in another site. This could have been due to either the sites having different intrinsic characteristics leading to different local optima, or our algorithm was omitting certain parts of the parameter space. We consequently selected 40 of the parameter points providing best performance in each site and launched simulations with the subset of these points that were not run in a given site. This resulted in implementing round two with 41, 49 and 38 additional parameter points for Fond Boucher, Pointe Borgnesse and Ilet à Rats, respectively. None of these additional runs showed a better performance (*Appendix 3—figure 9*), confirming that the divergence in certain parameter values we observed between sites are due to intrinsic environmental differences. Analysis results at this stage revealed a consistently poorer performance of the simulations implementing values 0.01 and 0.001 of the parameter *otherProportions* (*Appendix 3—figure 10*). We consequently excluded these values in the next simulations. In addition, inspecting surface covers of the best fitted runs in each site revealed a poor match of the populations of algae. Grazing being one of the most important processes controlling these populations, we added more values to algae grazing probabilities. We then generated 2592 additional combinations of parameter values from which the algorithm sampled 202 of them (round three). The analysis of the simulated cover at this stage revealed a consistent incapacity of crustose coralline algae to maintain its population in all three sites. We decided to improve its persistence by enhancing its competitiveness against other algae (i.e. by adding the value 0.1 to *prob cover crustose coralline algae*) and reducing its palatability (i.e. by adding the value 0.05 to *prob grazing crustose coralline algae*). We launched the fourth round of simulations by

generating 72 new parameter points with these new values and a subset of the other parameter values that consistently provided a better performance (i.e. *bleaching model*: 3; *cyclone model*: 1, 2, 3; *grazing model*: 3, 4; *growth rate reduction interaction*: 2, 4, 8; *ratio overtop colony*: 2; *prob grazing macroalgae*: 0.7; *prob grazing allopathic macroalgae*: 0.3; *prob grazing Halimeda*: 0.3, 0.5; *prob grazing articulated coralline algae*: 0.7). We generated a total of 7716 different parameter combinations from which we sampled 672, 680 and 669 parameter points for Fond Boucher, Pointe Borgnesse and Ilet à Rats respectively.

## Related code

*Simulations/Rscripts/ Calibration.R* and *Calibration_functions.R* (*Carturan et al., 2020*).

**Appendix 3—table 1.** Description of parameters calibrated and their respective values considered.

| Parameters | Description | Values |
|---|---|---|
| *bleaching model* | Value of the coefficient φ (see Appendix 4: §2.2): a smaller φ increases the interspecific difference for the probability of bleaching when the thermal stress increases; a larger φ reduces this difference | 2; 3; 4 |
| *cyclone model* | Value of one of the three hydrodynamic regimes models displayed in **Appendix 3—figure 5** | 1; 2; 3 |
| *grazing model* | Value of one of the four grazing regimes models displayed in **Appendix 3—figure 6** | 1; 2; 3; 4 |
| *growth rate reduction interaction* | Lateral growth rate reduction coefficient to apply when one coral colony or an algae overgrows over other colonies or algae | 2; 3; 4; 6; 8 |
| *otherProportions* | The coefficient $p_o$, which is used to reduce the number of larvae produced by all the colonies present in the reef (Appendix 2: §7.2.1.1.d) | 0.0001; 0.001; 0.01 |
| *prob cover crustose coralline algae* | Probability that algae overgrow crustose coralline algae | 0.10; 0.25, 0.50; 0.75 |
| *ratio overtop colony* | Ratio needed for a branching or plating colony to overtop smaller colonies | 1.5; 2; 3 |
| *prob grazing macroalgae* | Probability of macroalgae being palatable | 0.3; 0.5; 0.7 |
| *prob grazing allopathic macroalgae* | Probability of allopathic macroalgae being palatable | 0.3; 0.5 |
| *prob grazing Halimeda* | Probability of *Halimeda* spp. being palatable | 0.3; 0.5; 1.0 |
| *prob grazing articulated coralline algae* | Probability of articulated coralline algae being palatable | 0.5; 0.7; 1.0 |
| *prob grazing crustose coralline algae* | Probability of crustose coralline algae being palatable | 0.05; 0.1; 0.2; 0.3; 1.0 |

## 3.4 Results

Performance values are bounded between 28 and 10 (where lower values are better) and are all below the lower 95% confidence limit of the null distribution of performance values (*Appendix 3—figure 9*). This shows that despite the model complexity, the high number of parameters, and the uncertainty around them, the model outputs population dynamics significantly closer to the real data as compared to randomly generated ones. The best performance values converged toward 10 in the three Caribbean sites (minimum values with standard error are 10.93 ± 3.677, 10.89 ± 2.872, 10.39 ± 3.119 for Fond Boucher, Pointe Borgnesse and Ilet à Rats, respectively) and were obtained during round three and four of the calibration (*Appendix 3—figure 9*). The first 257 points in round one were sampled uniformly in the parameter space, and expectedly show the highest ranges of performance values. The ranges from runs 258 to 357 narrow down drastically as sampling happened around the points providing best performance but the lower limits stayed constant. In round two, running in one site the combinations of parameter values that provided best performance in other sites increased the lower limit slightly for Fond Boucher, importantly for Pointe Borgnesse, and increased the higher limit for Ilet à Rats. The addition of alternative values for the probabilities of grazing the different groups of algae in round three lowered the lower range limits in all three sites. Finally, our attempt to support crustose coralline algae populations by increasing its competitiveness

and reducing its palatability in round four, did not change further the range of performance values (*Appendix 3—figure 9*), as crustose coralline algae cover remained quasi absent even in the runs having the best performance (*Appendix 3—figure 14*, *Appendix 3—figure 15*, *Figure 3*).

For six parameters, the same value was the most commonly observed among the best fitted runs in the three sites: *bleaching model* (2), *grazing model* (4), *otherProportions* (0.0001), *ratio overtop colony* (2, *Appendix 3—figure 10*), and to a lesser extent *prob grazing macroalgae* (0.7) and *prob grazing articulated coralline algae* (0.7, *Appendix 3—figure 11*).

Three parameters have contrasting values between sites: *growth rate reduction interaction* (eight for Fond Boucher and Ilet à Rats and two for Pointe Borgnesse), *prob grazing allopathic macroalgae* (0.3 for Pointe Borgnesse and either 0.3 or 0.5 for the other two sites), *prob grazing Halimeda* (0.5 for Pointe Borgnesse and 0.3 for Ilet à Rats). The differences observed for *prob grazing allopathic macroalgae* and *prob grazing Halimeda* (*Appendix 3—figure 11*) could suggest that the accurate value for both parameters lies between 0.3 and 0.5. The small *growth rate reduction* value calibrated at Pointe Borgnesse compared to the two other sites could be due to the difference of turf abundance. Probably less turf-coral direct interactions occur at Pointe Borgnesse because turf is less abundant and shares its dominance with allopathic macroalgae (*Figure 3*; *Appendix 3—figures 14* and *15*). Turf has the fastest growth rate among algae and coral species and is the most competitive algae when competing with corals. As a result, a higher growth rate reduction value might be necessary for coral populations to persist in Fond Boucher and Ilet à Rats compared to Pointe Borgnesse. These differences in the values providing best performance for these three parameters could be caused by contrasting algae growth rates between sites due to either different species composing the functional groups or differences in grazing regimes that we did not capture or other processes not implemented (e.g. effects of nutrient concentrations on growth rate).

Lastly, values considered for three parameters did not influence performance: *cyclone model*, *prob cover crustose coralline algae* and *prob grazing crustose coralline algae*. Two reasons might explain why none of the three hydrodynamic disturbance regimes simulated influence performance: (*i*) the coral species present were either too resistant or (*ii*) they did not reach a colony size large enough to be dislodged. Colony dislodgement due to cyclones and waves is determined from colony planar area and growth form (Appendix 2: §7.3.1). Several of the dominant species in the three sites are massive (i.e. *P. astreoides, O. faveolata, O. annularis, O. franksi*) and could withstand even the hardest regime (i.e. model 3 in *Appendix 3—figure 5*). The other relatively abundant species have more vulnerable growth forms and their fragility increases with colony size (i.e. branching: *M. mirabilis, P. furgata*; digitate: *M. decactis*; laminar: *A. agaricites*). Possibly, spatial competition in the model prevented these species from reaching colony sizes large enough to be dislodged even under the most intense regime.

Analyzing the distribution of the combinations of parameter values with their associated performance in the parameter space reveals one unique global optimum in each site as the parameter combinations having the highest performance are clustered together (*Appendix 3—figure 12*). The parameters that influence performance the most in all sites are *grazing model*, and *otherProportions*, *growth rate reduction interaction* (respectively arrow number 3, 5 and 4 in *Appendix 3—figure 12*). The probabilities of grazing the different algae clearly dissociate runs generated during round one and two from the ones generated during rounds three and four: in the latter, higher values of *prob grazing macroalgae* and lower values of *prob grazing allopathic macroalgae, Halimeda, articulated coralline algae* and *crustose coralline algae* were implemented (respectively arrow number 8, 9, 10, 11 and 12 in *Appendix 3—figure 12*). When compared all together, the combination of parameter values providing best performance in sites Fond Boucher and Ilet à Rats overlap (*Appendix 3—figure 13*). Pointe Borgnesse performance optimum is distant from the other sites due to opposite calibrated values for *growth rate reduction interaction* (*Appendix 3—figure 10*, arrow number four in *Appendix 3—figure 13*).

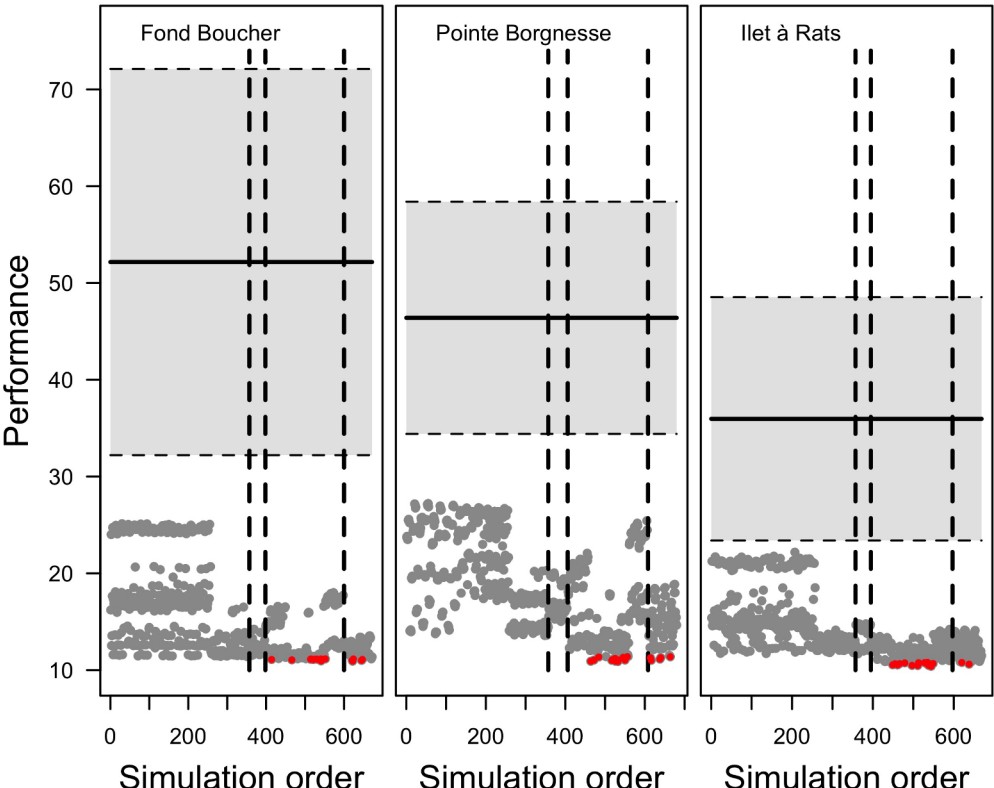

**Appendix 3—figure 9.** Evolution of the performance (smaller values show better performance) as a function of the order at which runs were launched for the three Caribbean sites (n = 672, 680 and 669 for Fond Boucher, Pointe Borgnesse and Ilet à Rats, respectively). Each grey dot represents the performance of a given parameter combination (averaged over five replicates). The 20 parameter points showing the best performance are shown in red. The vertical dashed lines separate the simulations launched respectively in round one, two, three and four. The horizontal black lines and grey areas show the mean and 95% confidence intervals of the null distributions of performance values (n = 1000).

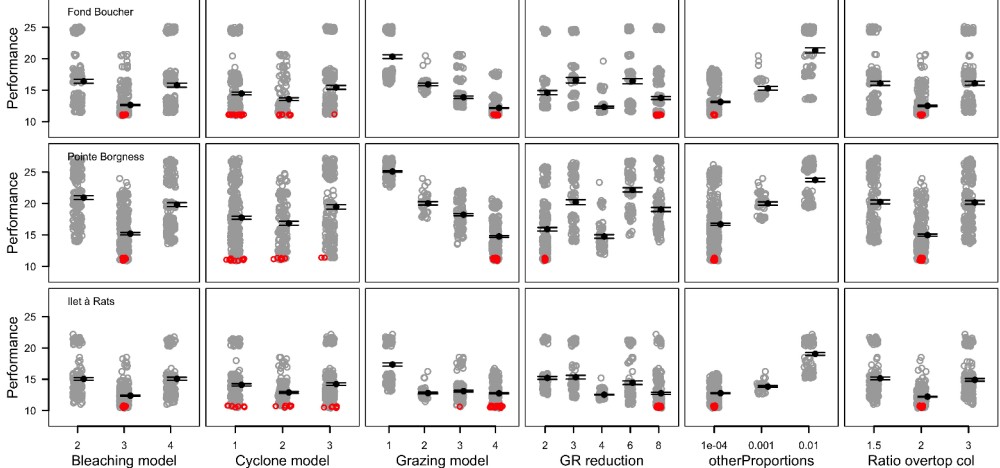

**Appendix 3—figure 10.** Performance comparison of all the combinations of parameter values simulated for six of the 12 parameters calibrated (n = 672, 680 and 669 for Fond Boucher, Pointe Borgnesse and Ilet à Rats, respectively). Grey circles represent the performance of a unique parameter combination (averaged over five replicates). Smaller values show better performance. Black dots and error bars show the mean ± standard error by parameter value. Red circles show the 20 parameter combinations providing the best performance.

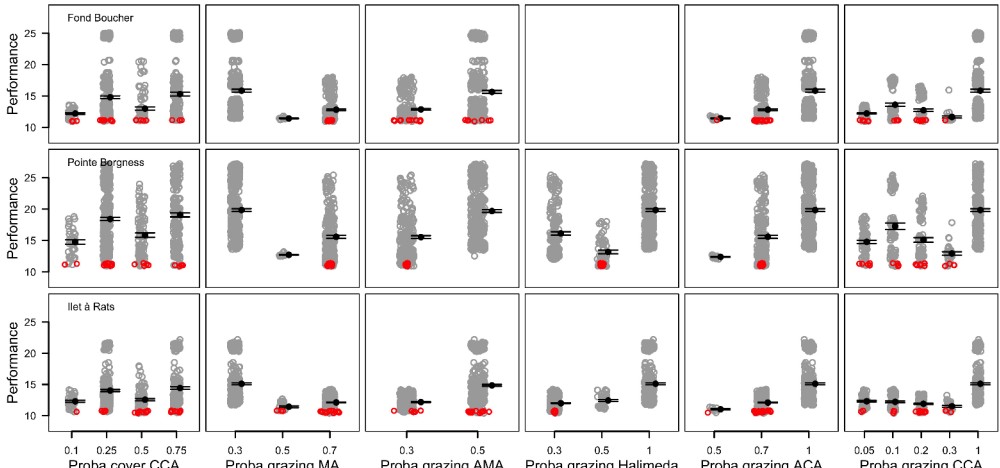

**Appendix 3—figure 11.** Performance comparison of all the combinations of parameter values simulated for six of the 12 parameters calibrated (n = 672, 680 and 669 for Fond Boucher, Pointe Borgnesse and Ilet à Rats, respectively). Grey circles represent the performance of a unique parameter combination (averaged over five replicates). Smaller values show better performance. Black dots and errors bars show the mean ± standard error by parameter value. Red circles show the 20 parameter combinations providing the best performance.

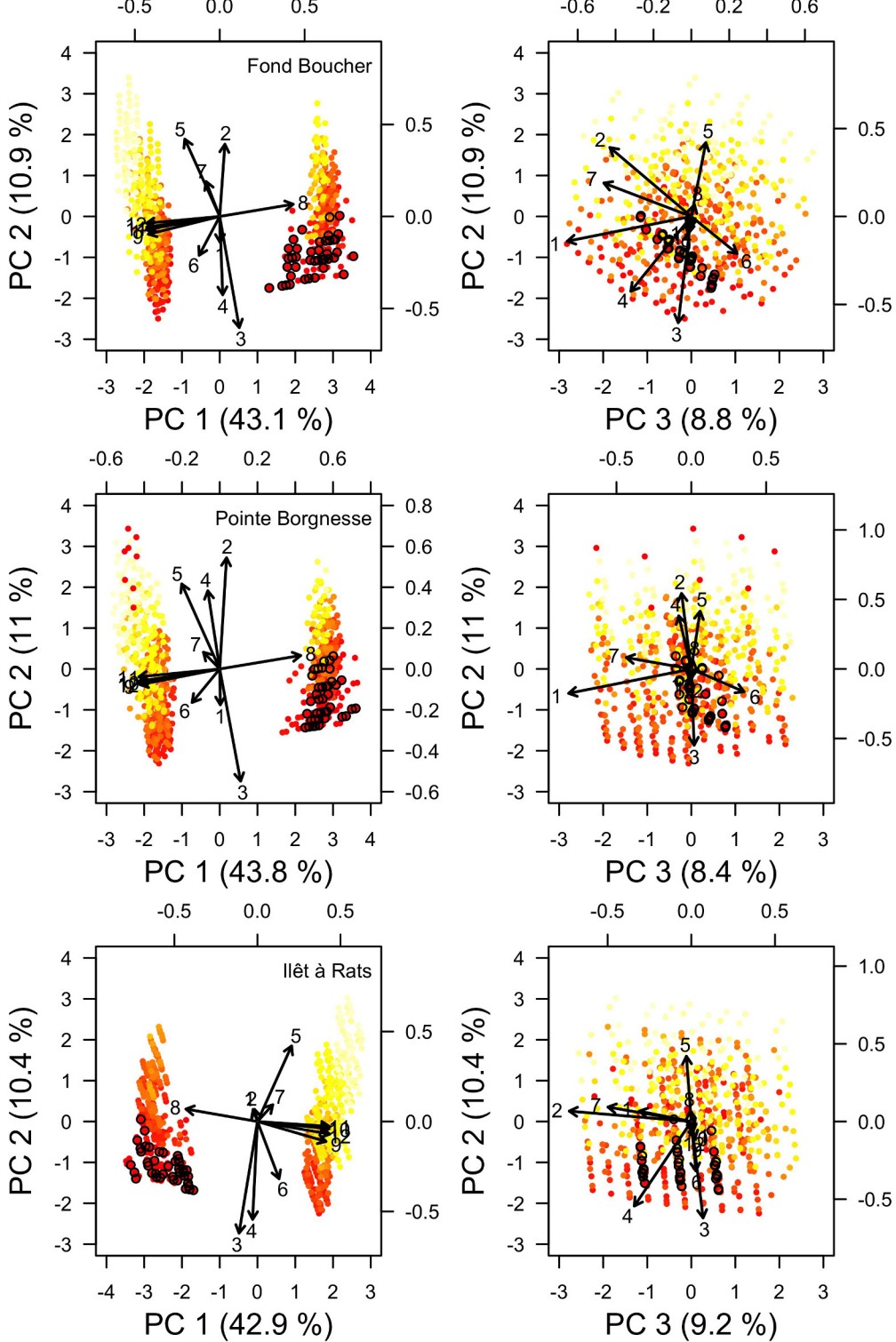

**Appendix 3—figure 12.** Projection in the parameter space of the combinations of parameter values selected during the calibration for each site (n = 672, 680 and 669 for Fond Boucher, Pointe Borgnesse and Ilet à Rats, respectively). Arrows indicate parameter loading; numbers designate parameters (coordinates values are displayed on the top and right sides): (1) *bleaching model*; (2) *cyclone model*; (3) *grazing model*; (4) *growth rate reduction*; (5) *otherProportions*; (6) *prob cover crustose coralline algae*; (7) *ratio overtop colony*; (8) *prob grazing macroalgae*; (9) *prob grazing allopathic macroalgae*; (10) *prob grazing Halimeda*; (11) *prob grazing articulated coralline algae*; (12)

*prob grazing crustose coralline*. Light yellow to red colour gradient indicates the performance values from lower to higher performance (averaged over five replicates). The 20 parameters points showing the best performance are circled in black.

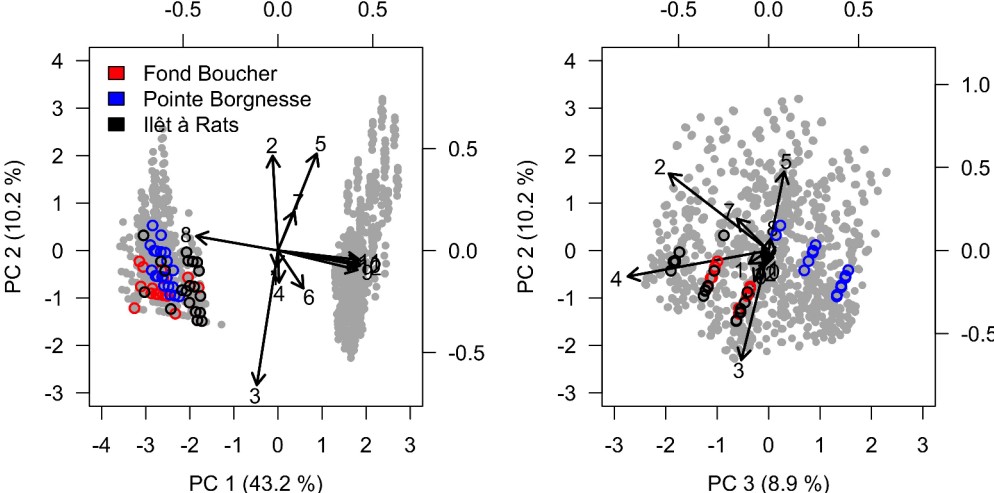

**Appendix 3—figure 13.** Projection in the parameter space of the combinations of parameter values selected during the calibration all sites confounded (n = 2021). Arrows indicate parameter loading; numbers designate parameters (coordinates values are displayed on the top and right sides): (1) *bleaching model*; (2) *cyclone model*; (3) *grazing model*; (4) *growth rate reduction*; (5) *otherProportions*; (6) *prob cover crustose coralline algae*; (7) *ratio overtop colony*; (8) *prob grazing macroalgae*; (9) *prob grazing allopathic macroalgae*; (10) *prob grazing Halimeda*; (11) *prob grazing articulated coralline algae*; (12) *prob grazing crustose coralline*. Grey dots represent individual combinations of parameter values. Circles highlight the 20 combinations providing best performance in each site.

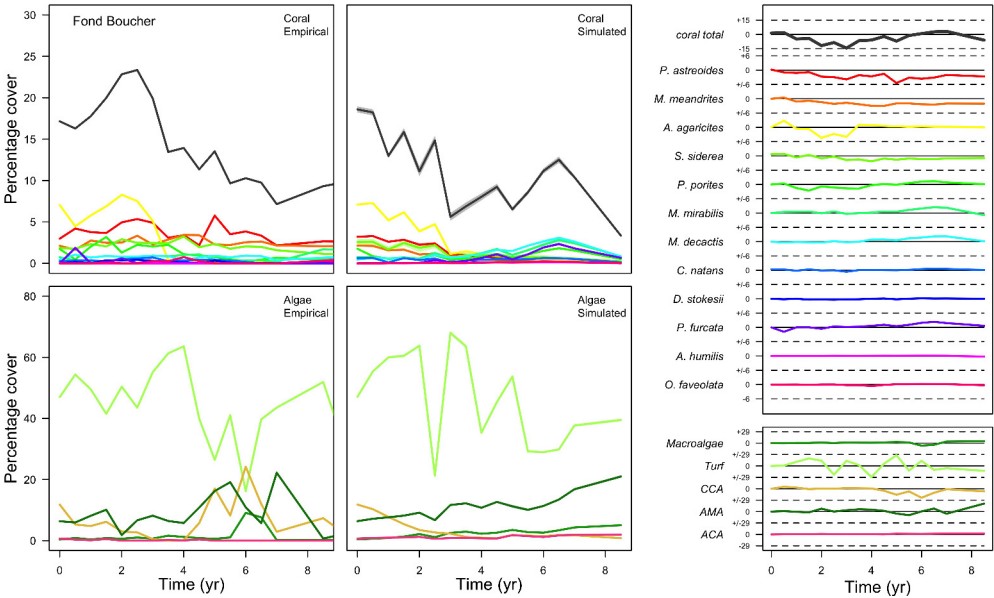

**Appendix 3—figure 14.** Comparison between empirical and simulated taxa cover for the combination of parameter values providing the best performance for site Fond Boucher. Solid lines in the simulated time series are the mean percentages cover (averaged over five replicates) and the shaded areas show the standard error. The right panels display the cover difference between simulated and empirical time series.

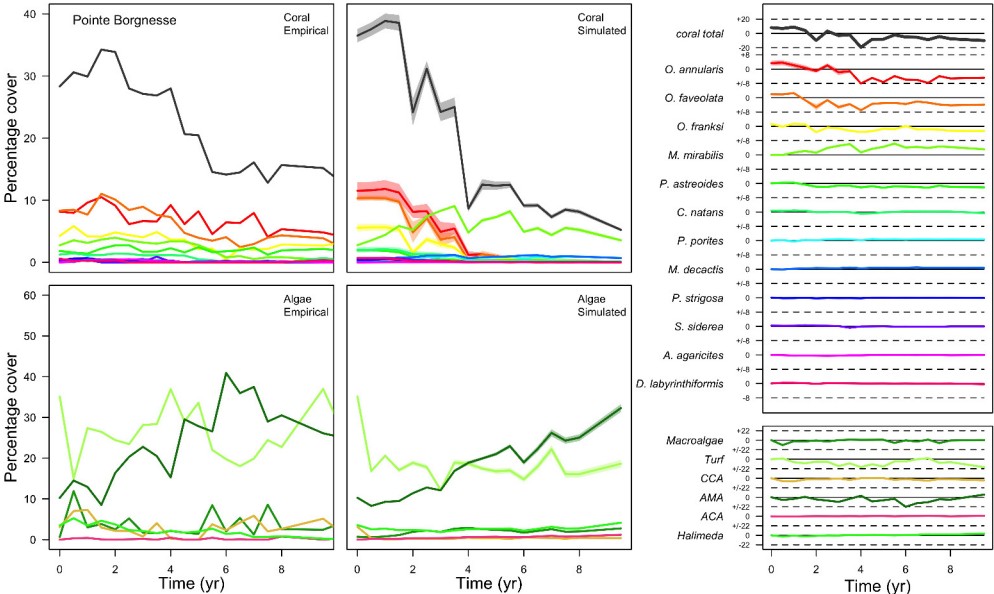

**Appendix 3—figure 15.** Comparison between empirical and simulated taxa cover for the combination of parameter values providing the best performance for site Pointe Borgnesse. Solid lines in the simulated time series are the mean percentages cover (averaged over five replicates) and the shaded areas show the standard error. The right panels display the cover difference between simulated and empirical time serie.

## Appendix 4

### Implementation of the bleaching response

The objective of this appendix is to show how we determined species-specific coral bleaching response models. We first defined an index of bleaching susceptibility expressed as a function of coral bleaching resistant traits. We then used this index to create species-specific logistic functions determining the bleaching probability of a colony as a function of thermal stress intensity. We parameterized these functions using an empirical dataset reporting impacts of bleaching events recorded in the Caribbean. Finally, we defined a non-species-specific logistic model determining the probability of bleaching-induced mortality as a function of thermal stress intensity. All the related code for the statistical analyses and production of figures is in *Manuscript/Rscripts/ Appendix S4 - Implementation bleaching response.R* (*Carturan et al., 2020*).

## 1. Species-specific index of bleaching susceptibility

### 1.1. General procedure

Our goal was to define an intrinsic (i.e. independent of environmental conditions) index of bleaching susceptibility expressed as a function of bleaching resistance traits. We established a statistical model between an empirical measure of bleaching susceptibility (dependent variables) and functional-traits (independent variables) on a subset of the 798 coral species for which the dependent variables had been measured. We then used the model to predict the bleaching susceptibility index value for the 798 coral species.

The dependent variable we selected is the bleaching response index (taxon-BRI), which represents the species-specific average percentage cover that bleached or died during an event. It was obtained for 374 taxa (304 if only considering species level values and after correcting for species names) from 2036 records concerning 316 sites, between 1982 and 2006 by *Swain et al. (2016b)*. 65% of the Taxon-BRI value is determined by intrinsic factors (i.e. coral biology), 6% by extrinsic factors (i.e., environmental conditions) and 29% by measurement uncertainty (*Swain et al., 2016b*).

Coral bleaching is a complex process involving numerous resistant traits. We initially selected five traits based on the number of species for which the traits were measured and on the degree to which they are implicated in the bleaching process: (i) colony maximum diameter, (ii) growth rate, (iii) microscopic reduced scattering coefficient, (iv) growth forms and (v) corallite area (*Carturan et al., 2018*). We used the imputed traits dataset in order to avoid having missing predictor values.

We generated numerous beta regression models, defined the confidence set (i.e. the subset of models being the most supported by the data), averaged the latter and predicted the index values for the 798 species.

### 1.2. Data exploration

We first converted the taxon-BRI's interval from [0100] to $]0,1[$. We then logit-transformed the taxon-BRI$_{]0,1[}$, as appropriate when modeling a proportional dependent variable (*Warton and Hui, 2011*). We log-transformed the numerical traits in order to reduce the skewness of their distributions and improve linearity of their association with taxon-BRI$_{]0,1[}$ (*Appendix 4—figure 1*).

The independent variables do show significant covariation (*Appendix 4—figure 2*): colony maximum diameter with corallite area (Spearman $r_s = -0.21$, $p<0.001$), growth form (Spearman $r_s = -0.18$, $p<0.01$), growth rate (Spearman $r_s = 0.25$, $p<0.001$) and microscopic reduced scattering coefficient (Spearman $r_s = 0.33$, $p<0.001$); corallite area with growth form (Spearman $r_s = 0.55$, $p<0.001$), growth rate (Spearman $r_s = -0.73$, $p<0.001$); and growth form with growth rate (Spearman $r_s = -0.75$, $p<0.001$). These covariations need to be considered when defining the full model (see §1.3). (We produced *Appendix 4—figure 2* using the *pairs.panels* function from the R package `psych` 1.8.3.3; *Revelle, 2017*).

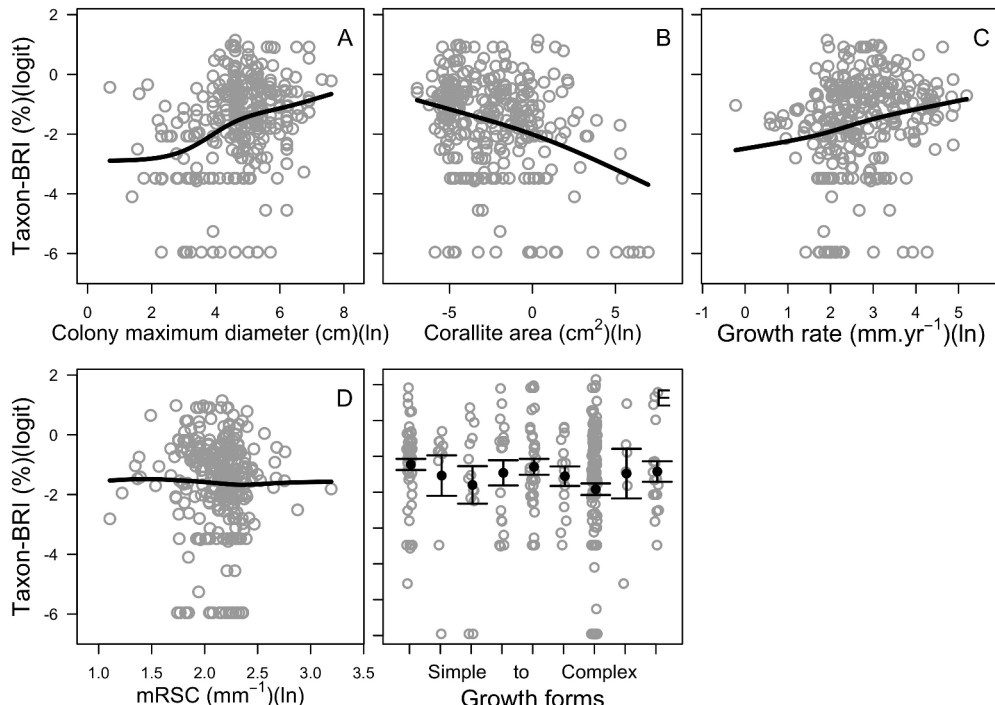

**Appendix 4—figure 1.** Relationships between the taxon-BRI$_{]0,1[}$ (logit-transformed) and the five bleaching resistance traits initially selected (n = 304). Black lines are smoothing splines and are used for visual aid. Growth forms are ranked from the most complex to the simplest: branching, table/plate, corymbose, digitate, laminar, columnar, massive, encrusting long upright, encrusting. Each gray circle represents the trait value averaged by species, the black points are the averaged trait value over all the species by category (E), and the error bars extend to ± one standard error.

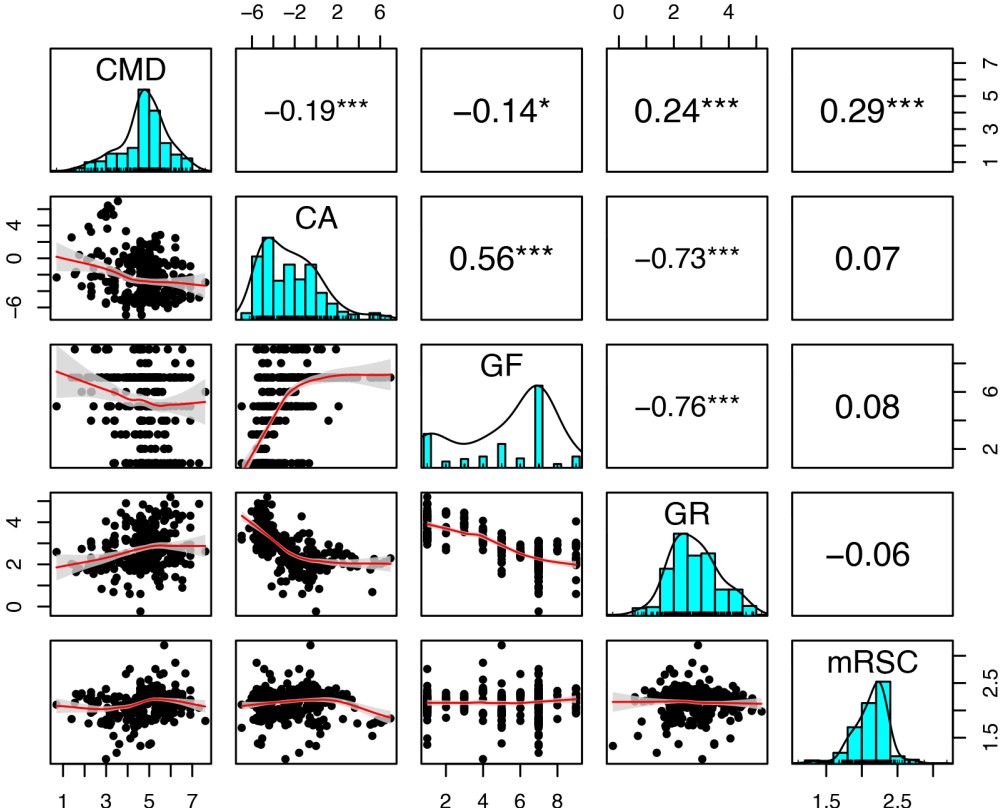

**Appendix 4—figure 2.** Correlation analyses between the bleaching resistance traits used in the full regression model (n = 304): colony maximum diameter (CMD), coralline algae (CA), growth forms (GF), growth rate (GR), and microscopic reduced scattering coefficient (mRSC). Growth forms are ranked from the most complex to the simplest: branching, table/plate, corymbose, digitate, laminar, columnar, massive, encrusting long upright, encrusting. Black points represent the trait value averaged by species, the red line is the locally estimated scatterplot smoothing fit and the grey area it 95% confidence interval. The upper panels show the values of the Spearman's rank correlation coefficient. Asterisks indicate the test statistics' significance: *p < 0.05; **p < 0.01; ***p < 0.001.

## 1.3. Beta regression: the global model

Proportional data often display asymmetric distributions, which contradict the normality assumption required in classic regression models. Contrastingly, beta regression models allow for the prediction of response variables having any unimodal distribution in the interval ]0,1[ (i.e. 'beta' distributions; *Ferrari and Cribari-Neto, 2004*). Beta density distributions are defined by two parameters: the mean $\mu$ and the precision parameter $\phi$ and are mathematically expressed as $B(\mu,\phi)$. Obtaining the final beta regression model requires following several steps: (i) define the 'global' model, (ii) remove the observations (species) that are too influential on the model parameters (based on the Cook's distance), (iii) chose the appropriate link function for $\mu$, (iv) chose the appropriate link function for $\phi$, (v) define the confidence set of models (i.e. data dredging), and eventually (vi) average these models. These steps are described in detail below:

### The initial 'global' model

We first defined the 'global' (also referred as 'full') model by including as predictors all the bleaching resistance traits at the first and second degree polynomials as well as their significant interactions based on *Appendix 4—figure 2*. We chose the logit link function as the initial link function for the 'location' submodel (i.e. to model the mean μ) because the logit function is appropriate for modeling proportional data (*Warton and Hui, 2011*). We chose a constant $\phi$ to model dispersion. We used the *betareg* function of the R package `betareg` 3.1–0 (*Cribari-Neto and Zeileis, 2010*) to define the beta regression models. The global model is:

$$logit(E(taxon-BRI]0,1[i)) = \alpha \quad +\beta 1 \times ln(colony\ maximum\ diameter)_i$$
$$+\beta 2 \times ln(corallite\ area)_i$$
$$+\beta 3 \times ln(growth\ rate)_i$$
$$+\beta 4 \times ln(mRSC)_i$$
$$+\beta 5 \times growth\ form_i$$
$$+\beta 6 \times ln(colony\ maximum\ diameter)_i : ln(corallite\ area)_i$$
$$+\beta 7 \times ln(colony\ maximum\ diameter)_i : growth\ form_i$$
$$+\beta 8 \times ln(colony\ maximum\ diameter)_i : ln(growth\ rate)_i$$
$$+\beta 9 \times ln(colony\ maximum\ diameter\ )_i : ln(mRSC)_i$$
$$+\beta 10 \times ln(corallite\ area)_i : growth\ form_i$$
$$+\beta 11 \times ln(corallite\ area)_i : ln(growth\ rate)_i$$
$$+\beta 12 \times growthform_i : ln(growth\ rate)_i$$
$$+\beta 13 \times ln(colony\ maximum\ diameter)_i 2$$
$$+\beta 14 \times ln(corallite\ area)_i 2$$
$$+\beta 15 \times ln(growth\ rate)_i 2$$
$$+\beta 16 \times ln(mRSC)_i 2$$
$$+\epsilon i$$

with *taxon-BRI$_{]0,1[i}$* ~ *B($\mu_i$, $\phi$)* (i.e. beta distribution with mean $\mu_i$ and dispersion $\phi$), E(*taxon-BRI$_{]0,1[}$*) = $\mu$, VAR(*taxon-BRI$_{]0,1[}$*) = $\mu(1 - \mu)/(1 + \phi)$, $\alpha$ the intercept, $\varepsilon_i$ the residual associated with the $i^{th}$ observation (i.e. species) and $\beta_j$ the respective coefficient for each of the bleaching resistance traits.

## Removal of influential observations

We used the Cook's distance (**Cook, 1977**) to identify observations (i.e. species) being the most influential on the model parameters. Observations with a Cook's distance superior to one should be removed (**Johnson and Omland, 2004** and reference therein). We conserved all the observations because none of them had a large Cook's distance (**Appendix 4—figure 3**). We calculated the Cook's distance using the *cooks.distance* function from the package stats (**R Development Core Team, 2017**).

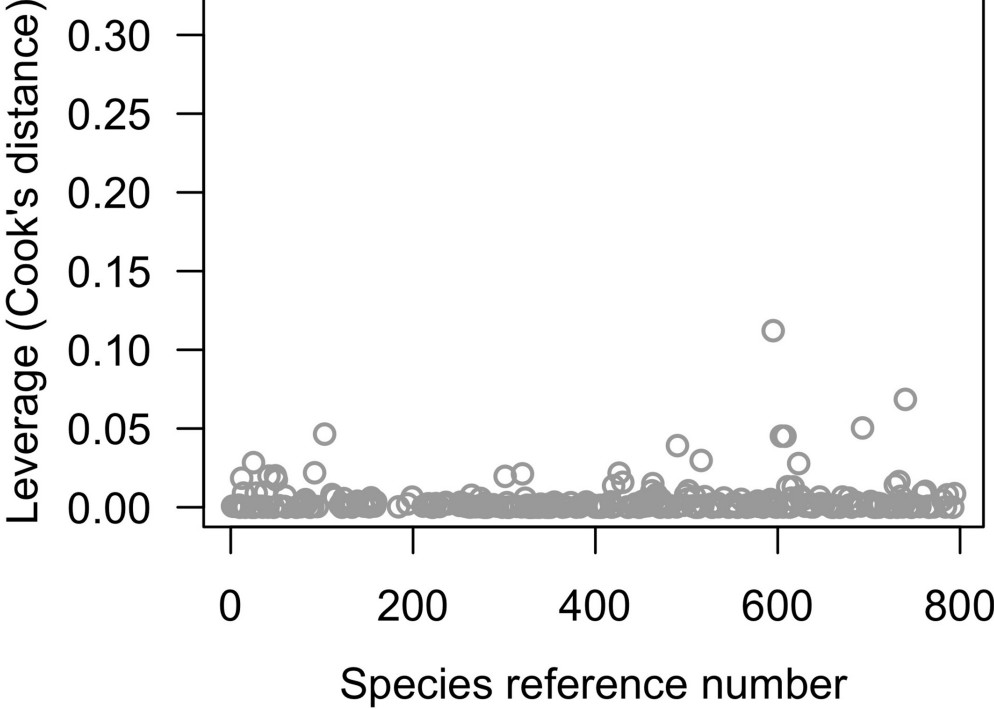

**Appendix 4—figure 3.** Cook's distance of the species (n = 304) for the global beta regression model.

## Selection of the link function for μ

Different link functions can be used to model the distribution of the response variable (e.g. logit, loglog, etc.). We created a global beta regression for each of the link functions available in the function *betareg* (*Appendix 4—figure 4*). We then selected the link function based on (1) fit maximization—assess with the adjusted $R^2$ and the Akaike information criterion (AIC)—and (2) the distribution of the residuals (*Ferrari and Cribari-Neto, 2004*; *Zuur et al., 2009*). *Johnson and Omland (2004)* argued that AIC should be used over the adjusted $R^2$ in model selection because the latter does not account for the complexity of the model (i.e. the parsimony principal). However, in our situation, all the model candidates have the same complexity (i.e. number of predictors). It is consequently relevant to also consider this measure of fit. We assessed the homoscedasticity of the residuals by comparing Pearson residuals against predicted values (*Appendix 4—figure 5*). Finally, we assessed the distribution of the residuals by plotting the Pearson residuals against normal quantiles (*Appendix 4—figure 6*).

Each version of the global model satisfies the assumptions of homogeneity (*Appendix 4—figure 5*) and normality (*Appendix 4—figure 6*) of the residual distribution and no model seems to perform better in these aspects. We chose the model implementing the *cloglog* link function as it provides the highest fit ($R^2 = 0.23$; *Appendix 4—figure 4*).

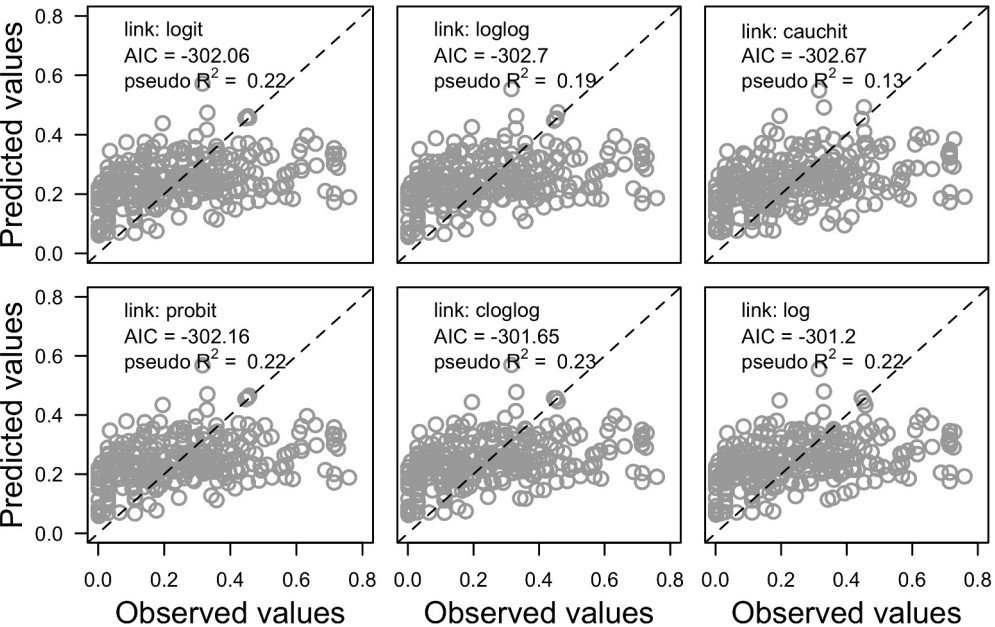

**Appendix 4—figure 4.** Associations between the observed and predicted values for different versions of the global model. Each version implements a different link function to estimate the parameter μ (n = 304). Also displayed are the pseudo $R^2$, the Akaike information criterion (AIC) and the identity line.

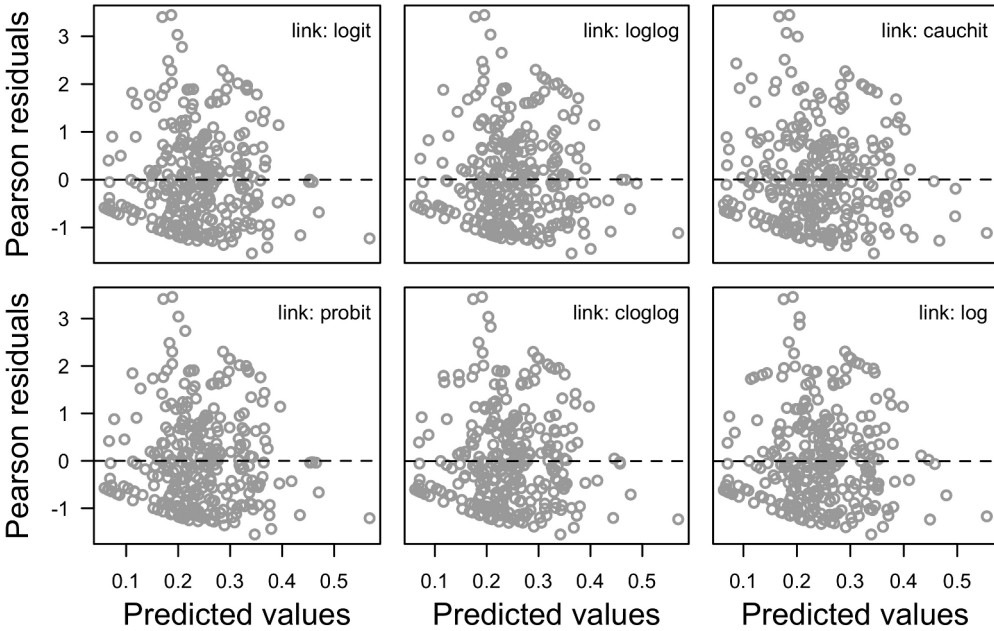

**Appendix 4—figure 5.** Associations between Pearson's residuals and predicted values for the different versions of the global model. Each version implements a different link function to estimate the parameter $\mu$ (n = 304).

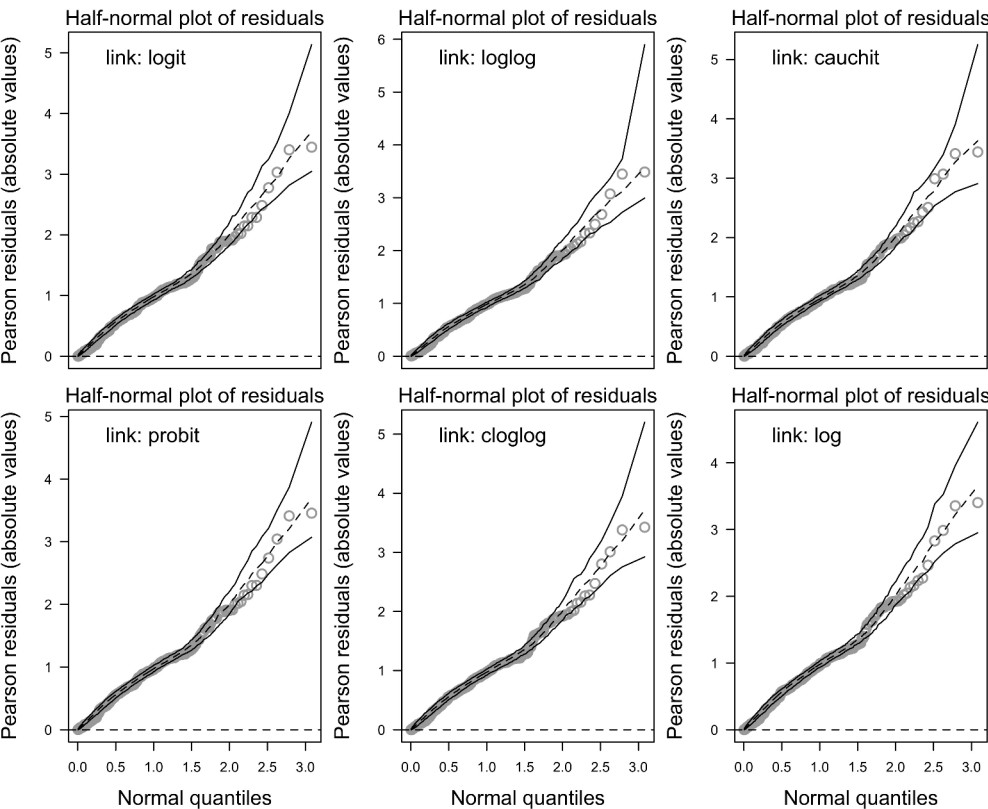

**Appendix 4—figure 6.** Half-normal plot with simulated envelopes for the different versions of the global model. Each version implements a different link function to estimate the parameter $\mu$ (n = 304).

## Selection of the link function for $\phi$

We implemented each factor individually with a given link function (i.e. *identity*, *log*, *sqrt*) and compared the resulting models with the one implementing $\phi$ as a constant using a likelihood-ratio test (*Cribari-Neto and Zeileis, 2010*). None of the models obtained yielded a significant p-value. We consequently kept $\phi$ as a constant. The likelihood-ratio test was performed with the *lrtest* function from the R package `lmtest` (*Zeileis and Hothorn, 2002*).

## Model selection

We generated 504 submodels (i.e. models implementing a subset of the global model's predictors, excluding models with only the second order of the polynomial for a given predictor) and selected the 95% confidence set of models using the Akaike weight (*Johnson and Omland, 2004*). The procedure selected 65 nested models. The predictors *ln(colony maximum diameter)* and *ln(corallite area)* are present in all the models whereas *growth form* is not present in any of them (*Appendix 4—table 1*, *Appendix 4—table 2*). We used the *dredge* function for generating and ranking the different models and the *get.models* function for selecting the confidence set; both functions come from the R package `MuMIn` 1.40.0 (*Bartón, 2017*).

**Appendix 4—table 1.** Summary of the bleaching resistance traits (predictors) for the averaged beta regression model (from the 65 models of the 95% confidence set).
The relative importance of each predictor is calculated as a sum of the Akaike weights over all of the models in which the term appears.

| Bleaching resistance traits | Relative importance | No. models where present |
|---|---|---|
| *ln(colony maximum diameter)* | 1.00 | 65 |
| *ln(corallite area)* | 1.00 | 65 |
| *ln(mRSC)* | 1.00 | 65 |
| *ln(growth rate)* | 0.59 | 49 |
| *ln(colony maximum diameter): ln(corallite area)* | 0.57 | 37 |
| *ln(colony maximum diameter): ln(growth rate)* | 0.22 | 23 |
| *ln(colony maximum diameter)$^2$* | 0.27 | 29 |
| *ln(corallite area) $^2$* | 0.58 | 37 |
| *ln(mRSC) $^2$* | 0.26 | 29 |
| *ln(growth rate) $^2$* | 0.20 | 21 |

**Appendix 4—table 2.** Summary statistics of the ten best models (out of 65) belonging to the 95% confidence set (link function: *cloglog*).

| | Df | logLik | AIC | Delta | Weight |
|---|---|---|---|---|---|
| 1/2/8/9* | 6 | 169.759 | −327.517 | 0.000 | 0.073 |
| 1/2/5/8 | 6 | 169.742 | −327.484 | 0.033 | 0.071 |
| 1/2/3/5/8 | 7 | 170.434 | −326.868 | 0.649 | 0.052 |
| 1/2/5/8/9 | 7 | 170.284 | −326.568 | 0.950 | 0.045 |
| 1/2/3/8/9 | 7 | 170.005 | −326.011 | 1.506 | 0.034 |
| 1/2/3/5/8/9 | 8 | 171.000 | −326.000 | 1.517 | 0.034 |
| 1/2/3/8/9/10 | 8 | 170.948 | −325.896 | 1.622 | 0.032 |
| 1/2/8 | 5 | 167.844 | −325.689 | 1.829 | 0.029 |
| 1/2/3/5/6/8 | 8 | 170.825 | −325.650 | 1.867 | 0.029 |
| 1/2/4/5/8 | 7 | 169.793 | −325.585 | 1.932 | 0.028 |

*Each number corresponds to a predictor: *ln(colony maximum diameter)* (1); *ln(corallite area)* (2); *ln (growth rate)* (3); *ln(colony maximum diameter)2* (4); *ln(corallite area)2* (5); *ln(growth rate)2* (6); *ln(mRSC)2*

(7); *ln(mRSC)* (8); *ln(colony maximum diameter): ln(corallite area)* (9).

## Model averaging

We averaged the parameter values to obtain the final model. The procedure provides parameter values obtained from (i) the 'natural average', or 'conditional average', which we refer as the 'subset model' (i.e. the parameter estimate is averaged only from the parameter values of the model in which it is present) and (ii) the 'zero method' or 'full average', which we refer as the 'full model' (i.e., zeros are replaced as values of parameters in the models where the variable is not present). We used the 'subset model' in order to avoid shrinking the factors not present in all the models (*Grueber et al., 2011*), but we also present the results of the 'full model' for comparison (*Appendix 4—figure 7*, *Appendix 4—figure 8*, *Appendix 4—figure 9*, *Appendix 4—figure 10*, *Appendix 4—figure 11*). The averaged 'subset' model coefficients are presented in (*Appendix 4—table 3*). The diagnostic plots of the Pearson residuals for both models are presented in *Appendix 4—figure 7*.

For both models, residuals are evenly distributed along each predictor (*Appendix 4—figure 7*, *Appendix 4—figure 8*) and their normality is acceptable (*Appendix 4—figure 9*). Diagnostic residuals plots are similar between the 'subset' and the 'full' models.

We averaged the models using the *model.avg* function from the R package `MuMIn`. We obtained the pseudo-$R^2$ value by squaring the correlation between linear predictors and the cloglog-transformed $BRI_{]0,1[}$ (*Ferrari and Cribari-Neto, 2004*). We used the Pearson's coefficient as measure of correlation.

**Appendix 4—table 3.** Estimates of the parameters of the averaged 'subset' beta regression model (link function: *cloglog*).

| Model's parameters | Estimate | Std. error | Z value | Pr(>|z|) |
|---|---|---|---|---|
| (Intercept) | −1.242 | 1.0273 | 1.209 | 0.227 |
| *ln(colony maximum diameter)* | 0.187 | 0.1470 | 1.274 | 0.203 |
| *ln(corallite area)* | −0.123 | 0.0806 | 1.527 | 0.127 |
| *ln(mRSC)* | −0.668 | 0.8174 | 0.817 | 0.414 |
| *ln(growth rate)* | 0.024 | 0.2937 | 0.081 | 0.936 |
| *ln(colony maximum diameter): ln(corallite area)* | 0.024 | 0.0226 | 1.061 | 0.289 |
| *ln(colony maximum diameter): ln(growth rate)* | 0.063 | 0.0707 | 0.886 | 0.376 |
| *ln(colony maximum diameter)$^2$* | 0.001 | 0.0252 | 0.053 | 0.958 |
| *ln(corallite area)$^2$* | −0.001 | 0.0061 | 1.582 | 0.114 |
| *ln(mRSC)$^2$* | 0.135 | 0.3497 | 0.386 | 0.699 |
| *ln(growth rate)$^2$* | −0.034 | 0.0440 | 1.000 | 0.317 |
| *phi* | 4.267 | 0.3360 | 12.697 | <0.001 |
| Pseudo $R^2$ | 0.122 | | | |

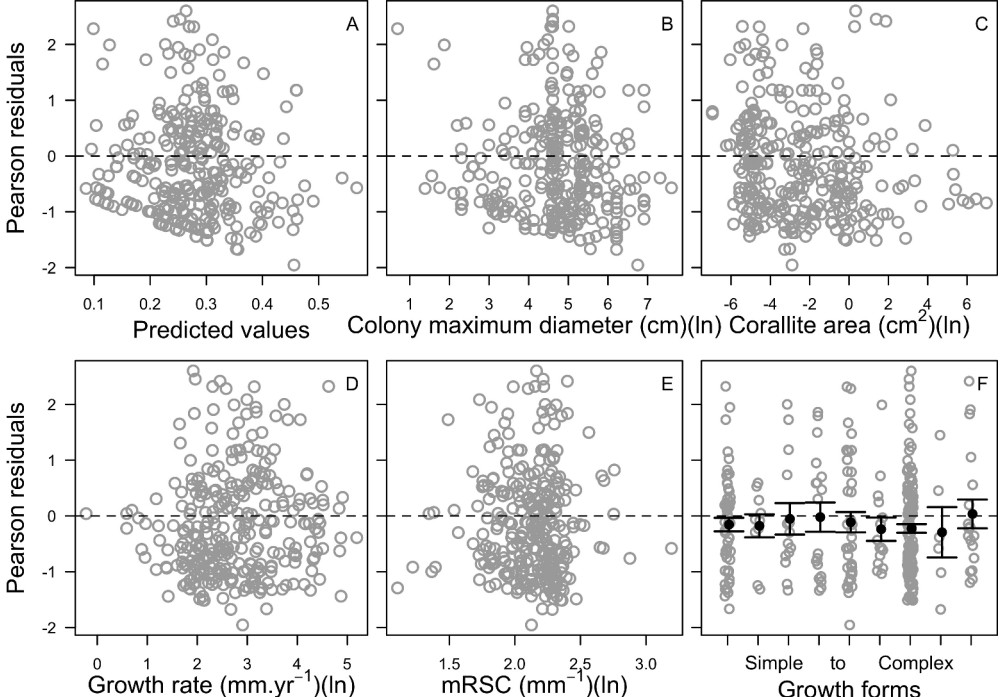

**Appendix 4—figure 7.** Relationship between Pearson residuals of the averaged 'subset' beta regression model and predicted values (**A**), and each of the potential bleaching resistant traits (**B–F**) (n = 304). Growth forms are ranked from the most complex to the simplest: branching, table/plate, corymbose, digitate, laminar, columnar, massive, encrusting long upright, encrusting. Each gray circle represents the trait value averaged by species, the black points are the averaged trait value over all the species by category (**E**), and the error bars extend to ± one standard error.

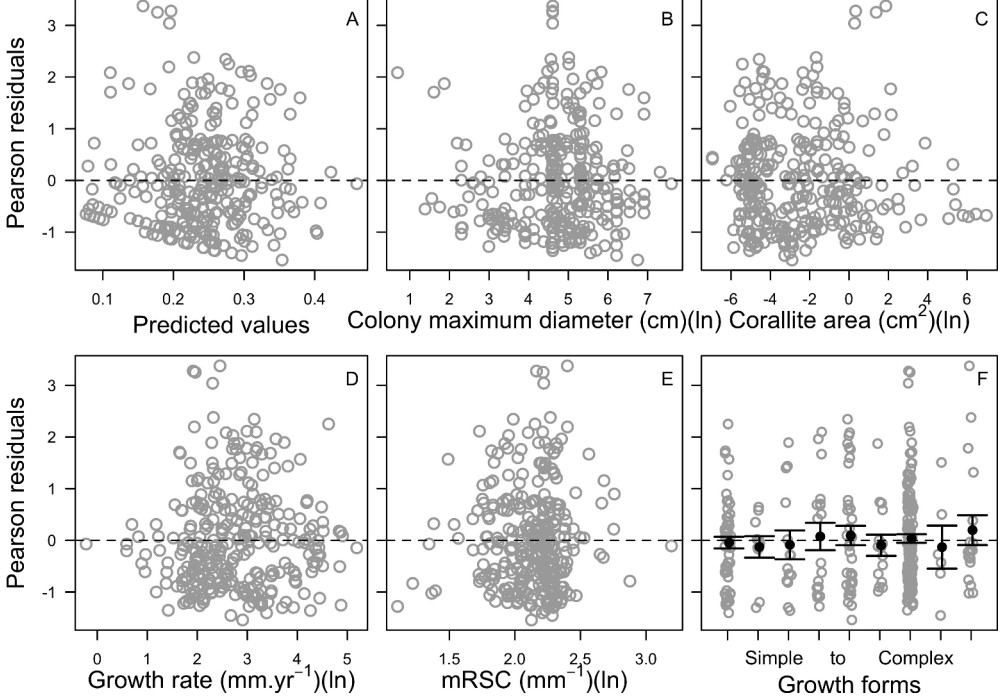

**Appendix 4—figure 8.** Relationship between Pearson residuals of the averaged 'full' beta regression model and predicted values (**A**), and each of the potential bleaching-resistant traits (**B–F**) (n = 304). Growth forms are ranked from the most complex to the simplest: branching, table/plate,

corymbose, digitate, laminar, columnar, massive, encrusting long upright, encrusting. Each gray circle represents the trait value averaged by species, the black points are the averaged trait value over all the species by category (**E**), and the error bars extend to ± one standard error.

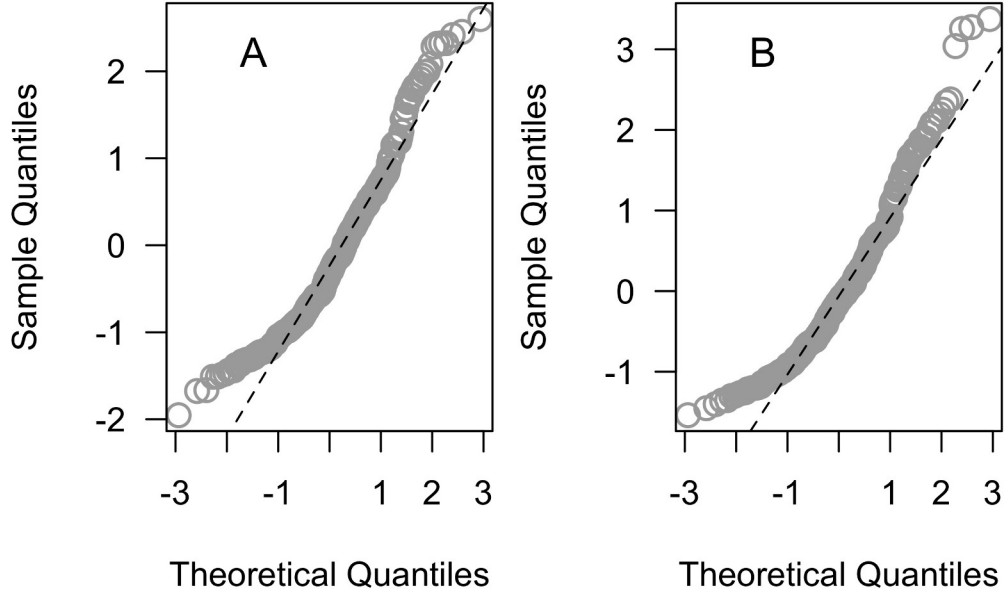

**Appendix 4—figure 9.** Normal quantile plot of the Pearson residuals of the averages 'subset' (**A**) and the 'full' (**B**) beta regression models (n = 304).

## Intrinsic probability of bleaching

We then obtained the intrinsic probability of bleaching for each of the 798 species using the final averaged beta regression model (*Appendix 4—figure 10*, *Appendix 4—figure 11*).

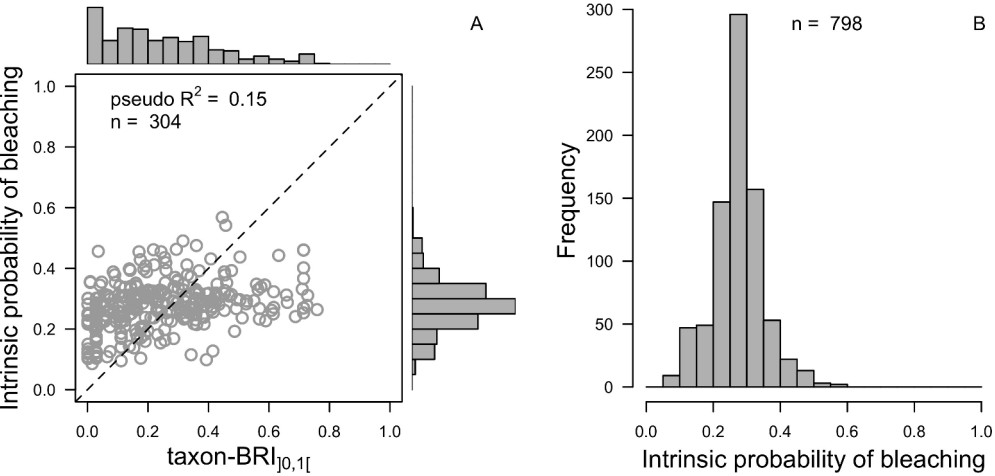

**Appendix 4—figure 10.** Comparison of distributions between the observed (taxon-BRI$_{]0,1[}$) and the predicted (intrinsic probability of bleaching) response variable (**A**) and the extrapolated intrinsic probability of bleaching (n = 798, **B**) for the averaged 'subset' beta regression model.

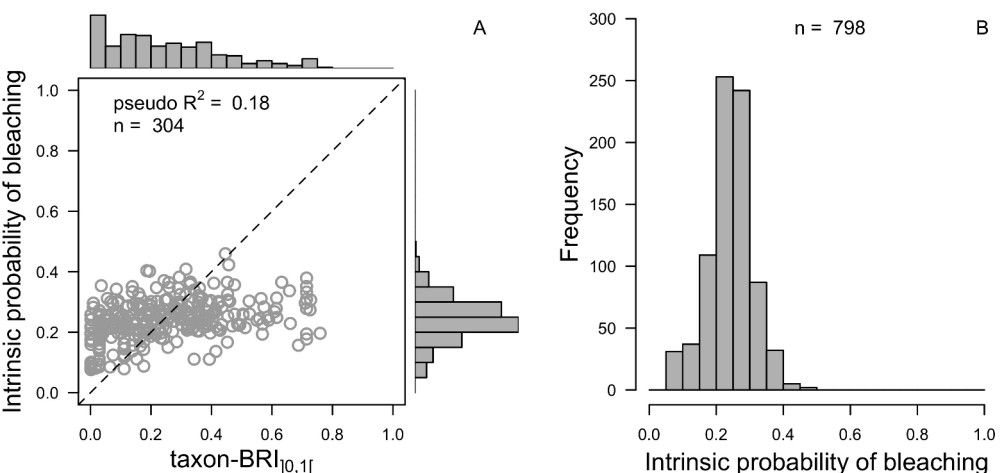

**Appendix 4—figure 11.** Comparison of distributions between the observed (taxon-BRI$_{]0,1[}$) and the predicted (intrinsic probability of bleaching) response variable (**A**) and the extrapolated intrinsic probability of bleaching (n = 798, **B**) for the averaged 'full' beta regression model.

## 2.Species-specific bleaching probability models

### 2.1. Dataset

*Eakin et al. (2010)* established an empirical linear model linking the mean proportion of colony bleaching within a reef and the intensity of the thermal stress expressed in degree heating weeks (DHW; the product of °C above the highest monthly mean sea-surface temperature for a location and its duration in weeks during the most recent 12-week period; coralreefwatch.noaa.gov; *Kayanne, 2017*). The relationship is based on 2575 bleaching surveys in the Caribbean between June 2005 and February 2006. *Eakin et al. (2010)* combined percentages of coral cover bleached and colonies bleached (i.e. the proportion of colonies bleached over the total number of colonies) to define the 'mean coral bleached (%)'. They looked at the association between the mean coral bleached (%) and (i) the 'observed DHW' and (ii) the '2005 annual maximum DHW'.

For consistency with our agent-based model, we only considered the cover bleached as the dependent variable. We used the 'observed DHW' as the independent variable because the bleaching response usually happens within the weeks following the onset of the disturbance.

### 2.2. Generalized linear mixed model

#### The global model

We defined a binomial generalized linear mixed model (GLMM) using a logistic link function to model the association between bleached coral cover ('cover bleached') and DHW. We considered each observation in *Eakin et al. (2010)*'s dataset as a single data point (several observations were made at a same site and date). We defined the variable 'time' as the number of days elapsed since the earliest sampling date in Eakin et al.,'s dataset (i.e. the first sampling date corresponds to day one). We established an initial model with DHW as fixed effect and site and time as random effects:

$$logit(E(cover\ bleached_{ijk})) = \alpha + \beta_1 \times DHW_i + site_j + time_k$$

where = $\alpha$ the intercept, $DHW_i$ = the fixed effect associated to the coefficient $\beta_1$ and the $i^{th}$ observation; $site_j$ and $time_k$ = the random intercepts associated to the $j^{th}$ site and $k^{th}$ time unit, respectively, with $site_j \sim N(0,\sigma^2_{site})$ and $timek_k \sim N(0,\sigma^2_{time})$ (i.e., $site_j$ and $time_k$ are normally distributed, with a mean of zero and variance $\sigma^2_{site}$ and $\sigma^2_{time}$, respectively).

#### Model selection

We tested the significance of each random effect individually using a likelihood ratio test to compare the goodness of fit of the first model with a model excluding alternatively site and time. The test showed that both site ($\chi^2$ = 14.56, p<0.001) and time ($\chi^2$ = 14.56, p=0.004) contribute significantly

to the goodness of fit of the model. We used the function *glmer* from the R package `lme4` 1.1–15 (***Bates et al., 2015***) to create the model and *lrtest* from the R package `lmtest` 0.9–35 (***Zeileis and Hothorn, 2002***) for the likelihood ratio test.

## Model validation

Residuals of logistic regressions are by nature curvilinear, not normally distributed and heteroscedastic. Residuals diagnostic plots are in consequence used to detect strong disqualifying patterns. No such patterns are found with the model: (i) no strong curvilinear trends are observed between the Pearson residuals and the fitted values; (ii) deviance residuals do not deviate greatly from normality; (iii) the distribution of residuals is not strongly heteroscedastic; (iv) there are no outliers (***Appendix 4—figure 12***).

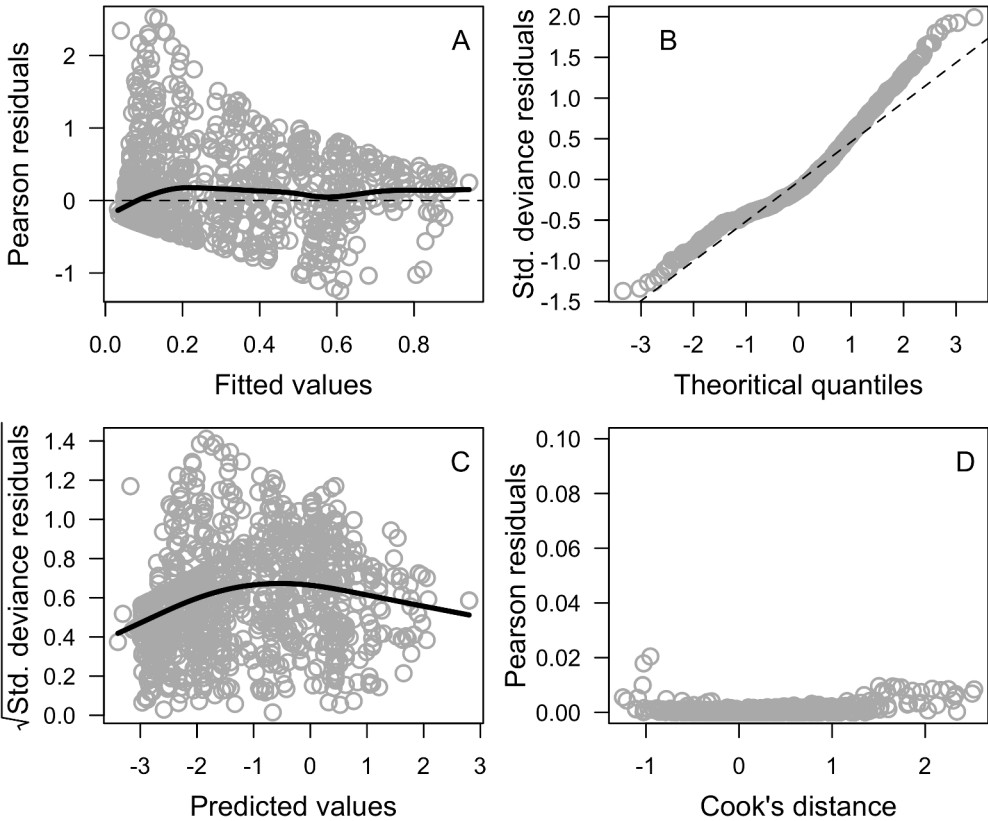

**Appendix 4—figure 12.** Diagnostic plots of the generalized linear mixed model (n = 1216). Black lines are smoothing splines and are used for visual aid.

## Model parameters

The parameters of the generalized linear mixed model are presented in ***Appendix 4—table 4*** and the fit of the model is displayed in ***Appendix 4—figure 13***.

**Appendix 4—table 4.** Parameters values of the generalized linear mixed model (binomial distribution and logit link function).
The model as fitted on ***Eakin et al. (2010)***'s dataset.

| Fixed effects | | | | | Random effects | |
|---|---|---|---|---|---|---|
| **Parameters** | **Estimate** | **SE** | **z-value** | **p-Value** | **Parameters** | **Variance** |
| $\alpha$ (intercept) | −2.78 | 0.191 | −14.55 | <0.001 | $site_j$ | 0.429 |
| $\beta_1$ (DHW) | 0.29 | 0.026 | 11.16 | <0.001 | $time_k$ | 0.567 |

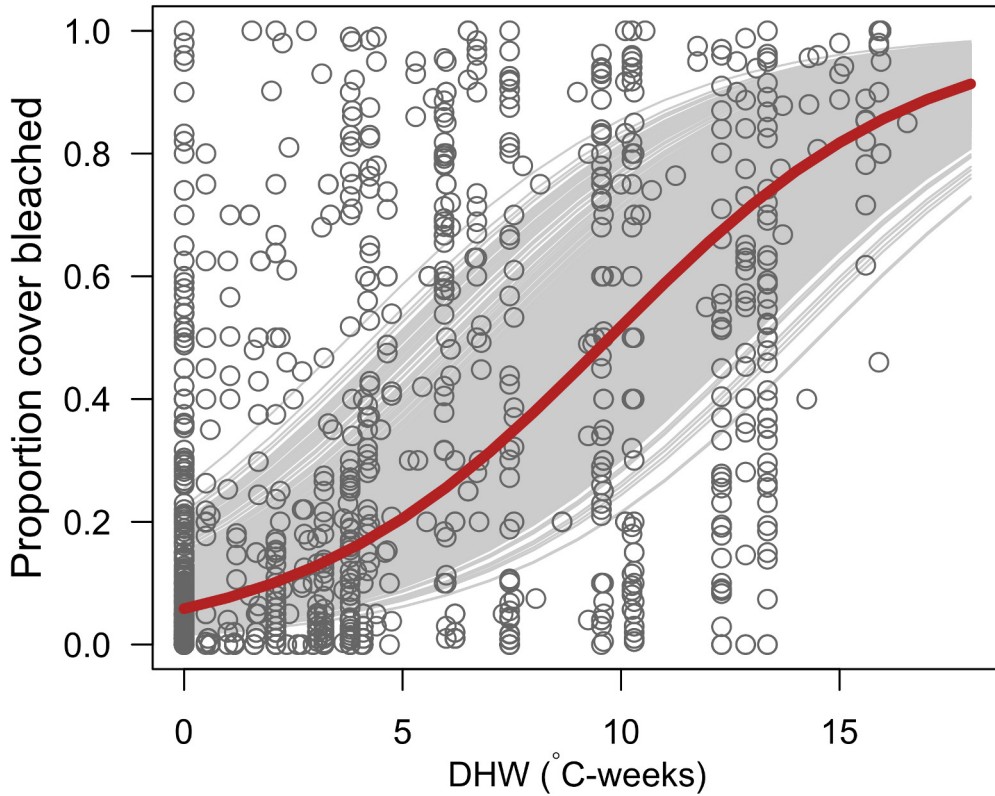

**Appendix 4—figure 13.** Relationship between the proportion of cover bleached in reefs and degree heating week (DHW). Grey circles represent individual observations (n = 1216) in a given site at a certain time (*Eakin et al., 2010*). The grey lines are the logistic regression curves of the general linear mixed model for each combination of sites and time (i.e. random effects; n = 771). The red curve is the logistic regression without random effects (y = 1/(1+exp(2.78–0.29 × DHW))).

## Species-specific bleaching response

We first defined the species-specific probability of bleaching ($P_B$) using the 'intrinsic probability of bleaching' ($IP_B$) and the logistic regression model defined in previous sections. We defined a species-specific logistic response function by adding the coefficient $\gamma_i$ as an intercept to the model. The coefficient $\gamma_i$ is the standardized distance between $IP_{Bi}$ and the averaged $IP_B$, so that the set of logistic regressions is centered around the GLM we established previously (i.e. red line in *Appendix 4—figure 13*). The model for the *i*th species can be expressed as:

$$logit(P_{Bi}) = \alpha + \beta_1 \times DHW + \gamma_i$$

where $\alpha$ and $\beta_1$ = the coefficients of the logistic regression established previously (*Appendix 4—table 4*). We defined $\gamma_i$ as being (1) only dependent on $IP_B$, or (2) also dependent on DHW so that the difference of response between species increases with DHW:

$$\gamma_i = \frac{IP_{Bi} - mean(IP_B)}{max(IP_B) - min(IP_B)} \tag{1}$$

$$\gamma_i = \frac{IP_{Bi} - mean(IP_B)}{max(IP_B) - min(IP_B)} \times \frac{DHW}{\varphi} \tag{2}$$

with *mean*($IP_B$) = 0.27, *min*($IP_B$) = 0.06, *and max*($IP_B$) = 0.60 and are the mean, maximum and minimum *bleaching_probablity* among the 798 coral species, respectively. The coefficient $\varphi \in\, ]0,+\infty[$; we chose $\varphi \in \{2,3,4\}$ as three reasonable values (*Appendix 4—figure 14*) and $\varphi = 2$ provided better fit with the empirical data during the model calibration (Appendix 3).

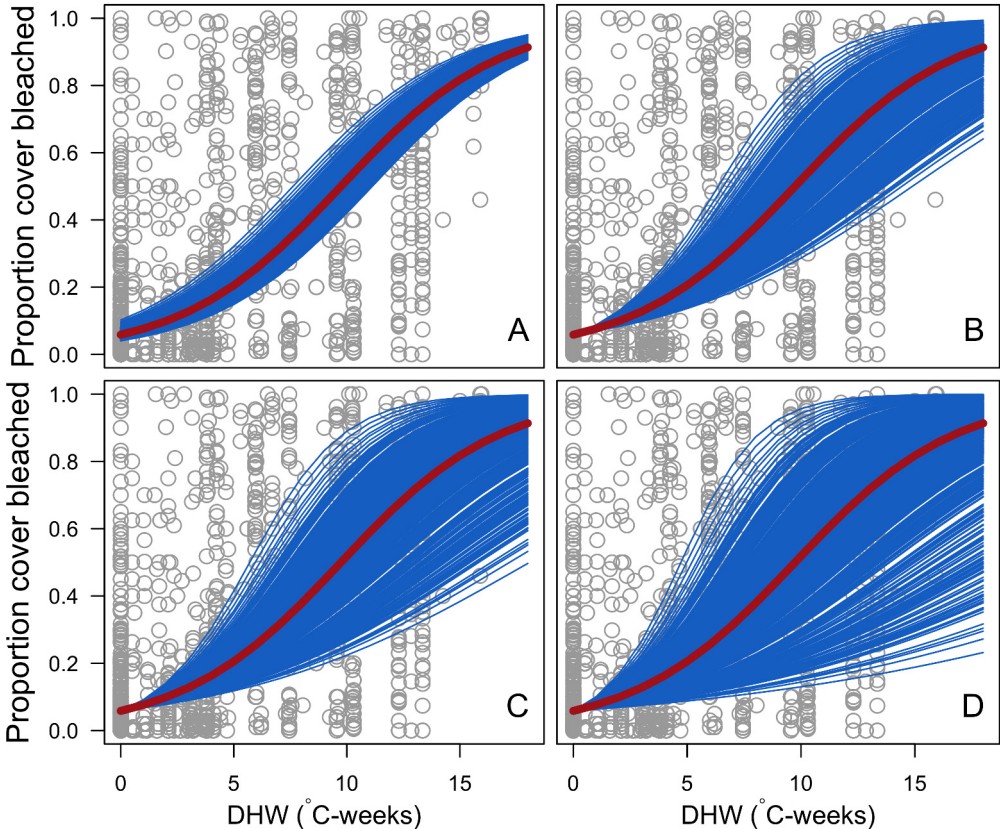

**Appendix 4—figure 14.** Species-specific logistic bleaching responses with $\gamma_i$ being independent of DHW (**A**), dependent of DHW and $\varphi = 4$ (**B**), $\varphi = 3$ (**C**) and $\varphi = 2$ (**D**). Grey circles are individual sampling proportions of coral cover bleached (n = 1216) from Eakin and colleagues' (2010) dataset. The solid red line represents the logistic regression fitting these observations (y = 1/(1+exp(2.78-0.29×DHW))). Blue lines represent the response of each 798 coral species.

## 3. Bleaching-induced mortality probability model

We define here another model to determine the probability that a bleached colony dies from the stress. *Eakin et al. (2010)*'s dataset does not allow to capture the relationship between the degree-heating weeks and the resulting percentage of dead coral cover. We consequently could not define a species-specific probability of mortality. Instead, we defined a unique logistic mortality response assuming the probability equals 0 for no stress (0˚C-week) and 0.85 for a high stress (16˚C-week, *Appendix 4—figure 15*). This latter value corresponds to the proportion of mortality in the most affected part of the Great Barrier Reef after the bleaching event of 2016 (ARC *ARC Center of Excellence For Coral Reef Studies, 2016*), for which degree heating week reached 16 ˚C-week (*Hughes et al., 2017*). The formula is:

$$P_{BD} = \frac{1}{1 + e^{-0.4 \times (DHW - 11.6)}}$$

The probability of mortality is multiplied by two in cases where the coral colony is already bleached when the bleaching event occurs.

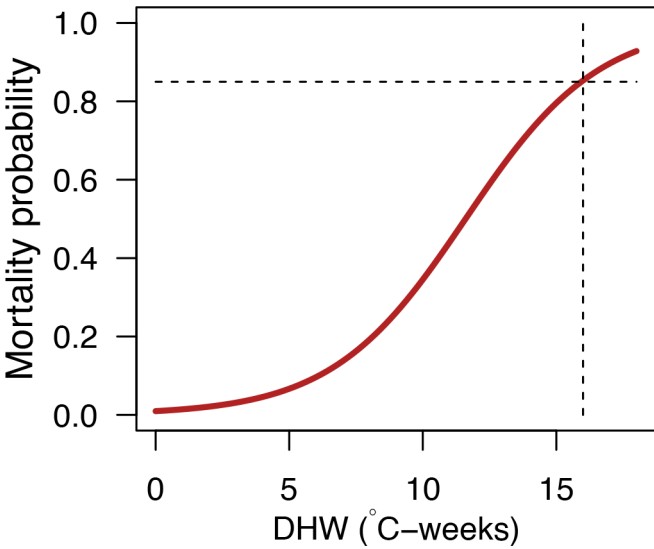

**Appendix 4—figure 15.** General mortality response of a bleached coral colony as a function of degree-heating weeks (DHW) (y = 1/(1 + exp(−0.4 * (x - 11.6)))).

# Appendix 5

## Hierarchically structured validation

We present here the hierarchically structured validation of our coral agent-based model, a procedure proposed by *Kubicek et al. (2015)* to further validate complex models. The goals are to show that the processes are implemented in an ecologically relevant manner, to discuss about eventual unexpected outputs, and more generally, to deeply understand the model behaviors. The procedure consists in assessing the correctness of each process implemented, starting from those happening at the lowest scales, to those affecting the whole system. Each process can be assessed differently: visual and qualitative inspection of patterns, expert knowledge or using statistical comparison with independent data.

The processes of the model we assessed here are: (i) vegetative growth, (ii) recruitment, (iii) responses to disturbances (i.e. resistance and recovery), (iv) larval connectivity and (v) grazing. Additionally, (vi) competition is assessed simultaneously with (iii), (iv) and (v) because competition outcomes are context dependent. For each process, we defined expectations based on ecological knowledge. We then compared the model outputs to these expectations and discuss the results. We based several of the expectations using *Grime (1977)*'s classification of life-history strategies into competitive, stress-tolerant and ruderal (or weedy) functional groups. *Darling et al. (2012)* adapted this classification to coral species: competitive species have large branching or plating growth forms able to overtop other colonies, fast growth rates, and low resistance to hydrodynamics and thermal disturbances; stress-tolerant species have resistant domed morphologies (e.g. massive growth form), slow growth rates and high fecundity; weedy species have for principal characteristic a small colony size and a brooding mode of larval development (weedy species tend to show a diversity of values in the other traits).

### Related code

*Manuscript/Rscripts/ Appendix S5 - Hierarchically structured validation.R*
*Simulations/Rscripts/Hierarchically_structured_validation_Input_dataset.R, Hierarchically_structured_validation.R, Hierarchically_structured_validation_functions.R and Measures_FD_model_functions. R (Carturan et al., 2020).*

## 4. Definition of the functional space

We compared coral interspecific functionality with the following traits: (i) aggressiveness; (ii) colony maximum diameter; (iii) corallite area; (iv) egg diameter; (v) polyp fecundity; (vi) growth rate; (vii) mode of larval development; (viii) sexual system; (ix) bleaching probability; (x) growth form. We converted categorical traits into numerical traits, so each dimension is continuous. For mode of larval development (vii), we attributed one to spawner and negative one to brooder and for sexual system (ix), we attributed one to hermaphrodite and negative one to gonochore.

We represented growth form (x) with two continuous dimensions, which are the first two principal components we obtained using our implemented relations between growth form and the three following processes: reproduction, resistance to cyclones and competition (Appendix 2: §7.2.1.1, 7.3.1.2, 7.5.2.2.b and 7.5.3.2.b, respectively). We first attributed a numerical value to each growth form for each of the three processes. For reproduction, we used the ratio between colony surface and planar area; for resistance to cyclones, we used the slope of the relationship between colony shape factor and colony planar area (on the logarithm scale) (*Appendix 2—table 14*); for competition, we attributed one to growth forms allowing to overtop (i.e. branching and plating) and zero otherwise. We then conducted the principal component analysis (*Appendix 5—figure 1*).

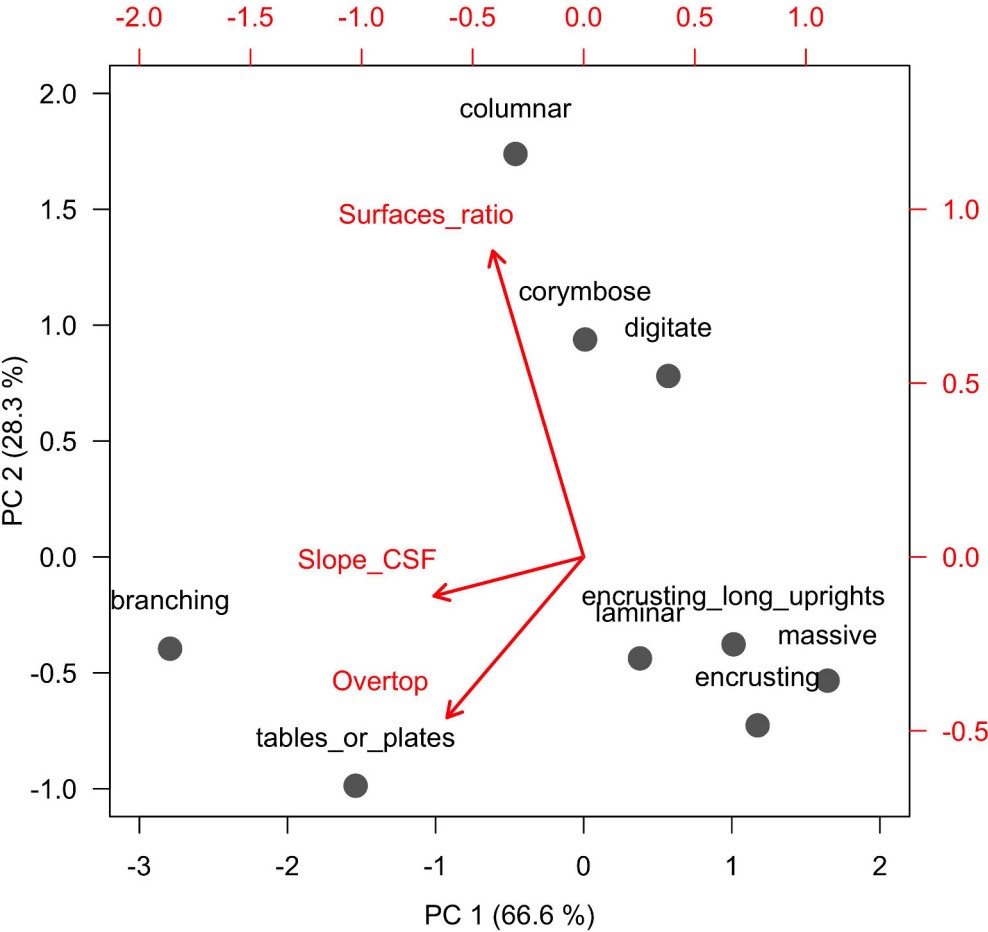

**Appendix 5—figure 1.** Principal component analysis used to quantify growth form on continuous dimensions. Grey dots represent growth forms and red arrows represent the load the different process-related variables (see text for details).

## 5. Vegetative growth and spatial saturation

### 5.1. Objective and procedure

Because the model smallest spatial scale is 1 cm$^2$, and taxa's growth rates have decimals, we implemented vegetative growth by sampling radius values within species-specific intervals and converting agents within this radius, so that continuous averaged growth rates emerge (see Appendix 2: §7.5.1). Here, we verified that (i) population growth rates (i.e., value averaged over all colony's growth rates) equals the species' growth rate, and (ii) colony growth is constrained by the amount of available surrounding space.

We disabled all processes except vegetative growth (i.e. no reproduction, disturbance, larvae connectivity, grazing and algae invasion). The reef contained only one species at a time, which initially covered 30% of total area. We ran simulations for 20 years and measured growth rate at the beginning, the middle and the end. Each time we calculated the averaged individual colony planar growth rates over 3 years. We calculated population growth rate by averaging colony's growth rates over all the colonies present. We repeated the procedure with three coral species having different growth rates (*Appendix 5—figure 2*). We avoided selecting branching or plating species so the colonies cannot overtop each other.

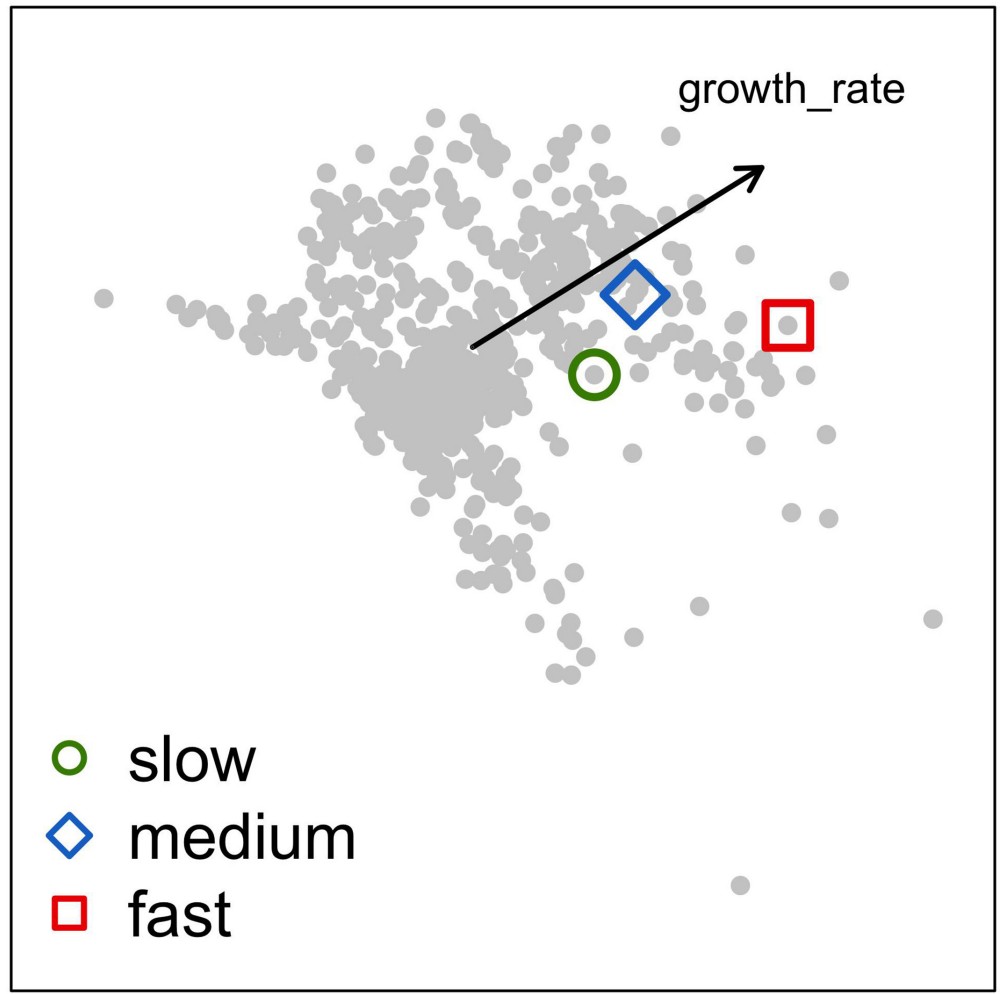

**Appendix 5—figure 2.** Principal component analysis of the coral functional space (shown with the first two principal components) and the three species selected to assess growth rate and space saturation processes: *Acropora pulchra* (red square; fast: 65.5 mm.yr$^{-1}$); *A. polystoma* (blue diamond; medium: 37.3 mm.yr$^{-1}$); *A. gemmifera* (green circle; slow: 12.0 mm.yr$^{-1}$). Each grey dot represents one of the 798 species. The black arrow represents the load of the trait growth rate.

## 5.2. Expected patterns

At first, colonies are small and most of them separated from each other. Population growth rate should consequently approximate species growth rate. As colonies grow, space becomes saturated, colonies touch each other, and population growth rate should decrease until eventually reaching zero. Fastest species should saturate the space more rapidly.

## 5.3. Results

Expectations are met: at the beginning of the simulation, population growth rates approximated species growth rates (*Appendix 5—figure 4*; left column). As colonies grew, space became saturated (*Appendix 5—figure 3*) and population growth rates decreased until eventually reaching zero for the fastest species (*Appendix 5—figure 4*).

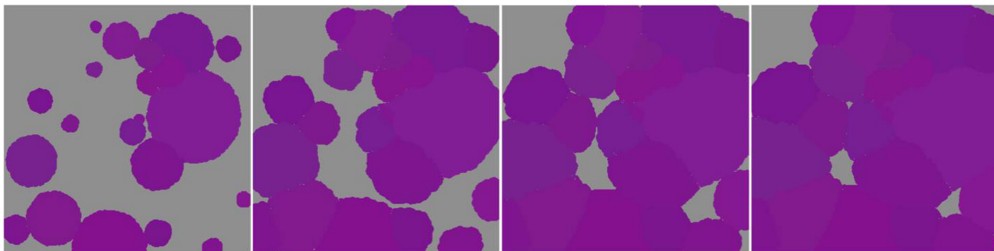

**Appendix 5—figure 3.** Space saturation due to the vegetative growth of a unique coral species in absence of any other processes (colonies are distinguished by different shades of purple). We took screen shots of the $2 \times 2$ m$^2$ at different time intervals.

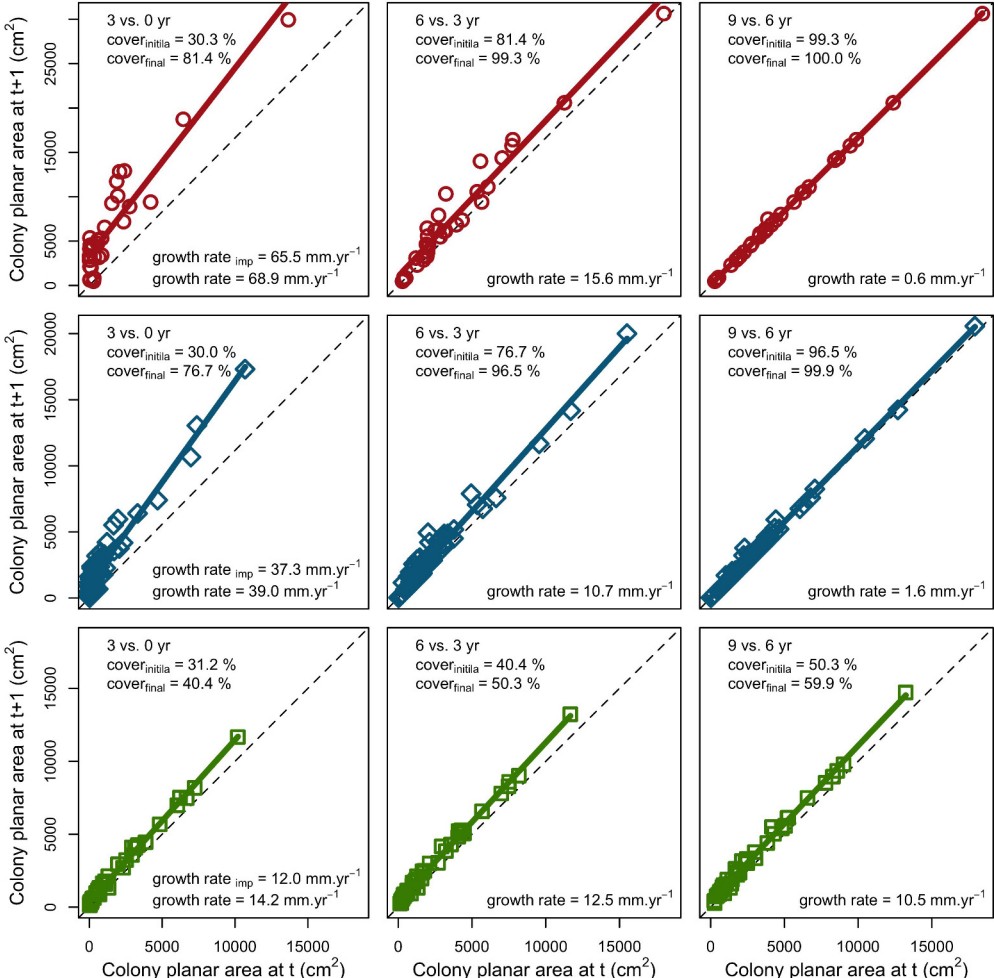

**Appendix 5—figure 4.** Effect of space saturation on coral colony growth rate saturation for three coral species, growing in a monospecific reef: *Acropora pulchra* (red square); *A. polystoma* (blue diamond) and *A. gemmifera* (green circle). The implemented growth rate (growth rate$_{imp}$) corresponds to the values found in the literature. The dashed lines indicate no growth.

## 6. Recruitment and spatial saturation

### 6.1. Objective and procedure

Here, we verified that population recruitment rate depends on the (i) species fecundity, (ii) mode of larval development (brooding species reproduce more often than spawning species), (iii) surface

covered, (iv) size distribution of the colonies (because smaller colonies have a lower proportion of mature polyps; see Appendix 2: §7.2.1.1), and (v) amount of available space for settlement.

We kept the same model configuration as in the previous section and enabled coral reproduction. The reef was occupied by only one coral species at a time, which initially covers 10% of the reef. We ran simulations for 20 years and recorded number of new coral recruits and percentage cover each time step (six months). We compared results between three species having different life history strategies, based on *Darling et al. (2012)* classification: (1) competitive *Acropora gemmifera*, (2) weedy *Agaricia tenuifolia*, (3) stress-tolerant *Echinophyllia orpheensis* (*Appendix 5—figure 5*). We replicated the simulations five times for each species.

To easily compare species fecundity, we defined 'colony fecundity' (number of eggs cm$^{-2}$ of colony planar surface area) such as:

$$colony\ fecundity = \frac{polyp\ fecundity \times ratio_{3D/2D} \times coefficient_{sexual\ system}}{corallite\ area}$$

with

$$coefficient_{sexual\ system} = \begin{cases} 0.5\ for\ gonochore \\ 1.0\ for\ hermaphrodite \end{cases}$$

The coefficient $ratio_{3D/2D}$ is the colony surface area divided by the colony planar surface area, based on the geometric formulae we used (*Appendix 2—table 5*) and assuming that planar surface area is circular.

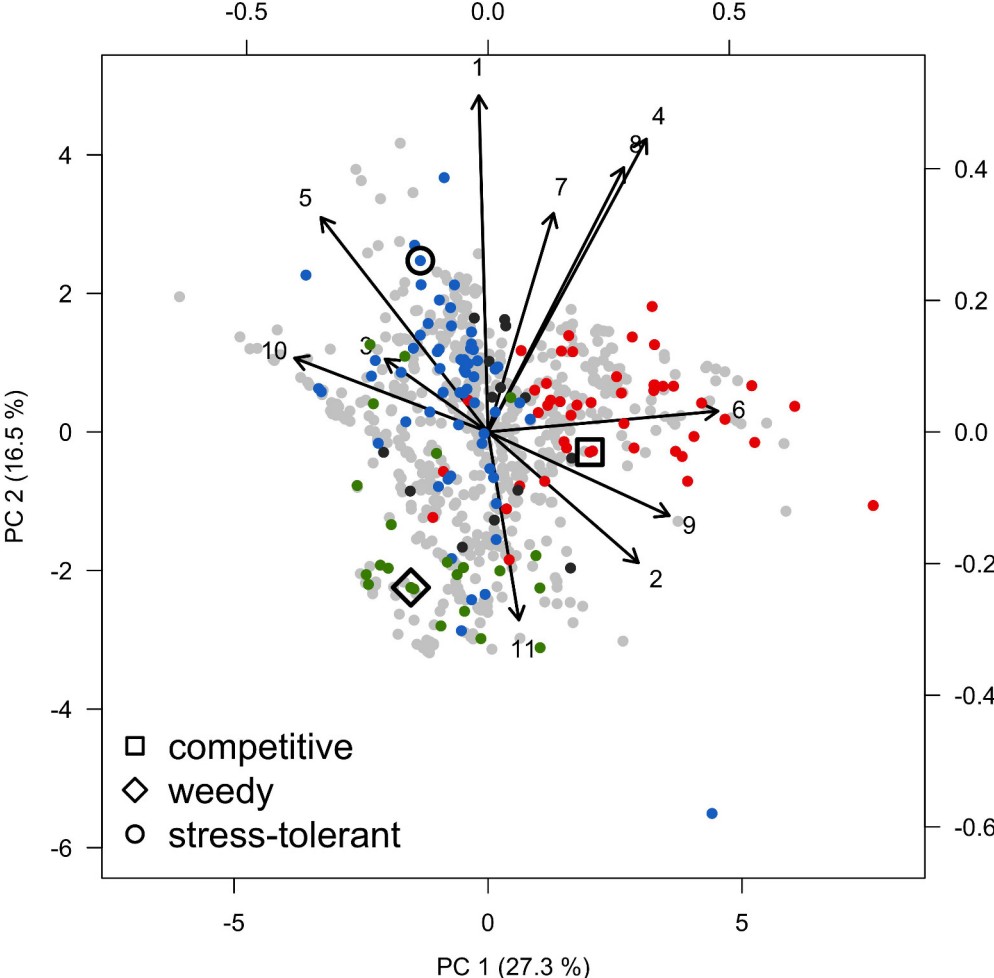

**Appendix 5—figure 5.** Principal component analysis of the coral functional space (shown with the

first two principal components) and the three species selected to assess recruitment rate and space saturation processes: *Acropora gemmifera* (competitive, red dot in square); *Agaricia tenuifolia* (weedy, green dot in diamond); *Echinophyllia orpheensis* (stress-tolerant, blue dot in circle). Each grey dot represents one of the 798 species, colored dots are species classified either as competitive (red), weedy (green), stress-tolerant (blue) generalist (black) by *Darling et al. (2012)*. Arrows indicate trait loadings; numbers correspond to: (1) aggressiveness; (2) colony maximum diameter; (3) corallite area; (4) egg diameter; (5) polyp fecundity; (6) growth rate; (7) mode of larval development; (8) sexual system; (9) bleaching probability; (10) growth form PC1; (11) growth form PC2.

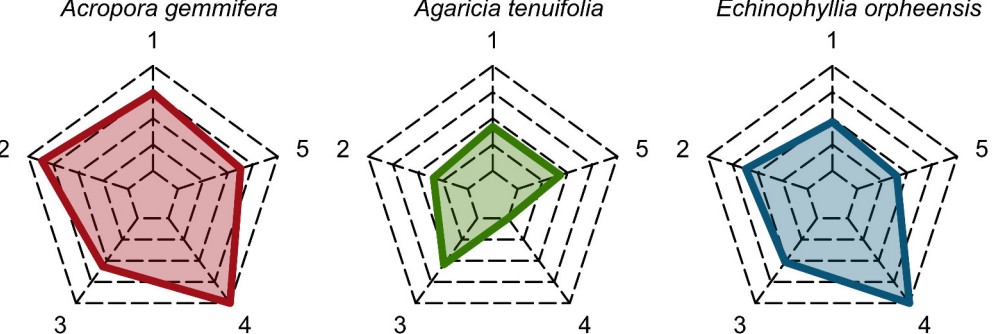

**Appendix 5—figure 6.** Functional characteristics of the three species selected to assess recruitment rate and space saturation processes (from left to right: competitive in red, weedy in green and stress-tolerant in blue). Each vertex of the web corresponds to a trait: (1) colony maximum diameter $(\log_{10})$; (2) egg diameter (mm); (3) colony fecundity (no. eggs.cm$^{-2}$, $\log_{10}$); (4) mode of larval development; (5) growth rate $(\log_{10})$. The colored polygons situate the trait value of the species in the range of values spanned by the 798 species, with highest values at the extremities and lowest values near the center of the web.

## 6.2. Expected patterns

At the beginning of the simulation, the percentage cover occupied is low and the colonies are smaller. Recruitment rate should increase as surface cover increase and colonies become larger and more fecund. After the population reaches a certain percentage cover, available space becomes more limited and recruitment rate decreases. The increase and decrease in recruitment rate should be proportional to growth rate because colonies reach larger sizes, higher fecundity and fill up available space faster. The initial colony size distribution, which depends on species' *maximum colony diameter* (Appendix 2: §5.2), and *colony fecundity* are positively associated with recruitment rate.

## 6.3. Results

Expectations are met. An increase followed by a decrease in recruitment is observed for *A. tenuifolia* (weedy) and *E. orpheensis* (stress-tolerant) but not for *A. gemmifera* (competitive), whose recruitment only decreased (*Appendix 5—figure 7*). All three species have similar colony fecundity (trait 3 in *Appendix 5—figure 6*) but *A. gemmifera* has by far the highest colony maximum diameter (trait 1 in *Appendix 5—figure 6*). Consequently, its initial colony size distribution is characterized by fewer but larger colonies, with one colony belonging to $[10^4, 10^{4.7}[$ and one to $[10^{4.7}, 10^5[$ cm$^2$; in comparison, the other species' colonies are all $<10^4$ cm$^2$ (*Appendix 5—figure 8*). Larger colonies containing a higher proportion of fecund polyps (Appendix 2: §7.2.1.1.b), *A. gemmifera* was able to recruit more the first year (*Appendix 5—figure 7*, *Appendix 5—figure 8*). In combination with a higher growth rate (trait 5 in *Appendix 5—figure 6*), the species was able to fill up the available space more rapidly (i.e. in six years). Stress-tolerant species *E. orpheensis* had initially a few more of the largest colonies compared to weedy *A. tenuifolia* (*Appendix 5—figure 8*), which is due to its slightly higher maximum colony diameter and results in a higher initial number of recruits (*Appendix 5—figure 7*). It surpassed the weedy species' maximum recruitment rate because it filled up space less rapidly. Indeed, the weedy species has a slightly faster vegetative growth rate (trait 5 in *Appendix 5—figure 6*) and reproduced twice a year due to its brooding model of larval development.

This section shows that colony size influences importantly recruitment rate. Consequently, expressing probability of polyp fecundity as a function of colony size in a non-species-specific model (Appendix 2: §7.2.1.1) is a potentially important limitation of the model. Future improvement should consist in defining realistic species-specific models, providing that empirical data and evidences will be available.

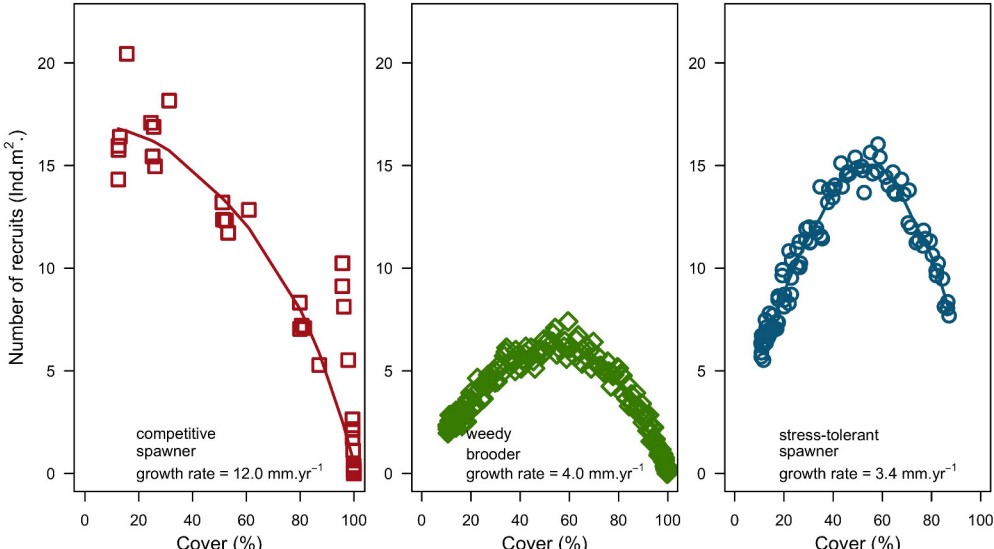

**Appendix 5—figure 7.** Recruitment rate and space saturation for three coral species, growing in a monospecific reef: *Acropora gemmifera* (competitive, red dot in square); *Agaricia tenuifolia* (weedy, green dot in diamond); *Echinophyllia orpheensis* (stress-tolerant, blue dot in circle). Each symbol indicates the cover and number of recruits measured at a given time interval. Simulations were replicated five times. Trend lines are shown for visual aid. Also displayed is the species radial growth rate.

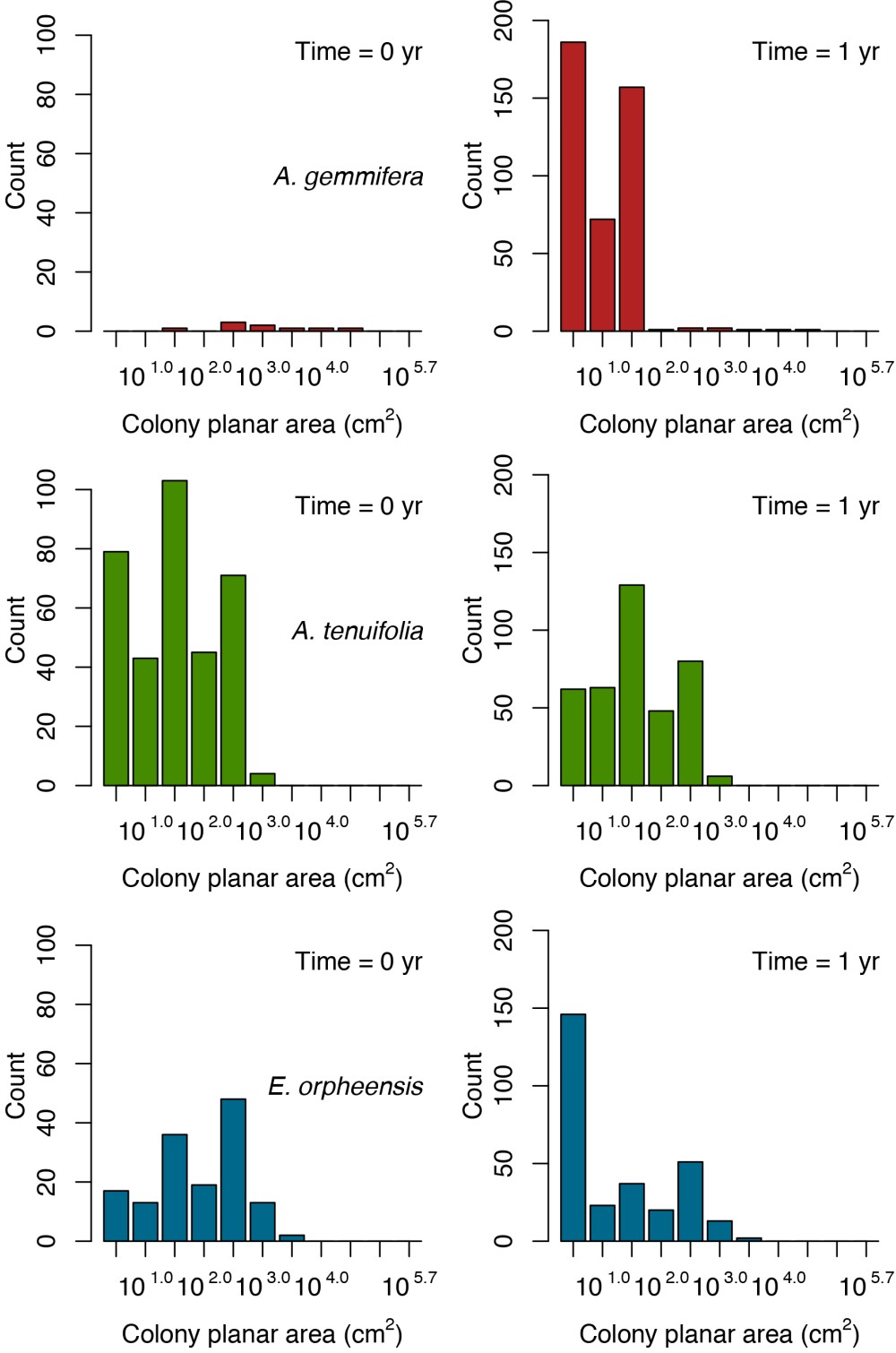

**Appendix 5—figure 8.** Initial colony size-frequency distributions of the three species grown in a monospecific reef for the recruitment rate and space saturation simulations. Frequencies were averaged over five replicates.

## 7. Disturbance intensity

### 7.1. Objective and procedure

Here, we verified that coral species fitness depends on their functional characteristics and the intensity of the disturbance regimes defining their environment. In particular, we want to identify the mechanisms generating the results and assess their ecological relevance.

We enable all the processes in the simulations. The reef was occupied by three coral species having different life history strategies, macroalgae, turf and crustose coralline algae, all covering initially 10% of the reef. We exposed the community to four different disturbance regimes, from low to high wave exposure ($DMT_{background}$) and bleaching intensity ($DHW_{bleaching}$): (1) $DMT_{background} = 120$ and $DHW_{bleaching} = 0$ °C-weeks; (2) $DMT_{background} = 100$ and $DHW_{bleaching} = 5$ °C-weeks; (3) $DMT_{background} = 80$ and $DHW_{bleaching} = 10$ °C-weeks; (4) $DMT_{background} = 60$ and $DHW_{bleaching} = 15$ °C-weeks. The other two factors are kept constant: grazing at 50% and larval connectivity at 'medium (10 km)' levels (7000 larvae.m$^{-2}$). We ran simulations for 20 years with a bleaching event happening at year seven. We replicated each treatment five times.

Because functional diversity is another factor influencing the model's outputs, we did the experiment with two different coral communities, which we assembled by selecting species from Western Atlantic and Eastern Pacific (*Appendix 5—figure 9*, *Appendix 5—figure 10*).

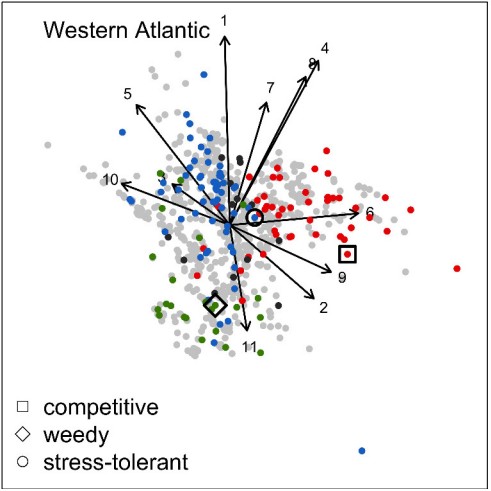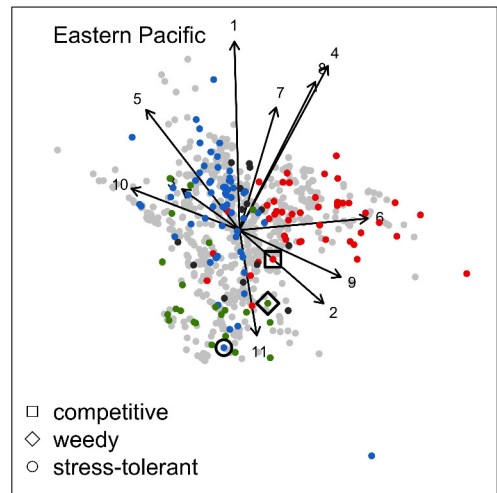

**Appendix 5—figure 9.** Principal component analyses of the coral functional space (shown with the first two principal components) and the three species selected to assess response to disturbances, larval connectivity, grazing and competition. Species from the Western Atlantic (left) are: *Acropora palmata* (competitive, red dot in square); *Madracis pharensis* (weedy, green dot in diamond); *Orbicella annularis* (stress-tolerant, blue dot in circle). Species from the Eastern Pacific (right) are: *Pocillopara elegans* (competitive, red dot in square); *P. damicornis* (weedy, green dot in diamond); *Porites lutea* (stress-tolerant, blue dot in circle). Each grey dot represents one of the 798 species, colored dots are species classified either as competitive (red), weedy (green), stress-tolerant (blue) or generalist (black) by *Darling et al. (2012)*. Arrows indicate trait loadings; trait numbers correspond to: (1) aggressiveness; (2) colony maximum diameter; (3) corallite area; (4) egg diameter; (5) polyp fecundity; (6) growth rate; (7) mode of larval development; (8) sexual system; (9) bleaching probability; (10) growth form PC1; (11) growth form PC2.

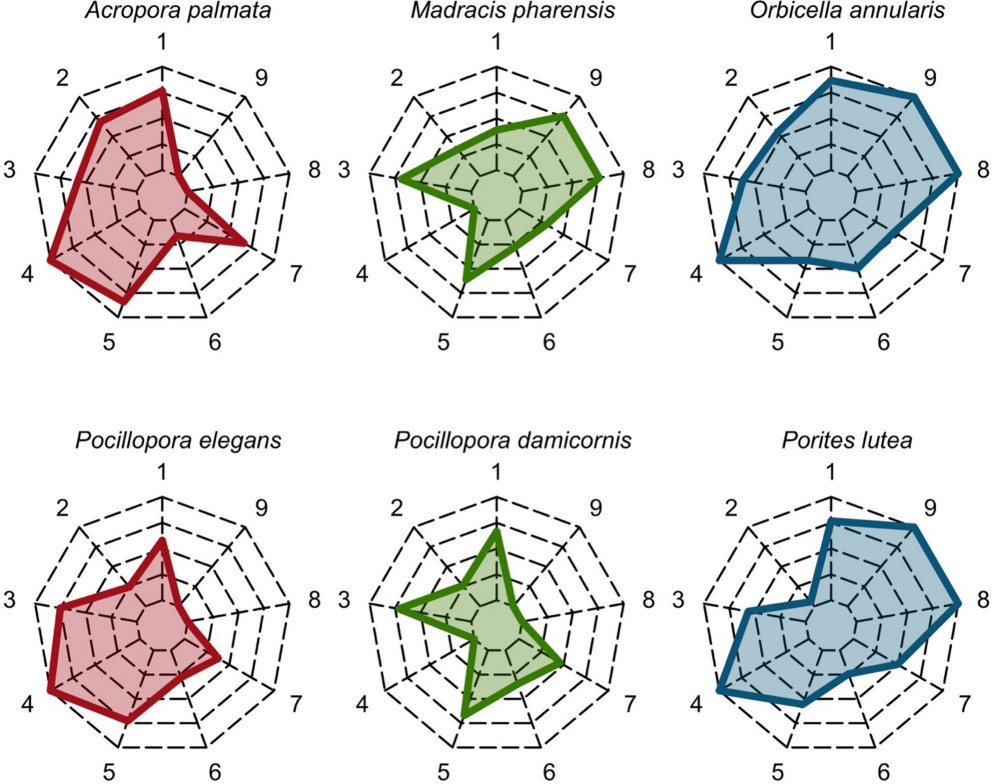

**Appendix 5—figure 10.** Functional characteristics of the species selected to assess recruitment rate and space saturation processes (the Western Atlantic and Eastern Pacific species are at the top and bottom, respectively; from left to right: competitive in red, weedy in green and stress-tolerant in blue). Each vertex of the web corresponds to a trait: (1) colony maximum diameter ($\log_{10}$); (2) egg diameter (mm); (3) colony fecundity (no. eggs.cm$^{-2}$, $\log_{10}$); (4) mode of larval development (i.e. one for spawner, zero for brooder); (5) growth rate ($\log_{10}$); (6) aggressiveness; (7) bleaching susceptibility; (8) growth for PC1 (i.e. resistance to hydrodynamics disturbances and capacity to overtop), (9) growth form PC2 (i.e. 3D to 2D surface ratio). The colored polygons situate the trait value of the species in in the range of values spanned by the 798 species, with highest values at the extremities and lowest values near the center of the web.

## 7.2. Expected patterns

Under low wave exposure, we expect the community to be dominated by the competitive species because of its faster growth rate and its capacity to overtop smaller colonies due to its potentially large and branching colonies. As wave intensity increases, the colonies of the competitive species are dislodged once they reach a certain size, preventing them from overtopping other colonies and providing more space to the other two species. Contrastingly, the cover of the stress-tolerant species should increase because of the high resistance of its colonies, which should become larger and more fecund. Finally, the fate of the weedy species under higher waves exposure depend on its capacity to compete with the other species. Competition can take different forms and involves different traits: (i) direct in the case of physical contact between colonies (aggressiveness and growth rate are involved), (ii) indirect, via competition with algae (growth rate and growth form are involved), pre-emption of available space due to (iii) vegetative growth (growth rate and growth form are involved), and (iv) via recruitment (polyp fecundity and size, growth form, mode of larval development and sexual system are involved).

The bleaching event happening at year 7 should affect more the species having the highest susceptibility, which in the model depends on the traits growth rate, corallite area (i.e. polyp size), microscopic reduced-scattering coefficient, and maximum colony diameter (Appendix 4). If the most bleaching susceptible species is the one dominating the community, we expect a switch of species composition after the bleaching event. However, such a change in species assembly dependents on

the intensity of the thermal stress, the level of larval connectivity and the amount of available space for larvae to settle (which also depends on grazing pressure).

## 7.3. Results

### Western Atlantic

The competitive species (*A. palmata*, in red) dominated the community under lower wave exposure (*Appendix 5—figure 11*). Having higher growth rate (trait 5 in *Appendix 5—figure 10*), its colonies rapidly pre-empted space and reached larger colony sizes, which hampered the other two species from growing and recruiting. In addition, many of its colonies were large enough to overtop other smaller colonies (*Appendix 5—figure 12*). By occupying a high surface cover and owning most of the largest colonies, *A. palmata* recruited importantly (*Appendix 5—figure 11*). The colony fecundity of the other two species is not superior enough to compensate for their slower growth rate (trait 3 in *Appendix 5—figure 10*).

Under more intense wave regimes, *A. palmata*'s largest colonies were systematically dislodged (*Appendix 5—figure 13*) due to the low resistance of branching growth forms (trait 8 in *Appendix 5—figure 10*). Its cover consequently decreased and weedy species *M. pharensis* became the most abundant species under the highest wave exposure (*Appendix 5—figure 11*) because of the wave-resistance of its digitate growth form and its relatively fast growth rate (respectively trait 8 and 5 in *Appendix 5—figure 10*). Reaching larger colonies, *M. pharensis* recruited more and could not be overtopped by *A. palmata* (*Appendix 5—figure 13*). Being more aggressive, *M. pharensis* also won most of its interactions with *A. palmata* (trait 6 in *Appendix 5—figure 10*).

Despite being the most resistant and aggressive species (respectively traits 8 and 6 in *Appendix 5—figure 10*), the stress-tolerant species *O. annularis* was only able to persist due to external larval supply (*Appendix 5—figure 11*). Its lower growth rate compared to the other two species (trait 5 in *Appendix 5—figure 10*) was a disadvantage for several reasons. First, *O. annularis* grows over less available surface when directly competing for space with the other coral species. Second, corals also compete against algae, especially with turf, which is the most dominant algae in the model (e.g., *Appendix 5—figure 20*, *Appendix 5—figure 25*). Turf has the fastest growth rate and wins 100% of its interactions with corals (Appendix 2: §7.5.3). Under the grazing regime imposed in these simulations (50%), *O. annularis* lost more surface in its competition with turf than it gained by growing vegetatively. Third, *O. annularis* recruited less (*Appendix 5—figure 11*) because of the smaller sizes of its colonies (*Appendix 5—figure 12*, *Appendix 5—figure 13*).

The bleaching disturbance happening at year seven affected coral species accordingly to their thermal sensitivity (*Appendix 5—figure 10*) — *A. palmata* cover decreased dramatically, even after a mild event, whereas *M. pharensis* and *O. annularis* were less affected in comparison (*Appendix 5—figure 11*). Recruitment rate increased just after the bleaching event because numerous dead colonies provided suitable larval settlement substrate.

Species recovered pre-cover level approximately one year after the disturbance, which is faster than the five to ten years usually observed in real ecosystems (*Pratchett et al., 2009* but see *Diaz-Pulido et al., 2009*). This is due to the high and constant number of larvae coming from the regional pool (7000 larvae m$^{-2}$). In reality, an intense bleaching event (>10 C°-weeks) would affect large geographic areas and consequently reduce larval supply and recovery rates (e.g. *Hughes et al., 2019*). If needed, the number of coral larvae to input at each time step can be defined before launching simulations.

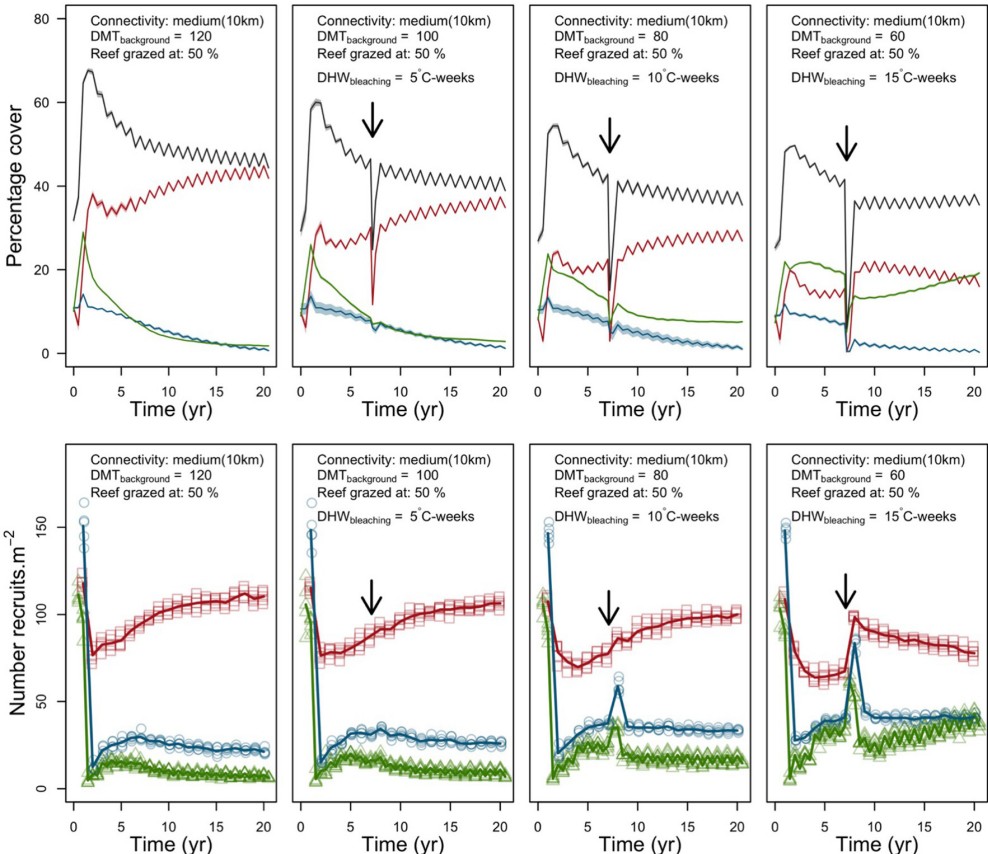

**Appendix 5—figure 11.** Evolution of cover (top) and number of recruits (bottom) of the three coral species from Western Atlantic (red for the competitive *Acropora palamata*, green for the weedy *Madracis pharensis*, blue for the stress-tolerant *Orbicella annularis* and black for total coral cover) for different disturbance regimes. The black arrow indicates when the bleaching event occurred. Solid lines represent mean values over five replicates; shaded areas at the top show ± SE; individual symbols at the bottom show the number of recruits at a given time for one replicate.

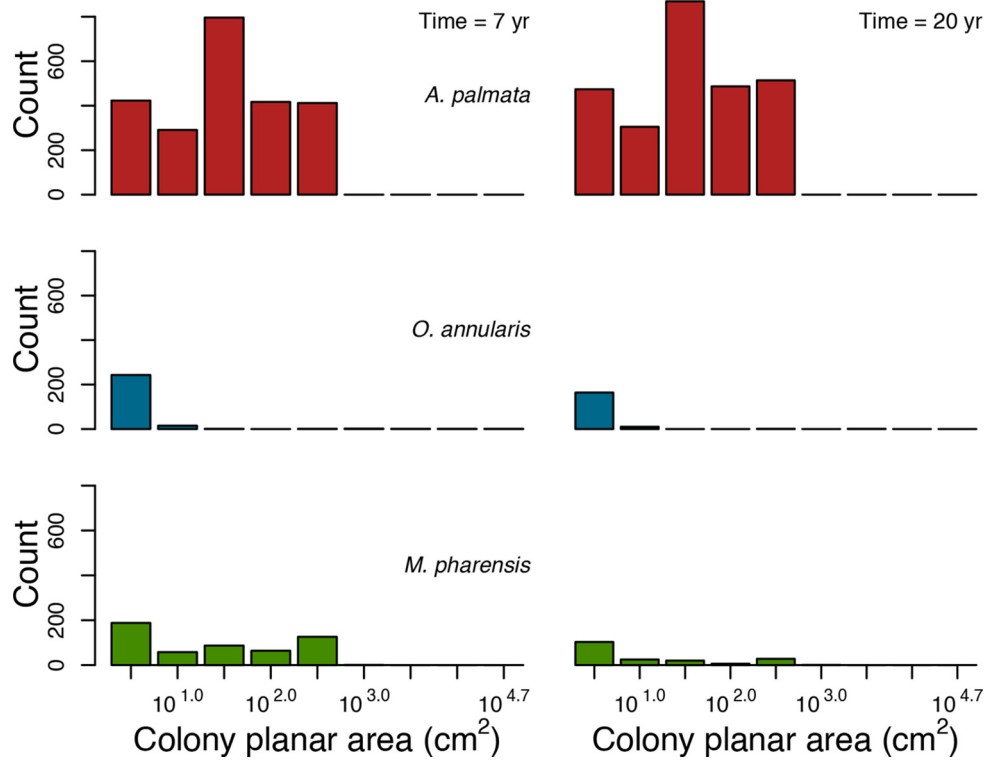

**Appendix 5—figure 12.** Colony size-frequency distributions of the Western Atlantic species at the lowest wave exposure ($DMT_{background}$ = 120) and with no thermal disturbance ($DHW_{bleaching}$ = 0 C°-weeks). Colonies with a colony planar area = 1 cm$^2$ are not displayed.

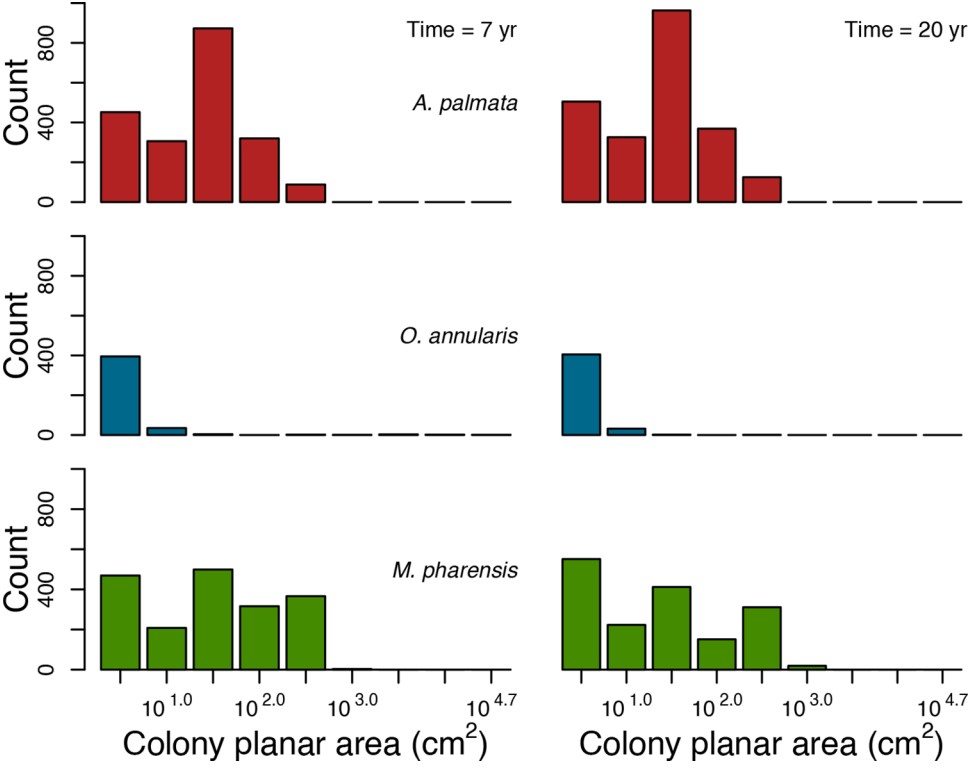

**Appendix 5—figure 13.** Colony size-frequency distributions of Western Atlantic species at the highest wave exposure ($DMT_{background}$ = 60) and thermal disturbance ($DHW_{bleaching}$ = 15 C°-weeks). Colonies with a colony planar area = 1 cm$^2$ are not displayed.

## Eastern pacific

The competitive species *P. elegans* dominated the community regardless of the intensity of the disturbance regime (*Appendix 5—figure 14*). Its cover decreased under higher wave regimes, but the species could recruit more than the two other species. The three species coexisted under the highest wave regime – weedy species *P. damiconis* reached a stable population size and structure and the cover of the stress-tolerant species *P. lutea* slowly increased (*Appendix 5—figure 16*).

The competitive species dominated the weedy species because of its mode of larval development – *P. elegans* is a spawning species and receives three times more larvae from the regional pool compared to *P. damicornis* (Appendix 2: §7.2.1.2), which explains its higher recruitment rate (*Appendix 5—figure 14*). Other traits played a minor role in the competition between the two species because of their high functional similarity (*Appendix 5—figure 10*).

The stress-tolerant species *P. lutea* did not establish permanently other than under the highest wave regime because of its lower growth rate and colony fecundity compared to the competitive species (respectively traits 5 and 3 in *Appendix 5—figure 10*). Unable to reach larger colony sizes (*Appendix 5—figure 15*), *P. lutea*'s population persisted because of the high input of larvae from the regional pool (*Appendix 5—figure 14*). Under the highest wave regime, the two other branching species could not reach large colony size (*Appendix 5—figure 16*), leaving more space to *Porites lutea*'s resistant colonies to grow (trait 8 in *Appendix 5—figure 10*). Having more large colonies (*Appendix 5—figure 16*), *P. lutea* recruitment rate slightly increased (*Appendix 5—figure 14*).

The three species have a similar bleaching susceptibility (trait 7 in *Appendix 5—figure 10*) and were similarly affected by the thermal disturbance (*Appendix 5—figure 14*). Compared to the two other species, *P. lutea* failed to rapidly recover pre-disturbance cover after bleaching because of its slow growth rates (traits 5 in *Appendix 5—figure 10*).

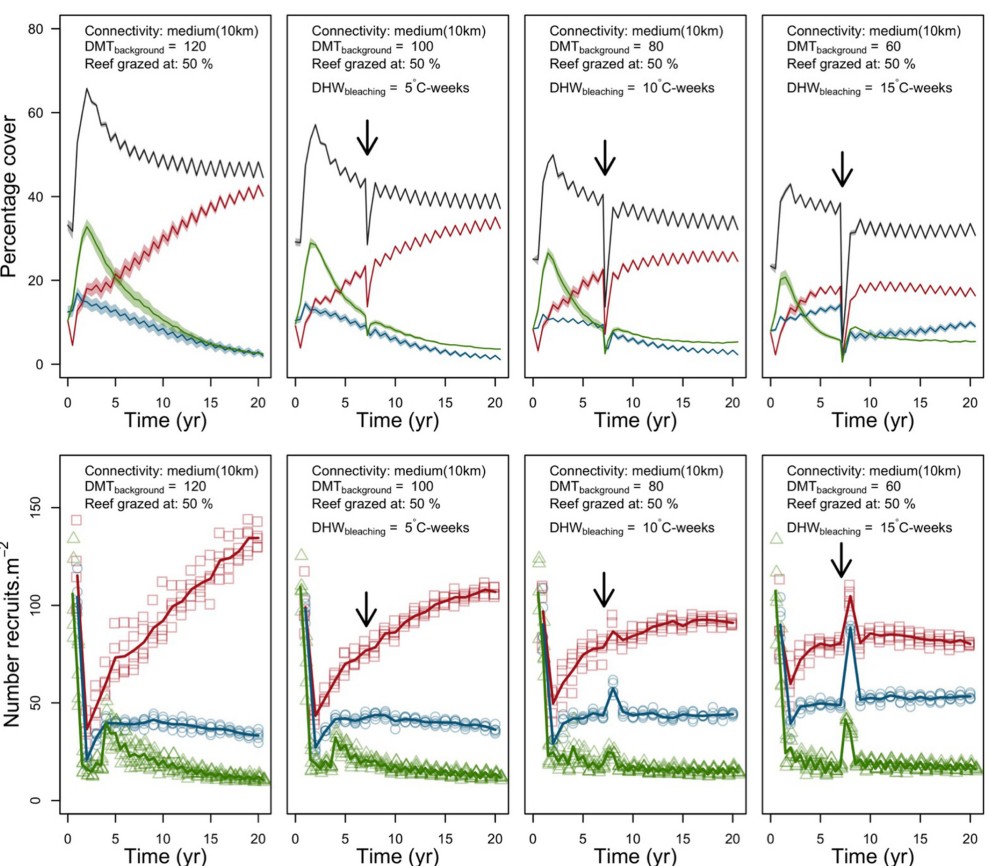

**Appendix 5—figure 14.** Evolution of the cover (top) and the number of recruits (bottom) of the three coral species from Eastern Pacific (red for the competitive *Pocillopara elegans*, green for the weedy *Pocillopora damicornis*, blue for the stress-tolerant *Porites lutea* and black for total coral cover) for different disturbance regimes. The black arrow indicates when the bleaching event

occurred. Solid lines represent mean values over five replicates; shaded areas at the top show ± standard error; individual symbols at the bottom show the number of recruits at a given time for one replicate.

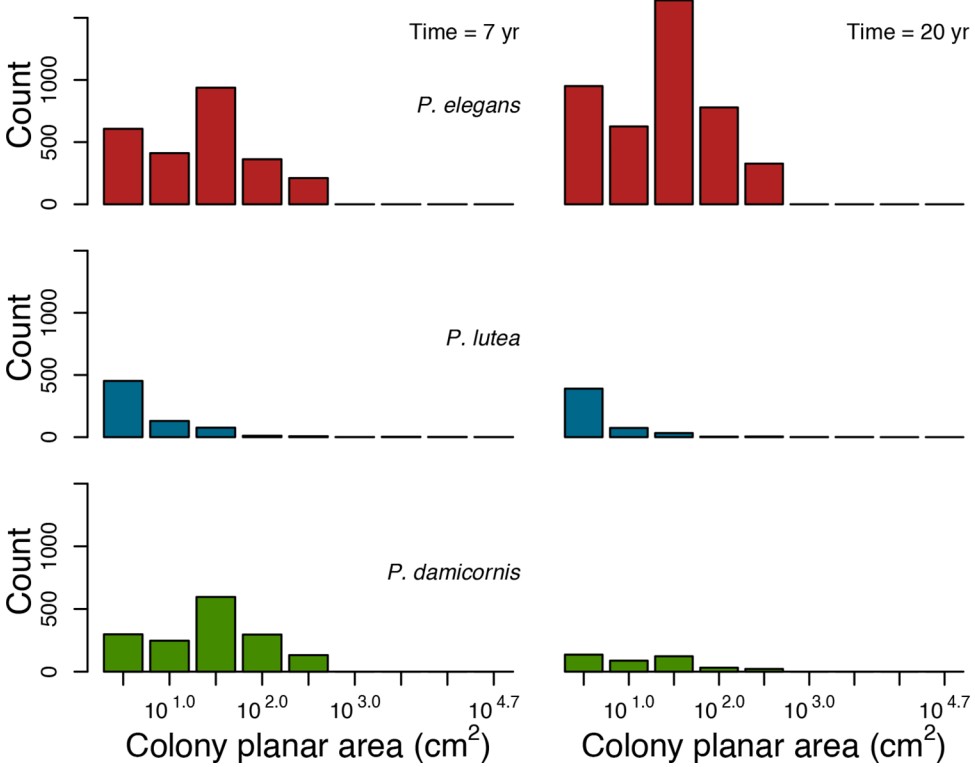

**Appendix 5—figure 15.** Colony size-frequency distributions of the Eastern Pacific species at the lowest wave exposure ($DMT_{background} = 120$) and with no thermal disturbance ($DHW_{bleaching} = 0$ C°-weeks). Colonies with a colony planar area = 1 cm² are not displayed.

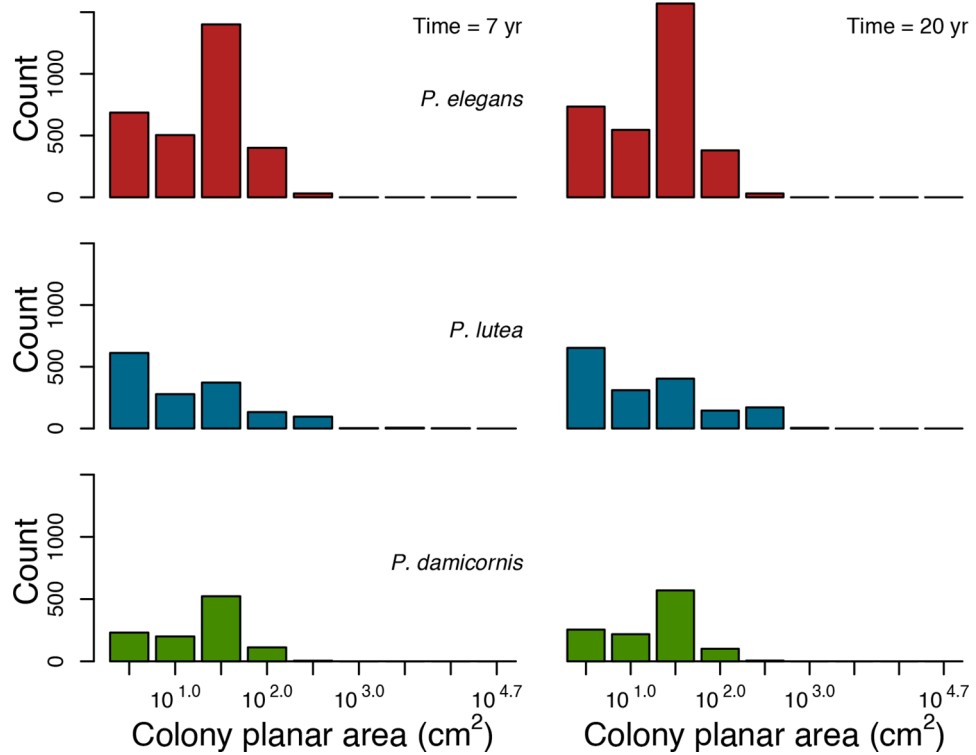

**Appendix 5—figure 16.** Colony size-frequency distributions of the Eastern Pacific species at the highest wave exposure ($DMT_{background}$ = 60) and thermal disturbance ($DHW_{bleaching}$ = 15 C°-weeks). Colonies with a colony planar area = 1 cm² are not displayed.

## 8. Connectivity

### 8.1. Objective and procedure

Here we assessed how larval connectivity influences coral population dynamics. We enabled all the processes and used the same coral species, and the same initial percentage cover as in the previous section. We ran simulations for 20 years and triggered the bleaching event at year 7. We exposed the communities to five different levels of larval connectivity: (i) 'no connectivity', (ii) 'isolated (100 km)' (66.5 larvae.m$^{-2}$), (iii) 'low (20 km)' (700.0 larvae.m$^{-2}$), (iv) 'medium (10 km)' (7000 larvae.m$^{-2}$); (5) 'high (5 km)' (35,000 larvae.m$^{-2}$). We kept constant grazing pressure, wave exposure and bleaching intensity (50%, $DMT_{background}$ = 80, $DHW_{bleaching}$ = 10 C°-weeks, respectively).

### 8.2. Expected patterns

Recruitment rate depends on the number of larvae settling in the reef and the amount of available space. The number of larvae a population produces locally depends on (i) the surface cover occupied, (ii) the size of the colonies, and (iii) the species investment in sexual reproduction. Corals have different strategies enhancing local recruitment rate, such as high colony fecundity, multiple reproductive cycles per year (a strategy mostly observed in brooding species, *Ritson-Williams et al., 2009*), and brooding larvae (which are ready to settle just after realise).

Under no or low larval connectivity, species having one or several of these strategies have an advantage recovering from a disturbance and maintaining their population under stressful conditions (provided they are resistant enough). Under higher larval connectivity, the capacity to self-recruit locally is less critical because larval supply is not uniquely supported by local populations. Other traits become more important, such as growth rate and the capacity to overtop other colonies in protected locations and resistance to disturbances in more exposed ones.

### 8.3. Results

### Western Atlantic

There was a clear contrast in species dominance between low and high larval connectivity (*Appendix 5—figure 17*). Under no or low connectivity, weedy species *M. pharensis* competitively excluded the other two species, mainly because it was better at recruiting locally. *Madracis pharensis* could reproduce twice a year, and all its larvae remained in the reef because of its brooding mode of larval development. Contrastingly, *A. palmata* and *O. annularis* reproduce once a year and only a portion of their larvae remains in the reef. This proportion depends on water retention time (4.69 days) and time to motility of the larvae, which is determined by egg diameter (Appendix 2: §7.2.1.1.d). With relatively large egg diameter (trait 2 in *Appendix 5—figure 10*), only one third of *A. palmata* and *O. annularis* remained in the reef.

Despite its higher growth rate, the competitive species *A. palmata* became competitively excluded because of its lower aggressiveness, colony fecundity, and resistance to waves (respectively traits 6, 3 and 8 in *Appendix 5—figure 10*). In comparison, the stress-resistant species *O. annularis* resisted better because of its higher aggressiveness but was eventually excluded because it was incapable of recruiting (*Appendix 5—figure 17*) and could not compensate for the cover loss in its competition with turf.

Under higher connectivity, producing a large number of larvae was not the most important strategy and *A. palmata* competitively excluded the other two species (*Appendix 5—figure 17*) because of its faster growth rate and its capacity to overtop smaller colonies (see §4.3 for more details).

*Madracis pharensis* and *A. palmata* recovered rapidly after bleaching under low and high connectivity, respectively, by using different strategies. *Madracis pharensis* is the least bleaching susceptible species (trait 7 in *Appendix 5—figure 10*) and consequently lost less than 50% cover after the disturbance; its unbleached surviving colonies were able to reproduce every 6 months after bleaching. Contrastingly, *A. palmata* was much more impacted and recovered due to the high input of larvae and its fast growth rate.

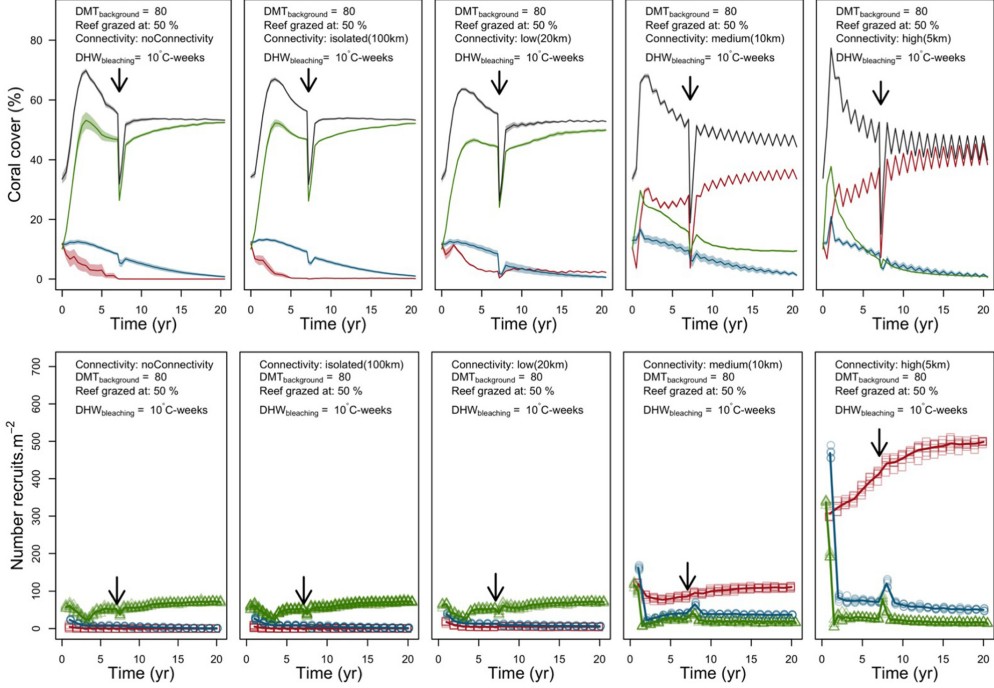

**Appendix 5—figure 17.** Evolution of cover (top) and number of recruits (bottom) of the three coral species from Western Atlantic (red for the competitive *Acropora palamata*, green for the weedy *Madracis pharensis*, blue for the stress-tolerant *Orbicella annularis* and black for total coral cover) for different larval connectivity levels. The black arrow indicates when the bleaching event occurred. Solid lines represent mean values over five replicates; shaded areas at the top show ± standard

error; individual symbols at the bottom show the number of recruits at a given time for one replicate.

## Eastern Pacific

Some of the results observed with this community contrast importantly with the ones described previously. Under no or low larval connectivity, the weedy species *P. damicornis* was not the dominant species, despite being also the only brooder species in the community. Instead, the stress-tolerant species *P. lutea* who competitively excluded both *P. damicornis* and the competitive species *P. elegans* (*Appendix 5—figure 18*). The superior competitiveness of *P. lutea* here is explained by its much higher resistance to waves (trait 8 in *Appendix 5—figure 10*). Both *P. damicornis* and *P. elegans* have branching colonies, which are dislodged by waves after reaching a certain size. *Porites lutea*, on the other hand, has massive colonies can sustain strong wave regimes, especially as they become larger. In contrast with the previous example, the capacity to recruit locally does not play an important role, and the recruitment rates remained low for all three species (*Appendix 5—figure 18*). The cover occupied by *P. lutea* increased mainly because of the vegetative growth of its colonies (*Appendix 5—figure 19*).

Under higher levels of larval connectivity, the competitive species excluded the other two species (*Appendix 5—figure 18*) for different reasons. *Pocillopora damicornis* was outcompeted because of its lower recruitment rate compared to the spawning species, while *P. lutea* has a lower growth rate and colony fecundity (traits 5 and 3 in *Appendix 5—figure 10*).

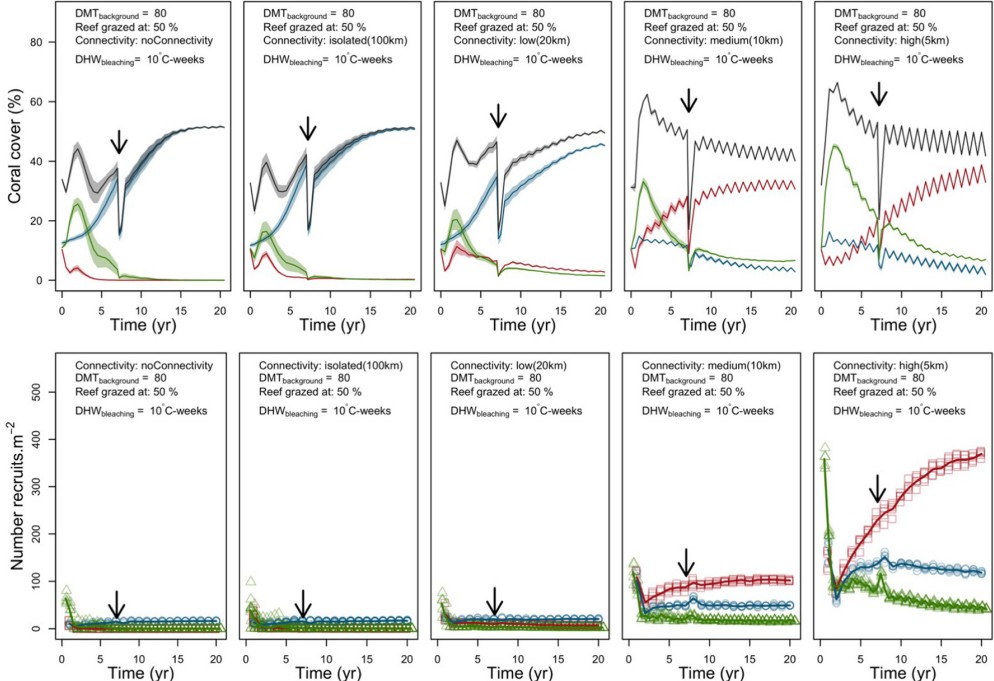

**Appendix 5—figure 18.** Evolution of the cover (top) and the number of recruits (bottom) of the three coral species from Eastern Pacific (red for the competitive *Pocillopara elegans*, green for the weedy *Pocillopora damicornis*, blue for the stress-tolerant *Porites lutea* and black for total coral cover) for different for different larval connectivity levels. The black arrows indicates when the bleaching event occurred. Solid lines represent mean values over five replicates; shaded areas at the top show ± standard error; individual symbols at the bottom show the number of recruits at a given time for one replicate.

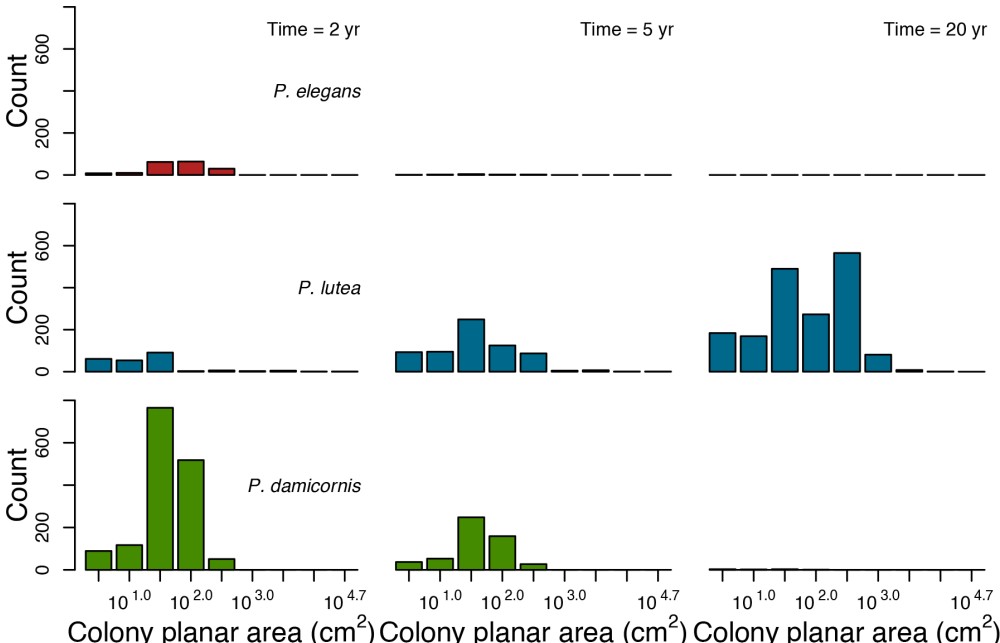

**Appendix 5—figure 19.** Colony size-frequency distributions of the Eastern Pacific species with no larval connectivity. Colonies with a colony planar area = 1 cm$^2$ are not displayed.

## 9. Grazing intensity

### 9.1. Objective and procedure

Here, we assessed the effect of grazing intensity on populations dynamics of both corals and algae. We enabled all the processes and used the same coral species, and initial percentage cover as in the previous section. We ran simulations for 20 years and triggered a bleaching event at year seven. We exposed the community to five different grazing regimes: 10, 30, 50, 70% and 90% reef cover permanently grazed. We kept constant larval connectivity, wave exposure and bleaching intensity (7000 larvae.m$^{-2}$, DMT$_{background}$ = 80 and DHW$_{bleaching}$ = 10 C°-weeks, respectively).

### 9.2. Expected patterns

A decrease of grazing pressure usually leads to an increase of algal biomass and cover and a decrease in coral cover (e.g. *Suchley and Alvarez-Filip, 2017*). The recovery of the coral community after a pulse disturbance should also be affected by a reduction of grazing intensity because less suitable space is available for vegetative growth and larval settlement (e.g. *Steneck et al., 2014*). Under prolonged insufficient grazing, new feedback processes establish and coral reefs switch to an alternative stable state dominated by algae (*Mumby, 2009*).

Under high grazing pressure, the algae community should be dominated principally by crustose coralline algae, because few herbivores species can consume its encrusting tissue. Turf algae can also persist in a cropped state, because their fast growth rate can compensate for their usually high palatability (e.g. *McClanahan, 1997*). With lower grazing pressure, a higher cover of turf cover and establishment of large macroalgae should be observed. Once established, these macroalgae can persist because of their lower palatability to many herbivores and higher longevity (*Steneck and Dethier, 1994*).

After a pulse disturbance, such as cyclone or bleaching, the available space is usually filled first by turf, followed by crustose coralline algae or macroalgae depending on the intensity of the grazing regime (*McClanahan et al., 2009*). The recovery of the coral community depends on the grazing regime, the coral cover surviving the pulse disturbance and larval connectivity (*Baker et al., 2008*). A high grazing pressure should provide enough available cover for corals to recover via vegetative growth and recruitment, provided that enough colonies survived locally and/or enough external larvae settle in the reef. Under more moderate grazing pressure, the fate of the coral community is

uncertain because a pulse disturbance can push the ecosystem into an alternative stable state dominated by algae. Even under substantial grazing pressure, a strong reduction of coral cover can lead to a large increase of algae cover, overwhelming herbivores ('dilution of herbivory') and allowing macroalgae to establish permanently (*Mumby et al., 2007*; *Williams et al., 2001*).

Predicting coral population dynamics in a community exposed to variable grazing regimes requires to consider coral-algae interactions at the species level because of the high functional diversity of the two taxa (even within functional groups), and the diverse mechanisms they use to compete (see Appendix 2: §7.5.3.1). Certain coral traits play a role in these interactions. Large colonies are for instance, less likely shaded or overtopped than smaller ones. Branching colonies entangles erected macroalgae and are consequently less affected by whiplash (*McCook et al., 2001*). Plating and branching colonies can overtop algae and avoid direct contact ('escape in height strategy'). Fast rates of calcification and tissue regeneration allow to outcompete certain algae (e.g. *Diaz-Pulido et al., 2009*). On the other hand, species having fast growth rates and small polyp size (i.e. characteristics of most competitive species) seem to be less competitive against many algae, potentially because of their limited capacity to increase heterotrophy (*Steneck et al., 2014*). Finally, disease-susceptible species seem to be more affected by allopathic algae (*Bonaldo and Hay, 2014*). Despite these coral traits being important to consider, coral-algae competitive outcomes might depend more on the functional characteristics of the algae (*Diaz-Pulido and McCook, 2008*).

## 9.3. Results

### Western Atlantic

As expected, we observed an algae and coral-dominated states under low and high grazing regimes, respectively (*Appendix 5—figure 20*). Under the lowest grazing pressure, fewer coral colonies surpassed 100 cm$^2$ (*Appendix 5—figure 22*) and the coral community relied essentially on external larval supply (*Appendix 5—figure 21*). Under higher grazing pressure, more coral colonies reached larger sizes due to the increase of available space (*Appendix 5—figure 23*, *Appendix 5—figure 24*). Coral recruitment rate was the highest at intermediate grazing level (*Appendix 5—figure 21*) because space was saturated by algae and coral colonies under the lowest and highest regimes, respectively. Finally, there was no hysteresis (as attested by the approximate equality between coral cover and surface grazed) because we did not implement feedback processes (*Appendix 5—figure 20*).

Turf dominated the algae community regardless of grazing pressure (*Appendix 5—figure 20*). Despite being the most palatable algae (see Appendix 2: §7.1.2), turf has the highest radial growth rate (250 mm.yr$^{-1}$) and consequently over-competed the other algae by pre-empting space more rapidly. In the model, all algae (except crustose coralline algae) cannot overgrow each other (see Appendix 2: §7.5.4.2). Macroalgae failed to establish because its lower palatability did not compensate for the difference in growth rate. In future, we can aim at solving the issue by, for instance, allowing macroalgae to overgrow turf or implementing a process of transition from turf to macroalgae similar to the one implemented by *Mumby et al. (2007)*. Crustose coralline algae failed to establish under all grazing regimes partly because of its very low radial growth rate (12 mm.yr$^{-1}$). We attempted to enhance crustose coralline algae competitiveness during the calibration by lowering its probabilities of being grazed and being covered by other algae but with no real improvement (see Appendix 3). In addition, crustose coralline algae is an inferior competitor against corals (Appendix 2: §7.5.3.2) and is a favorable substrate for coral settlement (Appendix 2: §7.2.1.3.b). Preventing crustose coralline algae from being grazed and overgrown by other algae or increasing its competitiveness against corals are solutions to investigate in future to help it reach realistic cover occupancy.

The available space created by bleaching-induced coral mortality was filled primely by coral recruits (*Appendix 5—figure 21*) rather than algae (*Appendix 5—figure 20*). This happened because *coral recruitment* precedes *growth* and *algae invasion* during a time step (*Figure 2*), which favors coral recovery, especially under high larval connectivity. Unfortunately, the model structure does not allow to implement these two processes simultaneously. Reducing further the competitiveness of coral recruits against algae could however solve the issue.

The competitive species *A. palmata* dominated the coral community under low to moderate grazing pressure (10% to 50%), because of its higher growth and recruitment rates (*Appendix 5—figure*

21). As discussed previously, algae indirectly mediate coral-coral competition in the model (see §4.3). There are more coral-algae interactions under lower grazing pressures, and *A. palmata*'s colonies can compensate better for the cover lost to algae due to their growth rate. In addition, *A. palmata* recruits more than (i) the brooding species *M. pharensis* because three times more larvae arrive form the regional pool and (ii) *O. annularis* because of a much higher growth rate, allowing *A. palmata* to reach more fecund colony sizes (*Appendix 5—figure 22*, *Appendix 5—figure 23*). Under higher grazing regimes (70% to 90%), *A. palmata* was over-competed by the weedy species *M. pharensis* (*Appendix 5—figure 20*), because of the relatively high wave exposure of the site and the higher resistance of *M. pharensis*' colonies (trait 8 in *Appendix 5—figure 10*). Precisely, *A. palmata*'s colonies were dislodged once reaching 66.7 cm$^2$ whereas *M. pharensis*' colonies were only limited by their colony maximum diameter (*Appendix 5—figure 24*). In addition, a lower algae cover implies a higher number of coral-coral interactions and *A. palmata* gradually conceded space to the other two species because it is the least aggressive species (trait 6 in *Appendix 5—figure 10*). The population of the stress-tolerant species *O. annularis* was stable because it is the most resistant and aggressive species but also the slowest.

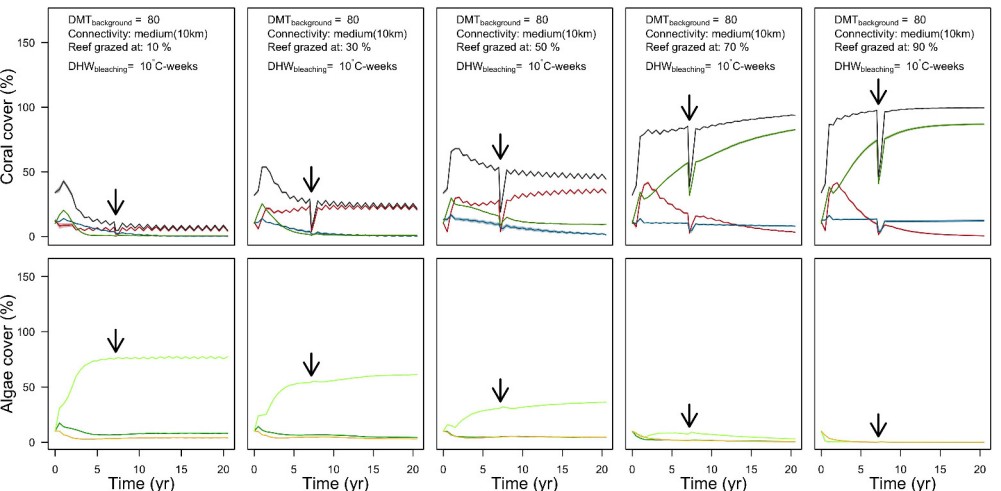

**Appendix 5—figure 20.** Evolution of cover of the three coral species from Western Atlantic (top: red for the competitive *Acropora palamata*, green for the weedy *Madracis pharensis*, blue for the stress-tolerant *Orbicella annularis* and black for total coral cover) and algae (bottom: light green for turf, dark green for macroalgae and orange for CCA) for different grazing intensities. The black arrow indicates when the bleaching event occurred. Solid lines represent mean values over five replicates; shaded areas at the top show ± standard error.

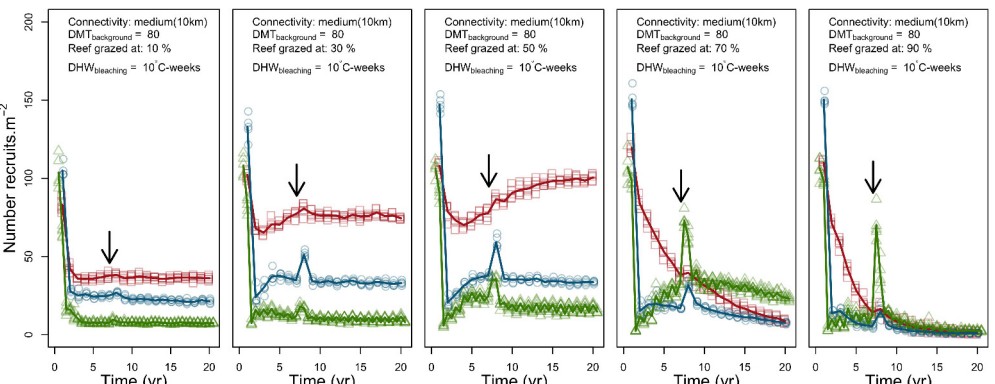

**Appendix 5—figure 21.** Evolution number of recruits of the three coral species from Western Atlantic (red for the competitive *Acropora palamata*, green for the weedy *Madracis pharensis*, blue for the stress-tolerant *Orbicella annularis* and black for total coral cover) for different grazing intensities.

The black arrow indicates when the bleaching event occurred. Solid lines represent mean values over five replicates; individual symbols show the number of recruits at a given time for one replicate.

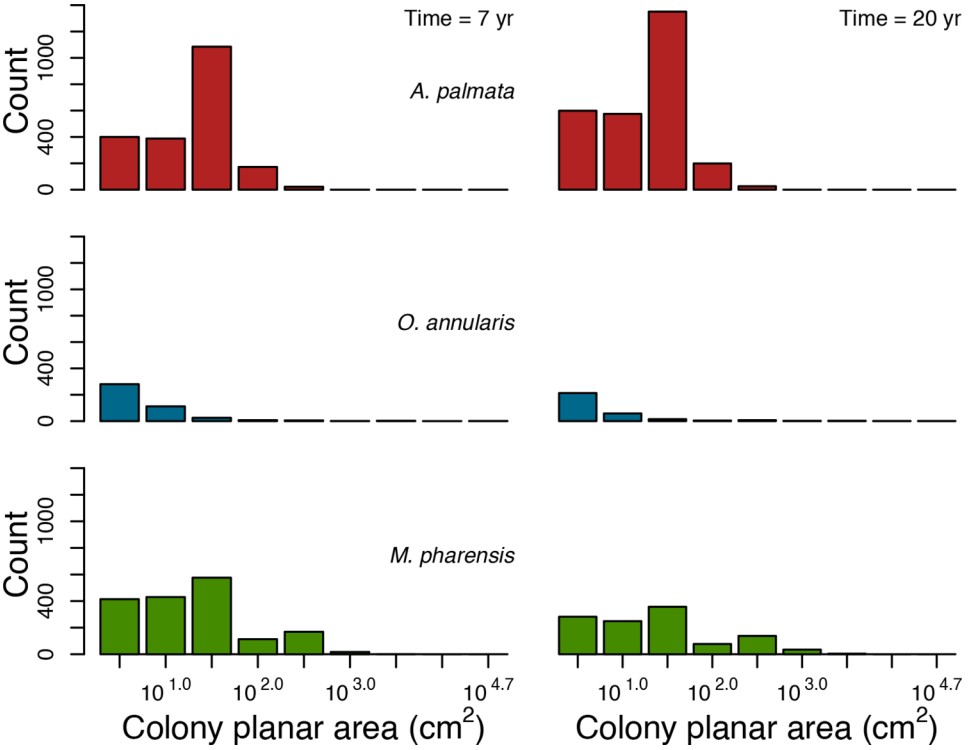

**Appendix 5—figure 22.** Colony size-frequency distributions of the Western Atlantic species at lowest grazing pressure (i.e. 10%). Colonies with a colony planar area = 1 cm$^2$ are not displayed.

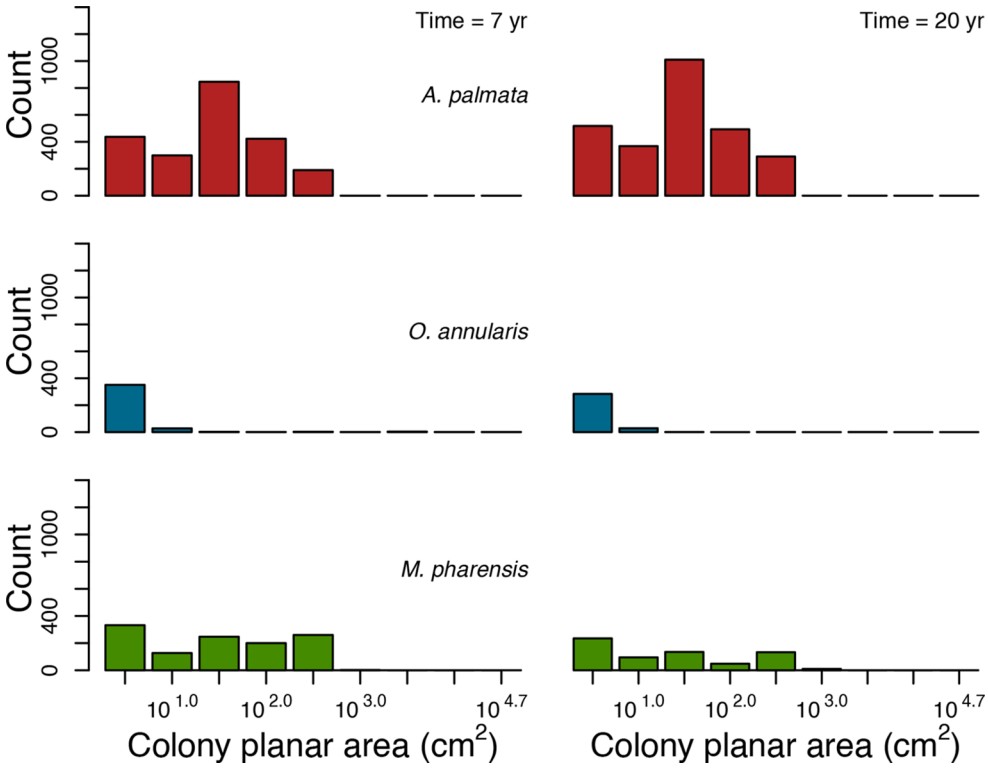

**Appendix 5—figure 23.** Colony size-frequency distributions of the Western Atlantic species at intermediate grazing pressure (i.e. 50%). Colonies with a colony planar area = 1 cm$^2$ are not displayed.

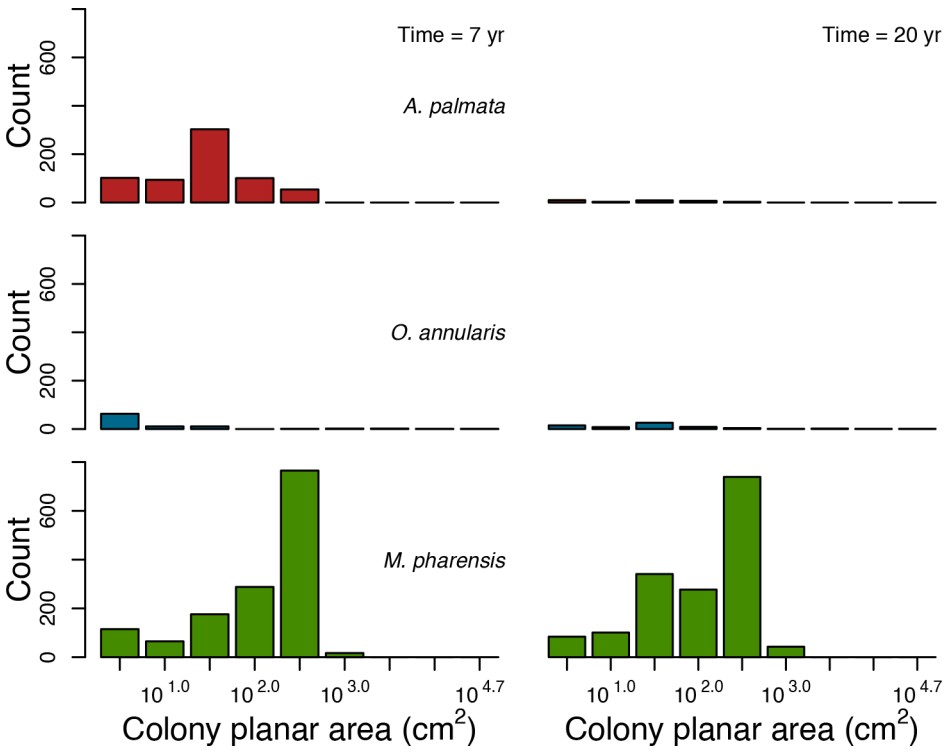

**Appendix 5—figure 24.** Colony size-frequency distributions of the Western Atlantic species at highest grazing pressure (i.e. 90%). Colonies with a colony planar area = 1 cm² are not displayed.

## Eastern Pacific

The results with the Eastern Pacific community are similar on many aspects to the previous results: turf and corals dominated under low and high grazing pressure, respectively (*Appendix 5—figure 25*); coral recruitment was higher at intermediate grazing levels (*Appendix 5—figure 26*); the increased available space after the disturbance was filled by coral recruits mainly; the competitive coral species *P. elegans* dominated the coral community under lower grazing regimes. Under higher grazing pressure, however, the stress-tolerant species *P. lutea* over-competed the other two species (*Appendix 5—figure 25*). Both *P. elegans* and the weedy species *P. damicornis* have branching colonies, which are dislodged once reaching a certain size. *Porites lutea* consequently progressively gained more surface cover by reaching higher colony size (*Appendix 5—figure 27*).

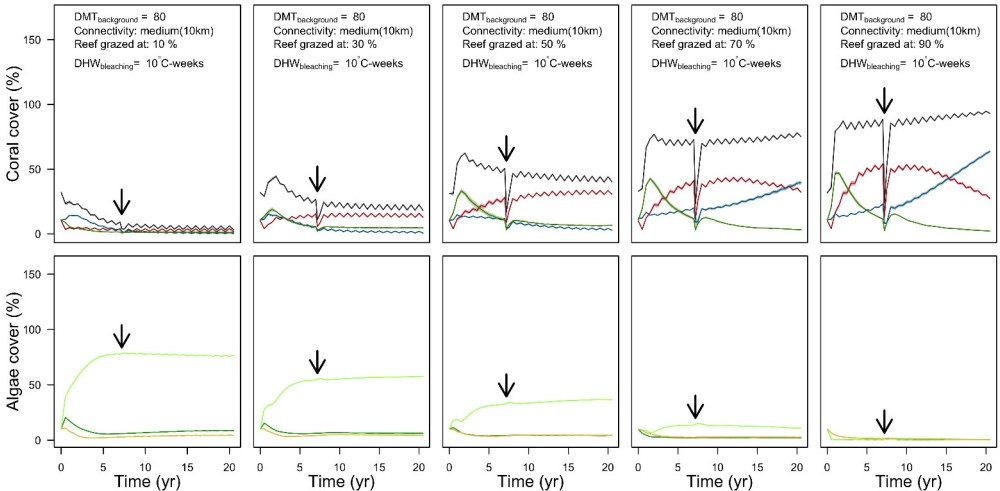

**Appendix 5—figure 25.** Evolution of cover of the three coral species from Eastern Pacific (red for the competitive *Pocillopara elegans*, green for the weedy *Pocillopora damicornis*, blue for the stress-

tolerant *Porites lutea* and black for total coral cover) and algae (bottom: light green for turf, dark green for macroalgae and orange for CCA) for different grazing intensities. The black arrow indicates when the bleaching event occurred. Solid lines represent mean values over five replicates; shaded areas at the top show ± standard error.

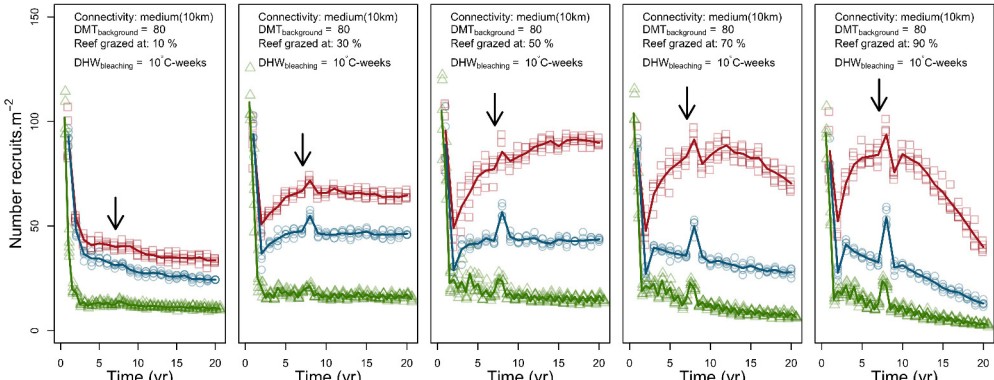

**Appendix 5—figure 26.** Evolution number of recruits of the three coral species from Eastern Pacific (red for the competitive *Pocillopara elegans*, green for the weedy *Pocillopora damicornis*, blue for the stress-tolerant *Porites lutea* and black for total coral cover) for different grazing intensities. The black arrows indicates when the bleaching event occurred. Solid lines represent mean values over five replicates; individual symbols show the number of recruits at a given time for one replicate.

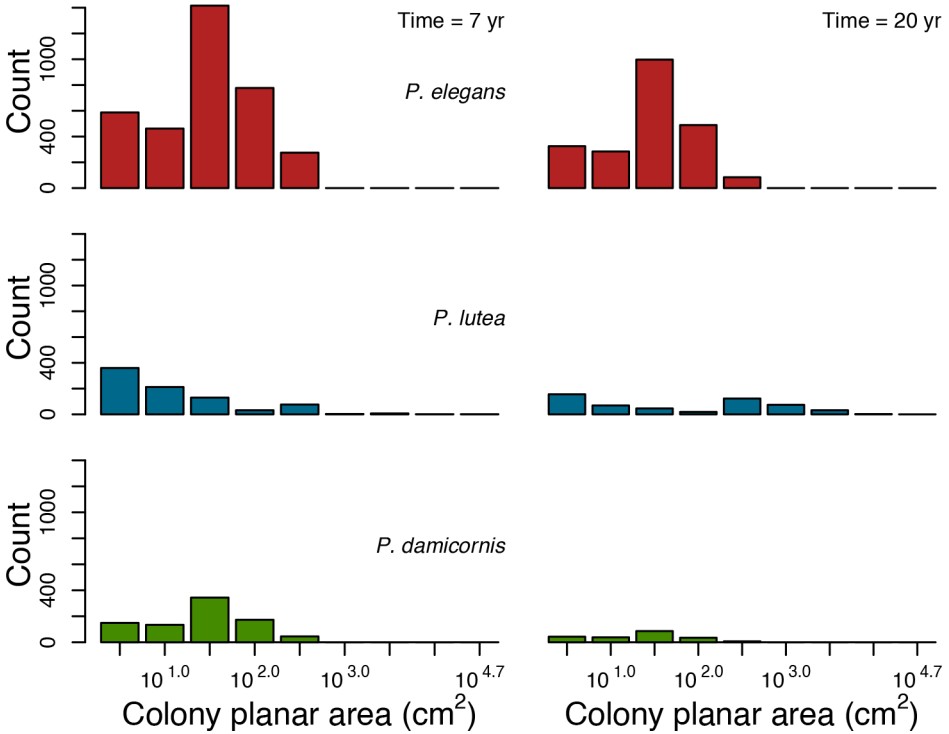

**Appendix 5—figure 27.** Colony size-frequency distributions of the Eastern Pacific species at highest grazing pressure (90%). Colonies with a colony planar area = 1 cm$^2$ are not displayed.

## Summary

The hierarchically structured validation showed that our implementation of the processes we selected yields ecologically relevant patterns. The patterns for vegetative growth and recruitment rate as a function of percentage cover and colony size distribution matched our expectations. The dynamics we observed at the community level are harder to validate with certainty because we,

ecologists, cannot predict precisely population dynamics because of to our lack of understanding of the numerous processes at play and their interactions. However, we provided ecologically relevant explanations for the emergent community dynamics we observed. For each factor tested (i.e. disturbance, grazing, connectivity), we could explain population dynamics considering simultaneously (i) functional differences among species, (ii) colony size distributions and (iii) the environmental context. The variations and magnitudes of the different variables we measured (i.e. % cover, number of recruits $m^{-2}$, colony size distributions) do not contrast with what we would expect in real systems. These proves that our model combines adequately functional traits and demographic approaches.

The validation procedure also revealed where the model can be improved. We summarize here the two main anomalies we observed. First, coral communities recovered pre-disturbance cover in 1 year, even in the most intense disturbance regime scenario. There are three modifications we could do: (1) reduce larval supply from the regional pool after the disturbance, (2) reduce coral larvae competitiveness against algae, and (3) implement feedback processes. Second, the algae community was dominated by turf algae, irrespectively of the grazing intensity. We could reduce turf competitiveness by decreasing the palatability of the other algae.

## Appendix 6

### 10. Method

We did a global sensitivity analysis on each of the three versions of the model that we calibrated with the Caribbean sites (Appendix 3). The goal was to estimate the model sensitivity during a process of recovery after a strong pulse-disturbance. We followed the recommendations of *Prowse et al. (2016)* for sampling the parameter space, choosing the number of replicates and the emulator and for the assessment of sampling sufficiency.

#### Related code

*Manuscript/Rscripts/ Appendix S6 - Global sensitivity analysis.R*
*Simulations/Rscripts/Global_sensitivity_analysis.R* and *Hierarchically_structured_validation_functions.R* (*Carturan et al., 2020*).

#### 10.1. Parameters, range of values and sampling

We selected ten of the twelve parameters considered in the calibration (we did not include *cyclone model* and *grazing model* because we used these to select site-specific disturbance regimes and are consequently not intrinsic model parameters; see Appendix 3). We added six parameters having potentially important effects in the execution of ecological processes implemented, and whose values are uncertain (*Appendix 6—table 1*). We defined for each parameter a range of continuous values centred on the nominal values (the calibrated values in the case of the ten first parameters) and we defined the size of the range considering the type of parameter (e.g. proportion, coefficient) and the results of the calibration (i.e. the range is wider when more than one value provided the best fitness during calibration). We then drew a Latin hypercube sample ($n$ = 1000) from the resulting parameter space using the *randomLHS* function from the R package `lhs` 0.16 (*Carnell, 2018*).

#### 10.2. Simulations

We ran the simulations for 10.5 years, with a bleaching event happening at year four at an intensity of 12°C-weeks. We kept constant grazing pressure (50%), wave hydrodynamic regime (dislodgement mechanical threshold = 120, which is equivalent to strong wave regimes that colonies experience at the reef crest), and larval input from regional pool (700 larvae m$^{-2}$). We defined the initial benthic composition as the one observed in the Caribbean sites during the first assessment. We ran each of the 1000 simulations only once (no replicate) as recommended (*Prowse et al., 2016*).

#### 10.3. Response variables

We assessed model sensitivity for five different, ecologically relevant response variables: (i) *coral cover* (total coral cover at 10.5 years); (ii) *difference cover* (difference between total coral cover at 10.5 years and that just after the bleaching event); (iii) *evenness* (Pielou's evenness at 10.5 years); (iv) *richness* (number of coral species having >1% cover at 10.5 years) and (v) *recruits* (total number of coral recruits m$^{-2}$ at 10.5 years).

#### 10.4. Emulators

We determined the influence of each parameter on the variability of the response variables by fitting boosted regression trees in the *gbm.step* function from the R package `dismo` 1.1–4 (*Hijmans et al., 2017*), setting the learning rate to 0.01, the bag fraction to 0.75, the tree complexity to 3, and optimized the number of fitted trees based on 10-fold cross-validation. We assumed a Gaussian error distribution for each of the five response variables.

#### 10.5. Estimation of sampling sufficiency

We estimated the sufficiency of the sample size by fitting boosted regression trees to random subsamples ($n$ = 100, 250, 500, 750) from the $n$ = 1000 set of parameter points generated. With higher sample sizes, we expected the influence of the parameters on the response variables to converge toward the same values. The difference of influence values for two consecutive sample sizes should

consequently decrease as sample size increases. To measure the difference of the parameters' influence between two samples, we used *De'ath (2012)*'s measure of community turnover:

$$D_\beta = e^{\sum_{j=1}^{2} \sum_{i=1}^{s} \frac{p_{ij}\ln(p_{ij})}{2} - \sum_{i=1}^{s} p_i\ln(p_i)}$$

where $j$ = one of the two consecutive samples, $i$ = one of the model parameters, $s$ = the total number of parameters, $p_{ij}$ = the influence of parameter $i$ obtained with sample $j$, and $p_i$ = the averaged value between $p_{i1}$ and $p_{i2}$. $D_\beta$ decreases asymptotically toward 1 as the parameters' influence converges towards equality. We considered that the sample size was sufficient if we observed $D_\beta$ asymptoting toward 1.

**Appendix 6—table 1.** Description of parameters used in the global sensitivity analysis and their respective ranges for each site (FB = Fond Boucher; PB = Pointe Borgnesse; IR = Ilet à Rats).

| Parameters | Description | Nominal value | Sampling interval |
|---|---|---|---|
| bleaching diff response | Value of the coefficient φ (see Appendix 4: §2.2): a smaller φ increases the interspecific difference for the probability of bleaching when the thermal stress increases; a larger φ reduces this difference | 3 | [2.5,3.5] |
| growth rate reduction interaction | Lateral growth rate reduction coefficient to apply when a coral colony or algae overgrows other colonies or algae | Eight for FB, IR two for PB | [6,10] for FB, IR [1,5] for PB |
| otherProportions | Coefficient $p_o$ reduces the number of larvae produced by all the colonies present in the reef (see Appendix 2: §7.2.1.1.d) | 0.0001 | [0.00005,0.00015] |
| prob cover crustose coralline algae | Probability that algae overgrow crustose coralline algae | 0.1, 0.25, 0.5, 0.75 | [0,0.05] |
| ratio overtop colony | Ratio needed for a branching or plating colony to overtop smaller colonies | 2 | [1.1,3] |
| prob grazing macroalgae | Probability of macroalgae being palatable | 0.7 for FB, PB 0.5, 0.7 for IR | [0.5,0.9] |
| prob grazing allopathic macroalgae | Probability of allopathic macroalgae being palatable | 0.3, 0.5 for FB, IR 0.3 for PB | [0.2,0.6] for FB, IR [0.1,0.5] for PB |
| prob grazing Halimeda | Probability of *Halimeda* spp. being palatable | 0.5 for PB 0.3 for IR | [0.2,0.7] for PB [0.1,0.5] for IR |
| prob grazing articulated coralline algae | Probability of articulated coralline algae being palatable | 0.7 | [0.5,0.9] |
| prob grazing crustose coralline algae | Probability of crustose coralline algae being palatable | 0.05, 0.1, 0.2, 0.3 | [0.01,0.1] |
| height big algae[*] | Height in cm of macroalgae, allopathic macroalgae and articulated coralline algae | 30 | [20,50] |
| height turf [*] | Height in cm of turf | 10 | [5,15] |
| height crustose coralline algae encrusting coral[*] | Height in cm of crustose coralline algae, encrusting and encrusting long upright corals | 2 | [0.5,4] |
| prob settle crustose coralline algae[*] | Probability of a coral larvae to settle successfully on crustose coralline algae agents | 0.5 | [0.2,0.8] |
| prob settle barren ground[*] | Probability of a coral larvae to settle successfully on barren ground agents | 0.5 | [0.2,0.8] |
| prob settle dead coral[*] | Probability of a coral larvae to settle successfully on dead coral agents | 0.5 | [0.2,0.8] |

*The six additional parameters not considered in the calibration.

## 11. Results

The parameters with the most important effects on the response variables were *growth rate reduction interaction* and *otherProportions* (*Appendix 6—figure 1*). They were followed by *prob settle barren ground, prob settle dead coral,* and in certain sites *prob grazing articulated macroalgae* and *prob grazing Halimeda*. The remaining six parameters did not have an important influence.

*Growth rate reduction interaction* mediates the impact that a superior taxon has on an inferior taxon when in direct competition by reducing the surface conceded. Consequently, higher *growth rate reduction interaction* is expected to increase *evenness* and *richness* in the case of coral-coral interactions. Algae such as turf, allopathic macroalgae and *Halimeda* spp. are stronger competitors compared to corals, so higher parameter values reduced their competitive advantage, which should have increased *coral cover* and *difference cover. Number of recruits* is expected to be positively correlated as well due to higher *coral cover*. As expected, *growth rate reduction interaction* was positively correlated with all response variables in all three sites, except for *evenness* at Pointe Borgnesse. This is explained by a nonlinear relationship between *evenness* and *growth rate reduction interaction* (i.e. the relationship is negative for *growth rate reduction interaction* <3.5 and positive when $\geq$3.5) and a range of smaller parameter values used for Pointe Borgnesse (*Appendix 6—table 1*; *Appendix 6—figure 2*).

The parameter *otherProportions* directly controls the number of larvae locally produced — higher values increase the proportion of competent larvae potentially setting in the reef. As expected, the parameter was positively correlated with *coral cover, cover difference* and *number of coral recruits*. It was negatively correlated with *richness* and *evenness* because higher values favoured a few (brooding) species that outcompeted most other species mainly because of a higher recruitment rate (*Appendix 6—figure 3*).

The probabilities of grazing algae were counterintuitively associated negatively with the response variables. These probabilities define the inter-algae difference of palatability and have to be compared to the grazing probability of turf algae, which was constant = 1. Increasing the grazing probability of algae consequently reduced the grazing pressure on turf, which affected coral colonies because turf is the most competitive algae with corals. There were between-site differences in the influence of some of these parameters: *prob grazing allopathic macroalgae* and *prob grazing Halimeda* had a stronger influence at Pointe Borgnesse and Ilet à Rats, respectively. These discrepancies arose because of the different ranges of values used between sites (*Appendix 6—table 1*) — a range of smaller values (e.g., *prob grazing allopathic macroalgae* at Pointe Borgnesse) allowed the corresponding population of algae to have a higher cover (*Appendix 6—figure 4*) and consequently, to influence the response variables more. Likewise, *prob grazing macroalgae, prob grazing articulated coralline algae* and *prob grazing articulated coralline algae* had little effect on the response variables because these algae occupied smaller portions of substratum (*Appendix 6—figure 4*).

Similar to *otherProportions*, the probabilities to settle on dead coral, barren ground and crustose coralline algae influence coral recruitment rate. We therefore expected them to be positively correlated with *coral cover, difference cover*, and *number recruits*, and negatively with *evenness* and *richness*. We observed these expected patterns with *prob settle barren ground* and *prob settle dead coral*, but crustose coralline algae cover was too low to allow *prob settle crustose coralline algae* to have had an effect (*Appendix 6—figure 4*).

The remaining parameters did not have much influence on the response variables because their implication in processes depended on certain population or community structures. In particular, *height big algae* and *height turf* did not have an effect because most of the branching colonies did not reach large enough sizes to overtop these algae, even under the lowest heights of algae (*Appendix 6—figure 5*). Likewise, branching colonies were too small to overtop other colonies, which prevented *ratio overtop colony* from affecting the response variables. *Height crustose coralline algae encrusting coral* and *prob cover crustose coralline algae* did not have an effect because crustose coralline algae was not abundant enough (*Appendix 6—figure 4*) and there was only one encrusting coral species present (only at Fond Boucher) and its cover remained close to zero (*Appendix 6—figure 6*).

Lastly, *bleaching diff response* ($\varphi$) had no effect on the response variables because in most simulations, most coral species had such small population sizes that they had little effect on the response

variables (e.g. averaging species cover at 10.5 years over the 1000 simulations revealed that only one or two coral species had a mean cover > 5%), and because the difference of bleaching susceptibility between the few dominant species was small with the range of values we defined for *bleaching diff response* (*Appendix 6—figure 7*).

Finally, we are confident that our sampling effort was sufficient because the measure of stability converged towards one for all five response variables and all three sites (*Appendix 6—figure 8*).

| | Coral cover | | | Evenness | | | Difference cover | | | Richness | | | Recruits | | |
|---|---|---|---|---|---|---|---|---|---|---|---|---|---|---|---|
| | FB | PB | IR | FB | PB | IR | FB | PB | IR | FB | PB | IR | FB | PB | IR |
| GR_reduction_interaction | 50.33 | 29.42 | 34.66 | 10.60 | 7.69 | 7.59 | 52.30 | 5.90 | 31.04 | 48.48 | 13.38 | 51.48 | 3.68 | 14.14 | 5.27 |
| otherProportions | 22.02 | 21.13 | 7.36 | 67.31 | 63.93 | 35.59 | 13.88 | 24.09 | 7.06 | 19.32 | 27.48 | 0.86 | 53.26 | 43.65 | 41.36 |
| proba_settle_Barren_ground | 12.55 | 11.49 | 8.85 | 8.55 | 19.36 | 37.12 | 2.64 | 10.25 | 4.63 | 10.28 | 6.38 | 7.56 | 19.27 | 19.04 | 23.92 |
| proba_grazing_AMA | 4.80 | 27.83 | 2.41 | 5.27 | 3.38 | 1.58 | 11.15 | 40.48 | 2.76 | 9.03 | 37.32 | 7.31 | 1.34 | 3.05 | 0.47 |
| proba_settle_Dead_coral | 4.58 | 7.60 | 5.31 | 1.24 | 2.70 | 6.07 | 7.50 | 14.14 | 7.38 | 1.44 | 3.31 | 0.80 | 20.88 | 18.14 | 26.16 |
| ratioAreaBranchingPlating_OvertopColonies | 2.83 | 0.19 | 0.38 | 0.82 | 0.93 | 0.56 | 2.52 | 0.27 | 0.69 | 0.75 | 1.46 | 0.64 | 0.58 | 0.17 | 0.26 |
| proba_Algae_coverCCA | 1.04 | 0.20 | 0.22 | 0.36 | 0.21 | 0.30 | 2.01 | 0.71 | 0.70 | 1.20 | 0.51 | 0.30 | 0.21 | 0.19 | 0.19 |
| proba_grazing_CCA | 0.96 | 0.15 | 0.18 | 0.70 | 0.06 | 0.59 | 2.14 | 0.21 | 0.70 | 4.30 | 1.22 | 0.56 | 0.16 | 0.16 | 0.23 |
| proba_grazing_MA | 0.17 | 0.15 | 0.25 | 0.86 | 0.18 | 0.35 | 1.14 | 0.55 | 0.33 | 0.76 | 1.08 | 0.43 | 0.12 | 0.19 | 0.09 |
| height_Turf | 0.13 | 0.12 | 0.14 | 1.30 | 0.09 | 0.61 | 0.58 | 0.18 | 0.30 | 0.85 | 0.94 | 0.23 | 0.07 | 0.15 | 0.04 |
| height_CCA_EncrustingCoral | 0.13 | 0.13 | 0.23 | 0.74 | 0.27 | 0.61 | 0.52 | 0.28 | 0.39 | 0.78 | 0.56 | 0.54 | 0.08 | 0.18 | 0.11 |
| height_BigAlgae | 0.12 | 0.15 | 0.14 | 0.99 | 0.28 | 0.50 | 0.45 | 0.21 | 0.33 | 0.56 | 1.30 | 0.14 | 0.07 | 0.22 | 0.14 |
| proba_settle_CCA | 0.12 | 0.11 | 0.21 | 0.25 | 0.18 | 0.30 | 0.94 | 0.19 | 0.28 | 1.35 | 0.86 | 0.60 | 0.08 | 0.10 | 0.23 |
| Bleaching_diff_response | 0.11 | 0.13 | 0.25 | 0.58 | 0.33 | 0.45 | 1.65 | 0.20 | 0.15 | 0.73 | 1.92 | 0.76 | 0.09 | 0.17 | 0.15 |
| proba_grazing_ACA | 0.10 | 0.13 | 0.14 | 0.42 | 0.17 | 0.18 | 0.58 | 0.30 | 0.32 | 0.17 | 0.78 | 0.11 | 0.11 | 0.12 | 0.16 |
| proba_grazing_Halimeda | NA | 1.06 | 39.27 | NA | 0.23 | 7.59 | NA | 2.04 | 42.93 | NA | 1.48 | 27.67 | NA | 0.33 | 1.20 |

**Appendix 6—figure 1.** Influence (%) of each model parameter (rows) on each of the five response variables (columns), for each site: Fond Boucher (FB), Pointe Borgnesse (PB) and Ilet à Rats. Red and blue colours represent positive and negative relationships between parameters and response variables, respectively. Colour saturation indicates the amount of influence of parameters.

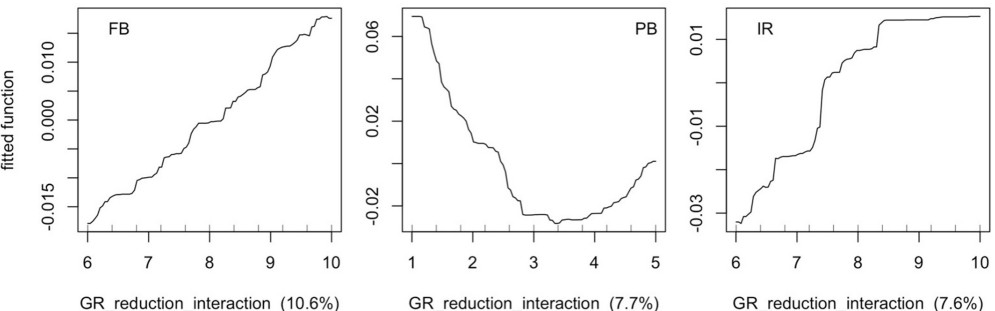

**Appendix 6—figure 2.** Relationship between the parameter *growth rate reduction interaction* and *evenness* predicted by the fitted boosted regression trees (fitted function) for each site (FB = Fond Boucher; PB = Pointe Borgnesse; IR = Ilet à Rats). Fitted functions are centred by subtracting their mean from each value.

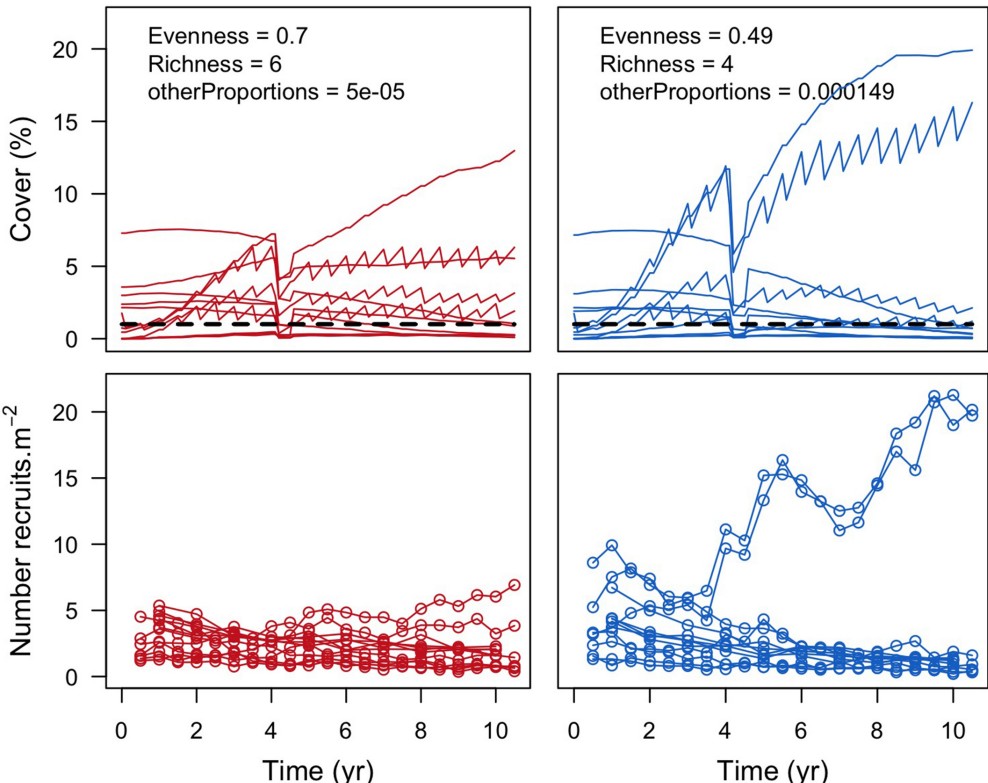

**Appendix 6—figure 3.** Example of a comparison of the effect of lower (left panels, red) and higher (right panels, blue) values of *otherProportions* on individual coral species cover (top panels) and number of recruits m$^{-2}$ at Fond Boucher. Each line represents the cover or number of recruits of one coral species (*n* = 12 species). The black dashed line in the top panels shows the 1% cover threshold below which species were not accounted for richness (at 10.5 years).

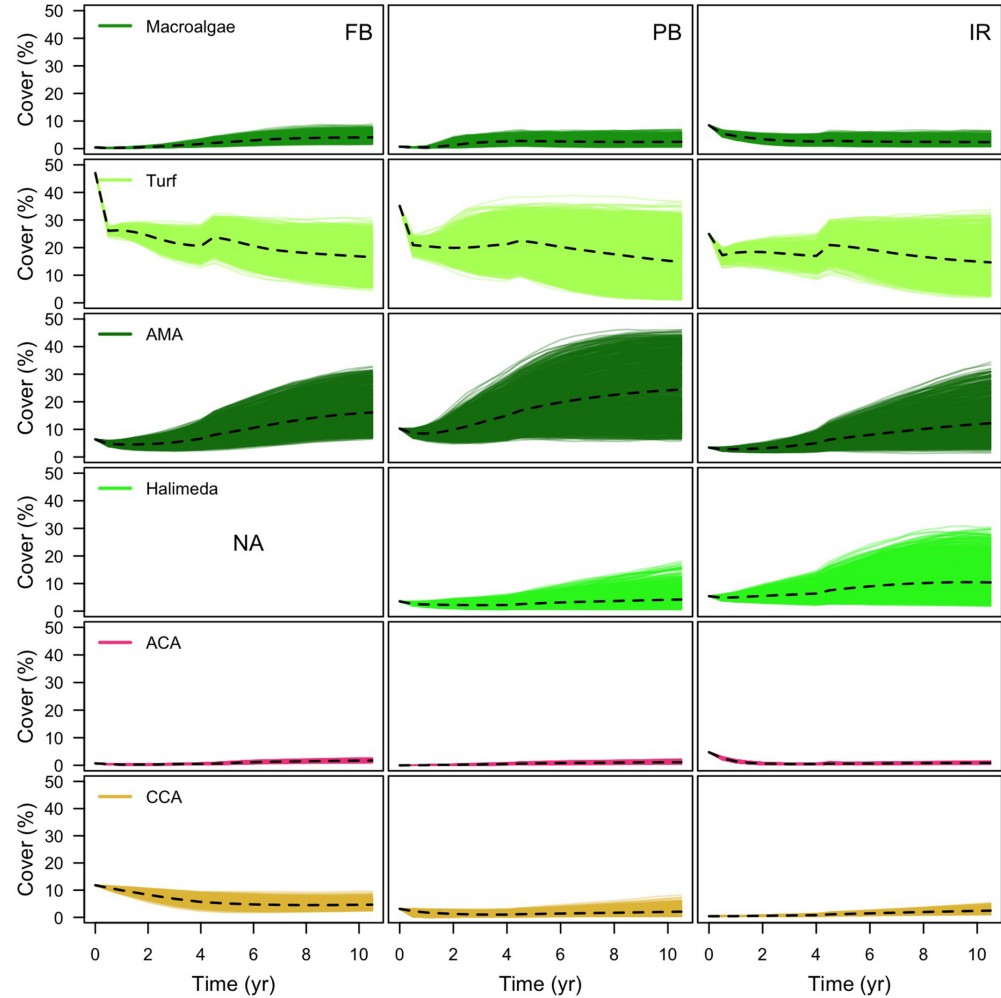

**Appendix 6—figure 4.** Comparison of algae cover between sites (FB = Fond Boucher; PB = Pointe Borgnesse; IR = Ilet à Rats). Each line represents the algal cover of one simulation; the dashed black line shoes the averaged cover over all the simulations (n = 1000).

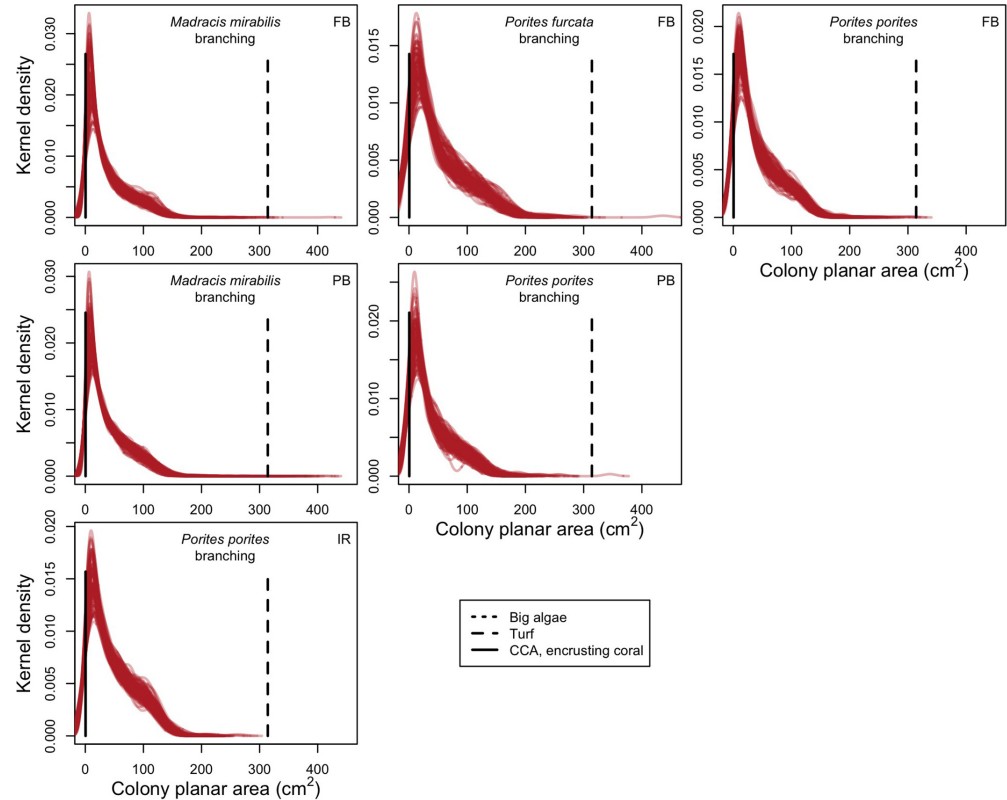

**Appendix 6—figure 5.** Examples of colony size distributions of branching species by sites (FB = Fond Boucher; PB = Pointe Borgnesse; IR = Ilet à Rats) at 8 years. Each red line represents a species' colony size distribution in one simulation; each plot shows 100 distributions that we randomly selected among the 1000 simulations. Horizontal black lines display the minimum colony planar area necessary to achieve for a colony to overtop the corresponding algae (CCA = crustose coralline algae; the line for *big algae* falls outside of the plots).

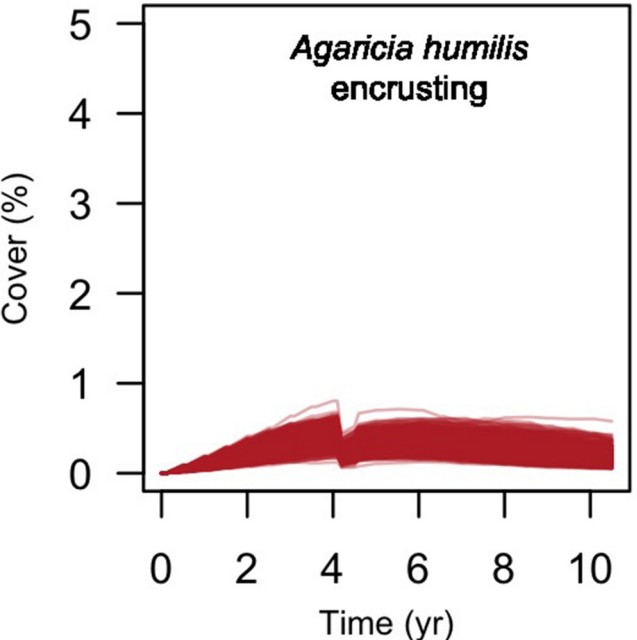

**Appendix 6—figure 6.** Cover of the encrusting coral species present at Fond Boucher. Each line represents the cover of one simulation (*n* = 1000).

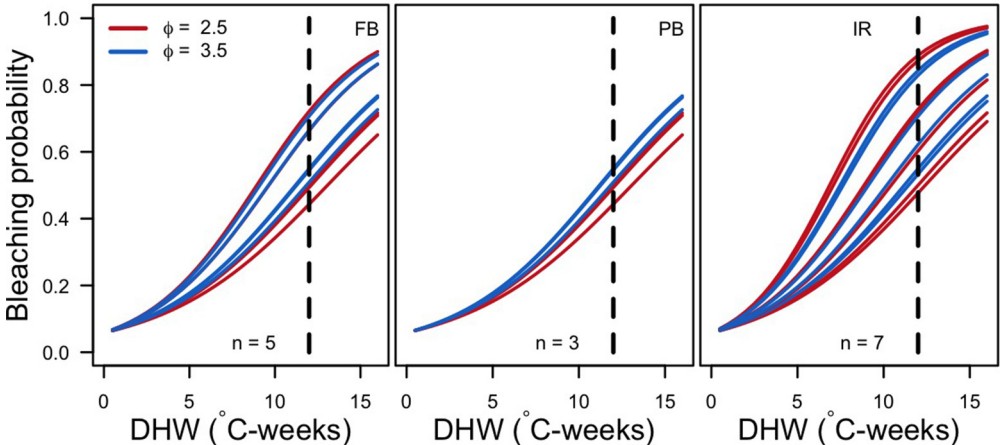

**Appendix 6—figure 7.** Bleaching probability of the most abundant species for the two extreme $\phi$ in each site (FB = Fond Boucher; PB = Pointe Borgnesse; IR = Ilet à Rats). We calculated the mean abundance at 10.5 years over the 1000 simulations and selected the $n$ species with a mean percentage cover >1%. The vertical dashed line shows the intensity of the thermal perturbation imposed in the simulations (12°C-weeks).

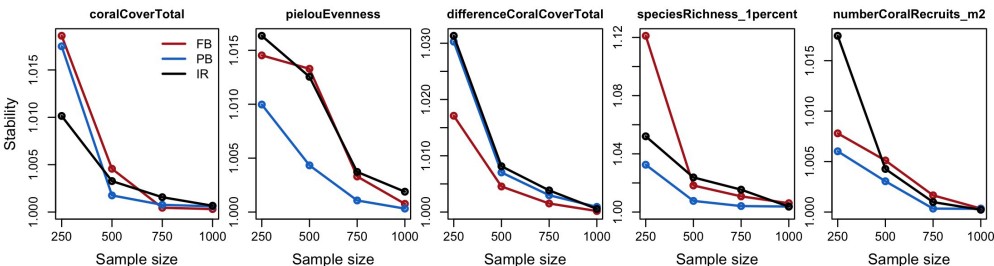

**Appendix 6—figure 8.** Measures of stability (turnover) between two consecutive sample sizes (i.e. 100–250; 250–500; 500–750; 750–1000) for each response variable and site (FB = Fond Boucher; PB = Pointe Borgnesse; IR = Ilet à Rats).

