## [Decision Letter]

**Acceptance summary:**

Your coral reef model uses traits and functional types of corals to represent not only taxonomic but also functional diversity. The supplement includes a comprehensive description of the design, calibration and testing of the model. Your work is an impressive demonstration of how it is possible to combine existing data in a systematic way, test a model at multiple levels, and thus demonstrate that trait-based agent-based models allow us to model the role of functional diversity. The model will be useful for addressing applied questions and, probably requiring some more development, for addressing theoretical questions regarding coexistence and the role of diversity for resilience.

**Decision letter after peer review:**

Thank you for submitting your article "A spatially explicit and mechanistic model for exploring coral reef dynamics" for consideration by *eLife*. Your article has been reviewed by three peer reviewers, including Volker Grimm as the Guest Editor and Reviewer #1, and the evaluation has been overseen by Ian Baldwin as the Senior Editor. The following individuals involved in review of your submission have agreed to reveal their identity: Hauke Reuter (Reviewer #2).

The reviewers have discussed the reviews with one another and the Guest Editor has drafted this decision to help you prepare a revised submission.

As the editors have judged, your manuscript is of interest, but as described below, additional information is required before it is published. Therefore, we would like to draw your attention to changes in our revision policy that we have made in response to COVID-19 (https://elifesciences.org/articles/57162). First, because many researchers have temporarily lost access to the labs, we will give authors as much time as they need to submit revised manuscripts. We are also offering, if you choose, to post the manuscript to bioRxiv (if it is not already there) along with this decision letter and a formal designation that the manuscript is 'in revision at *eLife*'. Please let us know if you would like to pursue this option. (If your work is more suitable for medRxiv, you will need to post the preprint yourself, as the mechanisms for us to do so are still in development.)

Summary:

This manuscript presents a new agent-based model of coral reefs that is designed to answer questions about the response of coral reefs to multiple stressors in a mechanistic, bottom-up way. The model uses traits and functional types of corals and algae to represent not only taxonomic but also functional diversity. The manuscript includes a very impressive description of the design, calibration and testing of a coral reef model. The authors have used the ODD protocol (to some degree), calibration of 12 model parameters for three empirical locations in the Caribbean, hierarchically structured validation, and global sensitivity analysis. Spatial interactions between corals and algae are represented in detail and allow to analyze relations between traits and functional responses and thus to depict realistic trajectories of reefs under different scenarios of external forcing.

Agent-based models are often criticized because of their complexity, which makes them difficult to parameterize, calibrate, test, and understand. This manuscript is an impressive demonstration of how it is possible to combine all relevant existing data in a systematic way, test a model at multiple levels, and thus demonstrate that, yes indeed, trait-based agent-based models allow us to model the role of diversity (see also this review: Zhakarova et al., 2019).

Essential revisions:

1) The Introduction takes a lot of space in discussing challenges to coral reefs. I guess virtually all papers about coral reefs start like this. It should be shortened, also because it raises the expectation that you are going to tackle these questions, which is not the case. Rather, this is a methods paper and you should come to this point more directly and perhaps list the challenges to ABMs for exploring diversity (see above) as the key challenge addressed in this manuscript.

2) If you say, in the Abstract, that the model “provides a virtual platform": Where can we download the software? Is there a manual describing the workflow needed for running the model and all its data scripts? Is the model description in the supplements complete? If not, this article would not really provide a tool. You might have a look at two examples where ABMs were presented, in journal articles, as tools. In both cases there was a full model description, a manual, and a download site: Becher et al., (2014) and Hradsky et al., (2019).

3) Subsection “Sources and software”: It is impressive to see all those packages and tools you used, but, ideally, you would also provide all, or the most important, scripts you wrote to run these packages and tools. If others are to use your virtual laboratory, they very likely would fail immediately because they would not know how to actually handle all those tools and data sources. I know that there is no culture yet to provide all relevant scripts, but – I think we should go there.

4) The ODD model description in the main text is not bad, but just a verbal summary description while the intention of ODD is to provide all information that is needed to re-implement the model. I understand that much of these details are in the Appendices, e.g. about Initialization and Submodels? It would be good if this link would be made more explicit by having a full ODD in the supplement, as a separate file. It would contain an augmented copy of the ODD of the main text and then just provide, in all detail, the information required for the seven elements of ODD. Why? Because the point of a standard is to follow it exactly so that readers, who either know the standard or learn about it, can easily find certain kinds of information at certain places in the model description. Currently, this is finding of relevant information is made unnecessarily complex. Examples of complete ODDs of complex model are provided by Ayllón et al., (2018) and Nabe-Nielsen et al., (2019).

For producing a complete ODD, please note that a new version of ODD has been published, which in particular has very detailed guidance, in the supplement, about ODD itself, summary ODDs, model narratives, etc.: Grimm et al., (2020). All that said, please note that we certainly do not require that you use ODD (because I am the main proponent of ODD), but any format, that compiles all information needed so that it is easy to find the kinds of information listed in ODD protocol, would be acceptable.

5) Scales: The model applications relate to a space of 5x5 m (25m^2^). I am not sure if such a small space allows for realistic dynamics if single corals grow large (> 2-3 m diameter) as then only a very low number of individuals would be present in the simulations potentially leading to artifacts in results. It is a pity that the spatial output of the model is not shown (except one specific figure in Appendix 5). I also see a discrepancy between the very high spatial (1cm) and the low temporal resolution (6 months). The time span within half a year could e.g. cover a mild bleaching event or other disturbances as well as processes of reef recovery leading to a different species composition and thus change the reef trajectory without being considered in the present model. I do not see that it is an argument, that the field data are only available in a low resolution of approx. 6 months. A comparison with model processes stays possible even if it is resolved higher.

6) It is apparent that all model runs cover only a very short time span of around ten years (21 simulation steps). This is extremely short for coral reefs which frequently undergo dynamics based on larger time scales. Thus, emerging dynamics and states, e.g., resulting from the sensitivity analysis, should be discussed with much care.

7) Overfitting? The model is very impressive, as it is possible to very closely possible represent the dynamics of measured reefs. However, I am not sure if this actually results from some overfitting. The model (runs) include some very strong and very specific influences of external drivers. For example, at the end of a time step certain values for grazing or sand cover are enforced. At least the impact of grazing results from a feedback with different reef processes. Thus, at least much of the trajectories in the model are the result of external drivers and it becomes difficult to analyze self-organization processes in the reef. In short: you cannot claim that a model is producing realistic dynamics due to a realistic representation of its internal organization if in fact the match between model output and observations is imposed by external drivers. A similar case occurred with honeybee colony models, where often the yearly time series of colony size was compared to data to claim that the model was realistic, but that time series was largely driven by the time series of the queen's egg-laying rate (Becher et al., 2013).

8) A major question thus is whether the authors believe that their model can better address large scale questions about coral reefs, such as their resilience to regime shifts from disturbances and climate change, than 'minimal' models, such as that of van de Leemput et al., (2016)?

9) In Carturan, Parrott and Pither, (2018) coral functional traits are classified as 'resistance' and 'recovery'. In the current manuscript, the terms 'stress tolerant', 'ruderal', and 'competitive' species (Grimes' classification) is used. Do 'resistance' species and 'recovery' species of Carturan et al., (2018) correspond to 'stress tolerant' and 'ruderal', respectively?

10) The Title is suboptimal: "mechanistic" and "spatially explicit" applies to hundreds of model, if not more, including coral reef models. The novelty of you work lies in merging the individual-based and trait-based approaches to represent functional diversity. The title should reflect this (but please observe *eLife*'s guidance on titles).

---

## [Author Response]

Summary:This manuscript presents a new agent-based model of coral reefs that is designed to answer questions about the response of coral reefs to multiple stressors in a mechanistic, bottom-up way. The model uses traits and functional types of corals and algae to represent not only taxonomic but also functional diversity. The manuscript includes a very impressive description of the design, calibration and testing of a coral reef model. The authors have used the ODD protocol (to some degree), calibration of 12 model parameters for three empirical locations in the Caribbean, hierarchically structured validation, and global sensitivity analysis. Spatial interactions between corals and algae are represented in detail and allow to analyze relations between traits and functional responses and thus to depict realistic trajectories of reefs under different scenarios of external forcing.Agent-based models are often criticized because of their complexity, which makes them difficult to parameterize, calibrate, test, and understand. This manuscript is an impressive demonstration of how it is possible to combine all relevant existing data in a systematic way, test a model at multiple levels, and thus demonstrate that, yes indeed, trait-based agent-based models allow us to model the role of diversity (see also this review: Zhakarova et al., 2019).Essential revisions:1) The Introduction takes a lot of space in discussing challenges to coral reefs. I guess virtually all papers about coral reefs start like this. It should be shortened, also because it raises the expectation that you are going to tackle these questions, which is not the case. Rather, this is a methods paper and you should come to this point more directly and perhaps list the challenges to ABMs for exploring diversity (see above) as the key challenge addressed in this manuscript.

While we agree that the introduction is lengthy for a methods paper, want to convince ecologists that we identified crucial research gaps that could be addressed using a modelling approach. We have, however, removed some of the ecological details and discussion of some of the bigger questions related to coral reef conservation that we do not specifically address in this paper.

As suggested, we have removed some ecological details at the end of the first paragraph of the Introduction. We did not modify the second (about the need to understand the effects of diversity on ecosystem functioning), third (about the need to understand the effects of diversity on ecosystem resilience), or the fourth paragraphs (about the need to develop a model like the one we present) because they each highlight the main research gaps that have motivated the development of our model.

As you suggested, we have now added a paragraph to the Introduction to justify our choice of using agent-based models to predict the behavior of coral diversity, and to present the challenges associated with them (i.e., they are difficult to parameterize, analyze and communicate), as well as the solutions (i.e., the development of trait databases, the overview, design concepts and details protocol, and the hierarchically structure validation).

2) If you say, in the Abstract, that the model “provides a virtual platform": Where can we download the software? Is there a manual describing the workflow needed for running the model and all its data scripts? Is the model description in the supplements complete? If not, this article would not really provide a tool. You might have a look at two examples where ABMs were presented, in journal articles, as tools. In both cases there was a full model description, a manual, and a download site: Becher et al., (2014) and Hradsky et al., (2019).

Yes, the model, the R scripts, the datasets used and produced, and instructions can be downloaded from the Open Source Framework (OSF) at http://dx.doi.org/10.17605/OSF.IO/CTQ43. The organization of the subdirectories, scripts and files allows one to reproduce the different simulations and analyses. We provided a README file that contains instructions about how to install the modelling platform (Repast Simphony), the model, R and how to reproduce the different simulations and output analyses.

We placed all the files on OSF soon after we submitted the manuscript to *eLife* and provided the URL after at the end of the manuscript. We now also provide the URL at the end of the Introduction.

3) Subsection “Sources and software”: It is impressive to see all those packages and tools you used, but, ideally, you would also provide all, or the most important, scripts you wrote to run these packages and tools. If others are to use your virtual laboratory, they very likely would fail immediately because they would not know how to actually handle all those tools and data sources. I know that there is no culture yet to provide all relevant scripts, but – I think we should go there.

As stated previously, we have provided all the R scripts we used to launch the model simulations and to analyse the results. At the top of each R script, we specified the goal of the script, the files used and produced.

To facilitate further reproducibility, we specified in each appendix the related R or Java scripts related to a given section.

4) The ODD model description in the main text is not bad, but just a verbal summary description while the intention of ODD is to provide all information that is needed to re-implement the model. I understand that much of these details are in the Supplement, e.g. about Initialization and Submodels? It would be good if this link would be made more explicit by having a full ODD in the supplement, as a separate file. It would contain an augmented copy of the ODD of the main text and then just provide, in all detail, the information required for the seven elements of ODD. Why? Because the point of a standard is to follow it exactly so that readers, who either know the standard or learn about it, can easily find certain kinds of information at certain places in the model description. Currently, this is finding of relevant information is made unnecessarily complex. Examples of complete ODDs of complex model are provided by Ayllón et al., (2018) and Nabe-Nielsen et al., (2019).For producing a complete ODD, please note that a new version of ODD has been published, which in particular has very detailed guidance, in the supplement, about ODD itself, summary ODDs, model narratives, etc.: Grimm et al., (2020). All that said, please note that we certainly do not require that you use ODD (because I am the main proponent of ODD), but any format, that compiles all information needed so that it is easy to find the kinds of information listed in ODD protocol, would be acceptable.

As suggested, we have created a complete and exhaustive overview, design concepts and details protocol (ODD) following Grimm and colleagues’ (2020) guide. Appendix 2 became the ODD (we placed all the information from the previous version in the sub-model section).

We also provided the path and name of the related Java scripts for each sub-model, so it is easier to find the code associated with a given process.

We updated the short ODD in the main manuscript and it now complies with the most recent instructions.

5) Scales: The model applications relate to a space of 5x5 m (25m^2^). I am not sure if such a small space allows for realistic dynamics if single corals grow large (> 2-3 m diameter) as then only a very low number of individuals would be present in the simulations potentially leading to artifacts in results. It is a pity that the spatial output of the model is not shown (except one specific figure in Appendix 5). I also see a discrepancy between the very high spatial (1cm) and the low temporal resolution (6 months). The time span within half a year could e.g. cover a mild bleaching event or other disturbances as well as processes of reef recovery leading to a different species composition and thus change the reef trajectory without being considered in the present model. I do not see that it is an argument, that the field data are only available in a low resolution of approx. 6 months. A comparison with model processes stays possible even if it is resolved higher.

The small spatial extent of the simulations we report is perhaps something of a limitation for exploring certain questions regarding coral community and metapopulation dynamics. However, we must stress that our model has no inherent limitation to the spatial extent of the modelled reef; larger extents are entirely possible and are limited only by available computational resources. We chose to use a small extent for the purposes of demonstrating and exploring the model’s behavior, because this size permitted a large number of reef configurations and model parameter combinations to be simulated with many repetitions within a reasonable amount of computation time.

Therefore, our choice was driven by pragmatic considerations related to computation time rather than any inherent limitation of the model itself. We do agree that in addition potentially to misrepresenting the population structure of large coral species, the small extent prevents implementing certain important processes such as larval connectivity along environmental gradients, which can facilitate species coexistence. We have now discussed these limitations at the end of the Discussion section.

As suggested, we have added one screen shot of a 25-m^2^ community in the main manuscript (Figure 1) and one in Appendix 2 (Appendix 2—figure 1). Note that a small screen shot of the spatial output is also in Figure 2 (previously, Figure 1). We have specified this in its figure caption.

We agree that a 6-month time step prevents modelling more subtle dynamics. We used a 6-month time step because this is the time period we had in the empirical dataset to validate the model; we kept this time step for the different simulations to be consistent and to save computation time. We highlight that a shorter time step can be selected (i.e., 3 or 4 months) (as specified in subsection “Entities, state variables, and scales”). It is completely possible to specify an even shorter time step; however, intervals of a few months seem reasonable considering that corals grow slowly and their reproductive cycles and disturbance regimes are seasonal. We discuss these choices and limitations in subsection “Entities, state variables, and scales”.

6) It is apparent that all model runs cover only a very short time span of around ten years (21 simulation steps). This is extremely short for coral reefs which frequently undergo dynamics based on larger time scales. Thus, emerging dynamics and states, e.g., resulting from the sensitivity analysis, should be discussed with much care.

This is a good observation. The model is designed to simulate complex community dynamics over short time periods, especially after a pulse-disturbance. We chose a short duration in the global sensitivity analysis (GSA) because we were focussing on the model sensitivity during the process of recovery. We estimated that six years was appropriate to measure the response variables—which describe diverse aspects of the diversity in communities—because the communities would more likely be at a different state compared to several years later (< 10 years), where most of them would be in a similar steady state. We added a sentence at the beginning the description of the global sensitivity analysis (subsection “Global sensitivity analysis”) and in Appendix 6 (Section 1. Method) to specify that goal of the GSA.

We also specified in the Discussion section that, more generally, the model should be used in the temporal scale for which it has been calibrated and that longer empirical time series are necessary to calibrate the model if one wants to predict over longer periods.

7) Overfitting? The model is very impressive, as it is possible to very closely possible represent the dynamics of measured reefs. However, I am not sure if this actually results from some overfitting. The model (runs) include some very strong and very specific influences of external drivers. For example, at the end of a time step certain values for grazing or sand cover are enforced. At least the impact of grazing results from a feedback with different reef processes. Thus, at least much of the trajectories in the model are the result of external drivers and it becomes difficult to analyze self-organization processes in the reef. In short: you cannot claim that a model is producing realistic dynamics due to a realistic representation of its internal organization if in fact the match between model output and observations is imposed by external drivers. A similar case occurred with honeybee colony models, where often the yearly time series of colony size was compared to data to claim that the model was realistic, but that time series was largely driven by the time series of the queen's egg-laying rate (Becher et al., 2013).

The main objective of our model is to represent realistic population dynamics of coral species and algae functional groups, including their interactions and response to pulse disturbances. We agree that imposing time series of environmental factors during the model calibration limits the range of possible community dynamics. However, the range of possibilities with 17 to 18 interacting populations is still considerable, especially considering the high number of ecological processes implemented. Given the range and variability of different patterns observed, we are confident that the model behavior is the emergent outcome of calibrated agent-level processes.

Furthermore, our global sensitivity analyses revealed that some of the most influential parameters on the emergent community dynamics were interaction-induced reductions in growth rate, a coefficient controlling the number of larvae locally produced and probabilities of successful larval settlement on different substrata. Thus, the fact that model predictions mainly derive from parameters associated to self-organization processes (interspecific competition and recruitment) suggest that over-fitting is not occurring.

We agree that implementing feedback processes related to certain environmental factors would increase the ecological realism of the model. We actually did implement the feedback process between the rugosity of the reef created by coral colonies and grazing pressure. But we decided to deactivate that process for the different analyses because we estimated that the data we had in hand were insufficient (validating the model with the feedback process activated would be possible with more reliable estimates of herbivorous fish population densities, and reef rugosity). We address this comment in the Discussion section.

Although not implemented in the simulations we report, we included the description of the rugosity-grazing feedback process in the subsection “sub-model” in the main manuscript and in Appendix 2 (subsection “The rugosity-grazing feedback (optional)”) for completeness. We also included the process in the flow chart (Figure 3).

8) A major question thus is whether the authors believe that their model can better address large scale questions about coral reefs, such as their resilience to regime shifts from disturbances and climate change, than 'minimal' models, such as that of van de Leemput et al., (2016)?

As previously discussed, we acknowledge that this version of the model has limitations due to the small spatial extent that can be represented and the short period for which it has been validated (addressed these limitations in the Discussion section). These limitations constrain the types of research questions that the model can address, and among these research questions, several have been addressed by minimal models. The principal advantage of our model is that it can account for the effects of diversity, and specifically functional diversity, on the system’s dynamics. By comparing our model’s results with those from minimal models, we could consequently assess if or which diversity-related ecological details are important to predict community dynamics and resilience to disturbances. We addressed this comment at the end of the Discussion section.

Another observation worth mentioning is that disturbance regimes in many coral reefs around the world are occurring at much higher frequencies than historically observed (e.g., the Great Barrier Reef has experienced three mass bleachings in the last five years). This most likely implies that ‘stability’ as such is rarely attained, so shorter-term recovery dynamics are more ecologically meaningful, as are the implications for managing these recoveries.

9) In Carturan, Parrott and Pither, (2018) coral functional traits are classified as 'resistance' and 'recovery'. In the current manuscript, the terms 'stress tolerant', 'ruderal', and 'competitive' species (Grimes' classification) is used. Do 'resistance' species and 'recovery' species of Carturan et al., (2018) correspond to 'stress tolerant' and 'ruderal', respectively?

Apologies for the confusion. No, “effect”, “resistance” and “recovery” qualify functional traits and not species. We used the effect, resistance and recovery traits framework to select the traits and processes to implement in the model. We used Grime’s competitive, stress-tolerant and ruderal (CSR) functional group classification to define a priori expectations of the community dynamics in the hierarchically structured validation. We clarified the difference between the two classifications and how we used them in the main text (subsection “Hierarchically structured model validation”).

10) The Title is suboptimal: "mechanistic" and "spatially explicit" applies to hundreds of model, if not more, including coral reef models. The novelty of you work lies in merging the individual-based and trait-based approaches to represent functional diversity. The title should reflect this (but please observe eLife's guidance on titles).

We agree. We have changed the title to: “Combining agent-based, trait-based and demographic approaches to model coral-community dynamics”. We added ‘demographic approaches’ because the resistance, fecundity and competitiveness of an individual colony in the model depend on its planar area. This is an important aspect, which is in alignment with a few foundational papers we cite at the beginning of the Discussion section (i.e., Edmunds et al., 2014; Salguero-Gómez et al., 2018; Violle et al., 2007; line 305).